# Provably Transformers Harness Multi-Concept Word Semantics for Efficient In-Context Learning

**Dake Bu**[1], **Wei Huang**[2]\*, **Andi Han**[2], **Atsushi Nitanda**[3,4], **Taiji Suzuki**[5,2],
**Qingfu Zhang**[1], **Hau-San Wong**[1]\*

[1]*Department of Computer Science, City University of Hong Kong, Hong Kong SAR*
[2]*Center for Advanced Intelligence Project, RIKEN, Japan*
[3]*CFAR and IHPC, Agency for Science, Technology and Research (A⋆STAR), Singapore*
[4]*College of Computing and Data Science, Nanyang Technological University, Singapore*
[5]*Department of Mathematical Informatics, the University of Tokyo, Japan*
dakebu2-c@my.cityu.edu.hk, {wei.huang.vr, andi.han}@riken.jp,
atsushi_nitanda@cfar.a-star.edu.sg, taiji@mist.i.u-tokyo.ac.jp,
{qingfu.zhang, cshswong}@cityu.edu.hk

## Abstract

Transformer-based large language models (LLMs) have displayed remarkable creative prowess and emergence capabilities. Existing empirical studies have revealed a strong connection between these LLMs' impressive emergence abilities and their in-context learning (ICL) capacity, allowing them to solve new tasks using only task-specific prompts without further fine-tuning. On the other hand, existing empirical and theoretical studies also show that there is a linear regularity of the multi-concept encoded semantic representation behind transformer-based LLMs. However, existing theoretical work fail to build up an understanding of the connection between this regularity and the innovative power of ICL. Additionally, prior work often focuses on simplified, unrealistic scenarios involving linear transformers or unrealistic loss functions, and they achieve only linear or sub-linear convergence rates. In contrast, this work provides a fine-grained mathematical analysis to show how transformers leverage the multi-concept semantics of words to enable powerful ICL and excellent out-of-distribution ICL abilities, offering insights into how transformers innovate solutions for certain unseen tasks encoded with multiple cross-concept semantics. Inspired by empirical studies on the linear latent geometry of LLMs, the analysis is based on a concept-based low-noise sparse coding prompt model. Leveraging advanced techniques, this work showcases the exponential 0-1 loss convergence over the highly non-convex training dynamics, which pioneeringly incorporates the challenges of softmax self-attention, ReLU-activated MLPs, and cross-entropy loss. Empirical simulations corroborate the theoretical findings.

## 1 Introduction

Recently, a variety of transformer-based large language models (LLMs) have demonstrated remarkable performance across a broad spectrum of machine learning tasks, including natural language understanding [1], symbolic reasoning [2], and even heuristics design [3, 4]. One crucial emerging ability of these models is their in-context learning (ICL) capacity [5], which allows them to learn from a few demonstrations and conduct predictions on new queries without requiring any further

---

\*Corresponding authors

38th Conference on Neural Information Processing Systems (NeurIPS 2024).

fine-tuning. However, the current theoretical understanding of the mechanisms underlying this ICL capability remains limited, leaving the reasons for the remarkable emergence and generalization power of transformer-based LLMs in unseen ICL tasks largely unexplained.

In line with traditional topic models [6], [7, 8] propose that latent concepts / topics underlie natural texts, providing a Bayesian inference framework to elucidate the ICL mechanism via Bayesian Model Averaging (BMA) approach. On the other hand, theoretical and empirical studies have shown that transformer-based models exhibit linear geometric regularities in their latent representations as a result of concept or topic learning [9, 10], where the representations *within-concept* have positive inner products while representations *cross-concepts* exhibit near-orthogonal relationships. This structured semantic geometry has been well-documented in recent research on pre-trained LLMs [11, 12, 10, 13]. However, the connection between this observed multi-concepts latent geometric structure and the LMs' remarkable ICL capabilities remains unclear. Separately, recent theoretical analyses have modeled ICL as a martingale process driven by latent "concept" variables [14, 15]. Yet, these studies have not incorporated the observed multi-concept semantic regularity into their analyses, nor have they discussed the strong out-of-distribution (OOD) ICL abilities exhibited by transformers.

Additionally, existing theoretical work on transformer has been conducted on unrealistic, oversimplified settings, such as linear or ReLU transformers [16, 17, 18, 19], MLP-free attention-only models [16, 20], QK-combined softmax attention [19, 20, 21, 22, 23], unrealistic infinite dimensional assumption [14, 19, 21, 24] and impractical loss functions like square loss [9, 16, 25, 20, 26] and hinge loss [27, 28]. Furthermore, existing works have only been able to derive linear or sub-linear convergence rates for the 0-1 loss.

Therefore, there is a need for a more advanced analysis that can bridge the understanding between the multi-concept semantic regularity and the mechanisms underlying transformer-based ICL. This naturally leads to the research question:

---
**Essential Questions**

Whether and how do the geometric regularity of the multi-concept-encoded representation facilitate transformer in conducting efficient ICL?

---

To answer the above question, following the meaningful data modeling ideas in [9, 29], we conduct theoretical analysis on a concept-specific sparse coding prompt distribution for classification tasks, where the sparse latent variable encodes the information denoting the word's belonging concept. Importantly, the features in both the word's and label's dictionaries exhibit concept-specific geometric properties - within-concept positive inner products and cross-concept orthogonal geometric properties - that aligns with the findings in [9, 10, 11]. Our main contributions are highlighted as below.

1. First, we provide a comprehensive analysis of the learning dynamics for a two-layer transformer model, comprising one attention layer followed by a ReLU-activated feed-forward network, which is trained using the cross-entropy loss via stochastic gradient descent over a concept-specific sparse coding prompt distribution. Leveraging advanced analytical techniques, we showcase the asymptotic properties governing the coupled learning dynamics of the attention and MLP layers.

2. To the best of our knowledge, we are the first to prove an exponential convergence of the 0-1 loss over this challenging setting. Despite the highly non-convex optimization landscape, we demonstrate that the transformer can achieve Bayes optimal test error with just a logarithmic number of iterations.

3. We provably show how the multi-concept encoded linear semantic geometry can enable transformer to efficiently perform certain out-of-distribution ICL tasks. This offers an intuitive explanation for why transformer-based LLMs are able to successfully leverage the polysemous nature of words to tackle diverse, unseen concept-specific tasks, aligning well with users' practical experiences. Furthermore, our analysis takes a step forward in providing a potential theoretical underpinning for the innovative capabilities of LLMs, encompassing their ability to achieve cross-concept knowledge intersection. We believe our findings provide an initial positive response to Question 5.1.4 in the ICML 2024 position paper [30], which asks whether the observed latent geometry of LLMs can explain their OOD extrapolation abilities.

## 2 Related Work

**Theory of Exponential Convergence Rate of Stochastic Gradient Descent.** Our analysis of the exponential convergence rate for the 0-1 loss builds upon prior work linking the excess risk and essential supremum norm to exponentially fast convergence under the "hard low-noise condition" [31, 32]. This phenomenon has been further explored in more recent studies analyzing the exponential convergence of stochastic gradient descent (SGD) [33, 34, 35, 36, 37], as well as in more generalized settings such as multiclass classification [38] and support vector machines [39].

**Feature Learning in Learning Theory.** Recent works in learning theory have extensively studied structured data from a *feature learning* perspective, examining NN's feature direction reconstruction and noise memorization as a proxy for training or 0-1 loss convergence [40, 41, 42]. While prior studies often assumed orthogonal features, recent efforts have analyzed non-orthogonal scenarios [43, 44]. Our work extends this line-of-research to challenging nonlinear Attention-MLP transformers with non-orthogonal structured data representations.

**Theory of Transformers and In-Context Learning** The literature on Transformers and ICL is wide-ranging, and we will selectively address the most relevant ones. Prior studies have analyzed how transformers learn topic/concept semantics [9], the origins and biases of LLM representations using latent variable models [10], and ICL from a model averaging perspective [14]. However, albeit incorporating concept variables, these works do not connect the geometric properties of concept-encoded representations to transformers' powerful ICL abilities. Another line of research has studied the learning dynamics of ICL, including analyses of linear transformers [17, 19], QK-combined attention-only models [45], and multi-head softmax attention over linear regression without MLP [25]. Though relevant, these works rely on simplifications and do not notice the connection between semantic regularity and powerful ICL. While [28] also analyzes the learning dynamics of transformers with softmax attention and ReLU MLPs for in-context classification tasks, making it the most relevant prior work, our analysis differs in several key aspects. Specifically, (i) they consider orthogonal dictionary learning with a single label vector, in contrast to our non-orthogonal concept-encoded dictionaries for both words and labels; (ii) their technique requires a large batch size (at least $\varepsilon^{-2}$, where $\varepsilon$ is the test error) and long context lengths, which are not required in our result; and (iii) they utilize an impractical hinge loss and only achieve linear convergence without a relation to $\varepsilon$, whereas we analyze the more practical cross-entropy loss and derive an exponential convergence rate in terms of the test error $\varepsilon$. However, we note that this is only an informal comparison due to the differences in the models and primary findings. A detailed Related Work Section is deferred to Appendix C.

## 3 Problem Setup

**Notations.** For $l_2$ and Frobenius norms we utilize $\| \cdot \|$ and $\| \cdot \|_F$ to denote their computations. Considering two series $a_n$ and $b_n$, we denote $a_n = O(b_n)$ if there exists positive constant $C > 0$ and $N > 0$ such that for all $n \geq N$, $|a_n| \leq C |b_n|$. Similarly, we denote $a_n = \Omega(b_n)$ if $b_n = O(a_n)$ holds, and $a_n = \Theta(b_n)$ if $a_n = O(b_n)$ and $a_n = \Omega(b_n)$ both hold. Our $\mathbb{1}(\cdot)$ is to denote the indicator variable of an event. In addition, we denote $\text{span}(v_1, v_2, \ldots, v_k)$ as the linear subspace spanned by the vectors $v_1, v_2, \ldots, v_k$, and $\text{conic}(v_1, v_2, \ldots, v_k)$ denotes the conic hull (the set of all non-negative linear combinations) of the vectors $v_1, v_2, \ldots, v_k$.

### 3.1 Data Distribution

The data distribution employed in this study draws inspiration from a range of empirical and theoretical research works [9, 10, 46, 47, 48]. This distribution captures context-awareness and can be viewed as a specialized prompt version of PLSA [49] and LDA [6]. In this distribution, each word and label has multiple feature embeddings, each embedding corresponding to a different concept. This is achieved through the use of a sparse latent concept/topic variable, which happened to be particularly adept at representing language polysemy [47]. Adhering to the LLM representation explored in [9, 10], the features in both the word and label dictionaries maintain orthogonality across concepts and positive inner products within concepts. Additionally, the distribution incorporates Gaussian noise accounting for linguistic ambiguity or the imperfection of the LLM's representation.

**Definition 1.** *Polysemous Word Model* $(\mathcal{D}_{\boldsymbol{x}}, \mathcal{D}_{\boldsymbol{y}}, \mathcal{D}_{\boldsymbol{z}}, \mathcal{D}_{\xi_{\boldsymbol{x}}}, \mathcal{D}_{\xi_{\boldsymbol{y}}})$. *We assume there exists $K_1$ task-relevant concepts, each characterized by two semantically-opposite word's feature vectors $\boldsymbol{\mu}_{k_1}^+$ and*

$\boldsymbol{\mu}_{k_1}^-$, and their corresponding label's feature vectors $\boldsymbol{q}_{k_1}^+$ and $\boldsymbol{q}_{k_1}^-$, $\forall k_1 \in [K_1]$. There are also $K_2$ task-irrelevant concepts denoted by $\nu_{k_2}$, $\forall k_2 \in [K_2]$. The word samples $\boldsymbol{x} \in \mathbb{R}^{d_{\mathcal{X}}}$ and their labels $\boldsymbol{y} \in \mathbb{R}^{d_{\mathcal{Y}}}$ are generated from distributions parameterized by a shared latent concept variable $\boldsymbol{z} = (z_1, \cdots, z_K) \in \{0,1\}^K (K < d_{\mathcal{X}})$ capturing the concept-specific information:

$$\boldsymbol{z} \sim \mathcal{D}_{\boldsymbol{z}}, \quad \xi_{\boldsymbol{x}} \sim \mathcal{D}_{\xi_{\boldsymbol{x}}} = \mathcal{N}(\boldsymbol{0}, \sigma_\xi^2 \mathbf{I}_{d_{\mathcal{X}}}), \quad \xi_{\boldsymbol{y}} \sim \mathcal{D}_{\xi_{\boldsymbol{y}}} = \mathcal{N}(\boldsymbol{0}, \sigma_\xi^2 \mathbf{I}_{d_{\mathcal{Y}}}),$$

$$\boldsymbol{x} = \mathbf{M}\boldsymbol{z} + \xi_{\boldsymbol{x}} \sim \mathcal{D}_{\boldsymbol{x}}, \quad \boldsymbol{y} = \mathbf{Q}\boldsymbol{z} + \xi_{\boldsymbol{y}} \sim \mathcal{D}_{\boldsymbol{y}},$$

where the feature dictionary $\mathbf{M} = [\boldsymbol{\mu}_1^+, \boldsymbol{\mu}_1^-, \boldsymbol{\mu}_2^+, \boldsymbol{\mu}_2^-, \cdots, \boldsymbol{\mu}_{K_1}^+, \boldsymbol{\mu}_{K_1}^-, \boldsymbol{\nu}_1, \boldsymbol{\nu}_2, \cdots, \boldsymbol{\nu}_{K_2}] \in \mathbb{R}^{d_{\mathcal{X}} \times K}$ exhibits positive inner products within concepts and orthogonality across concepts, and the label dictionary $\mathbf{Q} = [\boldsymbol{q}_1^+, \boldsymbol{q}_1^-, \boldsymbol{q}_2^+, \boldsymbol{q}_2^-, \cdots, \boldsymbol{q}_{K_1}^+, \boldsymbol{q}_{K_1}^-, 0, \cdots 0] \in \mathbb{R}^{d_{\mathcal{Y}} \times K}$ has similar geometric properties. Specifically, we have $\forall k_1 \in [K_1], k_2 \in [K_2], \|\boldsymbol{\mu}_{k_1}^{\pm}\| = \|\boldsymbol{\nu}_{k_2}\| = \|\mathbf{u}\|, \|\boldsymbol{q}_{k_1}^{\pm}\| = \|\mathbf{q}\|$, and there exist constants $0 < \kappa_{\boldsymbol{x}}, \kappa_{\boldsymbol{y}} < 1$ such that $0 < \langle \boldsymbol{\mu}_{k_1}^+, \boldsymbol{\mu}_{k_1}^- \rangle \le \kappa_{\boldsymbol{x}} \|\mathbf{u}\|^2$ and $0 < \langle \boldsymbol{q}_{k_1}^+, \boldsymbol{q}_{k_1}^- \rangle \le \kappa_{\boldsymbol{y}} \|\mathbf{q}\|^2$.

The detailed formal definition can be found in Appendix E. By this definition, a single word or label can possess different features corresponds to different concepts. The illustration of Figure 1 in [12] can be an example, where the "Dog" vector in the representation space of LLM is decomposed to a direct sum of orthogonal vectors: "[Animal] + [Mammal] + $\cdots$", and we can see "[Animal]" belongs to the concept "Organism's Category" categorized into labels "[Animal]" and "[Plant]", and "[Mammal]" belongs to the concept of "Animal's Category" characterized by labels "[Mammal]", "[Fish]", "[Bird]", "[Reptile]". Besides, Figure 1 in [46] can also be a good support for our modeling, where "Ferrari" vector consists of "[Cars] + [Italian] + $\cdots$".

The following definition models the contextual prompts via specifying the statistical property of $\boldsymbol{z}$ among in-context words, which is a special prompt version of PLSA [49] and LDA [6]. The detailed formal version is available in Appendix E.

**Definition 2.** *Concept-specific Contextual Prompt Distribution*[2]. *During training, each prompt sample $S = \boldsymbol{x}_1, \boldsymbol{y}_1, \cdots, \boldsymbol{x}_L, \boldsymbol{y}_L, \boldsymbol{x}_{L+1}$ would share at least one co-concept, which is drawn from a mixture distribution $\mathcal{D}_S$ defined as:*

$$\mathcal{D}_S = \sum_{k=1}^{K_1} \left( \pi_k^+ \mathcal{P}_{k,L+1}^+ + \pi_k^- \mathcal{P}_{k,L+1}^- \right), \tag{1}$$

*where $\mathcal{P}_{k,L+1}^{\pm}$ denotes the $k$-th concept-specific prompt distribution, and $\pi_k^{\pm} = (2K_1)^{-1}$ denotes the equal chance of a sample to belong to $\mathcal{P}_{k,L+1}^{\pm}$. Specifically, a sample $S_n \sim \mathcal{P}_{k,L+1}^e, e \in [\pm]$ means that the query's label $\boldsymbol{y}_{L+1}^n$ is $\boldsymbol{q}_k^e$, and we denote $y_{S_n} := e$ as the real value label of this prompt. In addition, every demonstration pairs $(\boldsymbol{x}_l^n, \boldsymbol{y}_l^n), l \in [L]$ in $\mathcal{P}_{k,L+1}^e$ contain either $(\boldsymbol{\mu}_k^+, \boldsymbol{q}_k^+)$ or $(\boldsymbol{\mu}_k^-, \boldsymbol{q}_k^-)$ with equal chance. Also, every $\boldsymbol{z}_l^n, l \in [L+1]$ would satisfy $\mathbb{P}(z_{l, \neg(2k-1 \vee 2k)}^n = 1) = K^{-1}$, denoting the equal chance to have diverse features other than the current co-concept of the $\mathcal{P}_{k,L+1}^e$.*

This definition suggests that for prompt $S$ sampling from $\mathcal{D}_S$, there exists $e \in [\pm]$, $k \in [K_1]$, such that all the word-label pairs in this prompt share the $k$-th concept as their co-concept, and the corresponding real value label of the query in this prompt is $e$. Besides, the real value label of each word-label pair in the demonstration would have equal chance to be $+1$ or $-1$.

### 3.2 Transformer Model

Following [17, 20, 28], our embedding $\mathbf{E}(\cdot)$ of prompt $S$ is formulated as $\mathbf{H}$:

$$\mathbf{H} = \mathbf{E}(S) = \begin{pmatrix} \boldsymbol{x}_1 & \boldsymbol{x}_2 & \cdots & \boldsymbol{x}_L & \boldsymbol{x}_{\text{query}} \\ \boldsymbol{y}_1 & \boldsymbol{y}_2 & \cdots & \boldsymbol{y}_L & \boldsymbol{0} \end{pmatrix} := (\mathbf{h}_1, \mathbf{h}_2, \cdots, \mathbf{h}_{\text{query}}) \in \mathbb{R}^{(d_{\mathcal{X}} + d_{\mathcal{Y}}) \times (L+1)},$$

The learning model is a single-head, one-layer Transformer with one self-attention layer and one two-layer perceptron. Mathematically, it can be expressed as follows:

$$f(\mathbf{H}; \Psi) = \mathbf{r}^\top \sigma_R \left( \mathbf{W}_O \, \text{attn}(\mathbf{H}; \Psi) \right),$$

$$\text{attn}(\mathbf{H}; \Psi) = \sum_{l=1}^{L} \mathbf{W}_V \mathbf{h}_l \sigma_S \left( (\mathbf{W}_K \mathbf{h}_l)^\top \mathbf{W}_Q \mathbf{h}_{\text{query}} \right),$$

---

[2]Our theory allows for a broader range of the probability settings stated in the training prompt distribution, but for the sake of simplicity in presentation, we here chose a feasible one.

where $\sigma_R(\cdot) := \text{Relu}(\cdot), \sigma_S(\cdot) := \text{softmax}(\cdot), \mathbf{W}_Q, \mathbf{W}_K \in \mathbb{R}^{m_{qk} \times (d_{\mathcal{X}} + d_{\mathcal{Y}})}, \mathbf{W}_V \in \mathbb{R}^{m_v \times (d_{\mathcal{X}} + d_{\mathcal{Y}})}$ are the embedding matrices for queries, keys, and values, respectively, and $\mathbf{W}_O \in \mathbb{R}^{m \times m_v}$ and $\mathbf{r} \in \mathbb{R}^m$ are parameters in the MLP layer. Typically, $\min(m_{qk}, m_v) \geq d_{\mathcal{X}} + d_{\mathcal{Y}}$. $\Psi := \{\mathbf{W}_Q, \mathbf{W}_K, \mathbf{W}_V, \mathbf{W}_O, \mathbf{r}\}$ denotes the set of all model weights.

**Training Setting**. We fix one layer in both the attention and MLP layers to scrutinize the training dynamics more rigorously. Specifically, we let

$$\mathbf{W}_Q = \begin{pmatrix} \mathbf{W}_Q^{\boldsymbol{x}} & * \\ * & * \end{pmatrix}, \quad \mathbf{W}_K = \begin{pmatrix} \mathbf{W}_K^{\boldsymbol{x}} & * \\ * & * \end{pmatrix}, \quad \mathbf{W}_V = \begin{pmatrix} * & * \\ * & \mathbf{W}_V^{\boldsymbol{y}} \end{pmatrix} \quad \mathbf{W}_O = (* \quad \mathbf{W}_O^{\boldsymbol{y}}),$$

where $\mathbf{W}_Q^{\boldsymbol{x}}, \mathbf{W}_K^{\boldsymbol{x}} \in \mathbb{R}^{d_{\mathcal{X}} \times d_{\mathcal{X}}}, \mathbf{W}_V^{\boldsymbol{y}} \in \mathbb{R}^{(m_v - d_{\mathcal{X}}) \times d_{\mathcal{Y}}}, \mathbf{W}_O^{\boldsymbol{y}} \in \mathbb{R}^{m \times d_{\mathcal{Y}}}$. Here, we set the elements other than $\mathbf{W}_Q^{\boldsymbol{x}}, \mathbf{W}_K^{\boldsymbol{x}}, \mathbf{W}_V^{\boldsymbol{y}}$ and $\mathbf{W}_O^{\boldsymbol{y}}$ to be zero. Besides, we fix $\mathbf{W}_V^{\boldsymbol{y}}$ to be $\mathbf{I}_{(m_v - d_{\mathcal{X}}) \times d_{\mathcal{Y}}}$. We sample $\mathbf{r}_i$ from a uniform distribution $\text{Unif}\{-1, 1\}$ and fixed during the training process. Based on this setting, the trainable part we need to consider is actually $\Psi' := \{\mathbf{W}_Q^{\boldsymbol{x}}, \mathbf{W}_K^{\boldsymbol{x}}, \mathbf{W}_O^{\boldsymbol{y}}\}$. This problem remains highly non-convex and challenging.

We utilize mini-batch with-replacement SGD to train the transformer model. The empirical cross-entropy loss for each batch $\mathcal{B}_t$ is written as

$$L_{\mathcal{B}_t}(\Psi) = L_{\mathcal{B}_t}(\Psi') := \frac{1}{B} \sum_{n \in \mathcal{B}_t} \ell(y_{S_n} \cdot f(\mathbf{H}; \Psi)) + \frac{\lambda}{2}\|\Psi'\|_F^2,$$

where $\ell(z) = \log(1 + \exp(-z))$, $y_{S_n}$ is the real value label of the prompt defined in Definition 2, and the term $\|\Psi'\|_F^2$ represents $\|\mathbf{W}_Q^{\boldsymbol{x}}\|_F^2 + \|\mathbf{W}_K^{\boldsymbol{x}}\|_F^2 + \|\mathbf{W}_O^{\boldsymbol{y}}\|_F^2$, which is the $L_2$ regularization term with $\|\cdot\|_F$ denoted as the Frobenius norm. The purpose of the regularization in this paper is to accelerate and stabilize the mini-batch with-replacement SGD. The learning step is set to be $\eta_t = \frac{2}{\lambda(\gamma + t)}$, where $\gamma$ is an offset parameter. This decaying schedule is standard and also used in prior work [34, 50, 51] studying convergence of SGD. The whole procedure is in Algorithm 1.

**Initialization Setting.** All initial values of $\mathbf{W}_O^{\boldsymbol{y}}$ are sampled from a i.i.d. Gaussian distributions with mean 0 and variance $\sigma_1^2$. The initialization of $\mathbf{W}_Q^{\boldsymbol{x}}$ and $\mathbf{W}_K^{\boldsymbol{x}}$ are diagonal matrices $\sigma_0 \mathbb{I}$, which are also adopted in other work that consider training $\mathbf{W}_Q$ and $\mathbf{W}_K$ separately [25, 28].

**Testing Setting**. The model performance is measured by 0-1 test error on a test prompt distribution $\mathcal{D}^*$:

$$L_{\mathcal{D}^*}^{0-1}(\Psi) := \mathbb{P}_{S \sim \mathcal{D}^*}[(y_S \cdot f(\mathbf{E}(S); \Psi)) < 0]. \tag{2}$$

---

**Algorithm 1** Training algorithm

---

**Input:** Training distribution $\mathcal{D}_S$, Test distribution $\mathcal{D}^*$, Batch size $B$, step size $\eta_t = \frac{2}{\lambda(\gamma + t)}$, stopping criterion $\varepsilon$ and total epochs $T$.
Initialize model parameters $\Psi'^{(0)}$.
**for** $t = 0, 1, \ldots, T - 1$ **do**
  If $L_{\mathcal{D}^*}^{0-1}(\Psi^{(t)}) \leq \varepsilon$ stop else continue.
  Randomly sample mini batches $\mathcal{B}_t$ of size $B$ from $\mathcal{D}_S$.
  Update model parameters: $\Psi'^{(t+1)} = \Psi'^{(t)} - \eta_t \nabla_{\Psi'} L_{\mathcal{B}_t}(\Psi'^{(t)})$.
**end for**

---

## 4   Theoretical Results

In this section, we present our main theoretical results, which is based on the following conditions. We consider the learning iterations $0 \leq t \leq T^*$, where $T^* = \Omega(m^{-1}\sigma_0^{-1}\sigma_1^{-1}m\lambda^{-2}K_1\|\mathbf{q}\|^2((L - 1)\|\mathbf{u}\|^2 + 1)\log(\varepsilon^{-1}))$ denotes the maximum admissible iteration.

**Condition 1.** *Suppose that there exists a sufficiently large constant $C$, such that the following hold:*

  *1.  $d_{\mathcal{X}}, d_{\mathcal{Y}} \geq \max\{C\log(KLBT^*/\delta), K\}, d_{\mathcal{Y}} \geq C\log(m/\delta), m \geq C\log(K/\delta)$.*

2. $\gamma \geq C \max\{\|\mathbf{q}\|^2/(mK_1\lambda), 10/\lambda\}, \lambda \leq \min\{(C\log(Km/\delta)\|\mathbf{q}\|)^{-1}, (C\sigma_0/2\|\mathbf{u}\|^2)^{-1}\}$

3. $K \geq \{CK_1, C\|\mathbf{u}\|/(\sigma_\xi\sqrt{d_{\mathcal{X}}})\}$.

4. $\sigma_\xi \leq \min\{\lambda m/(C\sqrt{d_{\mathcal{X}}}\|\mathbf{u}\|\|\mathbf{q}\|^{1/2}), \|\mathbf{q}\|/(C\sqrt{d_{\mathcal{Y}}})\}$.

5. $\sigma_0 \leq \sqrt{K^{-1}\log(\frac{\|\mathbf{u}\|^2}{\lambda K_1}\log(\frac{\|\mathbf{q}\|^2}{m\lambda K_1}))}/(C\|\mathbf{u}\|)$,
   $\sigma_1 \leq \min\{(C\sigma_0\|\mathbf{u}\|^4\|\mathbf{q}\|\sqrt{\log(5Km/\delta)}/K_1)^{-1}, w^{*2}/(Cm^{3/2}\|\mathbf{q}\|)\}$.

*Here,* $w^* = \dfrac{1 - e^{-\sigma_0{}^2(1-\kappa_{\boldsymbol{x}})^2\|\mathbf{u}\|^4/2}}{1 + e^{-\sigma_0{}^2(1-\kappa_{\boldsymbol{x}})^2\|\mathbf{u}\|^4/2}}.$

Note that we do not have any requirement upon demonstration length $L$ and batch size $B$ for training, thus the training can be really flexible compared with the strict requirement in [28]. The condition on dimensionality $d_{\mathcal{X}}, d_{\mathcal{Y}}$ and the network width $m$ ensure the learning problem is in a sufficiently overparameterized setting [41, 42, 52, 43]. The condition on $\gamma$ ensures the learning step to be small and thus learning process enjoys an approximation to gradient flow. The condition on the small $\lambda$ is to ensure the model's sufficient learning before being stuck by regularization [53]. The condition on $K$ is to control the impact of cross-concept contribution in the Attention's learning dynamic, which can actually be relaxed at the cost of a denser analysis. The condition on $\sigma_\xi$ is to ensure that the gradient flows be mildly influenced by the noise. Last but not least, the conditions on $\sigma_1$ guarantee that the initial beliefs of MLP is small and the gradients of SGD can update the model effectively. A more detailed discussion over the parameter settings is delayed to Appendix H.

**Theorem 2.** *Exponential Convergence of 0-1 loss. Under Condition 1, define*

$$\nu := \min\{2\sqrt{2}\sigma_1/(1+\kappa_{\boldsymbol{y}}), \sigma_0(1-\kappa_{\boldsymbol{x}})e^{-\log(5Km/\delta)\frac{\sigma_1^2\|\mathbf{u}\|^4(1+e^{-\sigma_0^2\|\mathbf{u}\|^2})}{(1-e^{-\sigma_0^2\|\mathbf{u}\|^2})}}\}.$$

*Then, for* $\forall \varepsilon > 0$ *there exist some positive constants* $C_1$ *and* $C_2$*, with probability no less than* $1-\delta$*, for* $T \geq \hat{T} = C_1\sigma_1 m\lambda K_1\gamma\sqrt{(1+\kappa_{\boldsymbol{y}})\log(5Km/\delta)}/w^{*2}(1-\kappa_{\boldsymbol{y}})\|\mathbf{q}\|$*, we have*

$$L_{\mathcal{D}^*}^{0-1}(\Psi^{(T)}) \leq \exp(-\frac{C_2\nu^2 m\lambda^2(\gamma+T)}{K_1\|\mathbf{q}\|^2((L-1)\|\mathbf{u}\|^2+1)}).$$

*Thus after*

$$T_\varepsilon = \frac{K_1\|\mathbf{q}\|^2((L-1)\|\mathbf{u}\|^2+1)}{C_2\nu^2 m\lambda^2}\log(\frac{1}{\varepsilon})$$

*iterations, we have* $L_{\mathcal{D}^*}^{0-1}(\Psi^{(T)}) \leq \varepsilon$.

Note that the bound is valid only when $T \geq \hat{T}$, a common threshold in prior convergence rate analyses [34, 33, 36]. Importantly, the existence of $\hat{T}$ does not affect the convergence rate as $\varepsilon \to 0$, since $\hat{T}$ is independent of $\varepsilon$. Our novel analysis generalizes these prior results to our realistic settings handling the challenges of self-attention, ReLU-MLP, and cross-entropy loss simultaneously. By considering extreme cases, our techniques relax the batch size requirement, enabling more general results. Consequently, the sample complexity for Bayes-optimal test error is $N = T_\varepsilon$.

Before introducing the next proposition, we highlight a key observation from the semantic geometry in Definition 1. For any $k_1 \in [K_1]$, defining $\boldsymbol{a}_{k_1} := (\boldsymbol{\mu}_{k_1}^+ + \boldsymbol{\mu}_{k_1}^-)/2$ and $\boldsymbol{b}_{k_1} := (\boldsymbol{\mu}_{k_1}^+ - \boldsymbol{\mu}_{k_1}^-)/2$, we find that for $k_1' \neq k_1$, $\{\boldsymbol{a}_{k_1}, \boldsymbol{b}_{k_1}\} \perp \{\boldsymbol{a}_{k_1'}, \boldsymbol{b}_{k_1'}\}$ and $\langle \boldsymbol{a}_{k_1}, \boldsymbol{b}_{k_1}\rangle = 0$. This structure is exemplified in Figure 1(b) of [12], where "[Bird]" consists of orthogonal steering vectors: "plant $\Rightarrow$ animal" and "mammal $\Rightarrow$ bird," corresponding to the concept feature $\boldsymbol{a}_k$ and semantic label features $\boldsymbol{b}_k$. Here, the term $e\boldsymbol{b}_{k_1}$ in $\boldsymbol{\mu}_{k_1}^e$ determines the label assignment. Similarly, defining $\boldsymbol{c}_{k_1} := (\boldsymbol{q}_{k_1}^+ + \boldsymbol{q}_{k_1}^-)/2$ and $\boldsymbol{d}_{k_1} := (\boldsymbol{q}_{k_1}^+ - \boldsymbol{q}_{k_1}^-)/2$ yields analogous properties. Detailed definitions are provided in Appendix I. The following proposition explores the model's ability to handle OOD unseen ICL tasks.

**Proposition 1.** *Out-of-Distribution-Generalization*[3]. *During testing, the learned model admits probability distribution shift on* $\mathcal{D}_{\boldsymbol{z}}^*$ *and data shift on* $\mathcal{D}_{\boldsymbol{x}}^* \times \mathcal{D}_{\boldsymbol{y}}^*$ *to generate a new prompt distribution*

---

[3]Here we do not consider the shift of $\mathcal{D}_{\xi_{\boldsymbol{x}}}, \mathcal{D}_{\xi_{\boldsymbol{y}}}$ for the ease of presentation. However, we assert that this can also be addressed by leveraging high-dimensional statistical analysis over other well-behaved noise distributions.

$\mathcal{D}_S^* = \sum_{k=1}^{K_1} \left( {\pi_k^+}^* \mathcal{P}_{k,L^*+1}^{+\,*} + {\pi_k^-}^* \mathcal{P}_{k,L^*+1}^{-\,*} \right)$. *Specifically, the new $\mathcal{D}_S^*$ satisfies the following properties.*

- *The prompt length $L^*$ can be any positive integer.*

- *$\mathcal{D}_z^*$ can enjoy arbitrary distribution, satisfying that each prompt has at least one co-concept $k \in [K_1]$, at least one pair shares the query word's co-concept's label, and still each word has equal chance to have positive or negative semantic labels over its concepts[4].*

- *$\mathcal{D}_x^* \times \mathcal{D}_y^*$ can enjoy a great family of data shift. $\forall k \neq k' \in [K_1], k_2 \in [K_2]$, we can have new $\mathbf{M}^*$ and $\mathbf{Q}^*$ such that $\boldsymbol{\mu}_k^{\pm *} = \boldsymbol{a}_k^* \pm \boldsymbol{b}_k^*$, $\boldsymbol{q}_k^{\pm *} = \boldsymbol{c}_k^* \pm \boldsymbol{d}_k^*$, $\boldsymbol{\nu}_{k_2} = \boldsymbol{\nu}_{k_2}^*$. Here, $\boldsymbol{a}_k^*, \boldsymbol{b}_k^*, \boldsymbol{c}_k^*, \boldsymbol{d}_k^*$ are any vectors belong to the conic hulls of $\{\boldsymbol{a}_k\}_{k=1}^{K_1}, \{\boldsymbol{b}_k\}_{k=1}^{K_1}, \{\boldsymbol{c}_k\}_{k=1}^{K_1}, \{\boldsymbol{d}_k\}_{k=1}^{K_1}$ respectively, satisfying $\|\boldsymbol{b}_k^*\| \geq \|\boldsymbol{a}_k^*\| = \Theta(\|\mathbf{u}\|)$ and $\|\boldsymbol{d}_k^*\| \geq \|\boldsymbol{c}_k^*\| = \Theta(\|\mathbf{q}\|)$. $\boldsymbol{\nu}_{k_2}^* = \Theta(\|\mathbf{u}\|)$ are any vectors from the complement space of $\mathrm{span}(\mathbf{M})$.*

*Again, the learned model satisfies $L_{\mathcal{D}_S^*}^{0-1}(\Psi^{(T^*)}) \leq \varepsilon$.*

This proposition demonstrates the strong Out-of-Distribution Generalization ability of transformer utilizing multi-concept semantics, suggesting the efficiency transformer to conduct unseen ICL tasks just by its learned "Knowledge" on the high-level concept and low-level label semantic information from the two non-orthogonal dictionaries. The admit of shift for $\mathcal{D}_z^*$ denotes that each prompt can enjoy multi-co-concepts and each word-label pair can appear in at least $\|\boldsymbol{z}\|_0$ concept-specific prompts/tasks' distribution, which aligns the real-world cases. On the other hand, we also believe the admit of shift for $\mathcal{D}_x^* \times \mathcal{D}_y^*$ is inspiring, suggesting that transformer can conduct specific cross-concept semantic "Knowledge Intersection". As such, this lemma suggest that the transformer can master the regularity of unseen ICL tasks' "structure" in the presence the multi-concept encoded representation.

**Remark 1.** ***Comparison with Related Work****. Theorem 3.4 in [28] and Theorem 2 in [54] address the transformer's OOD capability in specific structured ICL classification and regression tasks. Our results differ by focusing on compositional generalization of learned concepts, grounded in the concept-specific linear latent geometry observed in LLMs.*

## 5 Proof Idea

In a big picture, we simply extend standard expectation-variance reduction techniques [34] to our setting. Section 5.1 defines coefficients to examine NN's expected projection along feature directions. Section 5.2 provides the convergence of the expected estimator through the lens of coefficient evolution; Section 5.3 showcase the exponential convergence by treating the conditional expectations of the NNs as Doob martingales and exploiting the property of the tails under low-noise conditions.

### 5.1 Idempotent Operator Techniques

**Idempotent Operator Trick**. Define $\mathbb{U} := \mathrm{span}(\mathbf{M})$ and its complement space $\mathbb{U}^\perp$. By definition, we know that $\dim(\mathbb{U}) = K$ and $\dim(\mathbb{U}^\perp) = d_\mathcal{X} - K$. Then we can let $\{\{\boldsymbol{a}_{k_1}\}_{k_1=1}^{K_1}, \{\boldsymbol{b}_{k_1}\}_{k_1=1}^{K_1}, \{\boldsymbol{\nu}_{k_2}\}_{k_2=1}^{K_2}, \{\boldsymbol{u}_w\}_{w=1}^{d_\mathcal{X}-K}\}$ be the set of standard orthogonal basis for $\mathbb{R}^{d_\mathcal{X}}$, where $\boldsymbol{u}_1^\perp, \cdots, \boldsymbol{u}_{d_\mathcal{X}-K}^\perp$ are the standard orthogonal basis of $\mathbb{U}^\perp$.

Then we can derive an idempotent decomposition of the identity matrix

$$\sum_{s=1}^{K_1} \frac{\boldsymbol{a}_s \boldsymbol{a}_s^\top}{\|\boldsymbol{a}_s\|^2} + \sum_{s=1}^{K_1} \frac{\boldsymbol{b}_s \boldsymbol{b}_s^\top}{\|\boldsymbol{b}_s\|^2} + \sum_{r=1}^{K_2} \frac{\boldsymbol{\nu}_r \boldsymbol{\nu}_r^\top}{\|\mathbf{u}\|^2} + \sum_{w=1}^{d_\mathcal{X}-K} \boldsymbol{u}_w^\perp \boldsymbol{u}_w^{\perp\top} = \mathbf{I}_{d_\mathcal{X} \times d_\mathcal{X}}. \tag{3}$$

Similar techniques are also applied to the label's dictionary: $\mathbb{Q} := \mathrm{span}(\mathbf{Q})$, where we define $\boldsymbol{q}_1^\perp, \cdots, \boldsymbol{q}_{d_\mathcal{Y}-K_1}^\perp$ as the standard orthogonal basis of the complement space $\mathbb{Q}^\perp$. In our subsequent derivation, the expectation $\mathbb{E}[\cdot]$ is taken over the stochastic gradient descent. Similar to the idea in [34, 33, 36], we first serve to see how $\mathbb{E}(\Psi^{(t)})$ evolves. For $\mathbb{E}(\Psi^{(t)})$, every gradient descent update by all concept's samples within a soft "weight", and thus the analysis is equivalent to gradient descent

---

[4]The requirement of $\mathcal{D}_z^*$ could be relax with a stricter requirement on $L^*$ and a denser analyses.

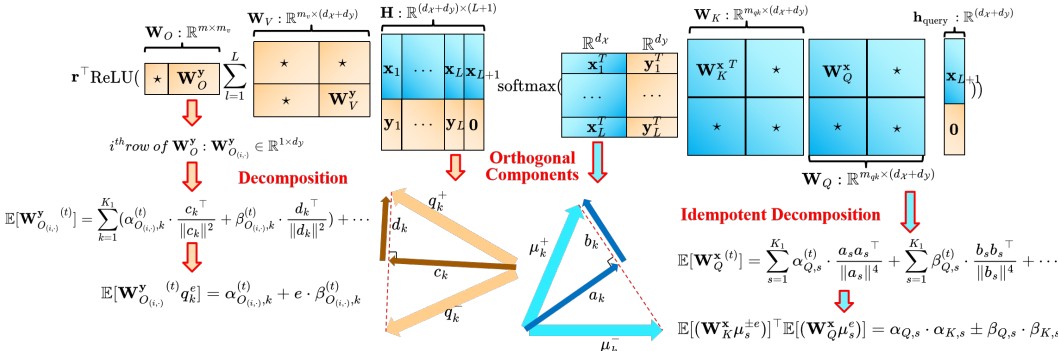

Figure 1: Illustration of our Idempotent Operator Techniques. This allows us to focus on analyzing the evolving coefficients, which are key to the expected 0-1 loss convergence.

with an ideally-balanced prompt set. Leveraging the symmetry of the prompt distribution, as well as the symmetry of $\mathbf{W}_Q^{(0)}$ and $\mathbf{W}_K^{(0)}$, we introduce the following decompositions.

**Lemma 1.** *We can decompose* $\mathbb{E}[\mathbf{W}_Q^{\boldsymbol{x}}]$, $\mathbb{E}[\mathbf{W}_K^{\boldsymbol{x}}]$ *and the $i$-th row of* $\mathbb{E}[\mathbf{W}_O^{\boldsymbol{y}}]$ $(i \in [m])$ *via the following (scaled) projection matrices and projection directions.*

$$\mathbb{E}[\mathbf{W}_Q^{\boldsymbol{x}\,(t)}] = \sum_{s=1}^{K_1} \alpha_{Q,s}^{(t)} \cdot \frac{\boldsymbol{a}_s \boldsymbol{a}_s^\top}{\|\boldsymbol{a}_s\|^4} + \sum_{s=1}^{K_1} \beta_{Q,s}^{(t)} \cdot \frac{\boldsymbol{b}_s \boldsymbol{b}_s^\top}{\|\boldsymbol{b}_s\|^4} + \sum_{r=1}^{K_2} \tau_{Q,r}^{(t)} \cdot \frac{\boldsymbol{\nu}_r \boldsymbol{\nu}_r^\top}{\|\mathbf{u}\|^4} + \sum_{w=1}^{d_{\mathcal{X}}-K} \rho_{Q,w}^{(t)} \cdot \boldsymbol{u}_w^\perp \boldsymbol{u}_w^{\perp\top},$$

$$\mathbb{E}[\mathbf{W}_K^{\boldsymbol{x}\,(t)}] = \sum_{s=1}^{K_1} \alpha_{K,s}^{(t)} \cdot \frac{\boldsymbol{a}_s \boldsymbol{a}_s^\top}{\|\boldsymbol{a}_s\|^4} + \sum_{s=1}^{K_1} \beta_{K,s}^{(t)} \cdot \frac{\boldsymbol{b}_s \boldsymbol{b}_s^\top}{\|\boldsymbol{b}_s\|^4} + \sum_{r=1}^{K_2} \tau_{K,r}^{(t)} \cdot \frac{\boldsymbol{\nu}_r \boldsymbol{\nu}_r^\top}{\|\mathbf{u}\|^4} + \sum_{w=1}^{d_{\mathcal{X}}-K} \rho_{K,w}^{(t)} \cdot \boldsymbol{u}_w^\perp \boldsymbol{u}_w^{\perp\top},$$

$$\mathbb{E}[\mathbf{W}_{O_{(i,\cdot)}}^{\boldsymbol{y}\,(t)}] = \sum_{k=1}^{K_1} \alpha_{O_{(i,\cdot)},k}^{(t)} \cdot \frac{\boldsymbol{c}_k^\top}{\|\boldsymbol{c}_k\|^2} + \sum_{k=1}^{K_1} \beta_{O_{(i,\cdot)},k}^{(t)} \cdot \frac{\boldsymbol{d}_k^\top}{\|\boldsymbol{d}_k\|^2} + \sum_{w=1}^{d_{\mathcal{Y}}-K_1} \rho_{O_{(i,\cdot)},w}^{(t)} \cdot \boldsymbol{q}_w^{\perp\top}.$$

Here $\alpha_{Q,s}^{(t)}$, $\alpha_{K,s}^{(t)}$ and $\alpha_{O_{(i,\cdot)},k}^{(t)}$ represent the expected concept learning process, $\beta_{Q,s}^{(t)}$, $\beta_{K,s}^{(t)}$ and $\beta_{O_{(i,\cdot)},k}^{(t)}$ represent the expected concept-specific semantic learning process and $\tau_{Q,r}^{(t)}, \tau_{K,r}^{(t)}, \rho_{Q,w}^{(t)}, \rho_{K,w}^{(t)}$ and $\rho_{O_{(i,\cdot)},w}^{(t)}$ represent the expected memorization of the concept irrelevant noise. It holds that

$$\begin{aligned}
\mathbb{E}[(\mathbf{W}_K^{\boldsymbol{x}\,(t)}\boldsymbol{\mu}_s^{\pm e})]^\top \mathbb{E}[\mathbf{W}_Q^{\boldsymbol{x}\,(t)}\boldsymbol{\mu}_s^e] &= \alpha_{Q,s}^{(t)} \cdot \alpha_{K,s}^{(t)}/\|\boldsymbol{a}_s\|^2 \pm \beta_{Q,s}^{(t)} \cdot \beta_{K,s}^{(t)}/\|\boldsymbol{b}_s\|^2, \\
\mathbb{E}[\mathbf{W}_{O_{(i,\cdot)}}^{\boldsymbol{y}\,(t)}\boldsymbol{q}_k^e] &= \alpha_{O_{(i,\cdot)},k}^{(t)} + e \cdot \beta_{O_{(i,\cdot)},k}^{(t)},
\end{aligned} \tag{4}$$

for $\forall e \in [\pm], i \in [m], k \in [K_1]$ and for $\forall e' \in [\pm], s' \in [K_1], r \in [K_2], w \in [d_{\mathcal{X}} - K], \forall \mathbf{u} \in \{\boldsymbol{\mu}_{s'}^{e'}, \boldsymbol{\nu}_r, \boldsymbol{u}_w^\perp\}$, it holds that $\mathbb{E}[(\mathbf{W}_K^{\boldsymbol{x}\,(t)}\mathbf{u})]^\top \mathbb{E}[\mathbf{W}_Q^{\boldsymbol{x}\,(t)}\boldsymbol{\mu}_s^e] = 0$. Similar conclusions hold when the query vectors are $\boldsymbol{\nu}_r$ and $\boldsymbol{u}_w^\perp, \forall r \in [K_2], w \in [d_{\mathcal{X}} - K]$. As such, our remaining task is to scrutinize the coefficients evolution, which would be the key contributors to the expected 0-1 loss convergence.

### 5.2 Convergence of the Expectation

Denote $\mathcal{U}_{k,n}^{y_{S_n}}(t)$ and $\mathcal{W}_{k,n}^y(t) - \mathcal{U}_{k,n}^{y_{S_n}}(t)$ as the activated neuron set for $\{i \in [m] \mid \mathbf{r}_i y_{S_n} > 0\}$ and $\{i \in [m] \mid \mathbf{r}_i y_{S_n} < 0\}$ separately, and $\sum_{l \in S_{n,k}^{y_{S_n}}} (\sigma_S^{(t)})_l^n$ represents the correct attention weight, where the detailed definitions are delayed in Appendix E. We then introduce the following lemma.

**Lemma 2.** *Under Condition 1, when*

$$\left( \sum_{i \in \mathcal{U}_{k,n}^{y_{S_n}}(t)} - \sum_{i \in \mathcal{W}_{k,n}^{y_{S_n}}(t) - \mathcal{U}_{k,n}^{y_{S_n}}(t)} \right) \left( \alpha_{O_{(i,\cdot)},k}^{(t)} + (2 \sum_{l \in S_{n,k}^{y_{S_n}}} (\sigma_S^{(t)})_l^n - 1) y_{S_n} \beta_{O_{(i,\cdot)},k}^{(t)} \right) \geq 0, \tag{5}$$

*holds, we have* $L_{\mathcal{D}^*}^{0-1}(\mathbb{E}(\Psi'^{(t)})) = 0$.

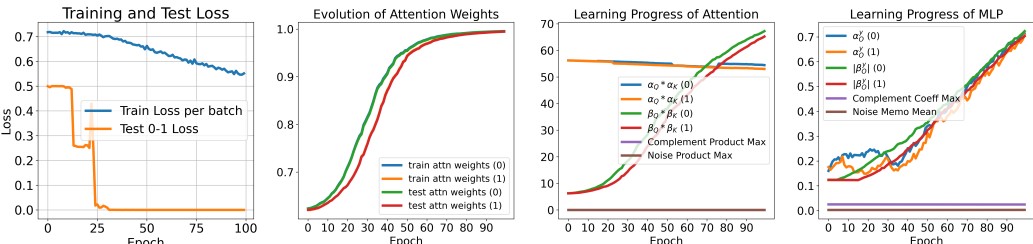

Figure 2: Learning dynamics: (i) training and test loss; (ii) correct attention weight; (iii) maximum values of $\alpha_{Q,s} \cdot \alpha_{K,s}$, $\beta_{Q,s} \cdot \beta_{K,s}$, maximum values of the complement products $\tau_{Q,r} \cdot \tau_{K,r}$ or $\rho_{Q,2} \cdot \rho_{K,2}$, and maximum values of product-with-noise $(\mathbf{W}_K^{\boldsymbol{x}} \xi_{\boldsymbol{x}})^\top \mathbf{W}_Q^{\boldsymbol{x}} \xi_{\boldsymbol{x}}$; (iv) maximum values of $\alpha_{O_{(i,\cdot)},k}$ and $|\beta_{O_{(i,\cdot)},k}|$, maximum values of the complement coefficients $\rho_{O_{(i,\cdot)},w}$ and maximum values of product-with-noise $\mathbf{W}_{O_{(i,\cdot)}}^{\boldsymbol{y}} \xi_{\boldsymbol{y}}$.

As such, the following lemmas show the learning outcomes of the $\mathbb{E}(\Psi^{(t)})$ along the iterations.

**Lemma 3.** *(Convergence of the Expectation). There exist constant $C_1 > 0$, $\forall t \geq \hat{T} = C_1\sigma_1 m\lambda K_1\gamma\sqrt{(1+\kappa_{\boldsymbol{y}})\log(5Km/\delta)}/w^{*2}(1-\kappa_{\boldsymbol{y}})\|\mathbf{q}\|$, we have $L_{\mathcal{D}^*}^{0-1}(\mathbb{E}(\Psi'^{(t)})) = 0$.*

**Lemma 4.** *(Regularizing the models). Under Condition 1, it holds that*

$$\alpha_{Q,k}^{(T^*)} = \alpha_{K,k}^{(T^*)} = O(\mathbb{E}[\alpha_{Q,k}^{(0)}]), \quad \beta_{Q,k}^{(T^*)} = \beta_{K,k}^{(T^*)} = \Theta(\|\mathbf{u}\|\sqrt{\log(\frac{\|\mathbf{u}\|^2}{\lambda K_1}\log(\frac{\|\mathbf{q}\|^2}{m\lambda K_1}))}),$$

$$\alpha_{O_{(i,\cdot)},k}^{(T^*)} \leq |\beta_{O_{(i,\cdot)},k}^{(T^*)}| = \Theta(\log(\frac{\|\mathbf{q}\|^2}{m\lambda K_1})), \mathbb{E}[(\sum_{j\in S_{n,k}^{y_{S_n}}}(\sigma_S^{(T^*)})_j^n)] = \Theta(\frac{1}{1+\frac{\lambda K_1}{\|\mathbf{u}\|^2}\log(\frac{m\lambda K_1}{\|\mathbf{q}\|^2})}).$$

In addition, our analysis provides three asymptotic properties of the coefficients evolution, which are delayed to Appendix I.1.3 and I.2 for room limitation.

### 5.3 Exponential Convergence of 0-1 loss

**Proposition 2.** *$\forall t \geq \hat{T}$, when $\|\Psi'^{(t)} - \mathbb{E}(\Psi'^{(t)})\|_F \leq \nu$ holds, we have $L_{\mathcal{D}^*}^{0-1}(\Psi'^{(t)}) = 0$. Here, $\|\Psi'\|_F^2 := \|\mathbf{W}_Q^{\boldsymbol{x}}\|_F^2 + \|\mathbf{W}_K^{\boldsymbol{x}}\|_F^2 + \|\mathbf{W}_O^{\boldsymbol{y}}\|_F^2$.*

By definition of 0-1 loss, then we only need to prove the 0-1 loss convergence by seeing the speed of $\Psi'^{(t)}$ converging to $\mathbb{E}(\Psi'^{(t)})$ with an error of $\nu$ in terms of $\|\cdot\|_F$.

Drawing insights from [34], we see $\mathcal{B}_0, \cdots, \mathcal{B}_{T-1}$ as a i.i.d. random variables following the same distribution. Then $\forall t \in \{0, \cdots, T\}$, it holds that

$$D_Q^t = \mathbb{E}[\mathbf{W}_Q^{\boldsymbol{x}(T+1)} \mid \mathcal{B}_0, \cdots, \mathcal{B}_t] - \mathbb{E}[\mathbf{W}_Q^{\boldsymbol{x}(T+1)} \mid \mathcal{B}_0, \cdots, \mathcal{B}_{t-1}],$$
$$D_K^t = \mathbb{E}[\mathbf{W}_K^{\boldsymbol{x}(T+1)} \mid \mathcal{B}_0, \cdots, \mathcal{B}_t] - \mathbb{E}[\mathbf{W}_K^{\boldsymbol{x}(T+1)} \mid \mathcal{B}_0, \cdots, \mathcal{B}_{t-1}] \tag{6}$$
$$D_O^t = \mathbb{E}[\mathbf{W}_O^{\boldsymbol{y}(T+1)} \mid \mathcal{B}_0, \cdots, \mathcal{B}_t] - \mathbb{E}[\mathbf{W}_O^{\boldsymbol{y}(T+1)} \mid \mathcal{B}_0, \cdots, \mathcal{B}_{t-1}],$$

are martingale difference sequences, and for $\forall X \in \{Q, K, O\}$ and its corresponding $\mathbf{W} \in \{\mathbf{W}_Q^{\boldsymbol{x}}, \mathbf{W}_K^{\boldsymbol{x}}, \mathbf{W}_O^{\boldsymbol{y}}\}$, we have $\sum_{t=0}^T D_X^t = \mathbf{W}^{(T+1)} - \mathbb{E}[\mathbf{W}^{(T+1)}]$. Then we utilize the following lemma in [34, 55] to give a bound over the variance.

**Lemma 5.** *Let $D_1, \cdots, D_{T-1}$ be a martingale difference sequence. Suppose $\exists c_T > 0$ such that $\sum_{t=0}^T \|D_t\|_\infty^2 \leq c_T^2$, where $\|\cdot\|_\infty$ is the essential supremum of $\|\cdot\|_F$. Then for $\forall \epsilon > 0$, we have*

$$\mathbb{P}\left[\sup_{s\in[T]} \|\sum_{t=0}^s D_t\|_F \geq \epsilon\right] \leq 2\exp(-\frac{\epsilon^2}{2c_T^2}).$$

Therefore, we need to see if there exists a decaying positive constant $c_T$ (with decaying rate $O(1/T^q), q > 0$), such that $\sum_{t=0}^T \|D_X^t\|_\infty^2 \leq c_T^2, \forall X \in \{Q, K, O\}$, where $\|\cdot\|_\infty$ is the essential

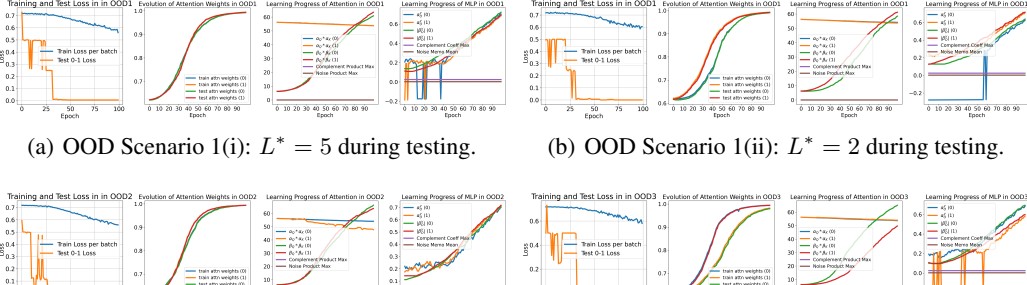

(a) OOD Scenario 1(i): $L^* = 5$ during testing.  (b) OOD Scenario 1(ii): $L^* = 2$ during testing.

(c) OOD Scenario 2: 0.8 fraction for concept 0 and 0.2 (d) OOD Scenario 3: Shift the data as $\boldsymbol{\mu}_1^{\pm*} = \boldsymbol{a}_1 \pm \boldsymbol{b}_2$
fraction for concept 1 during testing.  and $\boldsymbol{\mu}_2^{\pm*} = \boldsymbol{a}_2 \pm \boldsymbol{b}_1$ during testing.

Figure 3: Learning dynamic in three OOD scenarios. The training settings and plotting methods are identical to those used in Figure 2, and the testing settings are: (a-b) utilizes different prompt lengths; (c) adopts a skewed distribution over $\boldsymbol{z}$; (d) switches the concept-specific semantic features.

supremum of $\|D_X^t\|_F$. Subsequently, by controlling the martingale sequence norm tail similarly in [34, 55], we can obtain an exponential convergence rate after $T_1$.

For $\mathbf{W} \in \{\mathbf{W}_Q^{\boldsymbol{x}}, \mathbf{W}_K^{\boldsymbol{x}}, \mathbf{W}_O^{\boldsymbol{y}}\}$, to check the decaying $c_T$, we adopt the techniques of [34, 33, 36] in the following manner. Let $\mathcal{B}_t'$ be an independent variable from $\mathcal{B}_0, \cdots, \mathcal{B}_T$ and let $\mathbf{W}_t^{(T+1)}$ be an output of the algorithm depending on $(\mathcal{B}_0, \cdots, \mathcal{B}_{t-1}, \mathcal{B}_t', \mathcal{B}_{t+1}, \cdots, \mathcal{B}_T)$. Then we have

$$\|D_X^t\|_\infty \leq \mathbb{E}[\|\mathbf{W}^{(T+1)} - \mathbf{W}_t^{(T+1)}\|_\infty \mid \mathcal{B}_0, \cdots, \mathcal{B}_t].$$

Therefore, one may estimate $c_X^{T\,2}$ by bounding $\|\mathbf{W}^{(T+1)} - \mathbf{W}_t^{(T)}\|_\infty^2$ uniformly w.r.t. $\mathcal{B}_0, \cdots, \mathcal{B}_{T-1}$. Such a bound can be derived utilizing stability property of stochastic gradient descent [34, 56]. For the OOD scenario, since we require the data shift to be via conic combination, the new words and labels in each prompt will share the positive/negative real-valued label without any self-conflict. The norm requirements and constraints on $\mathcal{D}_{\boldsymbol{z}}^*$ would ensure the Gaussian noise, concepts other than the co-concepts, and probability shifts have limited influence on the prediction compared with the considerable scale of coefficients by Lemma 4, laying the groundwork for the proof.

## 6 Experiments

In this section, we demonstrate the validity of our theoretical analysis through simulations of Algorithm 1. We use the following parameter settings in Figure 2: The parameter settings are: the length $L = 4$, the number of co-concepts $K_1 = 2$, dictionary size $K = 104$, the number of test instances $n_{\text{test}} = 5000$, dimension $d_{\mathcal{X}} = d_{\mathcal{Y}} = 1000$, MLP width $m = 50$, feature strengths $\|\mathbf{u}\| = \|\mathbf{q}\| = 10, \forall k \in [K_1]$, the cosine $\langle \boldsymbol{\mu}_k^+, \boldsymbol{\mu}_k^- \rangle / \|\mathbf{u}\|^2 = \langle \boldsymbol{q}_k^+, \boldsymbol{q}_k^- \rangle / \|\mathbf{q}\|^2 = 0.5$, the initialization parameters $\sigma_0 = 0.1$, $\sigma_1 = 0.01$, and the noise deviation $\sigma_\xi = 0.01$. For the optimization, we use $\lambda = 0.002$, $B = 16$, $\gamma = 10000$, and the total training epochs is $100$. Figure 3 (a-d) uses the same training settings, but during testing, it applies different configurations: (a) $L^* = 5$, (b) $L^* = 2$, (c) a $0.8$ fraction for the first concept and a $0.2$ fraction for the second concepts, and (d) $\boldsymbol{\mu}_1^{\pm*} = \boldsymbol{a}_1 \pm \boldsymbol{b}_2, \boldsymbol{\mu}_2^{\pm*} = \boldsymbol{a}_2 \pm \boldsymbol{b}_1$. Figure 2 validates our Theorem 2 and Lemma 4, which showcases the fast convergence rate and the evolution of coefficients. Figure 3 validates Proposition 1, where the learned model permits certain data shifts.

## 7 Conclusion

This work provides the first exponential convergence analysis of 0-1 loss for transformers with softmax attention and ReLU-MLP, trained on a non-orthogonal concept-specific prompt distribution by practical cross-entropy loss. Furthermore, the results demonstrate transformers can perform certain OOD ICL tasks by leveraging the multi-concept semantic linearity, highlighting their innovative potential. An important future direction is to extend the analysis to more complex scenarios.

# 8  Acknowledgment

We thank the anonymous reviewers for their instrumental comments. D.B. and H.W. are supported in part by the Research Grants Council of the Hong Kong Special Administration Region (Project No. CityU 11206622). W.H. is supported in part by JSPS KAKENHI (24K20848). A.N. is supported in part by National Research Foundation, Singapore and Infocomm Media Development Authority under its Trust Tech Funding Initiative, the Centre for Frontier Artificial Intelligence Research, Institute of High Performance Computing, A*Star, and the College of Computing and Data Science at Nanyang Technological University. T.S. is supported in part by JSPS KAKENHI (24K02905) and JST CREST (JPMJCR2115, JPMJCR2015).

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

# A    Limitation and Broader Impact

The theoretical analysis provided in this work introduces novel perspectives on optimization and generalization, but the data model employed may require additional refinements to better align with practical scenarios, such as adding more layers of attention. The techniques and findings can inform future empirical and theoretical explorations of transformer architectures, though we do not foresee a direct social impact arising from the theoretical advancements presented.

# B    Additional Experiment Details

We implement our methods using PyTorch, ensuring consistent software and hardware environments. Specifically, the experiments are run on Linux servers with NVIDIA A100 graphics cards and CUDA 11.2, and can be completed within one hour.

# C    Additional Related Work

**Theory of Convergence Rate of Stochastic Gradient Descent**. Our analysis of the exponential convergence rate for the 0-1 loss builds upon a rich body of prior work. In the context of classification, the faster convergence rate mostly based on the excess of risk with some power of the essential supremum norm. Specifically, [31, 32] introduce the *Hard low-noise condition* over the margin. When there is a hard margin separating the classes, the test error can exhibit exponentially fast convergence as the number of training samples increases, even when the surrogate loss error only decreases polynomially. This phenomenon has been further explored in more recent studies. [33, 34, 35, 36, 37] have analyzed the exponential convergence of stochastic gradient descent under various settings. Meanwhile, [35] have investigated hard-margin and exponential rates in the context of structured prediction, which encompasses traditional classification as a special case. Besides, recent work also obtain the exponential rates in generalized settings such as Multi-class classification [38] and SVM [39]. Building upon this rich theoretical foundation, our work derives the first exponential convergence analysis for the 0-1 loss in the specific setting of transformer models with softmax attention and ReLU-activated MLP over the sparse coding data model, whose surrogate loss function is the cross-entropy loss.

**Theory of Feature Learning of GD-updated Neural Network**. A rich body of recent learning theory research has focused on the feature direction' recovery view of neural network representations [40, 41, 42, 43, 45, 52, 53, 57, 58, 59, 60, 61, 62, 63, 64, 65, 66, 67, 68, 69, 70]. Rather than directly examining the evolution of the 0-1 loss, this line of work explicitly studies the process of reconstruction of the data's feature directions and memorization of disrupted noise in the network's latent space as surrogate metrics. While most studies in this area have assumed (near) orthogonal data, recent efforts by [43] and [44] have made initial attempts to analyze non-orthogonal data scenarios. Building upon this foundation, our study extends this line of research to nonlinear attention-MLP transformers with within-concept positive inner products and cross-concept orthogonal data representations. The key to our analysis is the assumption of good initialization of attention matrices and a sufficiently low-noise condition, which is reasonable for modeling language rather than images. In this setting, SGD allows noise to have only a mild impact on shaping neural network matrices or influencing gradient flow.

**Theory of Transformers and In-Context Learning**. The literature on Transformers and ICL is wide-ranging, and we will selectively address the most relevant ones. Prior studies have analyzed how transformers learn topic/concept semantics [9], the origins and biases of LLM representations using latent variable models [10], and ICL from a model averaging perspective [14]. However, these works do not connect the geometric properties of concept-encoded representations to transformers' powerful ICL abilities. Another line of research has studied the learning dynamics of transformer, including analyses of linear-attention transformers [16, 17, 71, 72], QK-combined attention-only models [20, 21, 26, 54, 73, 74, 75, 76, 77, 78, 79], ReLU-free MLP [54, 80, 81] or without MLP [17, 25], impractical squared or hinge loss [25, 26, 27, 28]. Though relevant, these works rely on simplifications or do not connect the observed linear semantic representation of large model to the transformer's excelling OOD capability.

**Concept Learning in Deep Learning**. Hierarchical learning has long been regarded as a key factor behind the success of deep learning [82, 83, 84]. Recent research shows that large-scale generative models, such as diffusion models and transformers, effectively encode hierarchical concepts in their latent spaces [11, 12, 13, 46, 85, 86, 87]. Moreover, [73, 88, 89] show that transformers can capture hierarchical and compositional structures in data. From a Bayesian perspective, [7, 8, 14] interpret ICL as LLMs predicting outputs based on latent (concept) variable inference. Furthermore, studies reveal a linear structure in LLMs' latent space over independent interpretable concepts: representations of the same concept exhibit positive inner products, while statistically-independent concepts are nearly orthogonal [9, 10, 11, 12, 90]. Interestingly, aligning with the findings in [46, 90], Independent Component Analysis (ICA) is naturally more suitable than Principal Component Analysis (PCA) for obtaining meaningful feature or label vectors in our prompt modeling. This is because the features or labels are nearly statistically independent and of equal strength, especially with a large $K$, while the noise is

feeble in our modeling. Building on these insights, we explore in a theoretical context how the compositional nature of concept representations relates to transformers' ability to generalize to OOD tasks through a sparse coding modeling. We believe our OOD results are not only coincides with the transformer's compositional generalization ability on language tasks [89], but also consistent with other concept learning outcomes of diffusion and multi-model model: [87] shows that adjusting the length of semantic representations can directly affect image generation behaviors (see Figure 5), while [86] reveals that compositing different concepts enables OOD generalization (e.g. "blue square apples" in the Figure 1a in [86]).

# D Preliminary Lemmas

## D.1 Probablistic Lemmas on Concentration

**Lemma 6.** *Suppose that $\delta > 0$ and $\forall d \in \{d_{\mathcal{X}}, d_{\mathcal{Y}}\} = \Omega(\log(\frac{KNL}{\delta}))$, where $N = BT^*$. Then with probability at least $1 - \delta$,*

$$\frac{\sigma_\xi^2 d}{2} \leq \|\boldsymbol{\xi}_i\|_2^2 \leq 3\frac{\sigma_\xi^2 d}{2},$$

$$|\langle \boldsymbol{\xi}_i, \boldsymbol{\xi}_{i'} \rangle| \leq 2\sigma_\xi^2 \cdot \sqrt{d \log\left(\frac{6(N(L+1))^2}{\delta}\right)},$$

$$|\langle \boldsymbol{\xi}_i, \boldsymbol{\mu} \rangle| \leq \|\boldsymbol{\mu}\|_2 \sigma_\xi \cdot \sqrt{2 \log(\frac{6KN(L+1)}{\delta})}$$

*for all $\boldsymbol{\xi}_i, \boldsymbol{\xi}_i' \sim \mathcal{D}_{\xi_x}(\text{ or } \mathcal{D}_{\xi_y}), \boldsymbol{\mu} \in \mathcal{D}_{\boldsymbol{x}}(\text{ or } \mathcal{D}_{\boldsymbol{y}}), l \in \{1, 2\}$.*

*Proof.* See Lemma B.4 in [42] for a proof. $\square$

**Lemma 7.** *Suppose that $\delta > 0$, $d_{\mathcal{Y}} = \Omega(\log(m/\delta))$, $m = \Omega(\log(K/(\delta)))$. Then with probability at least $1 - \delta$, for $\forall i \in [m], k \in [K_1], w \in [d_{\mathcal{Y}} - K_1]$,*

$$\frac{\sigma_1^2 d_{\mathcal{Y}}}{2} \leq \|\mathbf{W}_{O_{(i,\cdot)}}^{\boldsymbol{y}}{}^{(0)}\|^2 \leq 3\frac{\sigma_1^2 d_{\mathcal{Y}}}{2},$$

$$\frac{|\alpha_{O_{(i,\cdot)},k}^{(0)}|}{\|\boldsymbol{c}_k\|}, \frac{|\beta_{O_{(i,\cdot)},k}^{(0)}|}{\|\boldsymbol{d}_k\|}, |\rho_{O_{(i,\cdot)},w}^{(0)}| \leq \sqrt{2\log(\frac{5Km}{\delta})} \cdot \sigma_1, \tag{7}$$

$$\sigma_1/2 \leq \max_{i \in [m]}\{\frac{|\alpha_{O_{(i,\cdot)},k}^{(0)}|}{\|\boldsymbol{c}_k\|}, \frac{|\beta_{O_{(i,\cdot)},k}^{(0)}|}{\|\boldsymbol{d}_k\|}, |\rho_{O_{(i,\cdot)},w}^{(0)}|\} \leq \sqrt{2\log(\frac{5Km}{\delta})} \cdot \sigma_1,$$

*Moreover, for some $\zeta \in (0, 1]$ for $\forall e \neq e', \in [\pm], \exists \omega_\zeta \in (0, \omega_\zeta')$ where $\omega_\zeta' < 1$,*

$$\left|\left|\{i \in [m] \mid \mathbf{r}_i = \frac{e}{m}, \alpha_{O_{(i,\cdot)},k}^{(0)} + e'\zeta\beta_{O_{(i,\cdot)},k}^{(0)} > 0\}\right| - \frac{m}{4}\right| \leq \sqrt{\frac{m \log(10K_1/\delta)}{2}},$$

$$\left|\left|\{i \in [m] \mid \alpha_{O_{(i,\cdot)},k}^{(0)} + e'\zeta\beta_{O_{(i,\cdot)},k}^{(0)} > 0, \mathbf{r}_i e \cdot \beta_{O_{(i,\cdot)},k}^{(0)} > 0\}\right| - \frac{m}{4}\right| \leq \sqrt{\frac{m \log(10K_1/\delta)}{2}},$$

$$\left|\left|\{i \in [m] \mid \mathbf{r}_i = \frac{e}{m}, \alpha_{O_{(i,\cdot)},k}^{(0)} \pm \zeta\beta_{O_{(i,\cdot)},k}^{(0)} > 0\}\right| - \frac{(1+\omega_\zeta)m}{8}\right| \leq \sqrt{\frac{m \log(10K_1/\delta)}{2}} \leq \frac{(\omega_\zeta' - \omega_\zeta)m}{8},$$

$$\left|\left|\{i \in [m] \mid \mathbf{r}_i = \frac{e}{m}, \alpha_{O_{(i,\cdot)},k}^{(0)} + \zeta\beta_{O_{(i,\cdot)},k}^{(0)} > 0, \alpha_{O_{(i,\cdot)},k}^{(0)} - \zeta\beta_{O_{(i,\cdot)},k}^{(0)} < 0\}\right| - \frac{(1-\omega_\zeta)m}{8}\right| \leq \sqrt{\frac{m \log(10K_1/\delta)}{2}},$$

$$\left|\sum_{i \in \{i \in [m] \mid \mathbf{r}_i = \frac{e}{m}, \alpha_{O_{(i,\cdot)},k}^{(0)} + e'\zeta\beta_{O_{(i,\cdot)},k}^{(0)} > 0\}} \mathbf{r}_i \cdot (\alpha_{O_{(i,\cdot)},k}^{(0)} + e'\zeta\beta_{O_{(i,\cdot)},k}^{(0)}) - 0\right| \leq \sqrt{2\log(\frac{5Km}{\delta})} \cdot \frac{5\sigma_1(\|\boldsymbol{c}_k\| + \zeta\|\boldsymbol{d}_k\|)}{16},$$

$$\left|\sum_{i \in \{i \in [m] \mid \alpha_{O_{(i,\cdot)},k}^{(0)} + e'\zeta\beta_{O_{(i,\cdot)},k}^{(0)} > 0\}} \mathbf{r}_i \cdot \beta_{O_{(i,\cdot)},k}^{(0)} - 0\right| \leq \sqrt{2\log(\frac{5Km}{\delta})} \cdot \frac{5\sigma_1\|\boldsymbol{d}_k\|}{16}. \tag{8}$$

*In addition, for a sufficient large $m = \Omega(\log(K/(\delta))/(1 - \omega_\zeta))$ the lower bound inequalities regarding maximum value in Eq.(7) hold at any above index set of $i$ in Eq.(8). For example, there exist $i \in \{i \in [m] \mid \mathbf{r}_i = \frac{e}{m}, \alpha_{O_{(i,\cdot)},k}^{(0)} + \zeta\beta_{O_{(i,\cdot)},k}^{(0)} > 0, \alpha_{O_{(i,\cdot)},k}^{(0)} - \zeta\beta_{O_{(i,\cdot)},k}^{(0)} < 0\}$, such that $\alpha_{O_{(i,\cdot)},k}^{(0)} \leq -\sigma_1/2\|\boldsymbol{c}_k\|$.*

*Proof.* First, notice that $\mathbf{W}^{\boldsymbol{y}}_{O_{(i,\cdot)}}{}^{(0)} \sim \mathcal{N}(\mathbf{0}, \sigma_1 \mathbb{I}_{d_{\mathcal{Y}}})$, then by Bernstein's inequality as well as $d_{\mathcal{Y}} = \Omega(\log(m/\delta))$, with probability at least $1 - \delta/(5m)$, for $\forall i \in [m]$

$$|\|\mathbf{W}^{\boldsymbol{y}}_{O_{(i,\cdot)}}{}^{(0)}\|^2 - \sigma_1 d_{\mathcal{Y}}| \leq O(\sigma_1^2 \cdot \sqrt{d_{\mathcal{Y}} \log(5m/\delta)}) \leq \sigma_1^2 d_{\mathcal{Y}}/2.$$

By union bound we can have the first inequality in the lemma hold with probability at least $1 - \delta/5$.

Next, we notice that

$$\frac{\alpha^{(0)}_{O_{(i,\cdot)},k}}{\|\boldsymbol{c}_k\|} = \langle \mathbf{W}^{\boldsymbol{y}}_{O_{(i,\cdot)}}{}^{(0)}, \frac{\boldsymbol{c}_k}{\|\boldsymbol{c}_k\|} \rangle, \quad \frac{\beta^{(0)}_{O_{(i,\cdot)},k}}{\|\boldsymbol{d}_k\|} = \langle \mathbf{W}^{\boldsymbol{y}}_{O_{(i,\cdot)}}{}^{(0)}, \frac{\boldsymbol{d}_k}{\|\boldsymbol{d}_k\|} \rangle, \quad \rho^{(0)}_{O_{(i,\cdot)},w} = \langle \mathbf{W}^{\boldsymbol{y}}_{O_{(i,\cdot)}}{}^{(0)}, \boldsymbol{q}^{\perp}_w \rangle$$

are all Gaussian random variable with mean 0 and variance $\sigma_1^2$. Then by Gaussian tail bound and union bound, with probability at least $1 - \delta/10$, for all $i \in [m]$ and $\mathbf{q} \in \bigcup_{k,w} \{ \frac{\boldsymbol{c}_k}{\|\boldsymbol{c}_k\|}, \frac{\boldsymbol{d}_k}{\|\boldsymbol{d}_k\|}, \boldsymbol{q}^{\perp}_w \}$, it holds that

$$|\langle \mathbf{W}^{\boldsymbol{y}}_{O_{(i,\cdot)}}{}^{(0)}, \mathbf{q} \rangle| \leq \sqrt{2\log(5Km/\delta)} \cdot \sigma_1.$$

Notice $\mathbb{P}(\sigma_1/2 > |\langle \mathbf{W}^{\boldsymbol{y}}_{O_{(i,\cdot)}}{}^{(0)}, \mathbf{q} \rangle|)$ is an positive constant, then following the techniques of Lemma B.5 in [42] and the condition $m = \Omega(\log(K/\delta))$, we have

$$\mathbb{P}(\sigma_1/2 \leq |\langle \mathbf{W}^{\boldsymbol{y}}_{O_{(i,\cdot)}}{}^{(0)}, \mathbf{q} \rangle|) = 1 - \mathbb{P}(\sigma_1/2 > \max\{|\langle \mathbf{W}^{\boldsymbol{y}}_{O_{(i,\cdot)}}{}^{(0)}, \mathbf{q} \rangle|\}),$$
$$= 1 - \mathbb{P}(\sigma_1/2 > |\langle \mathbf{W}^{\boldsymbol{y}}_{O_{(i,\cdot)}}{}^{(0)}, \mathbf{q} \rangle|)^{mK}$$
$$\geq 1 - \delta/10,$$

then with probability $1 - \delta/5$, the second and third inequality hold.

For $\zeta \in (0,1]$, we see that the variable $\alpha^{(0)}_{O_{(i,\cdot)},k} + e'\zeta\beta^{(0)}_{O_{(i,\cdot)},k} \sim \mathcal{N}(0, \sigma_1^2(\|\boldsymbol{c}_k\|^2 + \zeta\|\boldsymbol{d}_k\|^2))$, and it's independent to the event $\{\mathbf{r}_i = \frac{e}{m}\}, \forall e \in [\pm]$. Therefore, we can see the count of $\{i \in [m] \mid \mathbf{r}_i = \frac{e}{m}, \alpha^{(0)}_{O_{(i,\cdot)},k} + e'\zeta\beta^{(0)}_{O_{(i,\cdot)},k} > 0, e'\zeta\beta^{(0)}_{O_{(i,\cdot)},k} > 0\}$ as a binomial variable with $p = 1/4, n = m$, then by the property of binomial tail, condition $m = \Omega(\log(K/(\delta)))$ as well as Hoeffding's inequality, with probability at least $1 - \delta/5$ we have

$$\left| \frac{|\{i \in [m] \mid \mathbf{r}_i = \frac{e}{m}, \alpha^{(0)}_{O_{(i,\cdot)},k} + e'\zeta\beta^{(0)}_{O_{(i,\cdot)},k} > 0\}|}{m} - \frac{1}{4} \right| \leq \sqrt{\frac{\log(10K_1/\delta)}{2m}},$$

which completes the proof of the forth inequality. Similarly, for the fifth inequality we can utilize the same techniques to derive that it holds with probability at least $1 - \delta/5$.

For the event $\{i \in [m] \mid \mathbf{r}_i = \frac{e}{m}, \alpha^{(0)}_{O_{(i,\cdot)},k} \pm \zeta\beta^{(0)}_{O_{(i,\cdot)},k} > 0\}$, we have

$$\mathbb{P}(\mathbf{r}_i = \frac{e}{m}, \alpha^{(0)}_{O_{(i,\cdot)},k} \pm \zeta\beta^{(0)}_{O_{(i,\cdot)},k} > 0) = \mathbb{P}(\alpha^{(0)}_{O_{(i,\cdot)},k} + e'\zeta\beta^{(0)}_{O_{(i,\cdot)},k} > 0)$$
$$\cdot \mathbb{P}(\alpha^{(0)}_{O_{(i,\cdot)},k} - e'\zeta\beta^{(0)}_{O_{(i,\cdot)},k} > 0 \mid \alpha^{(0)}_{O_{(i,\cdot)},k} + e'\zeta\beta^{(0)}_{O_{(i,\cdot)},k} > 0)$$
$$= \frac{1}{2} \cdot \mathbb{P}(\alpha^{(0)}_{O_{(i,\cdot)},k} - e'\zeta\beta^{(0)}_{O_{(i,\cdot)},k} > 0 \mid \alpha^{(0)}_{O_{(i,\cdot)},k} + e'\zeta\beta^{(0)}_{O_{(i,\cdot)},k} > 0),$$
$$= \frac{1}{2} \cdot \frac{1 + \omega_\zeta}{2},$$

where $\frac{1 + \omega_\zeta}{2}$ is the probability of the conditional event $\{\alpha^{(0)}_{O_{(i,\cdot)},k} - e'\zeta\beta^{(0)}_{O_{(i,\cdot)},k} > 0 \mid \alpha^{(0)}_{O_{(i,\cdot)},k} + e'\zeta\beta^{(0)}_{O_{(i,\cdot)},k} > 0\}$, and $\omega_\zeta > 0$ due to the larger variance of $\alpha^{(0)}_{O_{(i,\cdot)},k}$ compared to $e'\zeta\beta^{(0)}_{O_{(i,\cdot)},k}$. We denote the probability with $\omega_\zeta$ since the true value is hard to compute. Subsequently, the event $\{i \in [m] \mid \mathbf{r}_i = \frac{e}{m}, \alpha^{(0)}_{O_{(i,\cdot)},k} \pm \zeta\beta^{(0)}_{O_{(i,\cdot)},k} > 0\}$ can be seen as a binomial variable with $p = \frac{1 + \omega_\zeta}{8}, n = m$, then we can have the sixth inequality hold with probability at least $1 - \delta/5$, utilizing the property of binomial tail, condition $m = \Omega(\log(K/(\delta)))$ as well as Hoeffding's inequality.

The seventh inequality is a natural inference of the third and forth inequality, where the $m = \Omega(\log(K_1/\delta))$ ensure $\sqrt{m\log(10K_1/\delta)/2} \leq m/16$, and the last inequality is then also a natural inference of the third and fifth inequality.

Therefore, by union bound, the proof is completed. $\square$

## D.2 Matrix Theories

**Lemma 8.** *(1.1.P5 in [91]) Let $A \in M_n$ be idempotent, that is, $A^2 = A$. Then, each eigenvalue of A equals to the rank of A, which is either 0 or 1. Beside, identity matrix $\mathbf{I}$ is the only nonsingular idempotent matrix.*

**Lemma 9.** *For a matrix $A = \sum_{i=1}^{d} \mu_i P_i$, where $P_i$ are symmetric idempotent matrices with $\operatorname{rank}(P_i) = 1$, and thus $\sum_{i=1}^{d} \mu_i P_i$ is the idempotent decomposition of matrix A by $P_i$. Then we see that $\|A\|_F = \sqrt{\operatorname{tr}(A^T A)} = \sqrt{\sum_{i=1}^{d} \mu_i^2} = \sqrt{\sum_{i=1}^{d} \lambda_i^2}$, where $\lambda_i$ are eigenvalues of A.*

*Proof.* By definition,

$$A^T A = \sum_{i=1}^{d} \mu_i^2 P_i^T P_i = \sum_{i=1}^{d} \mu_i^2 P_i P_i = \sum_{i=1}^{d} \mu_i^2 P_i.$$

Then, by Lemma 8 we have

$$\operatorname{tr}(A^T A) = \operatorname{tr}(\sum_{i=1}^{d} \mu_i^2 P_i) = \sum_{i=1}^{d} \mu_i^2 \operatorname{tr}(P_i) = \sum_{i=1}^{d} \mu_i^2 \operatorname{rank}(P_i) = \sum_{i=1}^{d} \mu_i^2 = \sum_{i=1}^{d} \lambda_i^2.$$

$\square$

## D.3 ODE Systems

**Lemma 10.** *(Lemma C.1 in [43]). Suppose that a sequence $a_t, t \geq 0$ follows the iterative formula*

$$a_{t+1} = a_t + \frac{c}{1 + be^{a_t}},$$

*for some $0 \leq c \leq 1$ and $b \geq 0$. Then it holds that*

$$x_t \leq a_t \leq \frac{c}{1 + be^{a_0}} + x_t$$

*for all $t \geq 0$. Here, $x_t$ is the unique solution of*

$$\frac{\mathrm{d}x_t}{\mathrm{d}t} = \frac{c}{1 + be^{x_t}}, \quad x_0 = a_0 \Leftrightarrow x_t + be^{x_t} = ct + a_0 + be^{a_0}.$$

**Lemma 11.** *(Coupled ODE System 1). Suppose that there are two coupled sequences $y_t$, $z_t$, $t \geq 0$ follows the iterative formula*

$$y_{t+1} = y_t + az_t y_t \frac{1}{2 + e^{-2y_t^2} + e^{2y_t^2}}, \qquad y_0 > 0, \qquad a > 0,$$

$$z_{t+1} = z_t + b, \qquad\qquad\qquad z_0 < 0, \qquad b > 0,$$

*for some $a, b \geq 0$. Then it holds that*

$$y(t) \leq y_t, \quad z(t) = z_t,$$

*for all $t \geq 0$. Here, $y(t)$, $z(t)$ are the unique solutions of the following ODE System respectively*

$$
\begin{aligned}
y'(t) &= \frac{a}{4} z(0) y(t), \quad y(0) = y_0, \\
z'(t) &= b, \qquad\qquad z(0) = z_0.
\end{aligned}
\tag{9}
$$

*As such, for $t_1 = \min\{t \in \mathbb{Z} \mid z_t \geq 0\}$, we have*

$$y_{t_1} \geq y(0) e^{\frac{-az(0)^2(1 + e^{-2y(0)^2})}{4b(1 - e^{-2y(0)^2})}},$$

*and $t_1 \geq \dfrac{-z(0)(1 + e^{-2y(0)^2})}{b(1 - e^{-2y(0)^2})}$.*

*Proof.* From the condition we see that $z_0 < 0$ and $z_t$ is an increasing sequence ($z_t \geq z_0$). Besides, as $y_0 > 0$, during the period where $z_t \leq 0$, we see that $y_t$ is monotonically decreasing. Then by $(2 + e^{-2y_t^2} + e^{2y_t^2})^{-1} \leq 1/4$ as well as Comparison Theorem, it's obvious that the continuous coupled ODE in Eq.(9) is the lower bound of $y_t$. Then one can readily obtain the result by solving the ODE. $\square$

**Lemma 12.** *(Coupled ODE System 2). Suppose that there are two coupled sequences $y_t$, $z_t$, which are the sequences after $t_1$ in Lemma 11, and $t \geq t_1$ follows the iterative formula*

$$y_{t+1} = y_t + a z_t y_t \frac{1}{2 + e^{-2y_t^2} + e^{2y_t^2}} \ell_t', \qquad y_{t_1} > 0, \qquad a > 0,$$

$$z_{t+1} = z_t + b \frac{1 - e^{-2y(t)^2}}{1 + e^{-2y(t)^2}} \ell_t', \qquad z_{t_1} \geq 0, \qquad b > 0,$$

*for some $a, b \geq 0$, and $c' \leq \ell_t' \leq 1$. Then it holds that*

$$\underline{y}(t) \leq y_t \leq \overline{y}(t), \quad \underline{z}(t) \leq z_t \leq \overline{z}(t),$$

*for all $t \geq t_1$. Here, $\overline{y}(t)$, $\underline{y}(t)$, $\overline{z}(t)$, $\underline{z}(t)$ are the unique solutions of the following ODE System respectively*

$$\frac{1}{2}\left(\mathrm{Ei}(2\underline{y}(t)^2) + \mathrm{Ei}(-2\underline{y}(t)^2) + 4\log(\underline{y}(t))\right) = abc'^2 \frac{1 - e^{-2y(t_1)^2}}{1 + e^{-2y(t_1)^2}} \frac{(t - t_1)^2}{2} + \frac{1}{2}\left(\mathrm{Ei}(2y_{t_1}^2) + \mathrm{Ei}(-2y_{t_1}^2)\right)$$
$$+ 4\log(y_{t_1}),$$

$$\underline{z}(t) = bc' \frac{1 - e^{-2y(t_1)^2}}{1 + e^{-2y(t_1)^2}}(t - t_1),$$

$$\frac{1}{2}\left(\mathrm{Ei}(2\overline{y}(t)^2) + \mathrm{Ei}(-2\overline{y}(t)^2) + 4\log(\overline{y}(t))\right) = \frac{ab(t - t_1)^2}{2} + \frac{1}{2}\left(\mathrm{Ei}(2y_{t_1}^2) + \mathrm{Ei}(-2y_{t_1}^2)\right) + 4\log(y_{t_1})$$

$$\overline{z}(t) = b(t - t_1),$$

*where*

$$\mathrm{Ei}(x) = \int_{-\infty}^{x} \frac{e^t}{t} dt = \gamma_{Euler} + \ln x + \exp(x/2) \sum_{n=1}^{\infty} \frac{(-1)^{n-1} x^n}{n! 2^{n-1}} \sum_{k=0}^{\lfloor (n-1)/2 \rfloor} \frac{1}{2k+1}.$$

*Proof.* We see that as $z_t \geq 0, t \geq t_1$, the $y_t$ is monotonically increasing. As such, by Comparison Theorem we see that the upper and lower bound of the coupled system would depends on $\frac{1-e^{-2y(t)^2}}{1+e^{-2y(t)^2}}$ and $\ell_t'$. Easy to see that

$$\frac{1 - e^{-2y(t_1)^2}}{1 + e^{-2y(t_1)^2}} \leq \frac{1 - e^{-2y(t)^2}}{1 + e^{-2y(t)^2}} \leq 1,$$

and then collaborating with $c' \leq \ell_t' \leq 1$ we can obtain the result by solving the ODE. Observing that

$$\frac{d\underline{y}(t)}{dt} = abc'^2 \frac{1 - e^{-2y(t_1)^2}}{1 + e^{-2y(t_1)^2}} \frac{(t - t_1)\underline{y}(t)dt}{1 + e^{2\underline{y}(t)^2} + e^{-2\underline{y}(t)^2}}$$

$$\Leftrightarrow \frac{1}{2}\left(\mathrm{Ei}(2\underline{y}(t)^2) + \mathrm{Ei}(-2\underline{y}(t)^2) + 4\log(\underline{y}(t))\right) = abc'^2 \frac{1 - e^{-2y(t_1)^2}}{1 + e^{-2y(t_1)^2}} \frac{(t - t_1)^2}{2} + \mathrm{const},$$

$$\underline{z}(t) = bc' \frac{1 - e^{-2y(t_1)^2}}{1 + e^{-2y(t_1)^2}}(t - t_1).$$

Thus by the monotonicity the system is unique, which is also ture for the upper bound ODE. The proof is completed. □

# E    Data Distribution

This section provided the detailed formal definitions of the prompt distribution.

**Definition 3.** *(**Polysemous Word Model** $(\mathcal{D}_x, \mathcal{D}_y, \mathcal{D}_z, \mathcal{D}_{\xi_x}, \mathcal{D}_{\xi_y})$ ). We assume there exists $K_1$ concepts of words totally. Specifically, each concept $k_1 \in [K_1]$ is characterized by two semantically-opposite feature vectors separately, denoted as $\boldsymbol{\mu}_{k_1}^+$ and $\boldsymbol{\mu}_{k_1}^-$, and the label vectors that describe their semantics under the co-concept are $\boldsymbol{q}_{k_1}^+$ and $\boldsymbol{q}_{k_1}^-$. Our word samples $\boldsymbol{x} \in \mathbb{R}^{d_{\mathcal{X}}}$ and their corresponding labels $\boldsymbol{y} \in \mathbb{R}^{d_{\mathcal{Y}}}$ are generated i.i.d. from distribution $\mathcal{D}_x$ and $\mathcal{D}_y$, which can be written as the following forms via reparameterization:*

$$\boldsymbol{z} \sim \mathcal{D}_{\boldsymbol{z}}, \quad \xi_{\boldsymbol{x}} \sim \mathcal{D}_{\xi_{\boldsymbol{x}}} = \mathcal{N}(\boldsymbol{0}, \sigma_\xi^2 \mathbf{I}_{d_{\mathcal{X}}}), \quad \xi_{\boldsymbol{y}} \sim \mathcal{D}_{\xi_{\boldsymbol{y}}} = \mathcal{N}(\boldsymbol{0}, \sigma_\xi^2 \mathbf{I}_{d_{\mathcal{Y}}}),$$

$$\boldsymbol{x} = \mathbf{M}\boldsymbol{z} + \xi_{\boldsymbol{x}} \sim \mathcal{D}_{\boldsymbol{x}}, \quad \boldsymbol{y} = \mathbf{Q}\boldsymbol{z} + \xi_{\boldsymbol{y}} \sim \mathcal{D}_{\boldsymbol{y}},$$

*where $\boldsymbol{z} \in \mathbb{R}^K (K < d_{\mathcal{X}})$. We denote $\boldsymbol{z}$ as the sparse latent signal and $\xi$ as the spurious dense noise, and each $\boldsymbol{x}$-$\boldsymbol{y}$ pair are reparameterized by one shared $\boldsymbol{z}$. We have the following assumptions on $\mathbf{M}, \boldsymbol{z}, \xi$ respectively:*

- *The sparse latent variable $\boldsymbol{z} = (z_1, \cdots, z_K) \in \{0,1\}^k$ is sampled from $\mathcal{D}_z$. $P(z_j = 1) = \Theta(\frac{\log \log K}{K})$.*

- $\mathbf{M} = [\boldsymbol{\mu}_1^+, \boldsymbol{\mu}_1^-, \boldsymbol{\mu}_2^+, \boldsymbol{\mu}_2^-, \cdots, \boldsymbol{\mu}_{K_1}^+, \boldsymbol{\mu}_{K_1}^-, \boldsymbol{\nu}_1, \boldsymbol{\nu}_2, \cdots, \boldsymbol{\nu}_{K_2}] = [M_1, \cdots, M_K] \in \mathbb{R}^{d_{\mathcal{X}} \times K}$ *is the feature dictionary matrix, where $\{\boldsymbol{\mu}_{k_1}^\pm\}_{k_1=1}^{K_1}$ are concept-relevant features, $\{\boldsymbol{\nu}_{k_2}\}_{k_2=1}^{K_2}$ are concept-irrelevant features, and $\forall k \in [K], \|M_k\| = \|\mathbf{u}\|$. We assume that features of the same concept have positive inner product: $\exists 0 < \kappa_{\boldsymbol{x}} < 1, \forall k_1 \in [K_1], 0 < \langle \boldsymbol{\mu}_{k_1}^+, \boldsymbol{\mu}_{k_1}^- \rangle \le \kappa_{\boldsymbol{x}} \|\mathbf{u}\|^2$. Meanwhile, we let the features of different concept be orthogonal: $\forall e \in [\pm], e' \in [\pm], s' \in [K_1], r \ne r' \in [K_2], \mathbf{u} \in \{\boldsymbol{\mu}_{s'}^{e'}, \boldsymbol{\nu}_r\}$, we have $\langle \boldsymbol{\mu}_s^e, \mathbf{u} \rangle = \langle \boldsymbol{\nu}_r, \boldsymbol{\nu}_{r'} \rangle = 0$.*

- $\mathbf{Q} = [\boldsymbol{q}_1^+, \boldsymbol{q}_1^-, \boldsymbol{q}_2^+, \boldsymbol{q}_2^-, \cdots, \boldsymbol{q}_{K_1}^+, \boldsymbol{q}_{K_1}^-, 0, \cdots 0] \in \mathbb{R}^{d_{\mathcal{Y}} \times K}$ *is the corresponding label dictionary matrix, where $\|\boldsymbol{q}_k^\pm\| = \|\mathbf{q}\|$, for $\forall k \in [K_1]$. Similarly, we let the labels of the same concept to have positive inner product: $\exists 0 < \kappa_{\boldsymbol{y}} < 1, \forall k_1 \in [K_1], 0 < \langle \boldsymbol{q}_{k_1}^+, \boldsymbol{q}_{k_1}^- \rangle \le \kappa_{\boldsymbol{y}} \|\mathbf{q}\|^2$, while the labels of different concept to be orthogonal: $\langle \boldsymbol{q}_k^\pm, \boldsymbol{q}_{k'}^\pm \rangle = 0, \forall k \ne k' \in [K_1]$.*

**Definition 4.** *(**Concept-specific Contextual Prompt Distribution**) We consider the case that each prompt is concept-specific (i.e., the multi-concept words in one prompt would at least share one co-concept). Specifically, the chance for selecting each concept as the co-concept of one particular prompt is $\Theta(K_1^{-1})$, and the chance for selecting the two semantically-opposite vectors of the same concept is $\frac{1}{2}$. During training, each prompt $S = \{\boldsymbol{x}_1, \boldsymbol{y}_1, \cdots, \boldsymbol{x}_L, \boldsymbol{y}_L, \boldsymbol{x}_{L+1}\}$ is sampled from the mixture distribution $\mathcal{D}_S$ defined as below.*

$$\mathcal{D}_S = \sum_{k=1}^{K_1} \left( \pi_k^+ \mathcal{P}_{k,L+1}^+ + \pi_k^- \mathcal{P}_{k,L+1}^- \right), \tag{10}$$

*where $\pi_k^+ = \pi_k^- = \frac{1}{2K_1}$, and the $\mathcal{P}_{k,L+1}^+$ and $\mathcal{P}_{k,L+1}^-$ are prompt distributions characterized by the k-th concept, defined as*

$$\mathcal{P}_{k,L+1}^+ = \Big\{ S \mid \boldsymbol{x} \sim \mathcal{D}_{\boldsymbol{x}}, \boldsymbol{y} \sim \mathcal{D}_{\boldsymbol{y}}, P_{L+1,2k-1} = 1, \forall l \in [L+1], j \ne \{2k-1, k\}, P_{l,j} = \frac{1}{K},$$

$$\{z_{l,2k-1} = 1\} \cup \{z_{l,2k} = 1\} = \Omega, \{z_{l,2k-1} = 1\} \cap \{z_{l,2k} = 1\} = \emptyset, \forall l \in [L], P_{l,2k-1} = P_{l,2k} = \frac{1}{2} \Big\},$$

$$\mathcal{P}_{k,L+1}^- = \Big\{ S \mid \boldsymbol{x} \sim \mathcal{D}_{\boldsymbol{x}}, \boldsymbol{y} \sim \mathcal{D}_{\boldsymbol{y}}, P_{L+1,2k} = 1, \forall l \in [L+1], j \ne \{2k-1, k\}, P_{l,j} = \frac{1}{K},$$

$$\{z_{l,2k-1} = 1\} \cup \{z_{l,2k} = 1\} = \Omega, \{z_{l,2k-1} = 1\} \cap \{z_{l,2k} = 1\} = \emptyset, \forall l \in [L], P_{l,2k-1} = P_{l,2k} = \frac{1}{2} \Big\},$$

*where $P_{l,j} := \mathbb{P}(z_{l,j} = 1)$. $\forall n \in [N]$ where N is the training size, if the training prompt $S_n$ is sampled from $\mathcal{P}_{k,L+1}^e, e \in [\pm], k \in [K_1]$, then by Definition 1, the label vector of the query should contain $\boldsymbol{q}_k^e$, and we call $y_{S_n} = e$ as the real value label of this k-th concept prompt. Specifically, for $\forall k \in [K_1]$ we define the index set of training prompts sharing the k-th co-concepts as*

$$\mathcal{V}_k = \mathcal{V}_k^+ \cup \mathcal{V}_k^-,$$

*where*

$$\mathcal{V}_k^+ = \{n \mid S_n \sim \mathcal{P}_{k,L+1}^+\},$$
$$\mathcal{V}_k^- = \{n \mid S_n \sim \mathcal{P}_{k,L+1}^-\}.$$

*For sample $\boldsymbol{x}_l$ where $n \in \mathcal{V}_k, k \in [K_1], l \in [L+1]$, we define the index set for its non-zero elements of $\boldsymbol{z}_l^n$ besides $z_{2k-1,l}^n$ and $z_{2k,l}^n$, namely $\mathcal{M}_l^n := \{k \in [K] \mid z_{l,k}^n = 1, k \notin \{2k-1, 2k\}\}$. Also, for each prompt sharing the k-th co-concept, we define the index set of demonstration in the context:*

$$S_{n,k}^+ = \{l \in [L] \mid n \in \mathcal{V}_k, z_{l,2k-1}^n = 1\}, \quad S_{n,k}^- = \{l \in [L] \mid n \in \mathcal{V}_k, z_{l,2k}^n = 1\},$$

# F  Model details: Attention Part

In this section, we provide several important definitions and compute the original gradients of attention.

**Lemma 13.** *(Contributing and Misleading Neurons)*

$$\mathcal{W}_{k,n}^+(t) = \{i \in [m] \mid n \in \mathcal{V}_k^+, \mathbb{1}_{O_{(i)}}^n{}^{(t)} > 0\}, \quad \mathcal{U}_{k,n}^+(t) = \{i \in [m] \mid n \in \mathcal{V}_k^+, \mathbf{r}_i \cdot \mathbb{1}_{O_{(i)}}^n{}^{(t)} > 0\},$$
$$\mathcal{W}_{k,n}^-(t) = \{i \in [m] \mid n \in \mathcal{V}_k^-, \mathbb{1}_{O_{(i)}}^n{}^{(t)} > 0\}, \quad \mathcal{U}_{k,n}^-(t) = \{i \in [m] \mid n \in \mathcal{V}_k^-, \mathbf{r}_i \cdot \mathbb{1}_{O_{(i)}}^n{}^{(t)} < 0\}.$$

$$\tag{11}$$

$\mathcal{W}_{k,n}(t) := \mathcal{W}_{k,n}^+(t) \cup \mathcal{W}_{k,n}^-(t)$ are neurons that can be activated, among which $\mathcal{U}_{k,n}(t) := \mathcal{U}_{k,n}^+(t) \cup \mathcal{U}_{k,n}^-(t)$ are neurons that correctly contribute to the prediction. The following lemma computes the original gradients.

**Lemma 14.** *(Gradient Update) Denote*

$$
\begin{aligned}
\mathbf{r}_i &= \mathbf{r}[i], \\
\ell_n'^{(t)} &= \ell'(y_{S_n} \cdot f(\mathbf{H}^n; \Psi^{(t)})), \\
(\sigma_S^{(t)})_l^n &= \mathrm{softmax}\left(\left(\mathbf{W}_K^{(t)} \mathbf{h}_l^n\right)^\top \mathbf{W}_Q^{(t)} \mathbf{h}_{L+1}^n\right), \\
\mathbb{1}_{O_{(i)}}^n{}^{(t)} &= \mathbb{1}(\mathbf{W}_{O_{(i,\cdot)}}^{(t)} \, \mathrm{attn}(\mathbf{H}^n; \Psi^{(t)}) > 0).
\end{aligned}
\tag{12}
$$

$\nabla_{\mathbf{W}_Q^{\boldsymbol{x}}{}^{(t)}} L_{\mathcal{B}_t}(\Psi^{(t)}) \in \mathbb{R}^{d_{\mathcal{X}} \times d_{\mathcal{X}}}$ *can be derived as*

$$
\frac{1}{B} \sum_{n \in \mathcal{B}_t} \left[ y_{S_n}^{(t)} \ell_n'^{(t)} \sum_{i=1}^m \mathbf{r}_i \mathbb{1}_{O_{(i)}}^n{}^{(t)} \sum_{l,j \in [L]} (\sigma_S^{(t)})_l^n (\sigma_S^{(t)})_j^n (\mathbf{W}_{O_{(i,\cdot)}}^{(t)} \mathbf{W}_V^{(t)} \mathbf{h}_l^n) \mathbf{W}_K^{\boldsymbol{x}}{}^{(t)} (\boldsymbol{x}_l^n - \boldsymbol{x}_j^n) \boldsymbol{x}_{L+1}^n{}^\top \right] + \lambda \mathbf{W}_Q^{\boldsymbol{x}}{}^{(t)}.
\tag{13}
$$

*Similarly,* $\nabla_{\mathbf{W}_K^{\boldsymbol{x}}{}^{(t)}} L_{\mathcal{B}_t}(\Psi^{(t)}) \in \mathbb{R}^{d_{\mathcal{X}} \times d_{\mathcal{X}}}$ *can be derived as*

$$
\frac{1}{B} \sum_{n \in \mathcal{B}_t} \left[ y_{S_n}^{(t)} \ell_n'^{(t)} \sum_{i=1}^m \mathbf{r}_i \mathbb{1}_{O_{(i)}}^n{}^{(t)} \sum_{l,j \in [L]} (\sigma_S^{(t)})_l^n (\sigma_S^{(t)})_j^n (\mathbf{W}_{O_{(i,\cdot)}}^{(t)} \mathbf{W}_V^{(t)} \mathbf{h}_l^n) \mathbf{W}_Q^{\boldsymbol{x}}{}^\top \boldsymbol{x}_{L+1}^n (\boldsymbol{x}_l^n - \boldsymbol{x}_j^n)^\top \right] + \lambda \mathbf{W}_K^{\boldsymbol{x}}{}^{(t)}.
\tag{14}
$$

Subsequently, we directly compute the update of the attention matrices along the feature directions as below.

**Lemma 15.** *(Concept Learning of Attention) For* $\forall \hat{k} \in [K_1]$, *we have the single step of learning of the concept part of the features:*

$$
\begin{aligned}
\boldsymbol{a}_{\hat{k}}^\top \mathbf{W}_Q^{\boldsymbol{x}}{}^{(t+1)} \boldsymbol{a}_{\hat{k}} - \boldsymbol{a}_{\hat{k}}^\top \mathbf{W}_Q^{\boldsymbol{x}}{}^{(t)} \boldsymbol{a}_{\hat{k}} &= -\eta_t \cdot \boldsymbol{a}_{\hat{k}}^\top \nabla_{\mathbf{W}_Q^{\boldsymbol{x}}{}^{(t)}} L_{\mathcal{B}_t}(\Psi^{(t)}) \boldsymbol{a}_{\hat{k}} \\
&= -\eta_t (I_{Q, \boldsymbol{a}_{\hat{k}}, chaos}^{(t)} + I_{Q, \boldsymbol{a}_{\hat{k}}, contri}^{(t)}) - \eta_t \lambda \boldsymbol{a}_{\hat{k}}^\top \mathbf{W}_Q^{\boldsymbol{x}}{}^{(t)} \boldsymbol{a}_{\hat{k}}, \\
\boldsymbol{a}_{\hat{k}}^\top \mathbf{W}_K^{\boldsymbol{x}}{}^{(t+1)} \boldsymbol{a}_{\hat{k}} - \boldsymbol{a}_{\hat{k}}^\top \mathbf{W}_K^{\boldsymbol{x}}{}^{(t)} \boldsymbol{a}_{\hat{k}} &= -\eta_t \cdot \boldsymbol{a}_{\hat{k}}^\top \nabla_{\mathbf{W}_K^{\boldsymbol{x}}{}^{(t)}} L_{\mathcal{B}_t}(\Psi^{(t)}) \boldsymbol{a}_{\hat{k}} \\
&= -\eta_t (I_{K, \boldsymbol{a}_{\hat{k}}, chaos}^{(t)} + I_{K, \boldsymbol{a}_{\hat{k}}, contri}^{(t)}) - \eta_t \lambda \boldsymbol{a}_{\hat{k}}^\top \mathbf{W}_K^{\boldsymbol{x}}{}^{(t)} \boldsymbol{a}_{\hat{k}},
\end{aligned}
\tag{15}
$$

*where* $I_{Q, \boldsymbol{a}_{\hat{k}}, chaos}^{(t)}$ *and* $I_{Q, \boldsymbol{a}_{\hat{k}}, contri}^{(t)}$ *are defined as below.*

$$
\begin{aligned}
I_{Q, \boldsymbol{a}_{\hat{k}}, chaos}^{(t)} &= \frac{1}{B} \sum_{\substack{k \neq \hat{k} \in [K_1] \\ e \in [\pm] \\ n \in \mathcal{V}_k^e \cap \mathcal{B}_t}} \left[ e \ell_n'^{(t)} \boldsymbol{a}_{\hat{k}}^\top (\xi_{\boldsymbol{x}, L+1}^n + \sum_{r \in \mathcal{M}_{L+1}^n} \mathbf{M}_r) \sum_{i \in \mathcal{W}_{k,n}^e(t)} \mathbf{r}_i \sum_{l,j \in [L]} (\sigma_S^{(t)})_l^n (\sigma_S^{(t)})_j^n \right. \\
&\qquad (\mathbf{W}_{O_{(i,\cdot)}}^{\boldsymbol{y}}{}^{(t)} (\boldsymbol{q}_k^{y_l^n} + \sum_{s \in \mathcal{M}_l^n} \mathbf{Q}_S + \xi_{\boldsymbol{y}, l}^n)) (\boldsymbol{a}_{\hat{k}}^\top \mathbf{W}_K^{\boldsymbol{x}}{}^{(t)} ((y_l^n - y_j^n) \boldsymbol{b}_k + \sum_{s \in \mathcal{M}_l^n} \mathbf{M}_s + \xi_{\boldsymbol{x}, l}^n - \sum_{s \in \mathcal{M}_j^n} \mathbf{M}_s - \xi_{\boldsymbol{x}, j}^n)) \right] \\
&\quad + \frac{1}{B} \sum_{\substack{\hat{e} \in [\pm] \\ n \in \mathcal{V}_{\hat{k}}^{\hat{e}} \cap \mathcal{B}_t}} \left[ \hat{e} \ell_n'^{(t)} (\|\boldsymbol{a}_{\hat{k}}\|^2 + \boldsymbol{a}_{\hat{k}}^\top (\xi_{\boldsymbol{x}, L+1}^n + \sum_{r \in \mathcal{M}_{L+1}^n} \mathbf{M}_r)) \sum_{i \in \mathcal{W}_{\hat{k}, n}^{\hat{e}}(t)} \mathbf{r}_i \{ \sum_{l,j \in [L]} (\sigma_S^{(t)})_l^n (\sigma_S^{(t)})_j^n \right. \\
&\qquad (\mathbf{W}_{O_{(i,\cdot)}}^{\boldsymbol{y}}{}^{(t)} (\sum_{s \in \mathcal{M}_l^n} \mathbf{Q}_S + \xi_{\boldsymbol{y}, l}^n)) (\boldsymbol{a}_{\hat{k}}^\top \mathbf{W}_K^{\boldsymbol{x}}{}^{(t)} ((y_l^n - y_j^n) \boldsymbol{b}_{\hat{k}} + \xi_{\boldsymbol{x}, l}^n - \xi_{\boldsymbol{x}, j}^n)) \right], \\
I_{Q, \boldsymbol{a}_{\hat{k}}, contri}^{(t)} &= \frac{1}{B} \sum_{\substack{\hat{e} \in [\pm] \\ n \in \mathcal{V}_{\hat{k}}^{\hat{e}} \cap \mathcal{B}_t}} \left[ \ell_n'^{(t)} (\|\boldsymbol{a}_{\hat{k}}\|^2 + \boldsymbol{a}_{\hat{k}}^\top (\xi_{\boldsymbol{x}, L+1}^n + \sum_{r \in \mathcal{M}_{L+1}^n} \mathbf{M}_r)) \sum_{i \in \mathcal{W}_{\hat{k}, n}^{\hat{e}}(t)} \mathbf{r}_i \mathbf{W}_{O_{(i,\cdot)}}^{\boldsymbol{y}}{}^{(t)} \right. \\
&\qquad \boldsymbol{d}_{\hat{k}} ( \sum_{l \in S_{n,\hat{k}}^{\hat{e}}} (\sigma_S^{(t)})_l^n - \sum_{l \in S_{n,\hat{k}}^{-\hat{e}}} (\sigma_S^{(t)})_l^n ) \boldsymbol{a}_{\hat{k}}^\top \mathbf{W}_K^{\boldsymbol{x}}{}^{(t)} (\hat{e}(y_l^n - y_j^n) \boldsymbol{b}_{\hat{k}} + \xi_{\boldsymbol{x}, l}^n - \sum_{j \in [L]} (\sigma_S^{(t)})_j^n \xi_{\boldsymbol{x}, j}^n) \right].
\end{aligned}
\tag{16}
$$

Similarly, $I_{K,\boldsymbol{a}_{\hat{k}},chaos}^{(t)}$ and $I_{K,\boldsymbol{a}_{\hat{k}},contri}^{(t)}$ are defined as below.

$$
\begin{aligned}
I_{K,\boldsymbol{a}_{\hat{k}},chaos}^{(t)} = \frac{1}{B} \sum_{\substack{k\neq\hat{k}\in[K_1] \\ e\in[\pm] \\ n\in\mathcal{V}_k^e\cap\mathcal{B}_t}} & \Big[ e\cdot\ell_n'^{\,(t)} \boldsymbol{a}_{\hat{k}}^\top \mathbf{W}_Q^{\boldsymbol{x}\,(t)}(\boldsymbol{a}_k+e\boldsymbol{b}_k+\xi_{\boldsymbol{x},L+1}^n + \sum_{r\in\mathcal{M}_{L+1}^n}\mathbf{M}_r) \sum_{i\in\mathcal{W}_{k,n}^e(t)}\mathbf{r}_i\cdot\sum_{l,j\in[L]}(\sigma_S^{(t)})_l^n \\
& (\sigma_S^{(t)})_j^n (\mathbf{W}_{O_{(i,\cdot)}}^{\boldsymbol{y}}{}^{(t)}(\boldsymbol{q}_k^{y_l^n}+\sum_{s\in\mathcal{M}_l^n}\mathbf{Q}_S+\xi_{\boldsymbol{y},l}^n))\boldsymbol{a}_{\hat{k}}^\top(\sum_{s\in\mathcal{M}_l^n}\mathbf{M}_s+\xi_{\boldsymbol{x},l}^n-\sum_{s\in\mathcal{M}_j^n}\mathbf{M}_s-\xi_{\boldsymbol{x},j}^n)\Big] \\
& + \frac{1}{B}\sum_{\substack{\hat{e}\in[\pm] \\ n\in\mathcal{V}_{\hat{k}}^{\hat{e}}\cap\mathcal{B}_t}} \Big[\hat{e}\ell_n'^{\,(t)}\boldsymbol{a}_{\hat{k}}^\top\mathbf{W}_Q^{\boldsymbol{x}\,(t)}(\boldsymbol{a}_{\hat{k}}+e\boldsymbol{b}_{\hat{k}}+\xi_{\boldsymbol{x},L+1}^n+\sum_{r\in\mathcal{M}_{L+1}^n}\mathbf{M}_r)\cdot\sum_{i\in\mathcal{W}_{\hat{k},n}^{\hat{e}}(t)}\mathbf{r}_i\cdot \\
& \sum_{l,j\in[L]}(\sigma_S^{(t)})_l^n(\sigma_S^{(t)})_j^n(\mathbf{W}_{O_{(i,\cdot)}}^{\boldsymbol{y}}{}^{(t)}(\sum_{s\in\mathcal{M}_l^n}\mathbf{Q}_S+\xi_{\boldsymbol{y},l}^n))\boldsymbol{a}_{\hat{k}}^\top(\xi_{\boldsymbol{x},l}^n-\xi_{\boldsymbol{x},j}^n)\Big],
\end{aligned}
$$

$$
\begin{aligned}
I_{K,\boldsymbol{a}_{\hat{k}},contri}^{(t)} = \frac{1}{B}\sum_{\substack{\hat{e}\in[\pm] \\ n\in\mathcal{V}_{\hat{k}}^{\hat{e}}\cap\mathcal{B}_t}} & \Big[\ell_n'^{\,(t)}\boldsymbol{a}_{\hat{k}}^\top\mathbf{W}_Q^{\boldsymbol{x}\,(t)}(\boldsymbol{a}_{\hat{k}}+e\boldsymbol{b}_{\hat{k}}+\xi_{\boldsymbol{x},L+1}^n+\sum_{r\in\mathcal{M}_{L+1}^n}\mathbf{M}_r)\sum_{i\in\mathcal{W}_{\hat{k},n}^{\hat{e}}(t)}\mathbf{r}_i\mathbf{W}_{O_{(i,\cdot)}}^{\boldsymbol{y}}{}^{(t)} \\
& \boldsymbol{d}_{\hat{k}}\{(\sum_{l\in S_{n,\hat{k}}^{\hat{e}}}(\sigma_S^{(t)})_l^n-\sum_{l\in S_{n,\hat{k}}^{-\hat{e}}}(\sigma_S^{(t)})_l^n)(\boldsymbol{a}_{\hat{k}}^\top\xi_{\boldsymbol{x},l}^n-\sum_{j\in[L]}(\sigma_S^{(t)})_j^n\boldsymbol{a}_{\hat{k}}^\top\xi_{\boldsymbol{x},j}^n)\}\Big].
\end{aligned}
$$

$$(17)$$

**Lemma 16.** *(Label Semantic Learning of Attention) Also, for* $\forall\hat{k}\in[K_1]$*, we have the single step of learning of the concept-specific semantically-opposite part of the features:*

$$
\begin{aligned}
\boldsymbol{b}_{\hat{k}}^\top\mathbf{W}_Q^{\boldsymbol{x}\,(t+1)}\boldsymbol{b}_{\hat{k}}-\boldsymbol{b}_{\hat{k}}^\top\mathbf{W}_Q^{\boldsymbol{x}\,(t)}\boldsymbol{b}_{\hat{k}} &= -\eta_t\cdot\boldsymbol{b}_{\hat{k}}^\top\nabla_{\mathbf{W}_Q^{\boldsymbol{x}\,(t)}}L_{\mathcal{B}_t}(\Psi^{(t)})\boldsymbol{b}_{\hat{k}} \\
&= -\eta_t(I_{Q,\boldsymbol{b}_{\hat{k}},chaos}^{(t)}+I_{Q,\boldsymbol{b}_{\hat{k}},contri}^{(t)})-\eta_t\lambda\boldsymbol{b}_{\hat{k}}^\top\mathbf{W}_Q^{\boldsymbol{x}\,(t)}\boldsymbol{b}_{\hat{k}}, \\
\boldsymbol{b}_{\hat{k}}^\top\mathbf{W}_K^{\boldsymbol{x}\,(t+1)}\boldsymbol{b}_{\hat{k}}-\boldsymbol{b}_{\hat{k}}^\top\mathbf{W}_K^{\boldsymbol{x}\,(t)}\boldsymbol{b}_{\hat{k}} &= -\eta_t\cdot\boldsymbol{b}_{\hat{k}}^\top\nabla_{\mathbf{W}_K^{\boldsymbol{x}\,(t)}}L_{\mathcal{B}_t}(\Psi^{(t)})\boldsymbol{b}_{\hat{k}} \\
&= -\eta_t(I_{K,\boldsymbol{b}_{\hat{k}},chaos}^{(t)}+I_{K,\boldsymbol{b}_{\hat{k}},contri}^{(t)})-\eta_t\lambda\boldsymbol{b}_{\hat{k}}^\top\mathbf{W}_K^{\boldsymbol{x}\,(t)}\boldsymbol{a}_{\hat{k}},
\end{aligned}
$$

$$(18)$$

*where* $I_{Q,\boldsymbol{b}_{\hat{k}},chaos}^{(t)}$ *and* $I_{Q,\boldsymbol{b}_{\hat{k}},contri}^{(t)}$ *are defined as below.*

$$
\begin{aligned}
I_{Q,\boldsymbol{b}_{\hat{k}},chaos}^{(t)} = \frac{1}{B}\sum_{\substack{k\neq\hat{k}\in[K_1] \\ e\in[\pm] \\ n\in\mathcal{V}_k^e\cap\mathcal{B}_t}} & \Big[e\ell_n'^{\,(t)}\cdot\boldsymbol{b}_{\hat{k}}^\top(\xi_{\boldsymbol{x},L+1}^n+\sum_{r\in\mathcal{M}_{L+1}^n}\mathbf{M}_r)\sum_{i\in\mathcal{W}_{k,n}^e(t)}\mathbf{r}_i\cdot\sum_{l,j\in[L]}(\sigma_S^{(t)})_l^n(\sigma_S^{(t)})_j^n \\
& (\mathbf{W}_{O_{(i,\cdot)}}^{\boldsymbol{y}}{}^{(t)}(\boldsymbol{q}_k^{y_l^n}+\sum_{s\in\mathcal{M}_l^n}\mathbf{Q}_S+\xi_{\boldsymbol{y},l}^n))(\boldsymbol{b}_{\hat{k}}^\top\mathbf{W}_K^{\boldsymbol{x}\,(t)}((y_l^n-y_j^n)\boldsymbol{b}_k+\sum_{s\in\mathcal{M}_l^n}\mathbf{M}_s+\xi_{\boldsymbol{x},l}^n-\sum_{s\in\mathcal{M}_j^n}\mathbf{M}_s-\xi_{\boldsymbol{x},j}^n))\Big] \\
& + \frac{1}{B}\sum_{\hat{e}\in[\pm]}\sum_{n\in\mathcal{V}_{\hat{k}}^{\hat{e}}\cap\mathcal{B}_t}\Big[\ell_n'^{\,(t)}(\|\boldsymbol{b}_{\hat{k}}\|^2+\hat{e}\boldsymbol{b}_{\hat{k}}^\top(\xi_{\boldsymbol{x},L+1}^n+\sum_{r\in\mathcal{M}_{L+1}^n}\mathbf{M}_r))\sum_{i\in\mathcal{W}_{\hat{k},n}^{\hat{e}}(t)}\mathbf{r}_i\cdot \\
& \{\sum_{l\in S_{n,\hat{k}}^+}\sum_{j\in S_{n,\hat{k}}^-}(\sigma_S^{(t)})_l^n(\sigma_S^{(t)})_j^n(\mathbf{W}_{O_{(i,\cdot)}}^{\boldsymbol{y}}{}^{(t)}(\sum_{s\in\mathcal{M}_l^n}\mathbf{Q}_S+\xi_{\boldsymbol{y},l}^n))\boldsymbol{b}_{\hat{k}}^\top\mathbf{W}_K^{\boldsymbol{x}\,(t)}(2\boldsymbol{b}_{\hat{k}}+\boldsymbol{b}_{\hat{k}}^\top(\xi_{\boldsymbol{x},l}^n-\xi_{\boldsymbol{x},j}^n)) \\
& + \sum_{l\in S_{n,\hat{k}}^-}\sum_{j\in S_{n,\hat{k}}^+}(\sigma_S^{(t)})_l^n(\sigma_S^{(t)})_j^n(\mathbf{W}_{O_{(i,\cdot)}}^{\boldsymbol{y}}{}^{(t)}(\sum_{s\in\mathcal{M}_l^n}\mathbf{Q}_S+\xi_{\boldsymbol{y},l}^n))\boldsymbol{b}_{\hat{k}}^\top\mathbf{W}_K^{\boldsymbol{x}\,(t)}(-2\boldsymbol{b}_{\hat{k}}+(\xi_{\boldsymbol{x},l}^n-\xi_{\boldsymbol{x},j}^n))\}\Big] \\
I_{Q,\boldsymbol{b}_{\hat{k}},contri}^{(t)} = \frac{1}{B}\sum_{\hat{e}\in[\pm]}\sum_{n\in\mathcal{V}_{\hat{k}}^{\hat{e}}\cap\mathcal{B}_t} & \Big[2\ell_n'^{\,(t)}(\|\boldsymbol{b}_{\hat{k}}\|^2+\hat{e}(\xi_{\boldsymbol{x},L+1}^n+\sum_{r\in\mathcal{M}_{L+1}^n}\mathbf{M}_r))\sum_{i\in\mathcal{W}_{\hat{k},n}^{\hat{e}}(t)}\mathbf{r}_i\mathbf{W}_{O_{(i,\cdot)}}^{\boldsymbol{y}}{}^{(t)}\boldsymbol{d}_{\hat{k}} \\
& \boldsymbol{b}_{\hat{k}}^\top\mathbf{W}_K^{\boldsymbol{x}\,(t)}\{2(\sum_{j\in S_{n,\hat{k}}^+}(\sigma_S^{(t)})_j^n)(\sum_{j\in S_{n,\hat{k}}^-}(\sigma_S^{(t)})_j^n)\boldsymbol{b}_{\hat{k}}+\sum_{e\in[\pm]}e\cdot(\sum_{j\in S_{n,\hat{k}}^{-e}}(\sigma_S^{(t)})_j^n)(\sum_{l\in S_{n,\hat{k}}^e}(\sigma_S^{(t)})_l^n)\xi_{\boldsymbol{x},l}^n\}\Big].
\end{aligned}
$$

$$(19)$$

Similarly, $I_{K,\boldsymbol{b}_{\hat{k}},chaos}^{(t)}$ and $I_{K,\boldsymbol{b}_{\hat{k}},contri}^{(t)}$ are defined as below.

$$I_{K,\boldsymbol{b}_{\hat{k}},chaos}^{(t)} = \frac{1}{B} \sum_{\substack{k\neq\hat{k}\in[K_1] \\ e\in[\pm] \\ n\in\mathcal{V}_k^e\cap\mathcal{B}_t}} \Big[ e\cdot\ell_n'^{(t)}\boldsymbol{b}_{\hat{k}}^\top\mathbf{W}_Q^{\boldsymbol{x}\,(t)}(\boldsymbol{a}_k + e\boldsymbol{b}_k + \xi_{\boldsymbol{x},L+1}^n + \sum_{r\in\mathcal{M}_{L+1}^n}\mathbf{M}_r)\sum_{i\in\mathcal{W}_{k,n}^e(t)}\mathbf{r}_i\sum_{l,j\in[L]}(\sigma_S^{(t)})_l^n(\sigma_S^{(t)})_j^n$$

$$(\mathbf{W}_{O_{(i,\cdot)}}^{\boldsymbol{y}}{}^{(t)}(\boldsymbol{q}_k^{y_l^n} + \sum_{s\in\mathcal{M}_l^n}\mathbf{Q}_S + \xi_{\boldsymbol{y},l}^n))(\boldsymbol{b}_{\hat{k}}^\top(\sum_{s\in\mathcal{M}_l^n}\mathbf{M}_s + \xi_{\boldsymbol{x},l}^n - \sum_{s\in\mathcal{M}_j^n}\mathbf{M}_s - \xi_{\boldsymbol{x},j}^n))\Big]$$

$$+ \frac{1}{B}\sum_{\hat{e}\in[\pm]}\sum_{n\in\mathcal{V}_{\hat{k}}^{\hat{e}}\cap\mathcal{B}_t}\Big[\ell_n'^{(t)}\boldsymbol{b}_{\hat{k}}^\top\mathbf{W}_Q^{\boldsymbol{x}\,(t)}(\hat{e}\boldsymbol{a}_{\hat{k}} + \boldsymbol{b}_{\hat{k}} + \hat{e}(\xi_{\boldsymbol{x},L+1}^n + \sum_{r\in\mathcal{M}_{L+1}^n}\mathbf{M}_r))\sum_{i\in\mathcal{W}_{\hat{k},n}^{\hat{e}}(t)}\mathbf{r}_i\cdot$$

$$\{\sum_{l\in S_{n,\hat{k}}^+}\sum_{j\in S_{n,\hat{k}}^-}(\sigma_S^{(t)})_l^n(\sigma_S^{(t)})_j^n(\mathbf{W}_{O_{(i,\cdot)}}^{\boldsymbol{y}}{}^{(t)}(\sum_{s\in\mathcal{M}_l^n}\mathbf{Q}_S + \xi_{\boldsymbol{y},l}^n))(2\|\boldsymbol{b}_{\hat{k}}\|^2 + \boldsymbol{b}_{\hat{k}}^\top(\xi_{\boldsymbol{x},l}^n - \xi_{\boldsymbol{x},j}^n))$$

$$+ \sum_{l\in S_{n,\hat{k}}^-}\sum_{j\in S_{n,\hat{k}}^+}(\sigma_S^{(t)})_l^n(\sigma_S^{(t)})_j^n(\mathbf{W}_{O_{(i,\cdot)}}^{\boldsymbol{y}}{}^{(t)}(\sum_{s\in\mathcal{M}_l^n}\mathbf{Q}_S + \xi_{\boldsymbol{y},l}^n))(-2\|\boldsymbol{b}_{\hat{k}}\|^2 + \boldsymbol{b}_{\hat{k}}^\top(\xi_{\boldsymbol{x},l}^n - \xi_{\boldsymbol{x},j}^n))\}\Big]$$

$$I_{K,\boldsymbol{b}_{\hat{k}},contri}^{(t)} = \frac{1}{B}\sum_{\hat{e}\in[\pm]}\sum_{n\in\mathcal{V}_{\hat{k}}^{\hat{e}}\cap\mathcal{B}_t}\Big[2\ell_n'^{(t)}\boldsymbol{b}_{\hat{k}}^\top\mathbf{W}_Q^{\boldsymbol{x}\,(t)}(\hat{e}\boldsymbol{a}_{\hat{k}} + \boldsymbol{b}_{\hat{k}} + \hat{e}(\xi_{\boldsymbol{x},L+1}^n + \sum_{r\in\mathcal{M}_{L+1}^n}\mathbf{M}_r))\sum_{i\in\mathcal{W}_{\hat{k},n}^{\hat{e}}(t)}\mathbf{r}_i\mathbf{W}_{O_{(i,\cdot)}}^{\boldsymbol{y}}{}^{(t)}\boldsymbol{d}_{\hat{k}}$$

$$\{2(\sum_{j\in S_{n,\hat{k}}^+}(\sigma_S^{(t)})_j^n)(\sum_{j\in S_{n,\hat{k}}^-}(\sigma_S^{(t)})_j^n)\|\boldsymbol{b}_{\hat{k}}\|^2 + \sum_{e\in[\pm]}e\cdot(\sum_{j\in S_{n,\hat{k}}^{-e}}(\sigma_S^{(t)})_j^n)(\sum_{l\in S_{n,\hat{k}}^e}(\sigma_S^{(t)})_l^n)\boldsymbol{b}_{\hat{k}}^\top\xi_{\boldsymbol{x},l}^n\}\Big]. \tag{20}$$

## G Model details: MLP Part

**Lemma 17.** *(Tensor Update)*

$$\mathbf{W}_{O_{(i,\cdot)}}^{\boldsymbol{y}}{}^{(t)}\boldsymbol{c}_{\hat{k}} = \alpha_{O_{(i,\cdot)},k}^{(0)} - \eta_t\sum_{t=0}^T\nabla_{\mathbf{W}_{O_{(i,\cdot)}}^{\boldsymbol{y}}{}^{(t)}}L_{\mathcal{B}_t}(\Psi^{(t)})\boldsymbol{c}_k,$$

$$\mathbf{W}_{O_{(i,\cdot)}}^{\boldsymbol{y}}{}^{(t)}\boldsymbol{d}_{\hat{k}} = \beta_{O_{(i,\cdot)},k}^{(0)} - \eta_t\sum_{t=0}^T\nabla_{\mathbf{W}_{O_{(i,\cdot)}}^{\boldsymbol{y}}{}^{(t)}}L_{\mathcal{B}_t}(\Psi^{(t)})\boldsymbol{d}_k, \tag{21}$$

**Lemma 18.** *(Gradient Update)* $\nabla_{\mathbf{W}_{O_{(i,\cdot)}}^{\boldsymbol{y}}{}^{(t)}}L_{\mathcal{B}_t}(\Psi^{(t)}) \in \mathbb{R}^{1\times(d_{\mathcal{X}}+d_{\mathcal{Y}})}$ *can be derived as*

$$\frac{1}{B}\sum_{\substack{k\neq\hat{k}\in[K_1] \\ e\in[\pm] \\ n\in\mathcal{V}_k^e\cap\mathcal{B}_t}}\Big[\ell_n'^{(t)}\mathbf{r}_i\mathbb{1}_{O_{(i)}}^{n}{}^{(t)}\{(2\sum_{l\in S_{n,k}^e}(\sigma_S^{(t)})_l^n - 1)\boldsymbol{d}_k^\top + e\sum_{l\in[L]}(\sigma_S^{(t)})_l^n(\boldsymbol{c}_k + \sum_{s\in\mathcal{M}_l^n}\mathbf{Q}_S + \xi_{\boldsymbol{y},l}^n)^\top\}\Big] + \lambda\mathbf{W}_{O_{(i,\cdot)}}^{\boldsymbol{y}}{}^{(t)}. \tag{22}$$

**Lemma 19.** *(Concept Learning of MLP) For* $\forall i\in[m], \hat{k}\in[K_1]$,

$$\mathbf{W}_{O_{(i,\cdot)}}^{\boldsymbol{y}}{}^{(t+1)}\boldsymbol{c}_{\hat{k}} - \mathbf{W}_{O_{(i,\cdot)}}^{\boldsymbol{y}}{}^{(t)}\boldsymbol{c}_{\hat{k}} = -\eta_t\cdot\nabla_{\mathbf{W}_{O_{(i,\cdot)}}^{\boldsymbol{y}}{}^{(t)}}L_{\mathcal{B}_t}(\Psi^{(t)})\boldsymbol{c}_{\hat{k}}$$

$$= -\eta_t(I_{O_{(i,\cdot)},\boldsymbol{c}_{\hat{k}},chaos}^{(t)} + I_{O_{(i,\cdot)},\boldsymbol{c}_{\hat{k}},contri}^{(t)}) - \eta_t\lambda\mathbf{W}_{O_{(i,\cdot)}}^{\boldsymbol{y}}{}^{(t)}\boldsymbol{c}_{\hat{k}}, \tag{23}$$

*where* $I_{O_{(i,\cdot)},\boldsymbol{c}_{\hat{k}},chaos}^{(t)}$ *and* $I_{O_{(i,\cdot)},\boldsymbol{c}_{\hat{k}},contri}^{(t)}$ *are defined as*

$$I_{O_{(i,\cdot)},\boldsymbol{c}_{\hat{k}},chaos}^{(t)} = \frac{1}{B}\sum_{k\neq\hat{k}\in[K_1]}\sum_{e\in[\pm]}\sum_{n\in\mathcal{V}_k^e\cap\mathcal{B}_t}\Big[e\cdot\ell_n'^{(t)}\mathbf{r}_i\cdot\mathbb{1}_{O_{(i)}}^{n}{}^{(t)}\sum_{l\in[L]}(\sigma_S^{(t)})_l^n(\sum_{s\in\mathcal{M}_l^n}\mathbf{Q}_S + \xi_{\boldsymbol{y},l}^n)^\top\boldsymbol{c}_{\hat{k}}\Big],$$

$$I_{O_{(i,\cdot)},\boldsymbol{c}_{\hat{k}},contri}^{(t)} = \frac{1}{B}\sum_{\hat{e}\in[\pm]}\sum_{n\in\mathcal{V}_{\hat{k}}^{\hat{e}}\cap\mathcal{B}_t}\Big[\hat{e}\cdot\ell_n'^{(t)}\mathbf{r}_i\cdot\mathbb{1}_{O_{(i)}}^{n}{}^{(t)}\|\boldsymbol{c}_{\hat{k}}\|^2\Big]. \tag{24}$$

**Remark 2.** *(Informal Discussions). Interestingly, the gradient of MLPs' Concept Learning is very large. We have the following situations.*

- When the neuron is activated (i.e., $\{n \in \mathcal{V}_k^{\hat{e}} \cap \mathcal{B}_t,$ if $(\mathbf{W}_K^{\boldsymbol{x}\,(t)} \boldsymbol{b}_k)^\top \mathbf{W}_Q^{\boldsymbol{x}\,(t)} \boldsymbol{b}_k > 0\}$ , and $\alpha_{O_{(i,\cdot)},\hat{k}}^{(t)} + \hat{e} \cdot (2 \sum_{l \in S_{n,\hat{k}}^{\hat{e}}} (\sigma_S^{(t)})_l^n - 1) \mathbf{W}_{O_{(i,\cdot)}}^{\boldsymbol{y}\,(t)} \boldsymbol{d}_{\hat{k}} > 0$), the neuron is likely to be activated ($i \in \mathcal{W}_{\hat{k},n}^{\hat{e}}(t)$).

  1. If (1) $\mathbf{r}_i \cdot \hat{e} > 0, i \in \mathcal{W}_{\hat{k},n}^{\hat{e}}(t) \Leftrightarrow i \in \mathcal{W}_{\hat{k},n}^{\hat{e}}(t) \cap \mathcal{U}_{\hat{k},n}^{\hat{e}}(t)$, the gradient will advance the $\mathbf{W}_{O_{(i,\cdot)}}^{\boldsymbol{y}\,(t)} \boldsymbol{c}_{\hat{k}}$;

  2. if (2) $\mathbf{r}_i \cdot \hat{e} < 0, i \in \mathcal{W}_{\hat{k},n}^{\hat{e}}(t) \Leftrightarrow i \in \mathcal{W}_{\hat{k},n}^{\hat{e}}(t) - \mathcal{U}_{\hat{k},n}^{\hat{e}}(t)$, the gradient will diminish the $\mathbf{W}_{O_{(i,\cdot)}}^{\boldsymbol{y}\,(t)} \boldsymbol{c}_{\hat{k}}$, thus help deactivate this neuron.

**Lemma 20.** *(Label Semantic Learning of MLP) For $\forall i \in [m], \hat{k} \in [K_1]$,*

$$
\begin{aligned}
\mathbf{W}_{O_{(i,\cdot)}}^{\boldsymbol{y}\,(t+1)} \boldsymbol{d}_{\hat{k}} - \mathbf{W}_{O_{(i,\cdot)}}^{\boldsymbol{y}\,(t)} \boldsymbol{d}_{\hat{k}} &= -\eta_t \cdot \nabla_{\mathbf{W}_{O_{(i,\cdot)}}^{\boldsymbol{y}\,(t)}} L_{\mathcal{B}_t}(\Psi^{(t)}) \boldsymbol{d}_{\hat{k}} \\
&= -\eta_t (I_{O_{(i,\cdot)},\boldsymbol{d}_{\hat{k}},chaos}^{(t)} + I_{O_{(i,\cdot)},\boldsymbol{d}_{\hat{k}},contri}^{(t)}) - \eta_t \lambda \mathbf{W}_{O_{(i,\cdot)}}^{\boldsymbol{y}\,(t)} \boldsymbol{d}_{\hat{k}},
\end{aligned}
\tag{25}
$$

*where $I_{O_{(i,\cdot)},\boldsymbol{d}_{\hat{k}},chaos}^{(t)}$ and $I_{O_{(i,\cdot)},\boldsymbol{d}_{\hat{k}},contri}^{(t)}$ are defined as*

$$
\begin{aligned}
I_{O_{(i,\cdot)},\boldsymbol{d}_{\hat{k}},chaos}^{(t)} &= \frac{1}{B} \sum_{k \in [K_1]} \sum_{e \in [\pm]} \sum_{n \in \mathcal{V}_k^e \cap \mathcal{B}_t} \Big[ e \cdot \ell_n'^{(t)} \mathbf{r}_i \cdot \mathbb{1}_{O_{(i)}}^{n\,(t)} \sum_{l \in [L]} (\sigma_S^{(t)})_l^n (\sum_{s \in \mathcal{M}_l^n} \mathbf{Q}_S + \xi_{\boldsymbol{y},l}^n)^\top \boldsymbol{d}_{\hat{k}} \Big], \\
I_{O_{(i,\cdot)},\boldsymbol{d}_{\hat{k}},contri}^{(t)} &= \frac{1}{B} \sum_{\substack{\hat{e} \in [\pm] \\ n \in \mathcal{V}_{\hat{k}}^{\hat{e}} \cap \mathcal{B}_t}} \Big[ \ell_n'^{(t)} \mathbf{r}_i \cdot \mathbb{1}_{O_{(i)}}^{n\,(t)} (\sum_{l \in S_{n,\hat{k}}^{\hat{e}}} (\sigma_S^{(t)})_l^n - \sum_{l \in S_{n,\hat{k}}^{-\hat{e}}} (\sigma_S^{(t)})_l^n) \|\boldsymbol{d}_{\hat{k}}\|^2 \Big].
\end{aligned}
\tag{26}
$$

## H Discussions over Parameter Settings

Note that we do not have any requirement upon demonstration length $L$ and batch size $B$ for training, thus the training can be really flexible compared with the strict requirement in [28]. The condition on dimensionality $d_{\mathcal{X}}, d_{\mathcal{Y}}$ and the network width $m$ ensure the learning problem is in a sufficiently overparameterized setting where the norm and the inner products of the Gaussian noise and initialized NN can be controlled within a certain range with high probability $1 - \delta$, which is standard requirements in recent *feature learning* line-of-research [41, 57, 53, 45, 58, 42, 52, 43]. The weak requirement on network width $m$ allows us to conduct a fine-grained analysis based on the network projection length, which is fundamentally differs from the NTK line of research [92] that requires an infinitely wide network to perform linear regression over a prescribed feature map. The condition on $\gamma$ ensures the learning step to be small and thus learning process enjoys an approximation to gradient flow rather than the challenging "Oscillation" regime [93], which is analyzable but not necessary in presenting our theory. The condition on the small $\lambda$ is to ensure that the learning dynamic of Attention and MLP would not stuck at the origin point, and ensure that we can analyze the expected learning dynamic with limited impact of the regularization at the initial stage, which is also adopted in [53]. The condition on $K$ is to control the impact of cross-concept contribution in the Attention's learning dynamic, which can actually be relaxed at the cost of a denser analysis. The condition on $\sigma_\xi$ is to ensure that the impact of the norms and inner-products involving the Gaussian Noise on the gradient cannot surpass those in the order of feature's norms, which ensures the gradient flows to be not too noisy and could converge to the expected gradient flow exponentially. Last but not least, the conditions on $\sigma_1$ guarantee that the initial beliefs of MLP is small and the gradients of SGD can update the model effectively. The condition of $\sigma_0$ is only used when discussing the OOD scenario.

## I Convergence of Expectation

In this section, we assume all the events in the Section D hold, denoted as $\Upsilon_{\text{Pre}}$.

We examine the evolution of $\mathbb{E}(\Psi'^t) := \{\mathbb{E}(\mathbf{W}_Q^{\boldsymbol{x}\,(t)}), \mathbb{E}(\mathbf{W}_K^{\boldsymbol{x}\,(t)}), \mathbb{E}(\mathbf{W}_{O_{(i,\cdot)}}^{(t)})\}$ at the whole iteration $0 \leq t \leq \underline{t}$, where the expectation $\mathbb{E}[\cdot]$ is taken over the stochastic batches. As such, we can see every stochastic gradient update within each batch as a gradient update upon noise-free and category-balanced concept-specific prompts.

**Lemma 21.** *For $\forall k_1 \in [K_1]$, we define $\boldsymbol{a}_{k_1} := \dfrac{\boldsymbol{\mu}_{k_1}^+ + \boldsymbol{\mu}_{k_1}^-}{2}$ and $\boldsymbol{b}_{k_1} := \dfrac{\boldsymbol{\mu}_{k_1}^+ - \boldsymbol{\mu}_{k_1}^-}{2}$. By definition, we then have*

$$
\begin{aligned}
&\boldsymbol{\mu}_{k_1}^+ = \boldsymbol{a}_{k_1} + \boldsymbol{b}_{k_1}, \quad \boldsymbol{\mu}_{k_1}^- = \boldsymbol{a}_{k_1} - \boldsymbol{b}_{k_1}, \\
&\langle \boldsymbol{a}_{k_1}, \boldsymbol{b}_{k_1} \rangle = 0, \quad \{\boldsymbol{a}_{k_1}, \boldsymbol{b}_{k_1}\} \perp \{\boldsymbol{a}_{k_1'}, \boldsymbol{b}_{k_1'}\}, \\
&\langle \boldsymbol{\mu}_{k_1}^+, \boldsymbol{\mu}_{k_1}^- \rangle = \|\boldsymbol{a}_{k_1}\|^2 - \|\boldsymbol{b}_{k_1}\|^2, \quad \|\boldsymbol{\mu}_{k_1}^{\pm}\|^2 = \|\boldsymbol{a}_{k_1}\|^2 + \|\boldsymbol{b}_{k_1}\|^2 = \|\mathbf{u}\|^2, \\
&\frac{1}{2}\|\mathbf{u}\|^2 < \|\boldsymbol{a}_{k_1}\|^2 \le \frac{\kappa_{\boldsymbol{x}} + 1}{2}\|\mathbf{u}\|^2, \quad \frac{-\kappa_{\boldsymbol{x}} + 1}{2}\|\mathbf{u}\|^2 \le \|\boldsymbol{b}_{k_1}\|^2 < \frac{1}{2}\|\mathbf{u}\|^2,
\end{aligned}
\tag{27}
$$

*for $\forall k_1' \ne k_1 \in [K_1]$.*

**Remark 3.** *We observe that, through this formulation, the shared component $\boldsymbol{a}_{k_1}$ can be interpreted as the "concept" part of the two features, while the terms $\pm \boldsymbol{b}_{k_1}$ represent their opposing semantic aspects. The relevance of this modeling is exemplified by Figure 1(b) in [12], where the concept "[Bird]" is composed of orthogonal steering vectors: "plant $\Rightarrow$ animal" and "mammal $\Rightarrow$ bird." These vectors correspond to the concept feature $\boldsymbol{a}_k$ and the semantic label features $\boldsymbol{b}_k$, respectively.*

**Idempotent Operator Trick**. Define $\mathbb{U} := \mathrm{span}(\mathbf{M})$ and its complement space $\mathbb{U}^\perp$. By definition, we know that $\dim(\mathbb{U}) = K$ and $\dim(\mathbb{U}^\perp) = d_{\mathcal{X}} - K$. Then we can have a set of standard orthogonal basis for $\mathbb{R}^d$, defined as

$$
\beta_{\mathbb{U} \oplus \mathbb{U}^\perp} = \{\frac{\boldsymbol{a}_1}{\|\boldsymbol{a}_1\|}, \frac{\boldsymbol{b}_1}{\|\boldsymbol{b}_1\|}, \frac{\boldsymbol{a}_2}{\|\boldsymbol{a}_2\|}, \frac{\boldsymbol{b}_2}{\|\boldsymbol{b}_2\|}, \cdots, \frac{\boldsymbol{a}_{K_1}}{\|\boldsymbol{a}_{K_1}\|}, \frac{\boldsymbol{b}_{K_1}}{\|\boldsymbol{b}_{K_1}\|}, \frac{\boldsymbol{\nu}_1}{\|\mathbf{u}\|}, \frac{\boldsymbol{\nu}_2}{\|\mathbf{u}\|}, \cdots, \frac{\boldsymbol{\nu}_{K_2}}{\|\mathbf{u}\|}, \boldsymbol{u}_1^\perp, \cdots, \boldsymbol{u}_{d_{\mathcal{X}}-K}^\perp\},
$$

where $\boldsymbol{u}_1^\perp, \cdots, \boldsymbol{u}_{d_{\mathcal{X}}-K}^\perp$ are the standard orthogornal basis of $\mathbb{U}^\perp$. Then we can derive that

$$
\sum_{s=1}^{K_1} \frac{\boldsymbol{a}_s \boldsymbol{a}_s^\top}{\|\boldsymbol{a}_s\|^2} + \sum_{s=1}^{K_1} \frac{\boldsymbol{b}_s \boldsymbol{b}_s^\top}{\|\boldsymbol{b}_s\|^2} + \sum_{r=1}^{K_2} \frac{\boldsymbol{\nu}_r \boldsymbol{\nu}_r^\top}{\|\mathbf{u}\|^2} + \sum_{w=1}^{d_{\mathcal{X}}-K} \boldsymbol{u}_w^\perp \boldsymbol{u}_w^{\perp\top} = \mathbf{I}_{d_{\mathcal{X}} \times d_{\mathcal{X}}}.
\tag{28}
$$

**Lemma 22.** *(Partial Statement of Lemma 1).* $\mathbb{E}[\mathbf{W}_Q^{\boldsymbol{x}}]$ *and* $\mathbb{E}[\mathbf{W}_K^{\boldsymbol{x}}]$ *are identical and symmetric during the whole iterations. We can decompose* $\mathbb{E}[\mathbf{W}_Q^{\boldsymbol{x}(t)}]$ *and* $\mathbb{E}[\mathbf{W}_K^{\boldsymbol{x}(t)}]$ *by (scaled) idempotent matrices.*

$$
\begin{aligned}
\mathbb{E}[\mathbf{W}_Q^{\boldsymbol{x}(t)}] &= \sum_{s=1}^{K_1} \alpha_{Q,s}^{(t)} \cdot \frac{\boldsymbol{a}_s \boldsymbol{a}_s^\top}{\|\boldsymbol{a}_s\|^4} + \sum_{s=1}^{K_1} \beta_{Q,s}^{(t)} \cdot \frac{\boldsymbol{b}_s \boldsymbol{b}_s^\top}{\|\boldsymbol{b}_s\|^4} + \sum_{r=1}^{K_2} \tau_{Q,r}^{(t)} \cdot \frac{\boldsymbol{\nu}_r \boldsymbol{\nu}_r^\top}{\|\mathbf{u}\|^4} + \sum_{w=1}^{d_{\mathcal{X}}-K} \rho_{Q,w}^{(t)} \cdot \boldsymbol{u}_w^\perp \boldsymbol{u}_w^{\perp\top}, \\
\mathbb{E}[\mathbf{W}_K^{\boldsymbol{x}(t)}] &= \sum_{s=1}^{K_1} \alpha_{K,s}^{(t)} \cdot \frac{\boldsymbol{a}_s \boldsymbol{a}_s^\top}{\|\boldsymbol{a}_s\|^4} + \sum_{s=1}^{K_1} \beta_{K,s}^{(t)} \cdot \frac{\boldsymbol{b}_s \boldsymbol{b}_s^\top}{\|\boldsymbol{b}_s\|^4} + \sum_{r=1}^{K_2} \tau_{K,r}^{(t)} \cdot \frac{\boldsymbol{\nu}_r \boldsymbol{\nu}_r^\top}{\|\mathbf{u}\|^4} + \sum_{w=1}^{d_{\mathcal{X}}-K} \rho_{K,w}^{(t)} \cdot \boldsymbol{u}_w^\perp \boldsymbol{u}_w^{\perp\top},
\end{aligned}
\tag{29}
$$

*where $\alpha_{Q,s}^{(t)}$ and $\alpha_{K,s}^{(t)}$ represent the concept learning process, $\beta_{Q,s}^{(t)}$ and $\beta_{K,s}^{(t)}$ represent the concept-specific semantic learning process and $\tau_{Q,r}^{(t)}, \tau_{K,r}^{(t)}, \rho_{Q,w}^{(t)}, \rho_{K,w}^{(t)}$ represent the memorization of the concept irrelevant noise.*

*Proof.* Apparently they hold at $t = 0$, suppose it holds at step $t$, thus

$$
\mathbb{E}[\mathbf{W}_K^{\boldsymbol{x}(t)}] = \mathbb{E}[(\mathbf{W}_K^{\boldsymbol{x}(t)})^\top] = \mathbb{E}[\mathbf{W}_Q^{\boldsymbol{x}(t)}] = \mathbb{E}[(\mathbf{W}_Q^{\boldsymbol{x}(t)})^\top],
$$

we examine $t + 1$. It holds that

$$
\mathbb{E}_{\mathcal{B}_t}[\mathbf{W}_K^{(t+1)} \mid \mathbb{E}(\Psi'^{(t)})] = \mathbb{E}[\mathbf{W}_K^{\boldsymbol{x}(t)}] - \eta_t \mathbb{E}_{\mathcal{B}_t}[\partial_{\mathbf{W}_K^{\boldsymbol{x}(t)}} L_{\mathcal{B}_t}(\mathbb{E}(\Psi'^{(t)}))]
$$

$$
\mathbb{E}_{\mathcal{B}_t}[\mathbf{W}_Q^{(t+1)} \mid \mathbb{E}(\mathbb{E}(\Psi'^{(t)}))] = \mathbb{E}[\mathbf{W}_Q^{\boldsymbol{x}(t)}] - \eta_t \mathbb{E}_{\mathcal{B}_t}[\partial_{\mathbf{W}_Q^{\boldsymbol{x}(t)}} L_{\mathcal{B}_t}(\mathbb{E}(\Psi'^{(t)}))]
$$

Here, we see $\mathbb{E}(\Psi'^{(t)})$ as fixed matrices and the expectation $\mathbb{E}_{\mathcal{B}_t}[\cdot]$ is taken over the stochastic batch at the time step $t$. As we are considering expectation over the isotropic prompt distribution, which can be seen as a noiseless distribution with an averaged categories of words and labels, the expected gradient form could be written as symmetric form:

$$
\begin{aligned}
\mathbb{E}_{\mathcal{B}_t}[\mathbf{W}_K^{\boldsymbol{x}(t+1)} \mid \mathbb{E}(\Psi'^{(t)})] - \mathbb{E}[\mathbf{W}_K^{\boldsymbol{x}(t)}] = &\sum_{s=1}^{K_1} (a_{K,s}^{(t)} \boldsymbol{a}_s \boldsymbol{a}_s^\top + b_{K,s}^{(t)} \boldsymbol{b}_s \boldsymbol{b}_s^\top) \\
&+ \lambda (\sum_{r=1}^{K_2} c_{Q,r}^{(t)} \boldsymbol{\nu}_r \boldsymbol{\nu}_r^\top + \sum_{w=1}^{d_{\mathcal{X}}-2K_1-K_2} d_{K,w}^{(t)} \cdot \boldsymbol{u}_w \boldsymbol{u}_w^\top)
\end{aligned}
$$

with some coefficients $a_{K,s}^{(t)}, b_{K,s}^{(t)}, c_{Q,r}^{(t)}, d_{K,w}^{(t)}, \forall s \in [K_1], r \in [K_2], w \in [d_{\mathcal{X}} - 2K_1 - K_2]$. It's direct to check that $\mathbb{E}[\mathbf{W}_Q^{\boldsymbol{x}(t)}]$ also has the exactly same outcome. The proof is completed. $\qquad\square$

Worth noting that

$$\boldsymbol{\mu}_s^{e\top}\mathbb{E}[\mathbf{W}_Q^{\boldsymbol{x}\,(t)}]\boldsymbol{\mu}_s^e = \alpha_{Q,s}^{(t)} + \beta_{Q,s}^{(t)}, \quad \boldsymbol{\mu}_s^{e\top}\mathbb{E}[\mathbf{W}_K^{\boldsymbol{x}\,(t)}]\boldsymbol{\mu}_s^e = \alpha_{K,s}^{(t)} + \beta_{K,s}^{(t)},$$
$$\boldsymbol{\mu}_s^{-e\top}\mathbb{E}[\mathbf{W}_Q^{\boldsymbol{x}\,(t)}]\boldsymbol{\mu}_s^e = \alpha_{Q,s}^{(t)} - \beta_{Q,s}^{(t)}, \quad \boldsymbol{\mu}_s^{-e\top}\mathbb{E}[\mathbf{W}_K^{\boldsymbol{x}\,(t)}]\boldsymbol{\mu}_s^e = \alpha_{K,s}^{(t)} - \beta_{K,s}^{(t)}, \tag{30}$$
$$\boldsymbol{\nu}_r^{\top}\mathbb{E}[\mathbf{W}_Q^{\boldsymbol{x}\,(t)}]\boldsymbol{\nu}_r = \tau_{Q,r}^{(t)}, \quad \boldsymbol{\nu}_r^{\top}\mathbb{E}[\mathbf{W}_K^{\boldsymbol{x}\,(t)}]\boldsymbol{\nu}_r = \tau_{K,r}^{(t)}.$$

We will also have

$$(\mathbb{E}[\mathbf{W}_K^{\boldsymbol{x}\,(t)}]\boldsymbol{\mu}_s^e)^{\top}\mathbb{E}[\mathbf{W}_Q^{\boldsymbol{x}\,(t)}]\boldsymbol{\mu}_s^e = \alpha_{Q,s}^{(t)}\cdot\alpha_{K,s}^{(t)}/\|\boldsymbol{a}_s\|^2 + \beta_{Q,s}^{(t)}\cdot\beta_{K,s}^{(t)}/\|\boldsymbol{b}_s\|^2,$$
$$(\mathbb{E}[\mathbf{W}_K^{\boldsymbol{x}\,(t)}]\boldsymbol{\mu}_s^{-e})^{\top}\mathbb{E}[\mathbf{W}_Q^{\boldsymbol{x}\,(t)}]\boldsymbol{\mu}_s^e = \alpha_{Q,s}^{(t)}\cdot\alpha_{K,s}^{(t)}/\|\boldsymbol{a}_s\|^2 - \beta_{Q,s}^{(t)}\cdot\beta_{K,s}^{(t)}/\|\boldsymbol{b}_s\|^2, \tag{31}$$

for $\forall e \in [\pm]$ and for $\forall e' \in [\pm], s' \in [K_1], r \in [K_2], w \in [d_{\mathcal{X}} - K], \forall \mathbf{u} \in \{\boldsymbol{\mu}_{s'}^{e'}, \boldsymbol{\nu}_r, \boldsymbol{u}_w^{\perp}\}$,

$$(\mathbb{E}[\mathbf{W}_K^{\boldsymbol{x}\,(t)}]\mathbf{u})^{\top}\mathbb{E}[\mathbf{W}_Q^{\boldsymbol{x}\,(t)}]\boldsymbol{\mu}_s^e = 0. \tag{32}$$

Similar conclusions hold when the query vectors are $\boldsymbol{\nu}_r$ and $\boldsymbol{u}_w^{\perp}, \forall r \in [K_2], w \in [d_{\mathcal{X}} - K]$.

**Definition 5.** *Define* $\mathbb{Q} \coloneqq span(\mathbf{Q})$ *and its complement space* $\mathbb{Q}^{\perp}$, *we can decompose* $i$-th *row of* $\mathbf{W}_O^{\boldsymbol{y}}$ *via the following decomposition:*

$$\mathbb{E}[\mathbf{W}_{O_{(i,\cdot)}}^{\boldsymbol{y}}{}^{(t)}] = \sum_{k=1}^{K_1}\alpha_{O_{(i,\cdot)},k}^{(t)}\cdot\frac{\boldsymbol{c}_k^{\top}}{\|\boldsymbol{c}_k\|^2} + \sum_{k=1}^{K_1}\beta_{O_{(i,\cdot)},k}^{(t)}\cdot\frac{\boldsymbol{d}_k^{\top}}{\|\boldsymbol{d}_k\|^2} + \sum_{w=1}^{d_{\mathcal{Y}}-K_1}\rho_{O_{(i,\cdot)},w}^{(t)}\cdot\boldsymbol{q}_w^{\perp\top} \in \mathbb{R}^{1\times d_{\mathcal{Y}}}., \tag{33}$$

*where* $\boldsymbol{q}_1^{\perp},\cdots,\boldsymbol{q}_{d_{\mathcal{Y}}-K_1}^{\perp}$ *are the standard orthogonal basis of the complement space* $\mathbb{Q}^{\perp}$. *Then we have*

$$\mathbb{E}[\mathbf{W}_{O_{(i,\cdot)}}^{\boldsymbol{y}}{}^{(t)}]\boldsymbol{q}_k^e = \alpha_{O_{(i,\cdot)},k}^{(t)} + e\cdot\beta_{O_{(i,\cdot)},k}^{(t)}, \tag{34}$$

*for* $\forall e \in [\pm], i \in [m], k \in [K_1]$.

**Lemma 23.** *At initialization, for some* $e \in [\pm]$ *and* $\forall k \in [K_1]$, *define*

$$\zeta_k^e \coloneqq 2\mathop{\mathbb{E}}_{n\in\mathcal{V}_k^e}\Big[\sum_{l\in S_{n,k}^e}(\sigma_S^{(0)})_l^n\Big] - 1 = \frac{\exp(\sigma_0^2\|\boldsymbol{b}_k\|^2) - \exp(-\sigma_0^2\|\boldsymbol{b}_k\|^2)}{\exp(\sigma_0^2\|\boldsymbol{b}_k\|^2) + \exp(-\sigma_0^2\|\boldsymbol{b}_k\|^2)},$$

*then we have some* $\omega_{\zeta_k^e} \in (0, \omega'_{\zeta_k^e})$ *where* $\omega'_{\zeta_k^e} < 1$, *the following will hold*

$$\frac{\alpha_{Q,k}^{(0)}}{\|\boldsymbol{a}_k\|^2} = \frac{\alpha_{K,k}^{(0)}}{\|\boldsymbol{a}_k\|^2} = \frac{\beta_{Q,k}^{(0)}}{\|\boldsymbol{b}_k\|^2} = \frac{\beta_{K,k}^{(0)}}{\|\boldsymbol{b}_k\|^2} = \frac{\tau_{Q,r}^{(0)}}{\|\mathbf{u}\|^2} = \frac{\tau_{K,r}^{(0)}}{\|\mathbf{u}\|^2} = \rho_{Q,w}^{(0)} = \rho_{K,w}^{(0)} = \sigma_0,$$

$$\mathop{\mathbb{E}}_{n\in\mathcal{V}_k^e}[|\mathcal{U}_{k,n}^e(0)|] = \Big|\{i \in [m] \mid \mathbf{r}_i = \frac{e}{m}, \alpha_{O_{(i,\cdot)},k}^{(0)} + e\zeta_k^e\cdot\beta_{O_{(i,\cdot)},k}^{(0)} > 0\}\Big| \geq \frac{m}{4} - \sqrt{\frac{m\log(\frac{10K_1}{\delta})}{2}} \geq \frac{m}{8},$$

$$\mathop{\mathbb{E}}_{n\in\mathcal{V}_k^e}[|\mathcal{W}_{k,n}^e(0) - \mathcal{U}_{k,n}^e(0)|] = \Big|\{i \in [m] \mid \mathbf{r}_i = -\frac{e}{m}, \alpha_{O_{(i,\cdot)},k}^{(0)} + e\zeta_k^e\beta_{O_{(i,\cdot)},k}^{(0)} > 0\}\Big| \leq \frac{m}{4} + \sqrt{\frac{m\log(\frac{10K_1}{\delta})}{2}}.$$

$$\mathop{\mathbb{E}}_{n\in\mathcal{V}_k^e}[|\mathcal{U}_{k,n}^e(0)\cap(\mathcal{W}_{k,n}^{-e}(0) - \mathcal{U}_{k,n}^{-e}(0))|] \leq \frac{(1+\omega_{\zeta_k^e})m}{8} + \sqrt{\frac{m\log(\frac{10K_1}{\delta})}{2}} \leq \frac{(1+\omega'_{\zeta_k^e})m}{8},$$

$$\mathop{\mathbb{E}}_{n\in\mathcal{V}_k^e}[|\mathcal{U}_{k,n}^e(0) - (\mathcal{W}_{k,n}^{-e}(0) - \mathcal{U}_{k,n}^{-e}(0))|] \geq \frac{(1-\omega_{\zeta_k^e})m}{8} - \sqrt{\frac{m\log(\frac{10K_1}{\delta})}{2}} \geq \frac{(1-\omega'_{\zeta_k^e})m}{8}.$$

*The parameter* $\omega'_{\zeta_k^e}$ *is determined by* $\sigma_0, \sigma_1, \|\boldsymbol{a}_k\|, \|\boldsymbol{b}_k\|, \|\boldsymbol{c}_k\|$ *and* $\|\boldsymbol{d}_k\|$.

*Proof.* We have that $\mathop{\mathbb{E}}_{n\in\mathcal{V}_k^e}[\mathcal{U}_{k,n}^e(0)\cap(\mathcal{W}_{k,n}^{-e}(0) - \mathcal{U}_{k,n}^{-e}(0))] \neq \varnothing$. By Lemma 7, we see that for

$$\zeta_k^e = 2\mathop{\mathbb{E}}_{n\in\mathcal{V}_k}\Big[\sum_{l\in S_{n,k}^{y_{S_n}}}(\sigma_S^{(0)})_l^n\Big] - 1 = \frac{\exp(\sigma_0^2\|\boldsymbol{b}_k\|^2) - \exp(-\sigma_0^2\|\boldsymbol{b}_k\|^2)}{\exp(\sigma_0^2\|\boldsymbol{b}_k\|^2) + \exp(-\sigma_0^2\|\boldsymbol{b}_k\|^2)},$$

we can have corresponding $\omega_{\zeta_k^e} \in (0, \omega'_{\zeta_k^e})$ where $\omega'_{\zeta_k^e} < 1$ to ensure the conclusion holds.

$\square$

**Lemma 24.** *(Coefficient Update) Denote* $\mathbb{E}(\Psi'^{(t)}) := \{\mathbb{E}(\mathbf{W}_Q^{x\,(t)}), \mathbb{E}(\mathbf{W}_K^{x\,(t)}), \mathbb{E}(\mathbf{W}_{O_{(i,\cdot)}}^{(t)})\}$, *where the expectation* $\mathbb{E}[\cdot]$ *is taken over the stochastic batches. We have*

$$\alpha_{Q,s}^{(T)} = \alpha_{Q,s}^{(0)} - \eta_t \sum_{t=1}^{T} \boldsymbol{a}_s^{\top} \nabla_{\mathbf{W}_Q^{x\,(t)}} \mathbb{E}_{\mathcal{B}_t}[L_{\mathcal{B}_t}(\Psi'^{(t)}) \mid \mathbb{E}(\Psi'^{(t-1)})] \boldsymbol{a}_s,$$

$$\alpha_{K,s}^{(T)} = \alpha_{K,s}^{(0)} - \eta_t \sum_{t=1}^{T} \boldsymbol{a}_s^{\top} \nabla_{\mathbf{W}_K^{x\,(t)}} \mathbb{E}_{\mathcal{B}_t}[L_{\mathcal{B}_t}(\Psi'^{(t)}) \mid \mathbb{E}(\Psi'^{(t-1)})] \boldsymbol{a}_s,$$

$$\beta_{Q,s}^{(T)} = \beta_{Q,s}^{(0)} - \eta_t \sum_{t=1}^{T} \boldsymbol{b}_s^{\top} \nabla_{\mathbf{W}_Q^{x\,(t)}} \mathbb{E}_{\mathcal{B}_t}[L_{\mathcal{B}_t}(\Psi'^{(t)}) \mid \mathbb{E}(\Psi'^{(t-1)})] \boldsymbol{b}_s,$$

$$\beta_{K,s}^{(T)} = \beta_{K,s}^{(0)} - \eta_t \sum_{t=1}^{T} \boldsymbol{b}_s^{\top} \nabla_{\mathbf{W}_K^{x\,(t)}} \mathbb{E}_{\mathcal{B}_t}[L_{\mathcal{B}_t}(\Psi'^{(t)}) \mid \mathbb{E}(\Psi'^{(t-1)})] \boldsymbol{b}_s,$$

$$\tau_{Q,r}^{(T)} = \tau_{Q,r}^{(0)} - \eta_t \sum_{t=1}^{T} \boldsymbol{\nu}_r^{\top} \nabla_{\mathbf{W}_Q^{x\,(t)}} \mathbb{E}_{\mathcal{B}_t}[L_{\mathcal{B}_t}(\Psi'^{(t)}) \mid \mathbb{E}(\Psi'^{(t-1)})] \boldsymbol{\nu}_r,$$

$$\tau_{K,r}^{(T)} = \tau_{K,r}^{(0)} - \eta_t \sum_{t=1}^{T} \boldsymbol{\nu}_r^{\top} \nabla_{\mathbf{W}_K^{x\,(t)}} \mathbb{E}_{\mathcal{B}_t}[L_{\mathcal{B}_t}(\Psi'^{(t)}) \mid \mathbb{E}(\Psi'^{(t-1)})] \boldsymbol{\nu}_r, \qquad (35)$$

$$\rho_{Q,w}^{(T)} = \rho_{Q,w}^{(0)} - \eta_t \sum_{t=1}^{T} \boldsymbol{u}_w^{\perp\,\top} \nabla_{\mathbf{W}_Q^{x\,(t)}} \mathbb{E}_{\mathcal{B}_t}[L_{\mathcal{B}_t}(\Psi'^{(t)}) \mid \mathbb{E}(\Psi'^{(t-1)})] \boldsymbol{u}_w^{\perp},$$

$$\rho_{K,w}^{(T)} = \rho_{K,w}^{(0)} - \eta_t \sum_{t=1}^{T} \boldsymbol{u}_w^{\perp\,\top} \nabla_{\mathbf{W}_K^{x\,(t)}} \mathbb{E}_{\mathcal{B}_t}[L_{\mathcal{B}_t}(\Psi'^{(t)}) \mid \mathbb{E}(\Psi'^{(t-1)})] \boldsymbol{u}_w^{\perp},$$

$$\alpha_{O_{(i,\cdot)},k}^{(T)} = \alpha_{O_{(i,\cdot)},k}^{(0)} - \eta_t \sum_{t=1}^{T} \nabla_{\mathbf{W}_{O_{(i,\cdot)}}^{(t)}} \mathbb{E}_{\mathcal{B}_t}[L_{\mathcal{B}_t}(\Psi'^{(t)}) \mid \mathbb{E}(\Psi'^{(t-1)})] \boldsymbol{c}_k,$$

$$\beta_{O_{(i,\cdot)},k}^{(T)} = \beta_{O_{(i,\cdot)},k}^{(0)} - \eta_t \sum_{t=1}^{T} \nabla_{\mathbf{W}_{O_{(i,\cdot)}}^{(t)}} \mathbb{E}_{\mathcal{B}_t}[L_{\mathcal{B}_t}(\Psi'^{(t)}) \mid \mathbb{E}(\Psi'^{(t-1)})] \boldsymbol{d}_k,$$

$$\rho_{O_{(i,\cdot)},w}^{(T)} = \rho_{O_{(i,\cdot)},w}^{(0)} - \eta_t \sum_{t=1}^{T} \nabla_{\mathbf{W}_{O_{(i,\cdot)}}^{(t)}} \mathbb{E}_{\mathcal{B}_t}[L_{\mathcal{B}_t}(\Psi'^{(t)}) \mid \mathbb{E}(\Psi'^{(t-1)})] \boldsymbol{q}_w^{\perp},$$

*where* $e \in [\pm], s \in [K_1], r \in [K_2], w \in [d_{\mathcal{X}} - K]$.

**Lemma 25.** *For* $\forall k_1 \in [K_1]$, *we define* $\boldsymbol{c}_{k_1} := \dfrac{\boldsymbol{q}_{k_1}^+ + \boldsymbol{q}_{k_1}^-}{2}$ *and* $\boldsymbol{d}_{k_1} := \dfrac{\boldsymbol{q}_{k_1}^+ - \boldsymbol{q}_{k_1}^-}{2}$. *By definition, we then have*

$$\begin{aligned}
&\boldsymbol{q}_{k_1}^+ = \boldsymbol{c}_{k_1} + \boldsymbol{d}_{k_1}, \quad \boldsymbol{q}_{k_1}^- = \boldsymbol{c}_{k_1} - \boldsymbol{d}_{k_1}, \\
&\langle \boldsymbol{c}_{k_1}, \boldsymbol{d}_{k_1} \rangle = 0, \quad \{\boldsymbol{c}_{k_1}, \boldsymbol{d}_{k_1}\} \perp \{\boldsymbol{c}_{k_1'}, \boldsymbol{d}_{k_1'}\}, \\
&\langle \boldsymbol{q}_{k_1}^+, \boldsymbol{q}_{k_1}^- \rangle = \|\boldsymbol{c}_{k_1}\|^2 - \|\boldsymbol{d}_{k_1}\|^2, \quad \|\boldsymbol{q}_{k_1}^{\pm}\|^2 = \|\boldsymbol{c}_{k_1}\|^2 + \|\boldsymbol{d}_{k_1}\|^2 = \|\mathbf{u}\|^2, \\
&\frac{1}{2}\|\mathbf{q}\|^2 < \|\boldsymbol{c}_{k_1}\|^2 \le \frac{\kappa_{\boldsymbol{y}} + 1}{2}\|\mathbf{q}\|^2, \quad \frac{-\kappa_{\boldsymbol{y}} + 1}{2}\|\mathbf{q}\|^2 \le \|\boldsymbol{d}_{k_1}\|^2 < \frac{1}{2}\|\mathbf{q}\|^2,
\end{aligned} \qquad (36)$$

*for* $\forall k_1' \neq k_1 \in [K_1]$.

Based on Lemma 22 and Lemma 24, the following two lemmas compute the update of attention's expected projection along non-feature and feature directions.

**Lemma 26.** *For* $t > 0$, *we have*

$$\begin{aligned}
&\tau_{Q,r}^{(t+1)} = (1 - \eta_t \lambda)\tau_{Q,r}^{(t)}, \quad \tau_{K,r}^{(t+1)} = (1 - \eta_t \lambda)\tau_{K,r}^{(t)}, \\
&\rho_{Q,w}^{(t+1)} = (1 - \eta_t \lambda)\rho_{Q,w}^{(t)}, \quad \rho_{K,w}^{(t+1)} = (1 - \eta_t \lambda)\rho_{K,r}^{(t)}, \\
&\rho_{O_{(i,\cdot)},\hat{w}}^{(t+1)} = (1 - \eta_t \lambda)\rho_{O_{(i,\cdot)},\hat{w}}^{(t)},
\end{aligned} \qquad (37)$$

*where* $r \in [K_2], w \in [d_{\mathcal{X}} - K], \hat{w} \in [d_{\mathcal{Y}} - K_1]$.

**Lemma 27.** *For $t > 0$, we have*

$$\alpha_{Q,k}^{(t+1)} = (1 - \eta_t\lambda)\alpha_{Q,k}^{(t)}, \quad \alpha_{K,k}^{(t+1)} = (1 - \eta_t\lambda)\alpha_{K,k}^{(t)},$$

$$\beta_{Q,k}^{(t+1)} = (1 - \eta_t\lambda)\beta_{Q,k}^{(t)}$$
$$- \frac{4\eta_t\beta_{K,k}^{(t)}\|\boldsymbol{b}_k\|^4}{K_1} \sum_{e\in[\pm]}\sum_{i\in[m]} \mathbf{r}_i\beta_{O_{(i,\cdot)},k}^{(t)} \mathop{\mathbb{E}}_{n\in\mathcal{V}_k^e}[\ell_n'^{(t)}\mathbb{1}_{O_{(i)}}^{n}{}^{(t)}(\sum_{j\in S_{n,k}^+}(\sigma_S^{(t)})_j^n)(\sum_{j\in S_{n,k}^-}(\sigma_S^{(t)})_j^n)],$$

$$\beta_{K,k}^{(t+1)} = (1 - \eta_t\lambda)\beta_{K,k}^{(t)}$$
$$- \frac{4\eta_t\beta_{Q,k}^{(t)}\|\boldsymbol{b}_k\|^4}{K_1} \sum_{e\in[\pm]}\sum_{i\in[m]} \mathbf{r}_i\beta_{O_{(i,\cdot)},k}^{(t)} \mathop{\mathbb{E}}_{n\in\mathcal{V}_k^e}[\ell_n'^{(t)}\mathbb{1}_{O_{(i)}}^{n}{}^{(t)}(\sum_{j\in S_{n,k}^+}(\sigma_S^{(t)})_j^n)(\sum_{j\in S_{n,k}^-}(\sigma_S^{(t)})_j^n)].$$

$$(38)$$

*Proof.* The deduction is direct by the symmetric property of prompt distribution in Lemma 22, and the gradient forms in Lemma 15 and Lemma 16. □

This lemma reveals that the attention layer mainly serves to learn the different semantic part of each concept, and hardly have interest in learning the shared co-concept part. Also, collaborating with Lemma 22, we see that $\beta_{Q,k}^{(t+1)} = \beta_{K,k}^{(t+1)}$, this indicates that the signal of $\beta_{Q,k}^{(t)} \cdot \beta_{K,k}^{(t)}$ would remain positive.

Also, by the symmetry property of learning progress denoted in Lemma 22, we see that $\forall k \in [K_1], \alpha_{Q,k}^{(t)} = \alpha_{K,k}^{(t)}, \beta_{Q,k}^{(t)} = \beta_{K,k}^{(t)}$. Observe that for $\forall k \in [K_1]$,

$$\mathop{\mathbb{E}}_{n\in\mathcal{V}_k^{y_{S_n}}}[\sum_{j\in S_{n,k}^{y_{S_n}}}(\sigma_S^{(t)})_j^n] = \frac{\exp(\beta_{Q,k}^{(t)} \cdot \beta_{K,k}^{(t)}/\|\boldsymbol{b}_k\|^2)}{\exp(\beta_{Q,k}^{(t)} \cdot \beta_{K,k}^{(t)}/\|\boldsymbol{b}_k\|^2) + \exp(-\beta_{Q,k}^{(t)} \cdot \beta_{K,k}^{(t)}/\|\boldsymbol{b}_k\|^2)},$$

$$\mathop{\mathbb{E}}_{n\in\mathcal{V}_k^{y_{S_n}}}[\sum_{j\in S_{n,k}^{-y_{S_n}}}(\sigma_S^{(t)})_j^n] = \frac{\exp(-\beta_{Q,k}^{(t)} \cdot \beta_{K,k}^{(t)}/\|\boldsymbol{b}_k\|^2)}{\exp(\beta_{Q,k}^{(t)} \cdot \beta_{K,k}^{(t)}/\|\boldsymbol{b}_k\|^2) + \exp(-\beta_{Q,k}^{(t)} \cdot \beta_{K,k}^{(t)}/\|\boldsymbol{b}_k\|^2)}.$$

$$(39)$$

We see from Lemma 7 that $\alpha_{Q,k}^{(0)} = \alpha_{K,k}^{(0)} = \sigma_0\|\boldsymbol{a}_k\|^2, \beta_{Q,k}^{(0)} = \beta_{K,k}^{(0)} = \sigma_0\|\boldsymbol{b}_k\|^2$. Therefore, for $t = 0, \forall k \in [K_1]$, we have

$$\mathop{\mathbb{E}}_{n\in\mathcal{V}_k^{y_{S_n}}}[\sum_{j\in S_{n,k}^{y_{S_n}}}(\sigma_S^{(0)})_j^n] = \frac{\exp(\sigma_0^2\|\boldsymbol{b}_k\|^2)}{\exp(\sigma_0^2\|\boldsymbol{b}_k\|^2) + \exp(-\sigma_0^2\|\boldsymbol{b}_k\|^2)},$$

$$\mathop{\mathbb{E}}_{n\in\mathcal{V}_k^{y_{S_n}}}[\sum_{j\in S_{n,k}^{-y_{S_n}}}(\sigma_S^{(0)})_j^n] = \frac{\exp(-\sigma_0^2\|\boldsymbol{b}_k\|^2)}{\exp(\sigma_0^2\|\boldsymbol{b}_k\|^2) + \exp(-\sigma_0^2\|\boldsymbol{b}_k\|^2)}.$$

$$(40)$$

Obviously, $\mathop{\mathbb{E}}_{n\in\mathcal{V}_k^{y_{S_n}}}[\sum_{j\in S_{n,k}^{y_{S_n}}}(\sigma_S^{(0)})_j^n] > 0.5 > \mathop{\mathbb{E}}_{n\in\mathcal{V}_k^{y_{S_n}}}[\sum_{j\in S_{n,k}^{-y_{S_n}}}(\sigma_S^{(0)})_j^n]$. Meanwhile we see that $\mathop{\mathbb{E}}_{n\in\mathcal{V}_k^{y_{S_n}}}[\sum_{j\in S_{n,k}^{y_{S_n}}}(\sigma_S^{(0)})_j^n] \approx 0.5$ due to the small $\sigma_0 = O(\|\mathbf{u}\|^{-2})$ by Condition 1.

The observation in Eq. (39), collaborating with the positiveness of $\beta_{Q,k}^{(t)} \cdot \beta_{K,k}^{(t)}$, we see that the inequality $\mathop{\mathbb{E}}_{n\in\mathcal{V}_k^{y_{S_n}}}[\sum_{j\in S_{n,k}^{y_{S_n}}}(\sigma_S^{(t)})_j^n] > \mathop{\mathbb{E}}_{n\in\mathcal{V}_k^{y_{S_n}}}[\sum_{j\in S_{n,k}^{-y_{S_n}}}(\sigma_S^{(t)})_j^n]$ will remain during whole iteration. Also, by Eq. (39), we know that

$$\mathbb{E}[(\sum_{j\in S_{n,k}^+}(\sigma_S^{(t)})_j^n)(\sum_{j\in S_{n,k}^-}(\sigma_S^{(t)})_j^n)] = \left(\exp(\beta_{Q,k}^{(t)} \cdot \beta_{K,k}^{(t)}/\|\boldsymbol{b}_k\|^2) + \exp(-\beta_{Q,k}^{(t)} \cdot \beta_{K,k}^{(t)}/\|\boldsymbol{b}_k\|^2)\right)^{-2}. \quad (41)$$

This observation under our expectation scenario greatly facilitate our analysis. Since $\ell_n'^{(t)} < 0$, it's obvious that the signal of $\mathbf{r}_i\beta_{O_{(i,\cdot)},k}^{(t)}$ will determine whether the neuron $i \in [m]$ will serve to increase or decrease the $\beta_{Q,k}^{(t)}$ and $\beta_{K,k}^{(t)}$ during the gradient update. We therefore start to analyze the MLP's update below based on Lemma 16.

**Lemma 28.** *For $t > 0$, we have*

$$\alpha_{O_{(i,\cdot)},k}^{(t+1)} = (1 - \eta_t \lambda)\alpha_{O_{(i,\cdot)},k}^{(t)} - \eta_t \underbrace{\frac{\|\boldsymbol{c}_k\|^2}{2K_1} \sum_{e \in [\pm]} [e\mathbf{r}_i \cdot \mathop{\mathbb{E}}_{n \in \mathcal{V}_k^e}({\ell_n'}^{(t)}\mathbb{1}_{O_{(i)}}^{n}{}^{(t)})]}_{\mathbb{E}(I_{O_{(i,\cdot)},\boldsymbol{c}_k,contri}^{(t)})}$$

$$- \eta_t \underbrace{\frac{(K_1 - 1)\|\boldsymbol{c}_k\|^2}{2K_1 K} \sum_{e \in [\pm]} [e\mathbf{r}_i \cdot \mathop{\mathbb{E}}_{n \in \mathcal{V}_{\neg k}^e}({\ell_n'}^{(t)}\mathbb{1}_{O_{(i)}}^{n}{}^{(t)})]}_{\mathbb{E}(I_{O_{(i,\cdot)},\boldsymbol{c}_k,chaos}^{(t)})}, \tag{42}$$

$$\beta_{O_{(i,\cdot)},k}^{(t+1)} = (1 - \eta_t\lambda)\beta_{O_{(i,\cdot)},k}^{(t)} - \frac{\eta_t\|\boldsymbol{d}_k\|^2\mathbf{r}_i}{2K_1}\sum_{e\in[\pm]}\mathop{\mathbb{E}}_{n\in\mathcal{V}_k^e}[{\ell_n'}^{(t)}\mathbb{1}_{O_{(i)}}^{n}{}^{(t)}(\sum_{l\in S_{n,k}^e}(\sigma_S^{(t)})_l^n$$

$$- \sum_{l\in S_{n,k}^{-e}}(\sigma_S^{(t)})_l^n)],$$

*where $k \in [K_1]$.*

*Proof.* The proof is direct by the symmetric property of prompt distribution in Lemma 22, and the gradient forms in Lemma 19 and Lemma 20. $\square$

An interesting fact is that the $\mathbb{E}(I_{O_{(i,\cdot)},\boldsymbol{c}_k,\text{chaos}}^{(t)})$ also contributes to the learning of $k$-th concept. This actually suits our intuition that if similar things appear in various fields (concepts), the learning process can help integrate and facilitate the learning. The following lemma demonstrate the lower bound of the attention assignment, which emerge from the good property of our expected attention.

**Lemma 29.** *For a certain iterations $t \in (0, T_1)$, for $\forall k \in [K_1], e \in [\pm]$, we have*

1. *The neuron set $\mathbb{E}[(\mathcal{W}_{k,n}^e(t) - \mathcal{U}_{k,n}^e(t)) - \mathcal{U}_{k,n}^{-e}(t)]$ is non-increasing, and all of this neuron will get deactivated. Additionally, both $\mathbb{E}[\alpha_{O_{(i,\cdot)},k}^{(t)}]$ and $e \cdot \beta_{O_{(i,\cdot)},k}^{(t)}$ would monotonically decrease. Also, it holds that $e \cdot \beta_{O_{(i,\cdot)},k}^{(t)} > 0$ and $|\alpha_{O_{(i,\cdot)},k}^{(t)}| \le e \cdot \beta_{O_{(i,\cdot)},k}^{(t)}$;*

2. *The neuron set $\mathbb{E}[\mathcal{U}_{k,n}^e(t) - (\mathcal{W}_{k,n}^{-e}(t) - \mathcal{U}_{k,n}^{-e}(t))]$ is non-increasing, and all neurons in it will turn into $\mathop{\mathbb{E}}_{n\in\mathcal{V}_k^e}[\mathcal{U}_{k,n}^e(t) \cap (\mathcal{W}_{k,n}^{-e}(t) - \mathcal{U}_{k,n}^{-e}(t))]$. Additionally, both $\mathbb{E}[\alpha_{O_{(i,\cdot)},k}^{(t)}]$ and $e \cdot \beta_{O_{(i,\cdot)},k}^{(t)}$ would monotonically increase. Also, it holds that $e \cdot \beta_{O_{(i,\cdot)},k}^{(t)} > 0$ and $|\alpha_{O_{(i,\cdot)},k}^{(t)}| \le e \cdot \beta_{O_{(i,\cdot)},k}^{(t)}$;*

3. *For $\mathop{\mathbb{E}}_{n\in\mathcal{V}_k^e}[\mathcal{U}_{k,n}^e(t) \cap (\mathcal{W}_{k,n}^{-e}(t) - \mathcal{U}_{k,n}^{-e}(t))]$, the $e \cdot \beta_{O_{(i,\cdot)},k}^{(t)}$ would monotonically increase. Besides, when there exists constant $C \ge 1$ such that*

$$\mathop{\mathbb{E}}_{n\in\mathcal{V}_k^e}({\ell_n'}^{(t)})] \le C \mathop{\mathbb{E}}_{n\in\mathcal{V}_k^{-e}}({\ell_n'}^{(t)})].$$

*the $\mathbb{E}[\alpha_{O_{(i,\cdot)},k}^{(t)}]$ would be contributed to increase, otherwise it will decrease. Also, $|\alpha_{O_{(i,\cdot)},k}^{(t)}| \ge \mathbb{E}[|e(2\sum_{l\in S_{n,k}^e}(\sigma_S^{(t)})_l^n - 1)\beta_{O_{(i,\cdot)},k}^{(t)}|]$ and $\mathbb{E}[\alpha_{O_{(i,\cdot)},k}^{(t)}] > 0$;*

4. *All the neurons in $\mathop{\mathbb{E}}_{n\in\mathcal{V}_k^e}[\mathcal{U}_{k,n}^e(t) \cap (\mathcal{W}_{k,n}^{-e}(t) - \mathcal{U}_{k,n}^{-e}(t))]$ will ultimately either have its coefficient update stuck due to regularization, or grow into a changing margin into $\mathbb{E}[\mathcal{U}_{k,n}^e(t) - (\mathcal{W}_{k,n}^{-e}(t) - \mathcal{U}_{k,n}^{-e}(t))]$ where*

$$\alpha_{O_{(i,\cdot)},k}^{(t)} \approx \mathbb{E}[(2\sum_{l\in S_{n,k}^e}(\sigma_S^{(t)})_l^n - 1)e\beta_{O_{(i,\cdot)},k}^{(t)}].$$

*Proof.* By Lemma 28, we see that $\forall i \in \mathbb{E}[\mathcal{U}_{k,n}^e(t)]$, $\alpha_{O_{(i,\cdot)},k}^{(t)}$ and $e\beta_{O_{(i,\cdot)},k}^{(t)}$ would be contributed by $\mathcal{V}_k^e$ to increase, and also $\forall i \in \mathbb{E}[\mathcal{W}_{k,n}^e(t) - \mathcal{U}_{k,n}^e(t)]$, $\alpha_{O_{(i,\cdot)},k}^{(t)}$ and $e\beta_{O_{(i,\cdot)},k}^{(t)}$ would be contributed by $\mathcal{V}_k^e$ to decrease. As such, the first and second point hold naturally by definition. The ultimate transformation of $\mathbb{E}[\mathcal{U}_{k,n}^e(t) - (\mathcal{W}_{k,n}^{-e}(t) - \mathcal{U}_{k,n}^{-e}(t))]$ into $\mathop{\mathbb{E}}_{n\in\mathcal{V}_k^e}[\mathcal{U}_{k,n}^e(t) \cap (\mathcal{W}_{k,n}^{-e}(t) - \mathcal{U}_{k,n}^{-e}(t))]$ attributes to the faster changing

speed of $\alpha^{(t)}_{O_{(i,\cdot)},k}$ compared to $e\beta^{(t)}_{O_{(i,\cdot)},k}$ in the neuron sets $\mathbb{E}[\mathcal{U}^e_{k,n}(t) - (\mathcal{W}^{-e}_{k,n}(t) - \mathcal{U}^{-e}_{k,n}(t))]$, whose learning speed ratio is at least $(\|\boldsymbol{c}_k\|/\|\boldsymbol{d}_k\|)^2$. Therefore, the absolute value of $\alpha^{(t)}_{O_{(i,\cdot)},k}$ will surpass that of $e\beta^{(t)}_{O_{(i,\cdot)},k}$, which indicates the neuron would be activated for opposite labels, then the proof is completed. Given that $(\sum_{l\in S^e_{n,k}} (\sigma^{(t)}_S)^n_l - \sum_{l\in S^{-e}_{n,k}} (\sigma^{(t)}_S)^n_l)$ will remain positive, the discussion over $e\beta^{(t)}_{O_{(i,\cdot)},k}$ is simple since it will always grow in $\boldsymbol{r}_i$'s direction, and thus the third and forth point hold.

Considering the growth of $\alpha^{(t)}_{O_{(i,\cdot)},k}$, by $\pi^+_k = \pi^-_k$, $P_{l,2k-1} = P^n_{l,2k} = \dfrac{1}{2}$, we know

$$\mathbb{E}_{n\in\mathcal{V}^e_k}[\sum_{l\in S^e_{n,k}} (\sigma^{(t)}_S)^n_l] = \mathbb{E}_{n\in\mathcal{V}^{-e}_k}[\sum_{l\in S^{-e}_{n,k}} (\sigma^{(t)}_S)^n_l],$$

hence if $i \in \mathbb{E}_{n\in\mathcal{V}^e_k}[\mathcal{U}^e_{k,n}(t) \cap (\mathcal{W}^{-e}_{k,n}(t) - \mathcal{U}^{-e}_{k,n}(t))]$, it indicates that

$$\mathbb{E}[\alpha^{(t)}_{O_{(i,\cdot)},k} \pm (2\sum_{l\in S^e_{n,k}} (\sigma^{(t)}_S)^n_l - 1)e\beta^{(t)}_{O_{(i,\cdot)},k}] \geq 0.$$

We see that for $\mathbb{E}_{n\in\mathcal{V}^e_k}[\mathcal{U}^e_{k,n}(t) \cap (\mathcal{W}^{-e}_{k,n}(t) - \mathcal{U}^{-e}_{k,n}(t))]$, the $\mathbb{E}[\mathcal{V}^e_k]$ will serve to increase the $\alpha^{(t)}_{O_{(i,\cdot)},k}$, but $\mathbb{E}[\mathcal{V}^{-e}_k]$ will serve to decrease the $\alpha^{(t)}_{O_{(i,\cdot)},k}$. The contribution will tend to be positive if

$$\mathbb{E}_{n\in\mathcal{V}^e_k}(\ell'^{(t)}_n (i\in\mathcal{U}^e_{k,n}(t) \cap (\mathcal{W}^{-e}_{k,n}(t) - \mathcal{U}^{-e}_{k,n}(t)))) ] \geq \mathbb{E}_{n\in\mathcal{V}^{-e}_k}(\ell'^{(t)}_n \mathbb{1}(i\in\mathcal{U}^e_{k,n}(t) \cap (\mathcal{W}^{-e}_{k,n}(t) - \mathcal{U}^{-e}_{k,n}(t)))) ].$$

Then, as $\mathbb{E}[(2\sum_{l\in S^e_{n,k}} (\sigma^{(t)}_S)^n_l - 1)e\beta^{(t)}_{O_{(i,\cdot)},k}]$ of the neurons in and $\mathbb{E}_{n\in\mathcal{V}^e_k}[\mathcal{U}^e_{k,n}(t) \cap (\mathcal{W}^{-e}_{k,n}(t) - \mathcal{U}^{-e}_{k,n}(t))]$ will continue to grow, and finally it will be comparable to the $\mathbb{E}[\alpha^{(t)}_{O_{(i,\cdot)},k}]$. Otherwise it will continue to grow while the evolving speed of $\mathbb{E}[\alpha^{(t)}_{O_{(i,\cdot)},k}]$ is comparatably feeble as it receive the contribution oppositely from $\mathbb{E}_{n\in\mathcal{V}^e_k}(\ell'^{(t)}_n \mathbb{1}(i\in\mathbb{E}[\mathcal{W}^e_{k,n}(t) - \mathcal{U}^e_{k,n}(t) \cap \mathcal{U}^{-e}_{k,n}(t)))) ]$ and $\mathbb{E}_{n\in\mathcal{V}^{-e}_k}(\ell'^{(t)}_n \mathbb{1}(i\in\mathbb{E}[\mathcal{W}^e_{k,n}(t) - \mathcal{U}^e_{k,n}(t) \cap \mathcal{U}^{-e}_{k,n}(t)))) ]$. Quantatively this is validated by our later results in Lemma 32 where the $\mathbb{E}_{n\in\mathcal{V}^e_k}(\ell'^{(t)}_n) - \mathbb{E}_{n\in\mathcal{V}^{-e}_k}(\ell'^{(t)}_n)$ would be controlled by the initialization. Interestingly, we see that as $\mathbb{E}[(2\sum_{l\in S^e_{n,k}} (\sigma^{(t)}_S)^n_l - 1)e\beta^{(t)}_{O_{(i,\cdot)},k}]$ grows up, its scale will surpass those of $\mathbb{E}[\alpha^{(t)}_{O_{(i,\cdot)},k}]$. Under this scenario, $\mathbb{E}_{n\in\mathcal{V}^e_k}[\mathcal{U}^e_{k,n}(t) \cap (\mathcal{W}^{-e}_{k,n}(t) - \mathcal{U}^{-e}_{k,n}(t))]$ will turn into $\mathbb{E}[\mathcal{U}^e_{k,n}(t) - (\mathcal{W}^{-e}_{k,n}(t) - \mathcal{U}^{-e}_{k,n}(t))]$, where $\mathbb{E}[\alpha^{(t)}_{O_{(i,\cdot)},k}]$ again continues to grow. Thus finally we have

$$\alpha^{(t)}_{O_{(i,\cdot)},k} \approx \mathbb{E}[(2\sum_{l\in S^e_{n,k}} (\sigma^{(t)}_S)^n_l - 1)e\beta^{(t)}_{O_{(i,\cdot)},k}].$$

Lemma 4 will show that the growing of $\mathbb{E}[(2\sum_{l\in S^e_{n,k}} (\sigma^{(t)}_S)^n_l - 1)e\beta^{(t)}_{O_{(i,\cdot)},k}]$ will stuck, and thus the growing of $\mathbb{E}[\alpha^{(t)}_{O_{(i,\cdot)},k}]$ will also stuck at the changing margin from $\mathbb{E}_{n\in\mathcal{V}^e_k}[\mathcal{U}^e_{k,n}(t) \cap (\mathcal{W}^{-e}_{k,n}(t) - \mathcal{U}^{-e}_{k,n}(t))]$ into $\mathbb{E}[\mathcal{U}^e_{k,n}(t) - (\mathcal{W}^{-e}_{k,n}(t) - \mathcal{U}^{-e}_{k,n}(t))]$.

The proof is completed. $\square$

*Proof. Proof of Lemma 2.* To examine the 0-1 loss, by definition, we know

$$L_{\mathcal{D}^*}^{0-1}(\mathbb{E}(\Psi^t)) = \mathbb{P}_{S_n \sim \mathcal{D}^*}(y_{S_n} \cdot f(\mathbf{E}(S_n), \mathbb{E}(\Psi^t)) \leq 0),$$

$$= \mathbb{P}_{S_n \sim \mathcal{D}_S}(y_{S_n} \cdot \sum_{e \in [\pm]} \frac{e}{m} \sum_{i \in \{\mathbf{r}_i = \frac{e}{m}\}} \mathbb{E}_{\Psi^{(t)}}[\sigma_R(\mathbf{W}_{O_{(i,\cdot)}}^{\boldsymbol{y}}{}^{(t)} \sum_{l \in [L]} (\sigma_S^{(t)})_l^n \boldsymbol{y}_l^n)] \leq 0),$$

$$= \mathbb{P}(\mathbb{E}[y_{S_n} \cdot \left( \sum_{e \in [\pm]} \frac{e}{m} \sum_{i \in \{\mathbf{r}_i = \frac{e}{m}\}} \sigma_R(\mathbf{W}_{O_{(i,\cdot)}}^{\boldsymbol{y}}{}^{(t)} \sum_{l \in [L]} (\sigma_S^{(t)})_l^n \boldsymbol{y}_l^n) \right)] \leq 0),$$

$$= \mathbb{P}(\mathbb{E}\Big[ \sum_{i \in \{\mathbf{r}_i = \frac{y_{S_n}}{m}\}} \sigma_R \left( \alpha_{O_{(i,\cdot)},k}^{(t)} + (2 \sum_{l \in S_{n,k}^{y_{S_n}}} (\sigma_S^{(t)})_l^n - 1) y_{S_n} \beta_{O_{(i,\cdot)},k}^{(t)} \right)$$

$$- \sum_{i \in \{\mathbf{r}_i = -\frac{y_{S_n}}{m}\}} \sigma_R \left( \alpha_{O_{(i,\cdot)},k}^{(t)} + (2 \sum_{l \in S_{n,k}^{y_{S_n}}} (\sigma_S^{(t)})_l^n - 1) y_{S_n} \beta_{O_{(i,\cdot)},k}^{(t)} \right) \Big] \leq 0)$$

$$= \mathbb{P}(\mathbb{E}\Big[ ( \sum_{i \in \mathcal{U}_{k,n}^{y_{S_n}}(t)} - \sum_{i \in \mathcal{W}_{k,n}^{y_{S_n}}(t) - \mathcal{U}_{k,n}^{y_{S_n}}(t)} )(\alpha_{O_{(i,\cdot)},k}^{(t)} + (2 \sum_{l \in S_{n,k}^{y_{S_n}}} (\sigma_S^{(t)})_l^n - 1) y_{S_n} \beta_{O_{(i,\cdot)},k}^{(t)}) \Big] \leq 0).$$

Therefore, a sufficient condition for $L_{\mathcal{D}^*}^{0-1}(\mathbb{E}(\Psi^t)) = 0$ is

$$\mathbb{E}[\sum_{i \in \mathcal{U}_{k,n}^e(t)} \alpha_{O_{(i,\cdot)},k}^{(t)} + (2 \sum_{l \in S_{n,k}^e} (\sigma_S^{(t)})_l^n - 1)e \cdot \beta_{O_{(i,\cdot)},k}^{(t)}] \geq \mathbb{E}[ \sum_{i \in \mathcal{W}_{k,n}^e(t) - \mathcal{U}_{k,n}^e(t)} \alpha_{O_{(i,\cdot)},k}^{(t)}$$

$$+ (2 \sum_{l \in S_{n,k}^e} (\sigma_S^{(t)})_l^n - 1)e \cdot \beta_{O_{(i,\cdot)},k}^{(t)}], \tag{43}$$

for $\forall k \in [K_1], e \in [\pm]$. $\qquad\square$

We know $\forall i \in \mathcal{U}_{k,n}^e(t)$, $\mathbb{E}[e \cdot \beta_{O_{(i,\cdot)},k}^{(t)}]$ in the left side of the inequality is increasing, and $\forall i \in \mathbb{E}[\mathcal{W}_{k,n}^e(t) - \mathcal{U}_{k,n}^e(t)]$, the $\mathbb{E}[e \cdot \beta_{O_{(i,\cdot)},k}^{(t)}]$ in the right side of the inequality is decreasing, which is a good news since we want the left side exceed the right side. By Lemma 29, we see that all the neurons in $\mathbb{E}[(\mathcal{W}_{k,n}^e(t) - \mathcal{U}_{k,n}^e(t)) - \mathcal{U}_{k,n}^{-e}(t)]$ will be deactivated, and all the neurons in $\mathbb{E}[\mathcal{U}_{k,n}^e(t) - (\mathcal{W}_{k,n}^{-e}(t) - \mathcal{U}_{k,n}^{-e}(t))]$ will turn into $\mathbb{E}_{n \in \mathcal{V}_k^e}[\mathcal{U}_{k,n}^e(t) \cap (\mathcal{W}_{k,n}^{-e}(t) - \mathcal{U}_{k,n}^{-e}(t))]$.

## I.1 First Stage: Growing of Coefficient

In this stage, the coefficient update dynamic is continually changing without being much influenced by the comparably feeble regularization. Also, the impact of the decaying learning step $\eta_t$ is under controlled during several periods, which can be safely done due to small initialization by a large $\gamma$, as well as the slow quadratic decaying nature of the derivative of $\eta_t'$. We see that at initialization, by Lemma 7 and Lemma 23, the $\mathbb{E}_{S_n \sim \mathcal{D}_S}[f(\mathbf{E}(S_n); \Psi^{(0)})]$ satisfies

$$\mathbb{E}_{S_n \sim \mathcal{D}_S}\Big[ \sum_{i \in \mathcal{W}_{k,n}^{y_{S_n}}(0)} \mathbf{r}_i \left( \alpha_{O_{(i,\cdot)},k}^{(0)} + (2 \sum_{l \in S_{n,k}^{y_{S_n}}} (\sigma_S^{(0)})_l^n - 1) y_{S_n} \beta_{O_{(i,\cdot)},k}^{(0)} \right) \Big] \geq - \sqrt{2 \log(\frac{5Km}{\delta})} \cdot$$

$$\frac{5\sigma_1(\|\boldsymbol{c}_k\| + \zeta_k^e \|\boldsymbol{d}_k\|)}{16}, \tag{44}$$

and our remaining job is to see when will $\mathbb{E}_{S_n \sim \mathcal{D}_S}[f(\mathbf{E}(S_n); \mathbb{E}(\Psi^{(t)}))]$ stay positive for some error tolerance. As such, we need to scrutinize the coefficients that would grow along the iterations. Therefore, we define

$$\mathbf{A}_t^{k,y_{S_n}} := \frac{1}{m}\Big[ \Big( \sum_{i \in \mathcal{U}_{k,n}^{y_{S_n}}(\tau)} - \sum_{i \in (\mathcal{W}_{k,n}^{y_{S_n}}(\tau) - \mathcal{U}_{k,n}^{y_{S_n}}(\tau)) \cap \mathcal{U}_{k,n}^{-y_{S_n}}(\tau)} \Big) \mathbf{W}_{O_{(i,\cdot)}}^{\boldsymbol{y}}{}^{(\tau)} \boldsymbol{c}_k$$

$$+ \Big( \sum_{i \in \mathcal{U}_{k,n}^{y_{S_n}}(\tau)} - \sum_{i \in (\mathcal{W}_{k,n}^{y_{S_n}}(\tau) - \mathcal{U}_{k,n}^{y_{S_n}}(\tau)) \cap \mathcal{U}_{k,n}^{-y_{S_n}}(\tau)} \Big) (2 \sum_{l \in S_{n,k}^{y_{S_n}}} (\sigma_S^{(\tau)})_l^n - 1) y_{S_n} \mathbf{W}_{O_{(i,\cdot)}}^{\boldsymbol{y}}{}^{(\tau)} \boldsymbol{d}_k \Big]\Big|_{\tau=t}^{\tau=0}.$$

We will see that the conditional expectation of this sequence (conditioned on $\mathbb{E}(\Psi'^{(t)})$, and the expectation is taken over $\mathcal{D}_S$) would grow up to conquer the small initialization and make $\underset{S_n \sim \mathcal{D}_S}{\mathbb{E}}[f(\mathbf{E}(S_n); \mathbb{E}(\Psi'^{(t)}))]$ stay positive. Consider the whole training duration $0 \leq t \leq T^*$, the evolving speed of $\beta_{Q,k}^{(t+1)}, \beta_{K,k}^{(t+1)}, \alpha_{O_{(i,\cdot)},k}^{(t+1)}$ and $\beta_{O_{(i,\cdot)},k}^{(t+1)}$ depends on $\mathbb{E}[\ell_n'^{(t)}], \mathbb{E}[\mathbb{1}_{O_{(i)}}^{n}{}^{(t)}]$ and $\mathbb{E}[\sum_{l \in S_{n,k}^e} (\sigma_S^{(t)})_l^n]$. Denote

$$\sigma_S^* := \cfrac{1}{1 + e^{-2^{-1}\sigma_0{}^2(1-\kappa_{\boldsymbol{x}})^2 \|\mathbf{u}\|^4 e^{-2\log(5Km/\delta)\frac{\sigma_1{}^2\|\mathbf{u}\|^4(1+e^{-\sigma_0^2\|\mathbf{u}\|^2})}{(1-e^{-\sigma_0^2\|\mathbf{u}\|^2})}}}},$$

$$\alpha := 4\log(T^*),$$

$$\kappa := 8\max_{i,k,w}\{|\alpha_{O_{(i,\cdot)},k}^{(0)}|, |\beta_{O_{(i,\cdot)},k}^{(0)}|\},$$

We will show that $\sigma_S^*$ is the lower bound of $\min_{t \in [T^*], k \in [K_1]}\{\underset{n \in \mathcal{D}_S}{\mathbb{E}}[\sum_{j \in S_{n,k}^{y_{S_n}}} (\sigma_S^{(t)})_j^n]\}$ along the whole iteration. By Lemma 7, $\kappa$ can be upper bounded by $8\sqrt{2\log(5Km/\delta)} \cdot \sigma_1(\sqrt{(1+\kappa_{\boldsymbol{y}})/2}\|\mathbf{q}\|)$, and lower bounded by $2\sqrt{2}\sigma_1\|\mathbf{q}\|$, which is a negligible term due to the small initialization by Condition 1.

**Lemma 30.** *Under Condition 1, for the whole iteration $0 \leq t \leq T^*$, for $\forall i \in [m], e \in [\pm], k \in [K_1], r \in [K_2], w \in [d_{\mathcal{X}} - K]$, we have that*

$$0 \leq \mathbb{E}[e \cdot \beta_{O_{(i,\cdot)},k}^{(t)}\mathbb{1}(i \in \mathcal{U}_{k,n}^e(t))] - e \cdot \beta_{O_{(i,\cdot)},k}^{(0)} \leq \sigma_S^{*-1}\alpha,$$

$$0 \geq \mathbb{E}[e \cdot \beta_{O_{(i,\cdot)},k}^{(t)}\mathbb{1}(i \in \mathcal{W}_{k,n}^e(t) - \mathcal{U}_{k,n}^e(t))] - e \cdot \beta_{O_{(i,\cdot)},k}^{(0)} \geq -\frac{\hat{C}\|\boldsymbol{c}_k\|^2}{\sigma_S^{*2}\|\boldsymbol{d}_k\|^2}\alpha$$

$$- \frac{\sigma_1(\sigma_S^{*2}\|\boldsymbol{d}_k\|^2 + \hat{C}\|\boldsymbol{c}_k\|^2)\sqrt{2\log(\frac{5Km}{\delta})}}{\sigma_S^{*2}\|\boldsymbol{d}_k\|},$$

$$0 \leq \mathbb{E}[|\alpha_{O_{(i,\cdot)},k}^{(t)}|] \leq \hat{C}\frac{\|\boldsymbol{c}_k\|^2}{\sigma_S^{*2}\|\boldsymbol{d}_k\|^2}\alpha,$$

(45)

**Lemma 31.** *Suppose Eq. (45) holds at iteration $t \leq T_2$, then we have*

$$\left|\underset{n \in \mathcal{V}_k}{\mathbb{E}}[y_{S_n}f(\mathbf{E}(S); \mathbb{E}(\Psi^{(t)}))] - \mathbb{E}[\mathbf{A}_{t+1}^{k,y_{S_n}}]\right| \leq \kappa/2.$$

*Proof.* By definition, we have

$$\mathbb{E}[y_{S_n}f(\mathbf{E}(S); \Psi^{(t)})] = \mathbb{E}[y_{S_n} \cdot \sum_{e \in [\pm]} \frac{e}{m} \sum_{i \in \{\mathbf{r}_i = \frac{e}{m}\}} \sigma_R(\mathbf{W}_{O_{(i,\cdot)}}^{\boldsymbol{y}}{}^{(t)} \sum_{l \in [L]} (\sigma_S^{(t)})_l^n \boldsymbol{y}_l^n)]$$

$$= \mathbb{E}\Big[\frac{1}{m}\Big(\sum_{i \in \mathcal{U}_{k,n}^{y_{S_n}}(t)} - \sum_{i \in \mathcal{W}_{k,n}^{y_{S_n}}(t) - \mathcal{U}_{k,n}^{y_{S_n}}(t)}\Big)\Big(\alpha_{O_{(i,\cdot)},k}^{(t)} + (2\sum_{l \in S_{n,k}^{y_{S_n}}} (\sigma_S^{(t)})_l^n - 1)y_{S_n}\beta_{O_{(i,\cdot)},k}^{(t)}\Big)\Big].$$

Observe that

$$\mathbb{E}\Big[\frac{1}{m}\sum_{i \in (\mathcal{W}_{k,n}^e(t) - \mathcal{U}_{k,n}^e(t))}\Big(\alpha_{O_{(i,\cdot)},k}^{(t)} + (2\sum_{l \in S_{n,k}^{y_{S_n}}} (\sigma_S^{(t)})_l^n - 1)y_{S_n}\beta_{O_{(i,\cdot)},k}^{(t)}\Big)\Big] - \frac{1}{m}\mathbb{E}\Big[$$

$$\sum_{i \in (\mathcal{W}_{k,n}^e(\tau) - \mathcal{U}_{k,n}^e(\tau)) \cap \mathcal{U}_{k,n}^{-e}(\tau)}\Big(\alpha_{O_{(i,\cdot)},k}^{(\tau)} + (2\sum_{l \in S_{n,k}^{y_{S_n}}} (\sigma_S^{(\tau)})_l^n - 1)y_{S_n}\beta_{O_{(i,\cdot)},k}^{(\tau)}\Big)\Big]\Big|_{\tau=t}^{\tau=0} \leq \kappa/4.$$

Here the inequality holds due to the fact that $\mathbb{E}[\alpha_{O_{(i,\cdot)},k}^{(t)}\mathbb{1}(i \in (\mathcal{W}_{k,n}^e(t) - \mathcal{U}_{k,n}^e(t)) - \mathcal{U}_{k,n}^{-e}(t))]$ is decreasing the initial value $\alpha_{O_{(i,\cdot)},k}^{(0)}\mathbb{1}(i \in (\mathcal{W}_{k,n}^e(0) - \mathcal{U}_{k,n}^e(0)) - \mathcal{U}_{k,n}^{-e}(0))$, and it's absolute value will not surpass that of $\mathbb{E}[e(2\sum_{l \in S_{n,k}^{y_{S_n}}} (\sigma_S^{(t)})_l^n - 1)\beta_{O_{(i,\cdot)},k}^{(t)}] \leq \kappa/8$, which is positive (by definition) and also decreasing by Lemma

29. On the other hand,

$$
\left| \mathbb{E}\Big[ \frac{1}{m} \sum_{i \in \mathcal{U}_{k,n}^{y_{S_n}}(t)} \Big( \alpha_{O_{(i,\cdot)},k}^{(t)} + (2 \sum_{l \in S_{n,k}^{y_{S_n}}} (\sigma_S^{(t)})_l^n - 1) y_{S_n} \beta_{O_{(i,\cdot)},k}^{(t)} \Big) \Big] - \mathbb{E}\Big[ \frac{1}{m} \sum_{i \in \mathcal{U}_{k,n}^{y_{S_n}}(\tau)} \alpha_{O_{(i,\cdot)},k}^{(\tau)} \right.
$$
$$
\left. - \frac{1}{m} \sum_{i \in \mathcal{U}_{k,n}^{y_{S_n}}(\tau)} (2 \sum_{l \in S_{n,k}^{y_{S_n}}} (\sigma_S^{(\tau)})_l^n - 1) y_{S_n} \beta_{O_{(i,\cdot)},k}^{(\tau)}] \Big|_{\tau=t}^{\tau=0} \right| \leq \kappa/4.
$$

Combining the two we can see the result is obtained. □

We then denote the last time when there still exists $\mathbb{E}[\mathbf{A}_t^{k,e}] \leq \kappa$ as $\hat{T}$, formally $\hat{T}$ is the last time where

$$
\bigcup_{k \in [K_1], e \in [\pm]} \{\mathbb{E}[\mathbf{A}_t^{k,e}] \leq \kappa\} \neq \emptyset.
$$

Latter we will show in Lemma 33 that

$$
\hat{T} = \frac{C_1 \sigma_1 m \lambda K_1 \gamma \sqrt{(1 + \kappa_{\boldsymbol{y}}) \log(5Km/\delta)}}{(2\sigma_S^* - 1)^2 (1 - \kappa_{\boldsymbol{y}}) \|\mathbf{q}\|}.
$$

We then denote the learning step at $\hat{T}$ as $\eta := \eta_{\hat{T}}$, and thus

$$
\eta = \eta_{\hat{T}} = \frac{2}{\lambda(\hat{T} + \gamma)}.
$$

By Lemma 31, actually it would hold that

$$
\mathbb{E}_{S_n \sim \mathcal{D}_S}[f(\mathbf{E}(S_n); \mathbb{E}(\Psi^{(\hat{T})}))] \geq \kappa/2 \geq 0.
$$

And thus the 0-1 loss converges to zero with an error tolerance by definition. Our following job is to find $\hat{T}$. The following lemma provides the continuous ODEs as the upper and lower bound of the sequence $\mathbf{A}_t^{k,e}$.

**Lemma 32.** *Under Condition 1, suppose Eq.(45) holds at any iteration $t \leq T^*$, then for $\forall t \leq T^*, \forall k \in [K_1], e \in [\pm]$, it holds that*

1. *The difference* $|\mathop{\mathbb{E}}_{n \in \mathcal{V}_k^e}(ef(\mathbf{E}(S); \mathbb{E}(\Psi^{(t)}))) - \mathop{\mathbb{E}}_{n \in \mathcal{V}_k^{-e}}(-ef(\mathbf{E}(S); \mathbb{E}(\Psi^{(t)})))|$ *is none-increasing.*

2. *The difference of the loss derivative is bounded by $O(\kappa)$:*

$$
|\mathbb{E}[\mathop{\mathbb{E}}_{n \in \mathcal{V}_k^e}(\ell_n'^{(t)}) - \mathop{\mathbb{E}}_{n \in \mathcal{V}_k^{-e}}(\ell_n'^{(t)})]| \leq \frac{\kappa}{8}.
$$

3. $\mathbb{E}[\mathbf{A}_t^{k,e}]$ *is non-decreasing. The lower and upper bounds of the gradient update have continuous ODE counterpart. Specifically, there exist positive constant $c_1, c_2$, we can define $\overline{c}^{k,e} = \frac{c_1 \eta_0 \|\mathbf{q}\|^2}{2mK_1}$, $\underline{c}^{k,e} = \frac{c_2 \eta_{T^*} (2\sigma_S^* - 1)^2 (1 - \kappa_{\boldsymbol{y}}) \|\mathbf{q}\|^2}{16mK_1}$, $\overline{b}^{k,e} = e^{-\kappa/2}$, $\underline{b}^{k,e} = e^{\kappa/2}$. Let $\overline{x}_t^{k,e}, \underline{x}_t^{k,e}$ be the unique solutions of*

$$
\overline{x}_t^{k,e} + \overline{b}^{k,e} e^{\overline{x}_t^{k,e}} = \overline{c}^{k,e} t + \overline{b}^{k,e}, \quad \underline{x}_t^{k,e} + \underline{b}^{k,e} e^{\underline{x}_t^{k,e}} = \underline{c}^{k,e} t + \underline{b}^{k,e},
$$

*then it holds that*

$$
\underline{x}_t^{k,e} \leq \mathbb{E}[\mathbf{A}_t^{k,e}] \leq \overline{x}_t^{k,e} + \frac{\overline{c}^{k,e}}{1 + \overline{b}^{k,e}}, \quad \frac{1}{1 + \overline{b}^{k,e} \overline{x}_t^{k,e}} \leq - \mathop{\mathbb{E}}_{n \in \mathcal{V}_k^e}(\ell_n'^{(t)}) \leq \frac{1}{1 + \underline{b}^{k,e} \underline{x}_t^{k,e}}.
$$

*Specifically, we have*

$$
\log(\frac{2\underline{c}^{k,e}}{3\underline{b}^{k,e}} + \frac{2}{3}) \leq \mathbb{E}[\mathbf{A}_t^{k,e}] \leq \log(\frac{\overline{c}^{k,e}}{\overline{b}^{k,e}} t + 1) + \frac{\overline{c}^{k,e}}{1 + \overline{b}^{k,e}}.
$$

*Proof.* Observe that $\mathbb{E}[\underset{n\in\mathcal{V}_k^e}{\mathbb{E}}(\ell_n'{}^{(t)}) - \underset{n\in\mathcal{V}_k^{-e}}{\mathbb{E}}(\ell_n'{}^{(t)})]$ equals to

$$\mathbb{E}[\frac{-1}{1 + e^{[-\frac{1}{m}(\sum_{i\in\mathcal{U}_{k,n}^e}(t) - \sum_{i\in\mathcal{W}_{k,n}^e}(t) - \mathcal{U}_{k,n}^e(t))\left(\alpha_{O_{(i,\cdot)},k}^{(t)} + (2\sum_{l\in S_{n,k}^e}(\sigma_S^{(t)})_l^n - 1)e\beta_{O_{(i,\cdot)},k}^{(t)}\right)]}}$$

$$- \frac{-1}{1 + e^{[-\frac{1}{m}(\sum_{i\in\mathcal{U}_{k,n}^{-e}}(t) - \sum_{i\in\mathcal{W}_{k,n}^{-e}}(t) - \mathcal{U}_{k,n}^{-e}(t))\left(\alpha_{O_{(i,\cdot)},k}^{(t)} + (2\sum_{l\in S_{n,k}^e}(\sigma_S^{(t)})_l^n - 1)e\beta_{O_{(i,\cdot)},k}^{(t)}\right)]}}]$$

$$= \mathbb{E}[\frac{e^{-[\frac{1}{m}(\sum_{i\in\mathcal{U}_{k,n}^y}(t) - \sum_{i\in\mathcal{W}_{k,n}^y}(t) - \mathcal{U}_{k,n}^y(t))\left(\alpha_{O_{(i,\cdot)},k}^{(t)} + (2\sum_{l\in S_{n,k}^y}(\sigma_S^{(t)})_l^n - 1)y\beta_{O_{(i,\cdot)},k}^{(t)}\right)]}\Big|_{y=e}^{y=-e}}{\prod_{y\in\{e,-e\}} 1 + e^{[-\frac{1}{m}(\sum_{i\in\mathcal{U}_{k,n}^y}(t) - \sum_{i\in\mathcal{W}_{k,n}^y}(t) - \mathcal{U}_{k,n}^y(t))\left(\alpha_{O_{(i,\cdot)},k}^{(t)} + (2\sum_{l\in S_{n,k}^y}(\sigma_S^{(t)})_l^n - 1)y\beta_{O_{(i,\cdot)},k}^{(t)}\right)]}}]$$

(46)

As cross-entropy loss is $L$-smooth with $L = 1$, one can bound the difference by

$$|\mathbb{E}[\underset{n\in\mathcal{V}_k^e}{\mathbb{E}}(\ell_n'{}^{(t)}) - \underset{n\in\mathcal{V}_k^{-e}}{\mathbb{E}}(\ell_n'{}^{(t)})]| \le |\underset{n\in\mathcal{V}_k^e}{\mathbb{E}}(ef(\mathbf{E}(S);\mathbb{E}(\Psi^{(t)})) - \underset{n\in\mathcal{V}_k^{-e}}{\mathbb{E}}(-ef(\mathbf{E}(S);\mathbb{E}(\Psi^{(t)})))|$$

(47)

$$= |\mathbb{E}[[\frac{1}{m}(\sum_{i\in\mathcal{U}_{k,n}^y(t)} - \sum_{i\in\mathcal{W}_{k,n}^y(t) - \mathcal{U}_{k,n}^y(t)})\left(\alpha_{O_{(i,\cdot)},k}^{(t)} + (2\sum_{l\in S_{n,k}^y}(\sigma_S^{(t)})_l^n - 1)y\beta_{O_{(i,\cdot)},k}^{(t)}\right)]\Big|_{y=e}^{y=-e}]|.$$

By Lemma 23, we see that for initialization, we have

$$|\underset{n\in\mathcal{V}_k^e}{\mathbb{E}}(ef(\mathbf{E}(S);\mathbb{E}(\Psi^{(0)})) - \underset{n\in\mathcal{V}_k^{-e}}{\mathbb{E}}(-ef(\mathbf{E}(S);\mathbb{E}(\Psi^{(0)})))| \le 2\sqrt{2\log(\frac{5Km}{\delta})} \cdot \frac{3\sigma_1(\|\mathbf{c}_k\| + \zeta_k^e\|\mathbf{d}_k\|)}{8}$$

$$\le \kappa/8.$$

Now we serve to show that the following expected difference

$$|\underset{n\in\mathcal{V}_k^e}{\mathbb{E}}(ef(\mathbf{E}(S);\mathbb{E}(\Psi^{(t)})) - \underset{n\in\mathcal{V}_k^{-e}}{\mathbb{E}}(-ef(\mathbf{E}(S);\mathbb{E}(\Psi^{(t)})))|$$

is non-increasing. Intuitively, this observation is due to the inherent nature of cross-entropy loss, which always pays more emphasis (has larger derivative) on those low value. Also, another important factor is the update of those ambiguous neurons' coefficient summation would also prefer the low-value one among $\underset{n\in\mathcal{V}_k^e}{\mathbb{E}}(ef(\mathbf{E}(S);\mathbb{E}(\Psi^{(t)}))), \forall e \in [m]$. To better present this observation, we define

$$e_t^* = \arg\min\{\underset{n\in\mathcal{V}_k^e}{\mathbb{E}}(ef(\mathbf{E}(S);\mathbb{E}(\Psi^{(t)}))), \underset{n\in\mathcal{V}_k^{-e}}{\mathbb{E}}(-ef(\mathbf{E}(S);\mathbb{E}(\Psi^{(t)})))\},$$

which further means that $e_t^*$ satisfies $\mathbb{E}[\underset{n\in\mathcal{V}_k^{e_t^*}}{\mathbb{E}}(\ell_n'{}^{(t)}) - \underset{n\in\mathcal{V}_k^{-e_t^*}}{\mathbb{E}}(\ell_n'{}^{(t)})] < 0$ due to the non-positive and non-increasing property of cross-entropy loss.

Recall the update rule, we have

$$\alpha_{O_{(i,\cdot)},k}^{(t+1)} = (1 - \eta_t\lambda)\alpha_{O_{(i,\cdot)},k}^{(t)} - \eta_t\frac{\|\mathbf{c}_k\|^2}{2K_1}\sum_{e\in[\pm]}[e\mathbf{r}_i \cdot \underset{n\in\mathcal{V}_k^e}{\mathbb{E}}(\ell_n'{}^{(t)}\mathbb{1}_{O_{(i)}}^n{}^{(t)})]$$

$$- \eta_t\frac{(K_1-1)\|\mathbf{c}_k\|^2}{2K_1K}\sum_{e\in[\pm]}[e\mathbf{r}_i \cdot \underset{n\in\mathcal{V}_{\neg k}^e}{\mathbb{E}}(\ell_n'{}^{(t)}\mathbb{1}_{O_{(i)}}^n{}^{(t)})],$$

$$\mathbb{E}[e\beta_{O_{(i,\cdot)},k}^{(t+1)} \mid \Psi^{(t)}] = (1 - \eta_t\lambda)e\beta_{O_{(i,\cdot)},k}^{(t)}$$

$$- \eta_t\frac{\|\mathbf{d}_k\|^2}{2K_1}\sum_{e\in[\pm]}\mathbf{r}_ie\underset{n\in\mathcal{V}_k^e}{\mathbb{E}}[\ell_n'{}^{(t)}\mathbb{1}_{O_{(i)}}^n{}^{(t)}(\sum_{l\in S_{n,k}^e}(\sigma_S^{(t)})_l^n - \sum_{l\in S_{n,k}^{-e}}(\sigma_S^{(t)})_l^n)].$$

Then we have

$$\mathbb{E}[\alpha_{O_{(i,\cdot)},k}^{(t+1)} + e(2\sum_{l\in S_{n,k}^e}(\sigma_S^{(t+1)})_l^n - 1)\beta_{O_{(i,\cdot)},k}^{(t+1)} \mid \Psi^{(t)}, \mathbb{E}[\sum_{l\in S_{n,k}^e}(\sigma_S^{(t+1)})_l^n]] = (1 - \eta_t\lambda)(\alpha_{O_{(i,\cdot)},k}^{(t)} + e(2$$

$$\sum_{l\in S_{n,k}^e}(\sigma_S^{(t)})_l^n - 1)\beta_{O_{(i,\cdot)},k}^{(t)}) - \frac{\eta_t}{2K_1}\sum_{e\in[\pm]}\mathbf{r}_ie\underset{n\in\mathcal{V}_k^e}{\mathbb{E}}[\ell_n'{}^{(t)}\mathbb{1}_{O_{(i)}}^n{}^{(t)}\left(\|\mathbf{c}_k\|^2 + \|\mathbf{d}_k\|^2(2\sum_{l\in S_{n,k}^e}(\sigma_S^{(t+1)})_l^n - 1)\right)$$

$$(2\sum_{l\in S_{n,k}^e}(\sigma_S^{(t)})_l^n - 1)) - \eta_t\frac{(K_1-1)}{2K_1K}\sum_{e\in[\pm]}\mathbf{r}_ie\underset{n\in\mathcal{V}_{\neg k}^e}{\mathbb{E}}[\ell_n'{}^{(t)}\mathbb{1}_{O_{(i)}}^n{}^{(t)}\|\mathbf{c}_k\|^2].$$

By Lemma 28 and Lemma 29, we see that the $\mathbb{E}[\sum_{i\in\mathcal{U}^e_{k,n}(\tau)-(\mathcal{W}^{-e}_{k,n}(\tau)-\mathcal{U}^{-e}_{k,n}(\tau))}\alpha^{(\tau)}_{O_{(i,\cdot),k}}]\Big|^{\tau=0}_{\tau=t},\forall e\in[\pm]$ is increasing such that

$$\mathbb{E}[\sum_{i\in\mathcal{U}^e_{k,n}(\tau)-(\mathcal{W}^{-e}_{k,n}(t)-\mathcal{U}^{-e}_{k,n}(\tau))}\alpha^{(\tau)}_{O_{(i,\cdot),k}}\mid\Psi^{(t)}]\Big|^{\tau=0}_{\tau=t+1}=\Theta(\sum_{i\in\mathcal{U}^e_{k,n}(\tau)-(\mathcal{W}^{-e}_{k,n}(t)-\mathcal{U}^{-e}_{k,n}(t))}\alpha^{(\tau)}_{O_{(i,\cdot),k}}\Big|^{\tau=0}_{\tau=t}$$
$$-\sum_{i\in\mathcal{U}^e_{k,n}(t)-(\mathcal{W}^{-e}_{k,n}(t)-\mathcal{U}^{-e}_{k,n}(t))}\frac{\eta_t\|\boldsymbol{c}_k\|^2}{2mK_1}\mathbb{E}_{n\in\mathcal{V}^e_k}(\ell'_n{}^{(t)})),$$
(48)

where we ignore the impact of cross-concept safely due to the large $K=\Omega(\eta_0 C(K_1-1)\|\mathbf{q}\|^2/(mK_1))$, as well as the impact of regularization term since $\lambda=O((C\log(Km/\delta)\|\mathbf{q}\|)^{-1})$ by Condition 1 in the first stage.

Similarly, suggest $\mathbb{E}[\sum_{l\in S^e_{n,k}}(\sigma^{(t+1)}_S)^n_l]$ is also given when considering the update for $t+1$, we see that

$$\mathbb{E}[\frac{1}{m}\sum_{i\in\mathcal{U}^e_{k,n}(\tau)}e(2\sum_{l\in S^e_{n,k}}(\sigma^{(t+1)}_S)^n_l-1)\beta^{(\tau)}_{O_{(i,\cdot),k}}\mid\Psi^{(t)},\mathbb{E}[\sum_{l\in S^e_{n,k}}(\sigma^{(t+1)}_S)^n_l]]\Big|^{\tau=0}_{\tau=t+1}=(2\sum_{l\in S^e_{n,k}}(\sigma^{(t+1)}_S)^n_l-1)$$
$$\Theta(\frac{1}{m}\sum_{i\in\mathcal{U}^e_{k,n}(\tau)}e\beta^{(\tau)}_{O_{(i,\cdot),k}}\Big|^{\tau=0}_{\tau=t}-\sum_{i\in\mathcal{U}^e_{k,n}(\tau)}\frac{\eta_t\|\boldsymbol{d}_k\|^2}{2mK_1}\mathbb{E}_{n\in\mathcal{V}^e_k}((2\sum_{l\in S^e_{n,k}}(\sigma^{(t)}_S)^n_l-1)\ell'_n{}^{(t)})).$$
(49)

Interestingly, by Eq.(39) we see that $(2\sum_{l\in S^e_{n,k}}(\sigma^{(t)}_S)^n_l-1)=(2\sum_{l\in S^{-e}_{n,k}}(\sigma^{(t)}_S)^n_l-1)$. Thus we can characterize that the magnitude of gradient update of the term in Eq.(48) and (49) of the $e^*_t$ would be larger than those of $-e^*_t$ due to the non-increasing nature of cross-entropy loss.

On the other hand, by Lemma 29 the monotonicity of

$$\mathbb{E}[\sum_{i\in\mathcal{U}^e_{k,n}(\tau)\cap(\mathcal{W}^{-e}_{k,n}(\tau)-\mathcal{U}^{-e}_{k,n}(\tau))}\alpha^{(\tau)}_{O_{(i,\cdot),k}}]\Big|^{\tau=0}_{\tau=t},\quad\mathbb{E}[\sum_{i\in(\mathcal{W}^e_{k,n}(\tau)-\mathcal{U}^e_{k,n}(\tau))\cap\mathcal{U}^{-e}_{k,n}(\tau)}\alpha^{(\tau)}_{O_{(i,\cdot),k}}]\Big|^{\tau=0}_{\tau=t}$$

depend on the signal of $\mathbb{E}[\mathbb{E}_{n\in\mathcal{V}^e_k}(\ell'_n{}^{(t)})-\mathbb{E}_{n\in\mathcal{V}^{-e}_k}(\ell'_n{}^{(t)})]$. Specifically, we see that

$$\mathbb{E}[\sum_{i\in\mathcal{U}^e_{k,n}(\tau)\cap(\mathcal{W}^{-e}_{k,n}(\tau)-\mathcal{U}^{-e}_{k,n}(\tau))}\alpha^{(\tau)}_{O_{(i,\cdot),k}}\mid\Psi^{(t)}]\Big|^{\tau=0}_{\tau=t+1}=\Theta(\sum_{i\in\mathcal{U}^e_{k,n}(\tau)\cap(\mathcal{W}^{-e}_{k,n}(\tau)-\mathcal{U}^{-e}_{k,n}(\tau))}\alpha^{(\tau)}_{O_{(i,\cdot),k}}\Big|^{\tau=0}_{\tau=t}$$
$$-\sum_{i\in\mathcal{U}^e_{k,n}(\tau)\cap(\mathcal{W}^{-e}_{k,n}(\tau)-\mathcal{U}^{-e}_{k,n}(\tau))}\frac{\eta_t\|\boldsymbol{c}_k\|^2}{2mK_1}[\mathbb{E}_{n\in\mathcal{V}^e_k}(\ell'_n{}^{(t)})-\mathbb{E}_{n\in\mathcal{V}^{-e}_k}(\ell'_n{}^{(t)})]);$$
$$\mathbb{E}[\sum_{i\in(\mathcal{W}^e_{k,n}(\tau)-\mathcal{U}^e_{k,n}(t))\cap\mathcal{U}^{-e}_{k,n}(\tau)}\alpha^{(\tau)}_{O_{(i,\cdot),k}}\mid\Psi^{(t)}]\Big|^{\tau=0}_{\tau=t+1}=\Theta(\sum_{i\in(\mathcal{W}^e_{k,n}(\tau)-\mathcal{U}^e_{k,n}(t))\cap\mathcal{U}^{-e}_{k,n}(\tau)}\alpha^{(t)}_{O_{(i,\cdot),k}}\Big|^{\tau=0}_{\tau=t}$$
$$+\sum_{i\in(\mathcal{W}^e_{k,n}(\tau)-\mathcal{U}^e_{k,n}(t))\cap\mathcal{U}^{-e}_{k,n}(\tau)}\frac{\eta_t\|\boldsymbol{c}_k\|^2}{2mK_1}[\mathbb{E}_{n\in\mathcal{V}^e_k}(\ell'_n{}^{(t)})-\mathbb{E}_{n\in\mathcal{V}^{-e}_k}(\ell'_n{}^{(t)})]);$$
(50)

where the contribution term is shared by the two sequences. Therefore, by Eq.(50) and (46), the evolution of

$$\mathbb{E}[\sum_{i\in\mathcal{U}^e_{k,n}(\tau)\cap(\mathcal{W}^{-e}_{k,n}(\tau)-\mathcal{U}^{-e}_{k,n}(\tau))}\alpha^{(\tau)}_{O_{(i,\cdot),k}}-\sum_{i\in(\mathcal{W}^e_{k,n}(\tau)-\mathcal{U}^e_{k,n}(\tau))\cap\mathcal{U}^{-e}_{k,n}(\tau)}\alpha^{(\tau)}_{O_{(i,\cdot),k}}]\Big|^{\tau=0}_{\tau=t}$$

will prefer to grow in the direction of $e^*_t$.

We then take a look on the decreasing coefficients based on Lemma 29.

$$
\mathbb{E}[\sum_{i\in(\mathcal{W}^e_{k,n}(\tau)-\mathcal{U}^e_{k,n}(\tau))-\mathcal{U}^{-e}_{k,n}(\tau)}\alpha^{(\tau)}_{O_{(i,\cdot)},k}\mid\Psi^{(t)}]\Big|^{\tau=0}_{\tau=t+1}=\Theta(\sum_{i\in(\mathcal{W}^e_{k,n}(\tau)-\mathcal{U}^e_{k,n}(\tau))-\mathcal{U}^{-e}_{k,n}(\tau)}\alpha^{(t)}_{O_{(i,\cdot)},k}\Big|^{\tau=0}_{\tau=t}
$$

$$
+\sum_{i\in(\mathcal{W}^e_{k,n}(t)-\mathcal{U}^e_{k,n}(t))-\mathcal{U}^{-e}_{k,n}(t)}\frac{\eta_t\|\boldsymbol{c}_k\|^2}{2mK_1}\mathbb{E}_{n\in\mathcal{V}^e_k}(\ell'^{(t)}_n)),
$$

$$
\mathbb{E}[\frac{1}{m}\sum_{i\in\mathcal{W}^e_{k,n}(\tau)-\mathcal{U}^e_{k,n}(\tau)}e(2\sum_{l\in S^e_{n,k}}(\sigma^{(t+1)}_S)^n_l-1)\beta^{(\tau)}_{O_{(i,\cdot)},k}\mid\Psi^{(t)},\mathbb{E}[\sum_{l\in S^e_{n,k}}(\sigma^{(t+1)}_S)^n_l]]\Big|^{\tau=0}_{\tau=t+1}=
$$

$$
(2\mathbb{E}[\sum_{l\in S^e_{n,k}}(\sigma^{(t+1)}_S)^n_l]-1)\Theta(\sum_{i\in\mathcal{W}^e_{k,n}(\tau)-\mathcal{U}^e_{k,n}(\tau)}\frac{e\beta^{(\tau)}_{O_{(i,\cdot)},k}}{m}\Big|^{\tau=0}_{\tau=t}
$$

$$
+\sum_{i\in\mathcal{U}^e_{k,n}(\tau)}\frac{\eta_t\|\boldsymbol{d}_k\|^2}{2mK_1}\mathbb{E}_{n\in\mathcal{V}^e_k}((2\sum_{l\in S^e_{n,k}}(\sigma^{(t)}_S)^n_l-1)\ell'^{(t)}_n)).
$$

(51)

As such, we have all preliminaries to characterize the first result of the lemma. We first utilize the induction to prove the following:

$$
|\mathbb{E}_{n\in\mathcal{V}^e_k}(ef(\mathbf{E}(S);\mathbb{E}(\Psi^{(t)}))-\mathbb{E}_{n\in\mathcal{V}^{-e}_k}(-ef(\mathbf{E}(S);\mathbb{E}(\Psi^{(t)})))|\le\kappa/8.\quad\forall e\in[\pm].
$$

This apparently hold at initialization. Suggest for any $t\le\widetilde{t}-1$ the result holds, then we only need to prove

$$
\mathbb{E}_{n\in\mathcal{V}^{e^*_{\widetilde{t}-1}}_k}(e^*_{\widetilde{t}-1}f(\mathbf{E}(S);\mathbb{E}(\Psi^{(\widetilde{t}-1)}))-\mathbb{E}_{n\in\mathcal{V}^{-e^*_{\widetilde{t}-1}}_k}(-e^*_{\widetilde{t}-1}f(\mathbf{E}(S);\mathbb{E}(\Psi^{(\widetilde{t}-1)})))\ge
$$

$$
\mathbb{E}_{n\in\mathcal{V}^{e^*_{\widetilde{t}-1}}_k}(e^*_{\widetilde{t}-1}f(\mathbf{E}(S);\mathbb{E}(\Psi^{(t_1)}))-\mathbb{E}_{n\in\mathcal{V}^{-e^*_{\widetilde{t}-1}}_k}(-e^*_{\widetilde{t}-1}f(\mathbf{E}(S);\mathbb{E}(\Psi^{(t_1)}))).
$$

By the condition of small $\eta_t$ in Condition 1, Lemma 31, Eq.(48) (49), (50) and (51), we see that

$$
\mathbb{E}_{n\in\mathcal{V}^{-e^*_{\widetilde{t}-1}}_k}(-e^*_{\widetilde{t}-1}f(\mathbf{E}(S);\mathbb{E}(\Psi^{(\widetilde{t}-1)})))-\mathbb{E}_{n\in\mathcal{V}^{e^*_{\widetilde{t}-1}}_k}(e^*_{\widetilde{t}-1}f(\mathbf{E}(S);\mathbb{E}(\Psi^{(\widetilde{t}-1)}))
$$

$$
-(\mathbb{E}_{n\in\mathcal{V}^{-e^*_{\widetilde{t}-1}}_k}(-e^*_{\widetilde{t}-1}f(\mathbf{E}(S);\mathbb{E}(\Psi^{(\widetilde{t})})))-\mathbb{E}_{n\in\mathcal{V}^{e^*_{\widetilde{t}-1}}_k}(e^*_{\widetilde{t}-1}f(\mathbf{E}(S);\mathbb{E}(\Psi^{(\widetilde{t})})))
$$

$$
\le\Theta(\mathbb{E}[\mathbb{E}_{n\in\mathcal{V}^{e^*_{\widetilde{t}-1}}_k}((\ell'^{(\widetilde{t}-1)}_n)-\mathbb{E}_{n\in\mathcal{V}^{-e^*_{\widetilde{t}-1}}_k}(\ell'^{(\widetilde{t}-1)}_n))\Big(\sum_{i\in\mathcal{W}^{e^*_{\widetilde{t}-1}}_{k,n}(\widetilde{t}-1)}\frac{\eta_{T^*}\|\boldsymbol{c}_k\|^2}{2mK_1}
$$

$$
+(2\sum_{l\in S^{e^*_{\widetilde{t}-1}}_{n,k}}(\sigma^{(\widetilde{t})}_S)^n_l-1)\sum_{i\in\mathcal{W}^{e^*_{\widetilde{t}-1}}_{k,n}(\widetilde{t}-1)}\frac{\eta_{T^*}\|\boldsymbol{d}_k\|^2}{2mK_1}\mathbb{E}_{n\in\mathcal{V}^{e^*_{\widetilde{t}-1}}_k}((2\sum_{l\in S^{e^*_{\widetilde{t}-1}}_{n,k}}(\sigma^{(\widetilde{t}-1)}_S)^n_l-1))\Big)])\le0,
$$

and thus we have

$$
|\mathbb{E}_{n\in\mathcal{V}^{e^*_{\widetilde{t}-1}}_k}(e^*_{\widetilde{t}-1}f(\mathbf{E}(S);\mathbb{E}(\Psi^{(\widetilde{t})}))-\mathbb{E}_{n\in\mathcal{V}^{-e^*_{\widetilde{t}-1}}_k}(-e^*_{\widetilde{t}-1}f(\mathbf{E}(S);\mathbb{E}(\Psi^{(\widetilde{t})})))|
$$

$$
\le|\mathbb{E}_{n\in\mathcal{V}^{e^*_{\widetilde{t}-1}}_k}(e^*_{\widetilde{t}-1}f(\mathbf{E}(S);\mathbb{E}(\Psi^{(\widetilde{t}-1)}))-\mathbb{E}_{n\in\mathcal{V}^{-e^*_{\widetilde{t}-1}}_k}(-e^*_{\widetilde{t}-1}f(\mathbf{E}(S);\mathbb{E}(\Psi^{(\widetilde{t}-1)})))|.
$$

Therefore, we complete the induction. Then we have

$$|\mathbb{E}[\underset{n\in\mathcal{V}_k^{e_{\tilde{t}-1}^*}}{\mathbb{E}}(\ell_n'^{(\tilde{t})}) - \underset{n\in\mathcal{V}_k^{-e_{\tilde{t}-1}^*}}{\mathbb{E}}(\ell_n'^{(\tilde{t})})]|$$

$$\leq |\underset{n\in\mathcal{V}_k^{e_{\tilde{t}-1}^*}}{\mathbb{E}}(e_{\tilde{t}-1}^* f(\mathbf{E}(S);\mathbb{E}(\Psi^{(\tilde{t})}))) - \underset{n\in\mathcal{V}_k^{-e_{\tilde{t}-1}^*}}{\mathbb{E}}(-e_{\tilde{t}-1}^* f(\mathbf{E}(S);\mathbb{E}(\Psi^{(\tilde{t})})))|$$

$$\leq |\underset{n\in\mathcal{V}_k^{e_{\tilde{t}-1}^*}}{\mathbb{E}}(e_{\tilde{t}-1}^* f(\mathbf{E}(S);\mathbb{E}(\Psi^{(\tilde{t}-1)}))) - \underset{n\in\mathcal{V}_k^{-e_{\tilde{t}-1}^*}}{\mathbb{E}}(-e_{\tilde{t}-1}^* f(\mathbf{E}(S);\mathbb{E}(\Psi^{(\tilde{t}-1)})))|$$

$$\leq \cdots \leq |\underset{n\in\mathcal{V}_k^e}{\mathbb{E}}(ef(\mathbf{E}(S);\mathbb{E}(\Psi^{(0)}))) - \underset{n\in\mathcal{V}_k^{-e}}{\mathbb{E}}(-ef(\mathbf{E}(S);\mathbb{E}(\Psi^{(0)})))|$$

$$\leq \sqrt{2\log(\frac{5Km}{\delta})} \cdot \frac{3\sigma_1(\|\boldsymbol{c}_k\| + \zeta_k^e\|\boldsymbol{d}_k\|)}{4} \leq \kappa/8.$$

This completes the proof of the first result.

To obtain the continuous ODE upper bound of $\mathbb{E}[\mathbf{A}_t^{k,e}]$, we first recall the update

$$\mathbb{E}[\alpha_{O_{(i,\cdot)},k}^{(t+1)} + e(2\sum_{l\in S_{n,k}^e}(\sigma_S^{(t+1)})_l^n - 1)\beta_{O_{(i,\cdot)},k}^{(t+1)} \mid \Psi^{(t)}, \mathbb{E}[\sum_{l\in S_{n,k}^e}(\sigma_S^{(t+1)})_l^n]] = (1-\eta_t\lambda)(\alpha_{O_{(i,\cdot)},k}^{(t)}+$$

$$e(2\sum_{l\in S_{n,k}^e}(\sigma_S^{(t)})_l^n - 1)\beta_{O_{(i,\cdot)},k}^{(t)}) - \eta_t\frac{1}{2K_1}\sum_{e\in[\pm]}\mathbf{r}_i e \underset{n\in\mathcal{V}_k^e}{\mathbb{E}}[\ell_n'^{(t)}\mathbb{1}_{O_{(i)}}^{n}{}^{(t)}\Big(\|\boldsymbol{c}_k\|^2 + \|\boldsymbol{d}_k\|^2(2\sum_{l\in S_{n,k}^e}(\sigma_S^{(t+1)})_l^n$$

$$-1)(2\sum_{l\in S_{n,k}^e}(\sigma_S^{(t)})_l^n - 1)\Big) - \eta_t\frac{(K_1-1)}{2K_1K}\sum_{e\in[\pm]}\mathbf{r}_i e \underset{n\in\mathcal{V}_{-k}^e}{\mathbb{E}}[\ell_n'^{(t)}\mathbb{1}_{O_{(i)}}^{n}{}^{(t)}\|\boldsymbol{c}_k\|^2].$$

Then, utilizing Lemma 31 and the fact $|\mathcal{W}_{k,n}^e(t)| \leq m$, we have constant $c_1 > 0$ such that

$$\mathbb{E}[\mathbf{A}_{t+1}^{k,e} \mid \Psi^{(t)}, \mathbb{E}(\sum_{l\in S_{n,k}^e}(\sigma_S^{(t+1)})_l^n)] \leq \mathbf{A}_t^{k,e} - c_1\big(\frac{\eta_t\|\mathbf{q}\|^2}{2mK_1} \cdot \underset{n\in\mathcal{V}_k^e}{\mathbb{E}}[\ell_n'^{(t)}]\big),$$

$$\leq \mathbf{A}_t^{k,e} + c_1\big(\frac{\eta_0\|\mathbf{q}\|^2}{2mK_1} \cdot \frac{1}{1+e^{-\kappa/2}e^{\mathbf{A}_t^{k,e}}}\big) \tag{52}$$

$$= \mathbf{A}_t^{k,e} + \frac{\bar{c}^{k,e}}{1+\bar{b}^{k,e}e^{\mathbf{A}_t^{k,e}}}.$$

where we also neglect the impact of cross-concept due to the large $K = \Omega(\eta_0 C(K_1-1)\|\mathbf{q}\|^2/(mK_1))$ in Condition 1 and appropriately chosen $c_1$.

To obtain the lower bound ODE couterpart, we examine the update of the correct contributor neurons, as shown in Eq.(48), (50) and (49).

In terms of the update of $\mathbb{E}[\alpha_{O_{(i,\cdot)},k}^{(t)}]$ where $i \in \mathbb{E}[\mathcal{U}_{k,n}^e(t) \cap (\mathcal{W}_{k,n}^{-e}(t) - \mathcal{U}_{k,n}^{-e}(t))]$, we see that its update is controlled by $\mathbb{E}[\underset{n\in\mathcal{V}_k^e}{\mathbb{E}}(\ell_n'^{(t)}) - \underset{n\in\mathcal{V}_k^{-e}}{\mathbb{E}}(\ell_n'^{(t)})]$:

$$\alpha_{O_{(i,\cdot)},k}^{(t+1)} = \Theta(\alpha_{O_{(i,\cdot)},k}^{(t)} - \frac{\eta_t\|\boldsymbol{c}_k\|^2}{2mK_1}[\underset{n\in\mathcal{V}_k^e}{\mathbb{E}}(\ell_n'^{(t)}) - \underset{n\in\mathcal{V}_k^{-e}}{\mathbb{E}}(\ell_n'^{(t)})]).$$

Then by the first result in this lemma we know $|\mathbb{E}[\underset{n\in\mathcal{V}_k^e}{\mathbb{E}}(\ell_n'^{(t)}) - \underset{n\in\mathcal{V}_k^{-e}}{\mathbb{E}}(\ell_n'^{(t)})]| \leq \frac{\beta}{4} \leq \frac{\kappa}{32}$, and thus

$$\alpha_{O_{(i,\cdot)},k}^{(t+1)} = \Theta(\alpha_{O_{(i,\cdot)},k}^{(t)} \pm \frac{\eta_t\kappa\|\boldsymbol{c}_k\|^2}{64mK_1}).$$

By the condition on the small initialization in Condition 1 such that $\sigma_1 = O(\frac{(2\sigma_S^*-1)^2}{Cm^{3/2}\|\mathbf{q}\|})$, due to the large $C$, we see that the $\kappa = O((2\sigma_S^* - 1)^2/m)$ is far more feeble. Thus the gradient contributions made by neuron set $\mathbb{E}[\alpha_{O_{(i,\cdot)},k}^{(t)}]$ where $i \in \mathbb{E}[\mathcal{U}_{k,n}^e(t) \cap (\mathcal{W}_{k,n}^{-e}(t) - \mathcal{U}_{k,n}^{-e}(t))]$ can be neglected compared to the increasing update of $\mathbb{E}[\alpha_{O_{(i,\cdot)},k}^{(t)}\mathbb{1}(i \in \mathcal{U}_{k,n}^e(t) - (\mathcal{W}_{k,n}^{-e}(t) - \mathcal{U}_{k,n}^{-e}(t)))]$ and $\mathbb{E}[e(2\sum_{l\in S_{n,k}^e}(\sigma_S^{(t)})_l^n - 1)\beta_{O_{(i,\cdot)},k}^{(t)}]$. Besides, we

see that $\mathbb{E}[\mathcal{W}_{k,n}^e(t)]$ will at least preserve the neurons of $\mathbb{E}[\mathcal{U}_{k,n}^e(0))]$, which will not be deactivated by Lemma 29.

Then there exists $c_2 > 0$, recall $\sigma_S^*$ is defined in Lemma 34 as the lower bound of $\min_{t,k}\{\underset{n \in \mathcal{D}_S}{\mathbb{E}}[\sum_{j \in S_{n,k}^{y_{S_n}}} (\sigma_S^{(t)})_j^n]\}$ and $\mathbb{E}[|\mathcal{U}_{k,n}^e(0))|] \geq m/8$, it holds that

$$
\begin{aligned}
\mathbb{E}[\mathbf{A}_{t+1}^{k,e} \mid \Psi^{(t)}, \mathbb{E}(\sum_{l \in S_{n,k}^e} (\sigma_S^{(t+1)})_l^n)] &\geq \mathbf{A}_t^{k,e} - c_2\left(\frac{\eta_t(2\sigma_S^*-1)^2\|\boldsymbol{d}_k\|^2}{8mK_1} \cdot \underset{n \in \mathcal{V}_k^e}{\mathbb{E}}[\ell_n'^{(t)}]\right) \\
&\geq \mathbf{A}_t^{k,e} + \left(\frac{c_2\eta_t(2\sigma_S^*-1)^2\|\boldsymbol{d}_k\|^2}{8mK_1} \cdot \frac{1}{1+e^{\kappa/2}e^{\mathbf{A}_t^{k,e}}}\right) \\
&\geq \mathbf{A}_t^{k,e} + \left(\frac{c_2\eta_{T^*}(2\sigma_S^*-1)^2(1-\kappa_{\boldsymbol{y}})\|\mathbf{q}\|^2}{16mK_1} \cdot \frac{1}{1+e^{\kappa/2}e^{\mathbf{A}_t^{k,e}}}\right) \\
&= \mathbf{A}_t^{k,e} + \frac{\underline{c}^{k,e}}{1+\underline{b}^{k,e}e^{\mathbf{A}_t^{k,e}}}.
\end{aligned}
\tag{53}
$$

where we ignore the impact of regularization term at this stage since $\lambda = O((C\log(Km/\delta)\|\mathbf{q}\|)^{-1})$ and appropriately chosen $c_2$. The third inequality is due to the definition of $\boldsymbol{d}_k$.

Collaborating with Lemma 10, the proofs are completed.

For the last results, following the techniques in [43], first it's easy to check that

$$
\overline{b}^{k,e}e^{\overline{x}_t^{k,e}} \leq \overline{x}_t^{k,e} + \overline{b}^{k,e}e^{\overline{x}_t^{k,e}} \leq 1.5\overline{b}^{k,e}e^{\overline{x}_t^{k,e}}, \quad \underline{b}^{k,e}e^{\underline{x}_t^{k,e}} \leq \underline{x}_t^{k,e} + \underline{b}^{k,e}e^{\underline{x}_t^{k,e}} \leq 1.5\underline{b}^{k,e}e^{\underline{x}_t^{k,e}},
$$

thus

$$
\log(\frac{2\overline{c}^{k,e}}{3\overline{b}^{k,e}} + \frac{2}{3}) \leq \overline{x}_t^{k,e} \leq \log(\frac{\overline{c}^{k,e}}{\overline{b}^{k,e}}t + 1), \quad \log(\frac{2\underline{c}^{k,e}}{3\underline{b}^{k,e}} + \frac{2}{3}) \leq \underline{x}_t^{k,e} \leq \log(\frac{\underline{c}^{k,e}}{\underline{b}^{k,e}} + 1).
$$

Thus

$$
\log(\frac{2\underline{c}^{k,e}}{3\underline{b}^{k,e}} + \frac{2}{3}) \leq \mathbb{E}[\mathbf{A}_t^{k,e}] \leq \log(\frac{\overline{c}^{k,e}}{\overline{b}^{k,e}}t + 1) + \frac{\overline{c}^{k,e}}{1+\overline{b}^{k,e}}.
$$

$\square$

***Proof of Lemma 30.*** We use induction to prove this lemma. All conclusion holds naturally at $t = 0$. Suppose there exists $\widetilde{T} \leq T^*$ such that the six conditions hold for any $0 \leq t \leq \widetilde{T} - 1$, we prove that these conclusions also hold for $t = \widetilde{T}$.

We now prove

$$
0 \leq \mathbb{E}[e \cdot \beta_{O_{(i,\cdot)},k}^{(t)}\mathbb{1}(i \in \mathcal{U}_{k,n}^e(t))] - e \cdot \beta_{O_{(i,\cdot)},k}^{(0)} \leq (\sigma_S^*)^{-1}\alpha.
\tag{54}
$$

Recall the update rule

$$
\beta_{O_{(i,\cdot)},k}^{(t+1)} = (1-\eta_t\lambda)\beta_{O_{(i,\cdot)},k}^{(t)} - \eta_t\frac{\|\boldsymbol{d}_k\|^2}{2K_1}\sum_{e \in [\pm]}\mathbf{r}_i \underset{n \in \mathcal{V}_k^e}{\mathbb{E}}[\ell_n'^{(t)}\mathbb{1}_{O_{(i)}}^{n\,(t)}(\sum_{l \in S_{n,k}^e} (\sigma_S^{(t)})_l^n - \sum_{l \in S_{n,k}^{-e}} (\sigma_S^{(t)})_l^n)].
$$

As we ignore the regularization term at the first stage, we can easily seen that $\mathbb{E}[e \cdot \beta_{O_{(i,\cdot)},k}^{(t)}\mathbb{1}(i \in \mathcal{U}_{k,n}^e(t))]$ increases with $t$. Assume $t_{\beta+,k}$ as the last time $\exists i \in \mathbb{E}[\mathcal{U}_{k,n}^e(t)]$ such that $\mathbb{E}[e \cdot \beta_{O_{(i,\cdot)},k}^{(t)}\mathbb{1}(i \in \mathcal{U}_{k,n}^e(t))] - e \cdot$

$\beta_{O_{(i,\cdot)},k}^{(0)} \le (\sigma_S^*)^{-1} \log(T^*)$, then for $i \in \mathbb{E}[\mathcal{U}_{k,n}^e(\tilde{t})]$ we have

$$
\begin{aligned}
\mathbb{E}[e \cdot \beta_{O_{(i,\cdot)},k}^{(\tilde{t})}] \le{}& \mathbb{E}[e \cdot \beta_{O_{(i,\cdot)},k}^{(t_{\beta+,k})}] \\
&- \eta_0 \frac{\|\boldsymbol{d}_k\|^2}{2K_1} \sum_{e \in [\pm]} \mathbf{r}_i \mathbb{E}[\ell_n'^{(t)} \mathbb{1}_{O_{(i)}}^{n}{}^{(t)} (\sum_{l \in S_{n,k}^e} (\sigma_S^{(t)})_l^n - \sum_{l \in S_{n,k}^{-e}} (\sigma_S^{(t)})_l^n)]\Big|_{t=t_{\beta+,k}} \\
&- \eta_0 \sum_{t_{\beta+,k} < t < \tilde{T}} \frac{\|\boldsymbol{d}_k\|^2}{2K_1} \sum_{e \in [\pm]} \mathbf{r}_i \mathbb{E}[\ell_n'^{(t)} (\sum_{l \in S_{n,k}^e} (\sigma_S^{(t)})_l^n - \sum_{l \in S_{n,k}^{-e}} (\sigma_S^{(t)})_l^n)] \\
\le{}& e \cdot \beta_{O_{(i,\cdot)},k}^{(0)} + (\sigma_S^*)^{-1} \log(T^*) + \frac{\eta_0 \|\boldsymbol{d}_k\|^2}{2mK_1} - \sum_{t_{\beta+,k} < t < \tilde{T}} \frac{\eta_0 \|\boldsymbol{d}_k\|^2}{2mK_1} \mathbb{E}[\ell_n'^{(t)}] \\
\le{}& e \cdot \beta_{O_{(i,\cdot)},k}^{(0)} + 2(\sigma_S^*)^{-1} \log(T^*) - \sum_{t_{\beta+,k} < t < \tilde{T}} \frac{\eta_0 \|\boldsymbol{d}_k\|^2}{2mK_1} \mathbb{E}[\ell_n'^{(t)} \mathbb{1}_{O_{(i)}}^{n}{}^{(t)}],
\end{aligned}
$$

where the first inequality is by the positive nature of regularization term as well as the contribution of the gradient; second inequality is by $\mathbb{E}[-\ell_n'^{(t_{\beta+,k})}] \le 1$ and $\mathbb{E}[(\sum_{l \in S_{n,k}^e} (\sigma_S^{(t)})_l^n - \sum_{l \in S_{n,k}^{-e}} (\sigma_S^{(t)})_l^n)] \le 1$; the third inequality is by the condition $\eta_0 = O(\frac{mK_1}{\|\mathbf{q}\|^2})$ and thus $\frac{\eta_0 \|\boldsymbol{d}_k\|^2}{2mK_1} \le 1 \le \log(T^*)$, as well as $(\sigma_S^*)^{-1} \ge 1$. The remaining job is to prove that

$$
-\sum_{t_{\beta+,k} < t < \tilde{T}} \frac{\eta_0 \|\boldsymbol{d}_k\|^2}{2mK_1} \mathbb{E}[\ell_n'^{(t)} \mathbb{1}_{O_{(i)}}^{n}{}^{(t)}] \le (\sigma_S^*)^{-1} \log(T^*).
$$

Observe that

$$
\begin{aligned}
|\mathbb{E}[\ell_n'^{(t)}]| = {}& \mathbb{E}[\frac{1}{1 + \exp(y_{S_n} \cdot (\sum_{e \in [\pm]} \frac{e}{m} \sum_{i \in \{\mathbf{r}_i = \frac{e}{m}\}} \sigma_R(\mathbf{W}_{O_{(i,\cdot)}}^{\boldsymbol{y}}{}^{(t)} \sum_{l \in [L]} (\sigma_S^{(t)})_l^n \boldsymbol{y}_l^n)))}] \\
\le{}& \mathbb{E}[\exp\left((\sum_{i \in \mathcal{W}_{k,n}^{y_{S_n}}(t) - \mathcal{U}_{k,n}^{y_{S_n}}(t)} - \sum_{i \in \mathcal{U}_{k,n}^{y_{S_n}}(t)})(\alpha_{O_{(i,\cdot)},k}^{(t)} + (2 \sum_{l \in S_{n,k}^{y_{S_n}}} (\sigma_S^{(t)})_l^n - 1) y_{S_n} \beta_{O_{(i,\cdot)},k}^{(t)})\right)] \\
\le{}& \mathbb{E}[\exp(\kappa/2 - \frac{1}{m} \sum_{i \in \mathcal{U}_{k,n}^{y_{S_n}}(t)} (2\sigma_S^* - 1) y_{S_n} \beta_{O_{(i,\cdot)},k}^{(t)})] \\
\le{}& 2 \exp(-\log(T^*)).
\end{aligned}
$$

Here the first inequality is by $1/(1 + \exp(z)) \le \exp(-z)$; the second inequality is by Lemma 31; the last inequality is by the feeble $\kappa/2$ and $\mathbb{E}[e \cdot \beta_{O_{(i,\cdot)},k}^{(t)} \mathbb{1}(i \in \mathcal{U}_{k,n}^e(t))] \ge (\sigma_S^*)^{-1} \log(T^*)$. Then we have that

$$
\begin{aligned}
-\sum_{t_{\beta+,k} < t < \tilde{T}} \frac{\eta_0 \|\boldsymbol{d}_k\|^2}{2mK_1} \mathbb{E}[\ell_n'^{(t)} \mathbb{1}_{O_{(i)}}^{n}{}^{(t)}] \le{}& \sum_{t_{\beta+,k} < t < \tilde{T}} \frac{\eta_0 \|\boldsymbol{d}_k\|^2}{2mK_1} \cdot 2 \exp(-\log(T^*)) \\
\le{}& \frac{\tilde{T} \eta_0 \|\boldsymbol{d}_k\|^2}{mK_1} \exp(-\log(T^*)) \le \frac{T^* \eta_0 \|\boldsymbol{d}_k\|^2}{T^* mK_1} \le (\sigma_S^*)^{-1} \log(T^*).
\end{aligned}
$$

We complete the proof that $0 \le \mathbb{E}[e \cdot \beta_{O_{(i,\cdot)},k}^{(t)} \mathbb{1}(i \in \mathcal{U}_{k,n}^e(t))] - e \cdot \beta_{O_{(i,\cdot)},k}^{(0)} \le 3(\sigma_S^*)^{-1} \log(T^*) \le (\sigma_S^*)^{-1} \alpha$.

We now prove a strong augmented hypothesis that there exist $i^* \in \mathbb{E}[\mathcal{U}_{k,n}^e(t)]$ for $\forall 0 \le t \le T^*$, we have

$$
\mathbb{E}[|\alpha_{O_{(i,\cdot)},k}^{(t)}| / (e \cdot \beta_{O_{(i^*,\cdot)},k}^{(t)})] \le \hat{C} \frac{\|\boldsymbol{c}_k\|^2}{\sigma_S^* \|\boldsymbol{d}_k\|^2}, \tag{55}
$$

where we set $\hat{C} = 2C' \sqrt{2 \log(\frac{5Km}{\delta})}$ for some constant $C'$. $i^*$ can be any element satisfies $|\beta_{O_{(i^*,\cdot)},k}^{(0)}| = \sigma_1/2 \|\boldsymbol{d}_k\|$, which exists at $t = 0$ by Lemma 7 as well as the fact that $\|\boldsymbol{c}_k\| > \|\boldsymbol{d}_k\|$ by their definition in Lemma 25.

Suppose Eq.(55) holds at $0 \le t \le \widetilde{T} - 1$, recall the update rule and the large $K$ condition, we can have a constant $C > 1$ such that

$$\mathbb{E}[e \cdot \beta^{(\tilde{t})}_{O_{(i,\cdot)},k} \mathbb{1}(i \in \mathcal{U}^e_{k,n}(\widetilde{T}-1))] = \mathbb{E}[(1 - \eta_{\widetilde{T}-1}\lambda)e \cdot \beta^{(\widetilde{T}-1)}_{O_{(i,\cdot)},k} \mathbb{1}(i \in \mathcal{U}^e_{k,n}(\widetilde{T}-1))]$$

$$- \eta_{\widetilde{T}-1} \frac{\|d_k\|^2}{2mK_1} \mathbb{E}[\ell'_n{}^{(\widetilde{T}-1)} (\sum_{l \in S^e_{n,k}} (\sigma^{(\widetilde{T}-1)}_S)^n_l - \sum_{l \in S^{-e}_{n,k}} (\sigma^{(\widetilde{T}-1)}_S)^n_l)],$$

$$\ge \mathbb{E}[(1 - \eta_{\widetilde{T}-1}\lambda)e \cdot \beta^{(\widetilde{T}-1)}_{O_{(i,\cdot)},k} \mathbb{1}(i \in \mathcal{U}^e_{k,n}(\widetilde{T}-1))] + \frac{\eta_{\widetilde{T}-1}\sigma^*_S\|d_k\|^2}{2mK_1}$$

$$\mathbb{E}[|\ell'_n{}^{(\widetilde{T}-1)}|],$$

$$\mathbb{E}[|\alpha^{(\tilde{t})}_{O_{(i,\cdot)},k}|] = \mathbb{E}\Big[|(1 - \eta_{\widetilde{T}-1}\lambda)\alpha^{(\widetilde{T}-1)}_{O_{(i,\cdot)},k}$$

$$- \eta_{\widetilde{T}-1} \frac{\|c_k\|^2}{2K_1} \sum_{e \in [\pm]} [e\mathbf{r}_i \cdot \mathop{\mathbb{E}}_{n \in \mathcal{V}^e_k}(\ell'_n{}^{(\widetilde{T}-1)} \mathbb{1}^n_{O_{(i)}}{}^{(\widetilde{T}-1)})] - \eta_{\widetilde{T}-1} \frac{(K_1-1)\|c_k\|^2}{2K_1 K} \sum_{e \in [\pm]} [e\mathbf{r}_i$$

$$\mathop{\mathbb{E}}_{n \in \mathcal{V}^e_{\neg k}}(\ell'_n{}^{(\widetilde{T}-1)} \mathbb{1}^n_{O_{(i)}}{}^{(\widetilde{T}-1)})]|\Big]$$

$$\le \mathbb{E}[|(1 - \eta_{\widetilde{T}-1}\lambda)\alpha^{(\widetilde{T}-1)}_{O_{(i,\cdot)},k}|] + \frac{C\eta_{\widetilde{T}-1}\|c_k\|^2}{2mK_1} \mathbb{E}[|\ell'_n{}^{(\widetilde{T}-1)}|].$$

where the first inequality is due to the definition of $\sigma^*_S$ and $\mathcal{U}^e_{k,n}(\widetilde{t})$; the second inequality is due to the large $K = \Omega(\eta_0 C(K_1 - 1)\|\mathbf{q}\|^2/(mK_1))$. Then we have

$$\frac{\mathbb{E}[|\alpha^{(\tilde{t})}_{O_{(i,\cdot)},k}|]}{\mathbb{E}[e \cdot \beta^{(\tilde{t})}_{O_{(i,\cdot)},k} \mathbb{1}(i \in \mathcal{U}^e_{k,n}(\widetilde{t}))]} \le \max\{\frac{\mathbb{E}[|\alpha^{(\widetilde{T}-1)}_{O_{(i,\cdot)},k}|]}{\mathbb{E}[e \cdot \beta^{(\widetilde{T}-1)}_{O_{(i,\cdot)},k} \mathbb{1}(i \in \mathcal{U}^e_{k,n}(\widetilde{T}-1))]}, \frac{C\|c_k\|^2}{\sigma^*_S\|d_k\|^2}\} \le \hat{C} \frac{\|c_k\|^2}{\sigma^*_S\|d_k\|^2},$$

where the last inequality is by the induction hypothesis and the $C'$ can be taken as $C$, which completes the induction.

We now prove

$$0 \ge \mathbb{E}[e \cdot \beta^{(t)}_{O_{(i,\cdot)},k} \mathbb{1}(i \in \mathcal{W}^e_{k,n}(t) - \mathcal{U}^e_{k,n}(t))] - e \cdot \beta^{(0)}_{O_{(i,\cdot)},k}$$

$$\ge -\frac{\hat{C}\|c_k\|^2}{\sigma^{*2}_S\|d_k\|^2}\alpha - \frac{\sigma_1(\sigma^{*2}_S\|d_k\|^2 + \hat{C}\|c_k\|^2)}{\sigma^{*2}_S\|d_k\|} \sqrt{2\log(\frac{5Km}{\delta})}.$$

Recall the update rule

$$\beta^{(t+1)}_{O_{(i,\cdot)},k} = (1 - \eta_t\lambda)\beta^{(t)}_{O_{(i,\cdot)},k} - \eta_t \frac{\|d_k\|^2}{2K_1} \sum_{e \in [\pm]} \mathbf{r}_i \mathop{\mathbb{E}}_{n \in \mathcal{V}^e_k}[\ell'_n{}^{(t)} \mathbb{1}^n_{O_{(i)}}{}^{(t)}(\sum_{l \in S^e_{n,k}} (\sigma^{(t)}_S)^n_l - \sum_{l \in S^{-e}_{n,k}} (\sigma^{(t)}_S)^n_l)].$$

Easy to see that $\mathbb{E}[e \cdot \beta^{(\tau)}_{O_{(i,\cdot)},k} \mathbb{1}(i \in \mathcal{W}^e_{k,n}(\tau) - \mathcal{U}^e_{k,n}(\tau))]\Big|_{\tau=t}^{\tau=0} \le 0$ and it's decreasing. As we know that the neuron $i \in \mathbb{E}[\mathcal{W}^e_{k,n}(t) - \mathcal{U}^e_{k,n}(t)]$ would be deactivated at $t + 1$ once

$$\mathbb{E}[\alpha^{(t+1)}_{O_{(i,\cdot)},k} + e \cdot (\sum_{l \in S^e_{n,k}} (\sigma^{(t+1)}_S)^n_l - \sum_{l \in S^{-e}_{n,k}} (\sigma^{(t+1)}_S)^n_l \beta^{(t+1)}_{O_{(i,\cdot)},k}] \le 0.$$

This indicates that for the neuron $i \in \mathbb{E}[\mathcal{W}^e_{k,n}(\widetilde{t}) - \mathcal{U}^e_{k,n}(\widetilde{t})]$,

$$\mathbb{E}[\alpha^{(\tilde{t})}_{O_{(i,\cdot)},k} + e \cdot (\sum_{l \in S^e_{n,k}} (\sigma^{(\tilde{t})}_S)^n_l - \sum_{l \in S^{-e}_{n,k}} (\sigma^{(\tilde{t})}_S)^n_l \beta^{(\tilde{t})}_{O_{(i,\cdot)},k}] \ge 0.$$

Now collaborating with Eq. (54) and Eq. (55), we now can have

$$\mathbb{E}[e \cdot (\sum_{l \in S_{n,k}^e} (\sigma_S^{(\tilde{t})})_l^n - \sum_{l \in S_{n,k}^{-e}} (\sigma_S^{(\tilde{t})})_l^n \beta_{O_{(i,\cdot)},k}^{(\tilde{t})}] \geq -\mathbb{E}[\alpha_{O_{(i,\cdot)},k}^{(\tilde{t})}]$$

$$\geq -\frac{\hat{C}\|\boldsymbol{c}_k\|^2}{\sigma_S^*\|\boldsymbol{d}_k\|^2}((\sigma_S^*)^{-1}\alpha + e \cdot \beta_{O_{(i,\cdot)}})$$

$$\geq -\frac{\hat{C}\|\boldsymbol{c}_k\|^2}{\sigma_S^*\|\boldsymbol{d}_k\|^2}((\sigma_S^*)^{-1}\alpha -$$

$$\sqrt{2\log(\frac{5Km}{\delta})}\sigma_1\|\boldsymbol{d}_k\|)$$

$$\Rightarrow \mathbb{E}[e \cdot \beta_{O_{(i,\cdot)},k}^{(t)} \mathbb{1}(i \in \mathcal{W}_{k,n}^e(t) - \mathcal{U}_{k,n}^e(t))] - e \cdot \beta_{O_{(i,\cdot)},k}^{(0)} \geq -\frac{\hat{C}\|\boldsymbol{c}_k\|^2}{\sigma_S^{*2}\|\boldsymbol{d}_k\|^2}\alpha$$

$$-\frac{\sigma_1(\sigma_S^{*2}\|\boldsymbol{d}_k\|^2 + \hat{C}\|\boldsymbol{c}_k\|^2)}{\sigma_S^{*2}\|\boldsymbol{d}_k\|}$$

$$\cdot \sqrt{2\log(\frac{5Km}{\delta})}.$$

The proof is completed. $\qquad\square$

### I.1.1 Expected 0-1 loss Convergence

**Lemma 33.** *Under Condition 1, there exist constant $C_1 > 0$, after at most*

$$\hat{T} = \frac{C_1\sigma_1 m\lambda K_1\gamma\sqrt{(1+\kappa_{\boldsymbol{y}})\log(5Km/\delta)}}{(2\sigma_S^* - 1)^2(1-\kappa_{\boldsymbol{y}})\|\mathbf{q}\|}.$$

*iterations, we have $L_{\mathcal{D}_S}^{0-1}(\mathbb{E}(\Psi^t)) = L_{\mathcal{D}^*}^{0-1}(\mathbb{E}(\Psi^t)) = 0$.*

*Proof.* For $t \leq \tilde{t}$, recall from Eq.(53) that for the period $t \leq \hat{T}$, it holds that

$$\mathbb{E}[\mathbf{A}_{t+1}^{k,e} \mid \Psi^{(t)}] \geq \mathbf{A}_t^{k,e} - (\frac{c_2\eta(2\sigma_S^* - 1)^2(1-\kappa_{\boldsymbol{y}})\|\mathbf{q}\|^2}{16mK_1} \cdot \underset{n \in \mathcal{V}_k^e}{\mathbb{E}}[\ell_n'^{(t)}]).$$

Note that by definition $\mathbf{A}_0^{k,e} = 0$, and we recursively use the equation t times

$$\mathbb{E}[\mathbf{A}_t^{k,e}] \geq \sum_{s=0}^{t-1} -\frac{c_2\eta(2\sigma_S^* - 1)^2(1-\kappa_{\boldsymbol{y}})\|\mathbf{q}\|^2}{16mK_1} \cdot \underset{n \in \mathcal{V}_k^e}{\mathbb{E}}[\ell_n'^{(s)}].$$

For each $k \in [K_1], e \in [\pm]$, denote by $\tilde{t}^{k,e}$ the last time in the period $[0, T^*]$ satisfying that $\mathbb{E}[\mathbf{A}_t^{k,e}] \leq \kappa$. Then by Lemma 31 we see that

$$\left| \underset{n \in \mathcal{V}_k^e}{\mathbb{E}}[ef(\mathbf{E}(S); \mathbb{E}(\Psi^{(t)}))]] \right| \leq 3\kappa/2.$$

Thus there exists a positive constant $\tilde{C}$ such that $-\underset{n \in \mathcal{V}_k^e}{\mathbb{E}}[\ell_n'^{(t)}] \geq \tilde{C}$ for $0 \leq t \leq \tilde{t}^{k,e}$. Then we have

$$\mathbb{E}[\mathbf{A}_t^{k,e}] \geq \frac{\tilde{C}c_2\eta(2\sigma_S^* - 1)^2(1-\kappa_{\boldsymbol{y}})\|\mathbf{q}\|^2 t}{16mK_1}.$$

Therefore we see that for $\forall k \in [K_1], e \in [\pm]$, $\mathbb{E}[\mathbf{A}_t^{k,e}]$ will reach $\kappa$ within $\frac{16mK_1\kappa}{\tilde{C}c_2\eta(2\sigma_S^* - 1)^2(1-\kappa_{\boldsymbol{y}})\|\mathbf{q}\|^2}$ epochs. Recall that in this first stage the impact of decaying learning rate is under controlled by a large $\gamma$ in Condition 1 as well as the slow quadratic decaying speed of $\eta_t$, under which we have $\eta = \Theta(\eta_0)$. By $\kappa \leq 8\sigma_1\|\mathbf{q}\|\sqrt{(1+\kappa_{\boldsymbol{y}})\log(5Km/\delta)})$, we see that there exist a positive constant $C_1 = \Theta(64/(\tilde{C}c_2))$, the threshold time can be

$$\hat{T} = \frac{C_1\sigma_1 m\lambda K_1\gamma\sqrt{(1+\kappa_{\boldsymbol{y}})\log(5Km/\delta)}}{(2\sigma_S^* - 1)^2(1-\kappa_{\boldsymbol{y}})\|\mathbf{q}\|}.$$

Then by definition of 0-1 loss we have

$$L_{\mathcal{D}^*}^{0-1}(\mathbb{E}(\Psi^{\hat{T}})) = \mathbb{P}_{S_n \sim \mathcal{D}^*}(y_{S_n} \cdot f(\mathbf{E}(S_n), \mathbb{E}(\Psi^{\hat{T}})) \leq 0)$$

$$\leq \mathbb{P}_{S_n \sim \mathcal{D}^*}(\mathbb{E}[\mathbf{A}_{\hat{T}}^{k,e}] - \kappa/2 \leq 0)$$

$$\leq \mathbb{P}_{S_n \sim \mathcal{D}^*}(\mathbb{E}[\kappa/2 \leq 0) = 0.$$

The proof is completed. $\qquad\square$

## I.1.2 Period 1: Decreasing Period of Correct Attention Score

We claim that if $\sum_{i\in[m]} \mathbf{r}_i \beta^{(0)}_{O_{(i,\cdot)},k} > 0$ during initialization, the expected attention score will not experience this decreasing period due to the expected gradient formula in Lemma 27. Our aim for this period is to examine the lower bound of the attention score during a limited number of iterations.

**Lemma 34.** *Under Condition 1, for $\forall k \in [K_1]$, after at most a certain iterations*

$$T_1 = \frac{C_3 \sigma_1 K_1 \gamma \sqrt{10\log(5Km/\delta)}(1 + e^{-2\sigma_0^2\|\boldsymbol{b}_k\|^2})}{2C_4\|\boldsymbol{d}_k\|(1 - e^{-2\sigma_0^2\|\boldsymbol{b}_k\|^2})},$$

*where $C_3$ is a positive constant, we would have the $\beta^{(t)}_{Q,k} = \beta^{(t)}_{K,k}$ be monotonically increasing during the remaining iterations $T_1 \leq t \leq T^*$. Besides, it holds that $\sigma_S^*$ is the lower bound of the lowest correct attention assignment along the whole iterations:*

$$\sigma_S^* \leq \min_{t\in[T^*],k\in[K_1]}\Big\{ \mathop{\mathbb{E}}_{n\in\mathcal{D}_S}\big[ \sum_{j\in S^{y_{S_n}}_{n,k}} (\sigma_S^{(t)})^n_j\big]\Big\}.$$

*Proof.* By Lemma 27, the $\beta^{(t+1)}_{Q,k} = \mathbb{E}[\beta^{(t+1)}_{K,k} \mid \Psi^{(t)}]$ will be contributed to increase by

$$\{i \in \mathbb{E}[\mathcal{W}^{\pm}_{k,n}(t)] \mid \mathbf{r}_i \cdot \beta^{(t)}_{O_{(i,\cdot)},k} > 0\}$$

and they will be contributed to decrease by

$$\{i \in \mathbb{E}[\mathcal{W}^{\pm}_{k,n}(t)] \mid \mathbf{r}_i \cdot \beta^{(t)}_{O_{(i,\cdot)},k} < 0\}$$

By the fifth inequality in Lemma 7, we know that

$$\left|\big|\{i \in \mathbb{E}[\mathcal{W}^{\pm}_{k,n}(0)] \mid \mathbf{r}_i \cdot \beta^{(0)}_{O_{(i,\cdot)},k} > 0\}\big| - \frac{m}{4}\right| \leq \frac{m}{16}, \left|\big|\{i \in \mathbb{E}[\mathcal{W}^{\pm}_{k,n}(0)] \mid \mathbf{r}_i \cdot \beta^{(0)}_{O_{(i,\cdot)},k} < 0\}\big| - \frac{m}{4}\right| \leq \frac{m}{16}.$$

As $\mathop{\mathbb{E}}_{n\in\mathcal{V}^e_k}[\ell'^{(t)}_n \mathbb{1}^{n~(t)}_{O_{(i)}} (\sum_{j\in S^+_{n,k}} (\sigma_S^{(t)})^n_j)(\sum_{j\in S^-_{n,k}} (\sigma_S^{(t)})^n_j)]$ is shared by all neurons, thus whether the $\beta^{(t)}_{Q,k}$ and $\beta^{(t)}_{K,k}$ will be contributed to increase or decrease depends on the signal of $\sum_{i\in\mathbb{E}[\mathcal{W}^{\pm}_{k,n}(t)]} \mathbf{r}_i \cdot e\beta^{(t)}_{O_{(i,\cdot)},k}$. By the last inequality in in Lemma 7, we see that at initialization,

$$\sum_{i\in\mathbb{E}[\mathcal{W}^{\pm}_{k,n}(0)]} \mathbf{r}_i \cdot \beta^{(0)}_{O_{(i,\cdot)},k} \geq -\sqrt{2\log(\frac{5Km}{\delta})} \cdot \frac{5\sigma_1\|\boldsymbol{d}_k\|}{16}. \tag{56}$$

By the expected gradient update in Lemma 28, the $e\beta^{(t)}_{O_{(i,\cdot)},k}$ will grow in $\mathbf{r}_i$'s direction along the whole iterations. As such, the values of $\mathbb{E}[\mathbf{r}_i \cdot \beta^{(t)}_{O_{(i,\cdot)},k}\mathbb{1}(i \in \mathcal{W}^{\pm}_{k,n}(t))], \forall k \in [K_1]$ will grow larger. Therefore, after a limited epochs we can have

$$\sum_{i\in\mathbb{E}[\mathcal{W}^{\pm}_{k,n}(t)]} \mathbf{r}_i \cdot e\beta^{(t)}_{O_{(i,\cdot)},k} \geq 0,$$

where the $\beta^{(t)}_{Q,k}$ and $\beta^{(t)}_{K,k}$ would be contributed positively and monotonically increase.

Now we serve to find the lower bound of the evolution of $\mathbb{E}[(\sum_{l\in S^e_{n,k}} (\sigma_S^{(t)})^n_l)]$, which is clearly to be the first iteration where the negative $\sum_{i\in\mathbb{E}[\mathcal{W}^{\pm}_{k,n}(t)]} \mathbf{r}_i \cdot e\beta^{(t)}_{O_{(i,\cdot)},k}$ has grown to surpass the 0. By the symmetry property denoted in Lemma 22 and Eq.(39) we have

$$\mathop{\mathbb{E}}_{n\in\mathcal{V}^{y_{S_n}}_k}\big[ \sum_{j\in S^{y_{S_n}}_{n,k}} (\sigma_S^{(t)})^n_j\big] = \frac{1}{1 + e^{-2\beta^{(t)~2}_{Q,k}/\|\boldsymbol{b}_k\|^2}}. \tag{57}$$

Recall that

$$\beta^{(t+1)}_{K,k} = \beta^{(t+1)}_{Q,k} = (1 - \eta_t\lambda)\beta^{(t)}_{Q,k} - \frac{4\eta_t\beta^{(t)}_{Q,k}\|\boldsymbol{b}_k\|^4}{K_1} \sum_{e\in[\pm]}\sum_{i\in[m]} \mathbf{r}_i\beta^{(t)}_{O_{(i,\cdot)},k} \mathop{\mathbb{E}}_{n\in\mathcal{V}^e_k}[\ell'^{(t)}_n \mathbb{1}^{n~(t)}_{O_{(i)}}$$

$$(\sum_{j\in S^+_{n,k}} (\sigma_S^{(t)})^n_j)(\sum_{j\in S^-_{n,k}} (\sigma_S^{(t)})^n_j)],$$

$$\beta^{(t+1)}_{O_{(i,\cdot)},k} = (1 - \eta_t\lambda)\beta^{(t)}_{O_{(i,\cdot)},k} - \eta_t\frac{\|\boldsymbol{d}_k\|^2}{2K_1} \sum_{e\in[\pm]} \mathbf{r}_i \mathop{\mathbb{E}}_{n\in\mathcal{V}^e_k}[\ell'^{(t)}_n \mathbb{1}^{n~(t)}_{O_{(i)}}(\sum_{l\in S^e_{n,k}} (\sigma_S^{(t)})^n_l - \sum_{l\in S^{-e}_{n,k}} (\sigma_S^{(t)})^n_l)],$$

and we also see that

$$\mathbb{E}[(\sum_{j\in S_{n,k}^+}(\sigma_S^{(t)})_j^n)(\sum_{j\in S_{n,k}^-}(\sigma_S^{(t)})_j^n)] = \left(\exp(\beta_{Q,k}^{(t)}\cdot\beta_{K,k}^{(t)}/\|\boldsymbol{b}_k\|^2)+\exp(-\beta_{Q,k}^{(t)}\cdot\beta_{K,k}^{(t)}/\|\boldsymbol{b}_k\|^2)\right)^{-2}\leq\frac{1}{4}.$$

(58)

As $\sum_{i\in\mathbb{E}[\mathcal{W}_{k,n}^\pm(t)]}\mathbf{r}_i\cdot e\beta_{O_{(i,\cdot)},k}^{(t)}$ will grow to surpass 0 in a limited number of iterations, we can claim that there exists a constant $C'$, such that for the limited decreasing period of $\mathbb{E}[(\sum_{l\in S_{n,k}^e}(\sigma_S^{(t)})_l^n)]$, we have $1\geq-\mathbb{E}(\ell_n'^{(t)})\geq\widetilde{C}$. Also, $m\geq\mathbb{E}[|\mathcal{U}_{k,n}^e(0))|]\geq m/8$ by Lemma 7, as well as the fact that $\mathbb{E}[\mathcal{W}_{k,n}^e(t)]$ will at least preserve the neurons of $\mathbb{E}[\mathcal{U}_{k,n}^e(0))]$ along the iterations, without being deactivated as discussed in Lemma 29. Also, we note that in this hypothesised decreasing period, the absolute value of the initially negative $\mathbb{E}[\sum_{i\in[m]}\mathbf{r}_i\beta_{O_{(i,\cdot)},k}^{(t)}]$ and initially positive $\beta_{Q,k}^{(t)}$ will all decreasing. Then by Condition 1 we see that the small initialization of MLP as well as the small regularization will make the decreasing order of $\beta_{Q,k}^{(t)}$ negligible, as

$$\max_k\{|-\lambda\beta_{Q,k}^{(0)}+\frac{4\beta_{Q,k}^{(0)}\|\boldsymbol{b}_k\|^4}{K_1}\sum_{e\in[\pm]}\sum_{i\in\mathbb{E}[\mathcal{W}_{k,n}^\pm(0)]}\mathbf{r}_i\cdot\beta_{O_{(i,\cdot)},k}^{(0)}\mathbb{E}[(\sum_{j\in S_{n,k}^+}(\sigma_S^{(0)})_j^n)(\sum_{j\in S_{n,k}^-}(\sigma_S^{(0)})_j^n)\ell_n'^{(t)}]|\}$$
$$\leq(\lambda+\frac{5\sigma_1\|\mathbf{u}\|^4\|\mathbf{q}\|}{32K_1}\sqrt{2\log(\frac{5Km}{\delta})}))\beta_{Q,k}^{(0)}$$
$$\leq O(1/C).$$

(59)

Here the second inequality is due to Eq.(58), (56) and the definition of the $\boldsymbol{b}_k$ in Eq.(27); the third inequality is by the condition $\lambda\leq(C\sigma_0/2\|\mathbf{u}\|^2)^{-1}$ and $\sigma_1\leq(C\sigma_0\|\mathbf{u}\|^4\|\mathbf{q}\|\sqrt{\log(5Km/\delta)}/K_1)^{-1}$. Therefore, it holds that during the decreasing period of $\beta_{Q,k}^{(t)}$ as well as the period where $\sum_{i\in\mathbb{E}[\mathcal{W}_{k,n}^\pm(t)]}\mathbf{r}_i\cdot\beta_{O_{(i,\cdot)},k}^{(t)}$ remain negative, we have

$$\sum_{i\in\mathbb{E}[\mathcal{W}_{k,n}^\pm(t+1)]}\mathbf{r}_i\cdot\beta_{O_{(i,\cdot)},k}^{(t+1)}\geq\sum_{i\in\mathbb{E}[\mathcal{W}_{k,n}^\pm(t)]}\mathbf{r}_i\cdot\beta_{O_{(i,\cdot)},k}^{(t)}+\frac{C_4\eta_0\|\boldsymbol{d}_k\|^2}{K_1}(2\frac{1}{1+e^{-2\beta_{Q,k}^{(0)\,2}/\|\boldsymbol{b}_k\|^2}}-1),$$

Here, by a appropriate chosen small $C_4$, we again ignore the regularization term at this period due to $\lambda=O((C\log(Km/\delta)\|\mathbf{q}\|)^{-1})$ for a large $C$ by Condition 1, and the impact of the learning rate is also controlled due to the slow quadratic decaying nature of $\eta_t'$ and a small initial $\eta_0\leq O(0.01C^{-1})$ by Condition 1, so as the changing amount of $1/(1+e^{-2\beta_{Q,k}^{(0)\,2}/\|\boldsymbol{b}_k\|^2})$ by Eq.(59).

Therefore, by Eq.(58) we have

$$\beta_{Q,k}^{(t+1)}\geq\beta_{Q,k}^{(t)}(1+\frac{C_4\eta_0\|\boldsymbol{b}_k\|^4}{K_1}\sum_{i\in\mathbb{E}[\mathcal{W}_{k,n}^\pm(t)]}\mathbf{r}_i\cdot\beta_{O_{(i,\cdot)},k}^{(t)}\cdot(\sum_{j\in S_{n,k}^+}(\sigma_S^{(t)})_j^n)(\sum_{j\in S_{n,k}^-}(\sigma_S^{(t)})_j^n))\qquad(60)$$

where the inequality is by the negative nature of $\sum_{i\in\mathbb{E}[\mathcal{W}_{k,n}^\pm(t)]}\mathbf{r}_i\cdot\beta_{O_{(i,\cdot)},k}^{(t)}$, and the decaying nature of $\eta_t$ and Eq.(58). Now we can see that there exists two surrogate sequences $\underline{\beta_{Q,k}^{(t)}}$ and $\underline{\sum_{i\in\mathbb{E}[\mathcal{W}_{k,n}^\pm(t)]}\mathbf{r}_i\cdot e\beta_{O_{(i,\cdot)},k}^{(t)}}$ as the lower bound sequence of the $\beta_{Q,k}^{(t)}$ and $\sum_{i\in\mathbb{E}[\mathcal{W}_{k,n}^\pm(t)]}\mathbf{r}_i\cdot e\beta_{O_{(i,\cdot)},k}^{(t)}$. These two former sequences's initial values are taken as the lower bounds of the latter two ($\sigma_0\|\boldsymbol{b}_k\|^2$ and $-\sqrt{2\log(5Km/\delta)}\cdot\frac{5\sigma_1\|\boldsymbol{d}_k\|}{16}$), and their update rule are

$$\underline{\beta_{Q,k}^{(t+1)}}=\underline{\beta_{Q,k}^{(t)}}+\underline{\beta_{Q,k}^{(t)}}\frac{C_4\eta_0\|\boldsymbol{b}_k\|^4}{K_1}\sum_{i\in\mathbb{E}[\mathcal{W}_{k,n}^\pm(t)]}\mathbf{r}_i\cdot\beta_{O_{(i,\cdot)},k}^{(t)}\cdot(\sum_{j\in S_{n,k}^+}(\sigma_S^{(t)})_j^n)(\sum_{j\in S_{n,k}^-}(\sigma_S^{(t)})_j^n),$$

$$\underline{\sum_{i\in\mathbb{E}[\mathcal{W}_{k,n}^\pm(t+1)]}\mathbf{r}_i\cdot e\beta_{O_{(i,\cdot)},k}^{(t+1)}}=\underline{\sum_{i\in\mathbb{E}[\mathcal{W}_{k,n}^\pm(t)]}\mathbf{r}_i\cdot e\beta_{O_{(i,\cdot)},k}^{(t)}}$$
$$+\frac{C_4\eta_0\|\boldsymbol{d}_k\|^2}{K_1}(2\frac{1}{1+e^{-2\beta_{Q,k}^{(0)\,2}/\|\boldsymbol{b}_k\|^2}}-1).$$

Then by Lemma 11, let $a=\frac{C_4\eta_0\|\boldsymbol{b}_k\|^4}{K_1}$, $b=\frac{C_4\eta_0\|\boldsymbol{d}_k\|^2}{K_1}(2\frac{1}{1+e^{-2\beta_{Q,k}^{(0)\,2}/\|\boldsymbol{b}_k\|^2}}-1)$, we have the maximum iterations $T_1=\frac{-z(0)(1+e^{-2y(0)^2})}{b(1-e^{-2y(0)^2})}=\frac{\sigma_1 K_1\gamma\sqrt{10\log(5Km/\delta)}(1+e^{-2\sigma_0^2\|\boldsymbol{b}_k\|^2})}{2C_4\|\boldsymbol{d}_k\|(1-e^{-2\sigma_0^2\|\boldsymbol{b}_k\|^2})}$, set $C_3=\sqrt{10}/(2C_4)$

we obtain the $T_1$ in the lemma. The lower bound of $\beta_{Q,k}^{(t)}$ along the decreasing period as

$$\underline{\beta_{Q,k}} = \sigma_0 \|\boldsymbol{b}_k\|^2 e^{-\log(5Km/\delta)\frac{25\sigma_1^2\|\mathbf{b}_k\|^4(1+e^{-2\sigma_0^2\|\mathbf{b}_k\|^2})}{1024(1-e^{-2\sigma_0^2\|\mathbf{b}_k\|^2})}},$$

Utilizing the scale bounding property $(-\kappa_{\boldsymbol{x}}+1)/2\|\mathbf{u}\|^2 \leq \|\boldsymbol{b}_{k_1}\|^2 < \|\mathbf{u}\|^2/2$ in Eq. (27) and (36), we can denoted the lower bound of all $\beta_{Q,k}^{(t)} = \beta_{K,k}^{(t)}$ for $\forall k \in [K_1]$ as $\underline{\beta_{QK}^-}$, which can be given as

$$\underline{\beta_{QK}^-} = \frac{\sigma_0(1-\kappa_{\boldsymbol{x}})\|\mathbf{u}\|^2}{2} e^{-\log(5Km/\delta)\frac{\sigma_1^2\|\mathbf{u}\|^4(1+e^{-\sigma_0^2\|\mathbf{u}\|^2})}{(1-e^{-\sigma_0^2\|\mathbf{u}\|^2})}},$$

Recall $\sigma_S^*$ is defined as

$$\sigma_S^* := \frac{1}{1+e^{-2^{-1}\sigma_0{}^2(1-\kappa_{\boldsymbol{x}})^2\|\mathbf{u}\|^4 e^{-2\sigma_1{}^2\log(5Km/\delta)\frac{\|\mathbf{u}\|^4(1+e^{-\sigma_0^2\|\mathbf{u}\|^2})}{(1-e^{-\sigma_0^2\|\mathbf{u}\|^2})}}}},$$

which is actually can be written as

$$\sigma_S^* = \frac{1}{1+e^{-2\underline{\beta_{QK}^-}{}^2}}.$$

Therefore, we see that $\sigma_S^*$ is the lower bound of $\min_{t\in[T^*],k\in[K_1]}\{\underset{n\in\mathcal{D}_S}{\mathbb{E}}[\sum_{j\in S_{n,k}^{y_{Sn}}}(\sigma_S^{(t)})_j^n]\}$.

$\square$

**Remark 4.** *As we see that in Lemma 33, we require that the lower bound given in Eq.(53) depends that the values of $\mathbb{E}[\mathbf{r}_i(2\sum_{l\in S_{n,k}^e}(\sigma_S^{(t)})_l^n - 1)\beta_{O_{(i,\cdot)},k}^{(t)})]$ surpasses $\kappa$, which naturally says that the value of $\mathbb{E}_{i\in\mathcal{U}_{k,n}^e(0))}[\mathbf{r}_i\beta_{O_{(i,\cdot)},k}^{(t)}]$ should surpass $\kappa$ since $\mathbb{E}[(2\sum_{l\in S_{n,k}^e}(\sigma_S^{(t)})_l^n - 1)] \leq 1$. Therefore $\mathbb{E}_{i\in\mathcal{U}_{k,n}^e(0))}[\mathbf{r}_i\beta_{O_{(i,\cdot)},k}^{(t)}]$ should surpass $0$ at $\hat{T}$ since $\kappa > 0$, which indicates that $\hat{T} > T_1$. We see that the initial period $t \leq T_1$ is where $\mathbb{E}_{i\in\mathcal{U}_{k,n}^e(0))}[\mathbf{r}_i\beta_{O_{(i,\cdot)},k}^{(t)}]$ grow to surpass the initial scale, whose upper bound is $\kappa/8$ by the definition of $\kappa$.*

### I.1.3  Period 2: Increasing Priod of Correct Attention Score

This period's analysis is based on Period 1 in Section I.1.2, or a good initialization such that

$$\sum_{i\in[m]}\mathbf{r}_i\beta_{O_{(i,\cdot)},k}^{(0)} > 0.$$

**Lemma 35.** *Under Condition 1, consider the duration after $T_1$ in Lemma 34, then for $\forall k \in [K_1]$, consider the period $T_1 \leq t \leq T_2 = C_5\min\{\frac{1+\gamma}{\lambda}, \frac{\|\mathbf{u}\|\|\mathbf{q}\|}{\lambda K_1\sqrt{m}}\}$, where $C_5$ is a small constant. Then the following holds that*

- *We have $\underline{y}(t)$, $\overline{y}(t)$, $\underline{z}(t)$, $\overline{z}(t)$ be the lower and upper bounds of the increasing $\beta_{Q,k}^{(t)} = \beta_{K,k}^{(t)}$ and $\sum_{i\in\mathbb{E}[\mathcal{W}_{k,n}^{\pm}(t)]}\mathbf{r}_i\cdot e\beta_{O_{(i,\cdot)},k}^{(t)}$ respectively. That is, there exists positive constants $c_{3-6}$, for*

$$\underline{a} = \frac{c_3(1-\kappa_{\boldsymbol{x}})\|\mathbf{u}\|^4}{\lambda\gamma K_1}, \overline{a} = \frac{c_4\|\mathbf{u}\|^4}{\lambda\gamma K_1}, \underline{b} = \frac{c_5(1-\kappa_{\boldsymbol{y}})\|\mathbf{q}\|^2}{\lambda\gamma K_1}), \overline{b} = \frac{c_6\|\mathbf{q}\|^2}{\lambda\gamma K_1}), c' = \widetilde{C}, \text{ it holds that}$$

$$\underline{y}(t) \leq \beta_{Q,k}^{(t)} = \beta_{K,k}^{(t)} \leq \overline{y}(t), \quad \underline{z}(t) \leq \sum_{i\in\mathbb{E}[\mathcal{W}_{k,n}^{\pm}(t)]}\mathbf{r}_i\cdot e\beta_{O_{(i,\cdot)},k}^{(t)} \leq \overline{z}(t),$$

*for all $t \geq T_1$. Here, $\overline{y}(t)$, $\underline{y}(t)$, $\overline{z}(t)$, $\underline{z}(t)$ are the unique solutions of the following ODE System respectively*

$$\frac{1}{2}(\mathrm{Ei}(2\underline{y}(t)^2) + \mathrm{Ei}(-2\underline{y}(t)^2) + 4\log(\underline{y}(t))) = \underline{abc}'^2(2\sigma_S^* - 1)\frac{(t-t_1)^2}{2}$$

$$+ \frac{1}{2}(\mathrm{Ei}(\log(\frac{\sigma_S^*}{1-\sigma_S^*})) + \mathrm{Ei}(\log(\frac{1-\sigma_S^*}{\sigma_S^*}))) + 4\log(\underline{\beta_{QK}^-}),$$

$$\underline{z}(t) = \underline{bc}'(2\sigma_S^* - 1)(t - T_1),$$

$$\frac{1}{2}(\mathrm{Ei}(2\overline{y}(t)^2) + \mathrm{Ei}(-2\overline{y}(t)^2) + 4\log(\overline{y}(t))) = \frac{\overline{a}\overline{b}t^2}{2} + \overline{a}\frac{\kappa}{8}t$$

$$+ \frac{1}{2}(\mathrm{Ei}(\frac{\sigma_0^2\|\mathbf{u}\|^4}{2}) + \mathrm{Ei}(-2\frac{\sigma_0^2\|\mathbf{u}\|^4}{2})) + 4\log(\sigma_0/2\|\mathbf{u}\|^2),$$

$$\overline{z}(t) = \overline{b}t + \frac{\kappa}{8},$$

*where*

$$\mathrm{Ei}(x) = \int_{-\infty}^x \frac{e^t}{t}\mathrm{d}t = \gamma_{Euler} + \ln x + \exp(x/2)\sum_{n=1}^{\infty}\frac{(-1)^{n-1}x^n}{n!2^{n-1}}\sum_{k=0}^{\lfloor (n-1)/2 \rfloor}\frac{1}{2k+1}.$$

- *For some limited constant $\triangle$ such that $\exists \overline{\triangle}$, $\sigma_S^* < \triangle \leq \overline{\triangle} < 1$. Then the $\beta_{Q,k}^{(t)} = \beta_{K,k}^{(t)}$ will grow to make the correct attention score $\underset{n \in \mathcal{V}_k^{yS_n}}{\mathbb{E}}[\sum_{j \in S_{n,k}^{yS_n}}(\sigma_S^{(t)})_j^n]$ achieve the $\triangle$ in at least a $\triangle(1-\triangle)$ scaled Gaussian rate such that*

$$\beta_{Q,k}^{(t)} \geq \exp(\frac{\underline{abc}'\triangle(1-\triangle)(2\sigma_S^* - 1)}{2}(t-T_1)^2 + \log(\underline{\beta_{QK}^-})).$$

*Proof.* By Remark 4, we see that at the initial phase during $t \geq T_1$, we have $\sum_{i \in [m]}\mathbf{r}_i\beta_{O_{(i,\cdot)},k}^{(0)} \leq \kappa/8$, and thus by Eq.(55) in Lemma 30 we see that $\alpha_{O_{(i,\cdot)},k}^{(0)} \leq \hat{C}\frac{\|\mathbf{c}_k\|^2}{\sigma_S^*\|\mathbf{d}_k\|^2}$. This indicates that $\mathbb{E}[\mathbf{A}_t^{k,e}] \leq \Theta(\alpha)$ and thus by Lemmar 31 we see that the scale of $|\underset{n \in \mathcal{V}_k^e}{\mathbb{E}}[ef(\mathbf{E}(S); \mathbb{E}(\Psi^{(t)}))]|$ is also $\Theta(\kappa)$. This suggest that there still exists a constant $\widetilde{C}$, during a certain amount of subsequent iterations we would still have that $\widetilde{C} \leq -\mathbb{E}[\ell'(t)] \leq 1$. Also, $m \geq \mathbb{E}[|\mathcal{U}_{k,n}^e(0)|] \geq m/8$ by Lemma 7, as well as the fact that $\mathbb{E}[\mathcal{W}_{k,n}^e(t)]$ will at least preserve the neurons of $\mathbb{E}[\mathcal{U}_{k,n}^e(0))]$ along the iterations, without being deactivated as discussed in Lemma 29. In addition, recall that in this first stage we also can control the impact of regularization and decaying learning rate by a small $\lambda$ and a big $\gamma$ by the sufficiently large $C$ in Condition 1, which indicates we now have

$$\beta_{Q,k}^{(t+1)} = \beta_{Q,k}^{(t)} + \Theta\Big(\beta_{Q,k}^{(t)}\frac{C_4\eta_0\|\mathbf{b}_k\|^4}{K_1}\sum_{i \in \mathbb{E}[\mathcal{W}_{k,n}^{\pm}(t)]}\mathbb{E}[\mathbf{r}_i\beta_{O_{(i,\cdot)},k}^{(t)} \cdot \ell'(t)\frac{1}{1+e^{-2\beta_{Q,k}^{(t)^2}/\|\mathbf{b}_k\|^2}}\frac{1}{1+e^{2\beta_{Q,k}^{(t)^2}/\|\mathbf{b}_k\|^2}}]\Big),$$

and

$$\sum_{i \in \mathbb{E}[\mathcal{W}_{k,n}^{\pm}(t+1)]}\mathbf{r}_i \cdot \mathbb{E}[e\beta_{O_{(i,\cdot)},k}^{(t+1)}] = \sum_{i \in \mathbb{E}[\mathcal{W}_{k,n}^{\pm}(t)]}\mathbf{r}_i \cdot \beta_{O_{(i,\cdot)},k}^{(t)}$$

$$+ \Theta\Big(\frac{C_4\eta_0\|\mathbf{d}_k\|^2}{K_1}\mathbb{E}[\ell'(t)(2\frac{1}{1+e^{-2\beta_{Q,k}^{(t)^2}/\|\mathbf{b}_k\|^2}} - 1)]\Big).$$

By Lemma 12, we see that the iteration satisfies the ODE System 2 with a positive initialization, where the parameters in Lemma 12 are $\ell'_t = -\mathbb{E}[\ell'(t)]$, $a = \Theta(\frac{C_4\eta_0\|\mathbf{b}_k\|^4}{K_1})$, $b = \Theta(\frac{C_4\eta_0\|\mathbf{d}_k\|^2}{K_1})$, $c' = \widetilde{C}$. Then by solving the coupled ODE systems, collaborating the scale bounding property $(-\kappa_x + 1)/2\|\mathbf{u}\|^2 \leq \|\mathbf{b}_{k_1}\|^2 < \|\mathbf{u}\|^2/2$, $-\kappa_y + 1/2\|\mathbf{q}\|^2 \leq \|\mathbf{d}_{k_1}\|^2 < \|\mathbf{q}\|^2/2$ in Eq. (27) and (36), as well as the Comparison Theorem with some constants $c_{3-6}$, we can have upper and lower bound of $\beta_{Q,k}^{(t)}$ and $\sum_{i \in \mathbb{E}[\mathcal{W}_{k,n}^{\pm}(t)]}$, which is the result in this lemma.

For the second result, given the $\triangle$, we can directly have a lower bound ODE $\underline{y}_{\triangle}(t)$ to be the lower bound of the $\beta_{Q,k}^{(t)}$ via Comparison Theorem, where $\underline{y}_{\triangle}(t)$ satisfies

$$\underline{y}'(t) \geq \underline{y}'_{\triangle}(t) = \underline{abc}'\triangle(1-\triangle)(2\sigma_S^* - 1))(t-T_1)\underline{y}_{\triangle}(t) \Rightarrow$$

$$\underline{y}(t) \geq \underline{y}_{\triangle}(t) = \exp(\frac{\underline{abc}'\triangle(1-\triangle)(2\sigma_S^* - 1)}{2}(t-T_1)^2 + \log(\underline{\beta_{QK}^-})),$$

where the inequality holds by the decaying nature of $g(x) = x(1-x)$ when $x > 1/2$. The proof is completed.

$\square$

**Lemma 36.** *(Asymptotic Property 1). In the first stage, the growing of $\beta_{Q,k}^{(t)} = \beta_{K,k}^{(t)}$ as well as the attention score enjoys the asymptotic property that*

$$\lim_{t\to+\infty} \frac{\mathbb{E}[\beta_{Q,k}^{(t)\,2}]}{\log(t)} = \Theta(1), \quad \lim_{t\to+\infty} \frac{\mathbb{E}[\mathop{\mathbb{E}}\limits_{n\in\mathcal{V}_k^{y_{S_n}}} [\sum_{j\in S_{n,k}^{y_{S_n}}} (\sigma_S^{(t)})_j^n]]}{\frac{t^4}{1+t^4}} = \Theta(1).$$

*Proof.* By the asymptotic property of $\mathrm{Ei}(x)$

$$\lim_{x\to+\infty} \frac{\mathrm{Ei}(x) + \mathrm{Ei}(-x)}{\frac{\exp(x)}{x}} = 1.$$

This suggest that when $y(t) \geq \underline{y}(t)$ is close to infinity, the lower bound ODE in Lemma 35 will approximately satisfies the following

$$\frac{\exp(2\underline{y}(t)^2)}{4\underline{y}(t)^2} + 2\log(\underline{y}(t)) \approx \underline{a}\underline{b}c'^2(2\sigma_S^* - 1)\frac{t^2}{2} + \mathrm{const},$$

This suggest that roughly

$$\lim_{t\to+\infty} \underline{y}(t)^2 / \log(t^2) = \Theta(1).$$

Then we see that as $y(t)$ goes to infinity, we have a lower bound

$$\lim_{t\to+\infty} \log\Big(\frac{\mathbb{E}[\sum_{l\in S_{n,k}^e} (\sigma_S^{(t)})_l^n]}{1 - \mathbb{E}[\sum_{l\in S_{n,k}^e} (\sigma_S^{(t)})_l^n]}\Big)/2\log(t^2) = \Theta(1) \Rightarrow \lim_{t\to+\infty} \mathbb{E}[\sum_{l\in S_{n,k}^e} (\sigma_S^{(t)})_l^n]/\frac{t^4}{1+t^4} = \Theta(1).$$

On the other hand, obtaining an upper bound over $y(t)$ is relatively easy. Since we have $2 + e^{-2\underline{y}(t)^2} + e^{2\underline{y}(t)^2} \leq 4$ and $(1 - e^{-2\underline{y}(t)^2})/(1 + e^{-2\underline{y}(t)^2}) \leq 1$, which gives the upper bound ODE over attention and MLP considering $z(0) > 0$

$$\frac{1}{2}\big(\mathrm{Ei}(2\overline{y}(t)^2) + \mathrm{Ei}(-2\overline{y}(t)^2) + 4\log(\overline{y}(t))\big) = \frac{\overline{a}\overline{b}t^2}{2} + \overline{a}\frac{\kappa}{8}t + \mathrm{const}.$$

$$\overline{z}(t) = bt + \mathrm{const},$$

where the term "const" ensure that $\overline{y}(0) = y(0)$. The asymptotic property of this ODE system is the same as the one of lower bound ODE. Then consider $t, y(t)$ both go to infinity, we have some $\hat{c}$ such that

$$\lim_{t\to+\infty} \mathbb{E}[\sum_{l\in S_{n,k}^e} (\sigma_S^{(t)})_l^n]/\frac{(t + \hat{c}t^{1/2})^4}{1 + (t + \hat{c}t^{1/2})^4} = \lim_{t\to+\infty} \mathbb{E}[\sum_{l\in S_{n,k}^e} (\sigma_S^{(t)})_l^n]/\frac{t^4}{1+t^4} = 1.$$

$\square$

## I.2   Second Stage: Regularizing the Model

As the $\beta_{Q,k}^{(t)} = \beta_{K,k}^{(t)}$ and $e\beta_{O_{(i,\cdot)},k}^{(t)}$ are continually growing up, we see that the decaying $-\mathbb{E}[\ell'(t)]$, as well as the decaying attention score products $(\sum_{j\in S_{n,k}^+} (\sigma_S^{(t)})_j^n)(1 - \sum_{j\in S_{n,k}^+} (\sigma_S^{(t)})_j^n)$ is becoming feeble and feeble, under which we can no longer ignore the regularization term safely when estimating the coefficient gradient dynamics. However, although the regularization can prevent the coefficients from growing, it will maintain their scales without decreasing them.

**Lemma 37.** *Under Condition 1, consider* $e = \mathbb{E}[y_{S_n}]$ *for all* $t \in [T_2, T^*]$ *it holds that*

$$e\beta_{O_{(i,\cdot)},k}^{(t)} = O(\frac{\|\mathbf{q}\|^2}{\lambda m K_1}),$$

$$e\beta_{O_{(i,\cdot)},k}^{(T^*)} = \Theta(\frac{\sigma_S^{*2}(1-\kappa_y)^2}{(1+\kappa_y)^2}\log(\frac{e^{-\kappa}\|\mathbf{q}\|^2(2\sigma_S^* - 1)}{\lambda m K_1})),$$

$$\beta_{Q,k}^{(t)} = O(\sqrt{\|\mathbf{u}\|\log(\frac{\|\mathbf{u}\|^2\|\mathbf{q}\|^2}{\lambda^2 m K_1^2})}),$$

$$\beta_{Q,k}^{(T^*)} = \Theta(\|\mathbf{u}\|\sqrt{\log(\frac{\|\mathbf{u}\|^2\sigma_S^{*2}(1-\kappa_y)^2}{\lambda K_1(1+\kappa_y)^2}\log(\frac{e^{-\kappa}\|\mathbf{q}\|^2(2\sigma_S^* - 1)}{\lambda m K_1}))}),$$

$$\mathbb{E}[(\sum_{j \in S_{n,k}^e}(\sigma_S^{(t)})_j^n)] = O(\frac{1}{1 + \frac{\lambda^2 m K_1^2}{2\|\mathbf{u}\|^2\|\mathbf{q}\|^2}}),$$

$$\mathbb{E}[(\sum_{j \in S_{n,k}^e}(\sigma_S^{(T^*)})_j^n)] = \Theta(\frac{1}{1 + \frac{\lambda K_1(1+\kappa_y)^2}{\|\mathbf{u}\|^2\sigma_S^{*2}(1-\kappa_y)^2}\log^{-1}(\frac{e^{-\kappa}\|\mathbf{q}\|^2(2\sigma_S^* - 1)}{\lambda m K_1})}),$$

*where* $e\beta_{O_{(i,\cdot)},k}^{(t)}$ *represents* $m\mathbf{r}_i\beta_{O_{(i,\cdot)},k}^{(t)}$. *That is, we consider the positive growth of* $\mathbb{E}[\beta_{O_{(i,\cdot)},k}^{(t)}]$.

*Proof.* We will prove the desired argument based on the following induction hypothesis:

$$e\beta_{O_{(i,\cdot)},k}^{(t)} = O(\frac{\|\mathbf{q}\|^2}{\lambda m K_1}),$$

$$\beta_{Q,k}^{(t)} = O(\|\mathbf{u}\|\sqrt{\log(\frac{2\|\mathbf{u}\|^2\|\mathbf{q}\|^2}{\lambda^2 m K_1^2})}),$$

We split the situations into two cases:

(i). $e\beta_{O_{(i,\cdot)},k}^{(t)} \leq \Theta(\frac{\sigma_S^{*2}(1-\kappa_y)^2}{(1+\kappa_y)^2}\log(\frac{e^{-\kappa}\|\mathbf{q}\|^2(2\sigma_S^* - 1)}{\lambda m K_1}))$,

and $\beta_{Q,k}^{(t)} \leq \Theta(\|\mathbf{u}\|\sqrt{\log(\frac{\|\mathbf{u}\|^2\sigma_S^{*2}(1-\kappa_y)^2}{\lambda K_1(1+\kappa_y)^2}\log(\frac{e^{-\kappa}\|\mathbf{q}\|^2(2\sigma_S^* - 1)}{\lambda m K_1}))})$;

(ii). $e\beta_{O_{(i,\cdot)},k}^{(t)} \geq \frac{\|\mathbf{q}\|^2}{2\lambda m K_1}$,

and $\beta_{Q,k}^{(t)} \geq \|\mathbf{u}\|\sqrt{\frac{1}{2}\log(\frac{6\|\mathbf{u}\|^2\|\mathbf{q}\|^2}{\lambda^2 m K_1^2})} \Rightarrow \mathbb{E}[(\sum_{j \in S_{n,k}^+}(\sigma_S^{(t)})_j^n)] \geq \frac{1}{1 + \frac{\lambda^2 m K_1^2}{6\|\mathbf{u}\|^2\|\mathbf{q}\|^2}}$. Easy to note that the scales'

orders of the case (i)'s quantities are less than those of case (ii), thus this split is plausible.

Recall

$$\beta_{O_{(i,\cdot)},k}^{(t+1)} = (1 - \eta_t\lambda)\beta_{O_{(i,\cdot)},k}^{(t)} - \eta_t\frac{\|\mathbf{d}_k\|^2}{2K_1}\sum_{e \in [\pm]}\mathbf{r}_i\mathop{\mathbb{E}}_{n \in \mathcal{V}_k^e}[\ell_n'^{(t)}\mathbb{1}_{O_{(i)}}^{n}{}^{(t)}(\sum_{l \in S_{n,k}^e}(\sigma_S^{(t)})_l^n - \sum_{l \in S_{n,k}^{-e}}(\sigma_S^{(t)})_l^n)].$$

Then it's easy to check that for case (i), as by Lemma 30 we see that the magnitude of $\mathbb{E}[|\alpha_{O_{(i,\cdot)},k}^{(t)}|]$ is controlled by some $\hat{C}\sigma_S^{*-2}(1+\kappa_y)^2(1-\kappa_y)^{-2}\beta_{O_{(i^*,\cdot)},k}^{(t)} \geq \hat{C}$. That means that the term $\mathbb{E}[\mathbf{A}_t^{k,e}]$ can be controlled by its contributor $\Theta(e\beta_{O_{(i,\cdot)},k}^{(t)})$. Then we have

$$\Theta(e\beta_{O_{(i,\cdot)},k}^{(t)}/\mathbb{E}[-\ell_n'^{(t)}]) \leq \Theta(e\beta_{O_{(i,\cdot)},k}^{(t)}(1 + e^\kappa e^{\sigma_S^{*-2}(1+\kappa_y)^2(1-\kappa_y)^{-2}e\beta_{O_{(i,\cdot)},k}^{(t)}}))$$

$$\leq \Theta(e^\kappa e^{\sigma_S^{*-2}(1+\kappa_y)^2(1-\kappa_y)^{-2}e\beta_{O_{(i,\cdot)},k}^{(t)}})$$

$$\leq \Theta(\frac{\|\mathbf{q}\|^2(2\sigma_S^* - 1)}{\lambda m K_1}),$$

where the first inequality is by the definition of $\mathbb{E}[-\ell_n'^{(t)}]$ (similar to the techniques in Lemma 32) and the definition of $\mathbb{E}[\mathbf{A}_t^{k,e}]$; the second inequality is by $g(x) = x < e^x - 1$ as well as $\sigma_S^{*-2}(1+\kappa_y)^2(1-\kappa_y)^{-2} > 1$; the second inequality is by the case (i) hypothesis. Then we would have

$$\lambda e\beta_{O_{(i,\cdot)},k}^{(t)} \leq \Theta(-\frac{\|\mathbf{d}_k\|^2}{2K_1}\sum_{e \in [\pm]}\mathbb{E}[\mathbf{r}_i\mathop{\mathbb{E}}_{n \in \mathcal{V}_k^e}[\ell_n'^{(t)}\mathbb{1}_{O_{(i)}}^{n}{}^{(t)}(\sum_{l \in S_{n,k}^e}(\sigma_S^{(t)})_l^n - \sum_{l \in S_{n,k}^{-e}}(\sigma_S^{(t)})_l^n)]]). \quad (61)$$

Thus the growing of $e\beta_{O_{(i,\cdot)},k}^{(t)}$ would be non-degenerated: $\mathbb{E}[e\beta_{O_{(i,\cdot)},k}^{(t+1)}] \geq \Theta(e\beta_{O_{(i,\cdot)},k}^{(t)})$, which directly suggest $\mathbb{E}[\beta_{O_{(i,\cdot)},k}^{(T^*)}] = \Theta(\frac{\sigma_S^{*2}(1-\kappa_{\boldsymbol{y}})^2}{(1+\kappa_{\boldsymbol{y}})^2}\log(\frac{e^{-\kappa}\|\mathbf{q}\|^2(2\sigma_S^*-1)}{\lambda m K_1}))$ holds since $T^*$ is the maximum admissible iterations.

Similarly, for the $\beta_{Q,k}^{(t)}$ in case (i), first recall that

$$\beta_{Q,k}^{(t+1)} = (1 - \eta_t\lambda)\beta_{Q,k}^{(t)} - \frac{4\eta_t\beta_{K,k}^{(t)}\|\boldsymbol{b}_k\|^4}{K_1} \sum_{e\in[\pm]}\sum_{i\in[m]} \mathbf{r}_i\beta_{O_{(i,\cdot)},k}^{(t)} \underset{n\in\mathcal{V}_k^e}{\mathbb{E}}[\ell_n'^{(t)}\mathbb{1}_{O_{(i)}}^{n}{}^{(t)}(\sum_{j\in S_{n,k}^+}(\sigma_S^{(t)})_j^n)$$

$$(1 - \sum_{j\in S_{n,k}^+}(\sigma_S^{(t)})_j^n)],$$

then, as we see that

$$\mathbb{E}[(\sum_{j\in S_{n,k}^+}(\sigma_S^{(t)})_j^n)(1 - \sum_{j\in S_{n,k}^+}(\sigma_S^{(t)})_j^n)] = \frac{1}{1+e^{-2\beta_{Q,k}^{(t)}{}^2/\|\boldsymbol{b}_k\|^2}}\frac{1}{1+e^{2\beta_{Q,k}^{(t)}{}^2/\|\boldsymbol{b}_k\|^2}}$$

$$= \frac{1}{2 + e^{2\beta_{Q,k}^{(t)}{}^2/\|\boldsymbol{b}_k\|^2} + e^{-2\beta_{Q,k}^{(t)}{}^2/\|\boldsymbol{b}_k\|^2}}$$

$$\leq \Theta(\frac{1}{2 + \frac{\|\mathbf{u}\|^2\sigma_S^{*2}(1-\kappa_{\boldsymbol{y}})^2}{\lambda K_1(1+\kappa_{\boldsymbol{y}})^2}\log(\frac{e^{-\kappa}\|\mathbf{q}\|^2(2\sigma_S^*-1)}{\lambda m K_1})})$$

$$= \Theta(\frac{\|\mathbf{u}\|^2\sigma_S^{*2}(1-\kappa_{\boldsymbol{y}})^2}{\lambda K_1(1+\kappa_{\boldsymbol{y}})^2}\log(\frac{e^{-\kappa}\|\mathbf{q}\|^2(2\sigma_S^*-1)}{\lambda m K_1}))^{-1},$$

where the first inequality is by the definition of $\mathbb{E}[(\sum_{j\in S_{n,k}^\pm}(\sigma_S^{(t)})_j^n)]$ and the induction hypothesis; the second inequality is by the small $\lambda$ by Condition 1 with a sufficiently large $C$. Then we see that

$$\Theta(-\frac{4\|\boldsymbol{b}_k\|^2}{K_1}\mathbb{E}[\sum_{e\in[\pm]}\sum_{i\in[m]}\mathbf{r}_i\beta_{O_{(i,\cdot)},k}^{(t)}\underset{n\in\mathcal{V}_k^e}{\mathbb{E}}[\ell_n'^{(t)}\mathbb{1}_{O_{(i)}}^{n}{}^{(t)}(\sum_{j\in S_{n,k}^+}(\sigma_S^{(t)})_j^n)(\sum_{j\in S_{n,k}^-}(\sigma_S^{(t)})_j^n)]]) \geq \Theta(\lambda).$$

Here the inequality is by the case (i) hypothesis upon $e\beta_{O_{(i,\cdot)},k}^{(t)}$, $-\mathbb{E}[\ell'(t)] \leq 1$, and

$$\mathbb{E}[\sum_{e\in[\pm]}\sum_{i\in[m]}\mathbf{r}_i\underset{n\in\mathcal{V}_k^e}{\mathbb{E}}[\mathbb{1}_{O_{(i)}}^{n}{}^{(t)}]] = \sum_{e\in[\pm]}\mathbb{E}[\sum_{i\in\mathcal{W}_{k,n}^e(t)}/m] \leq 2.$$

Thus we see that the growing of $\beta_{Q,k}^{(t)}$ would also be non-degenerated: $\beta_{Q,k}^{(t+1)} \geq \Theta(\beta_{Q,k}^{(t)})$. This also directly validates that for the maximum admissible iterations $T^*$, it holds that

$$\beta_{Q,k}^{(T^*)} = \Theta(\|\mathbf{u}\|\sqrt{\log(\frac{\|\mathbf{u}\|^2\sigma_S^{*2}(1-\kappa_{\boldsymbol{y}})^2}{\lambda K_1(1+\kappa_{\boldsymbol{y}})^2}\log(\frac{e^{-\kappa}\|\mathbf{q}\|^2(2\sigma_S^*-1)}{\lambda m K_1}))}),$$

$$\mathbb{E}[(\sum_{j\in S_{n,k}^e}(\sigma_S^{(T^*)})_j^n)] = \Theta(\frac{1}{1 + \frac{\lambda K_1(1+\kappa_{\boldsymbol{y}})^2}{\|\mathbf{u}\|^2\sigma_S^{*2}(1-\kappa_{\boldsymbol{y}})^2}\log^{-1}(\frac{e^{-\kappa}\|\mathbf{q}\|^2(2\sigma_S^*-1)}{\lambda m K_1})}).$$

For case (ii), we directly check that

$$\lambda e\beta_{O_{(i,\cdot)},k}^{(t)} \geq -\frac{\|\boldsymbol{d}_k\|^2}{2K_1}\sum_{e\in[\pm]}\mathbb{E}[\mathbf{r}_i\underset{n\in\mathcal{V}_k^e}{\mathbb{E}}[\ell_n'^{(t)}\mathbb{1}_{O_{(i)}}^{n}{}^{(t)}(\sum_{l\in S_{n,k}^e}(\sigma_S^{(t)})_l^n - \sum_{l\in S_{n,k}^{-e}}(\sigma_S^{(t)})_l^n)]].$$

Here the inequality is by $-\mathbb{E}[\ell'(t)] \leq 1$ and

$$\mathbb{E}[(\sum_{l\in S_{n,k}^e}(\sigma_S^{(t)})_l^n - \sum_{l\in S_{n,k}^{-e}}(\sigma_S^{(t)})_l^n)] = [\mathbb{E}(2\sum_{l\in S_{n,k}^e}(\sigma_S^{(t)})_l^n - 1)] \leq 1.$$

As a result, by the gradient form we see that $\beta_{Q,k}^{(t+1)} \leq \beta_{Q,k}^{(t)}$, and thus we prove the induction proving goal $\mathbb{E}[e\beta_{O_{(i,\cdot)},k}^{(t+1)}] = O(\frac{\|\mathbf{q}\|^2}{\lambda m K_1})$.

Similarly, as we now have

$$\mathbb{E}[(\sum_{j \in S_{n,k}^+} (\sigma_S^{(t)})_j^n)(1 - \sum_{j \in S_{n,k}^+} (\sigma_S^{(t)})_j^n)] = \frac{1}{1 + e^{-2\beta_{Q,k}^{(t)}{}^2/\|\boldsymbol{b}_k\|^2}} \frac{1}{1 + e^{2\beta_{Q,k}^{(t)}{}^2/\|\boldsymbol{b}_k\|^2}}$$

$$= \frac{1}{2 + e^{2\beta_{Q,k}^{(t)}{}^2/\|\boldsymbol{b}_k\|^2} + e^{-2\beta_{Q,k}^{(t)}{}^2/\|\boldsymbol{b}_k\|^2}}$$

$$\geq \frac{1}{3} \frac{1}{1 + e^{2\beta_{Q,k}^{(t)}{}^2/\|\boldsymbol{b}_k\|^2}/3}$$

$$\geq \frac{1}{3} \frac{1}{1 + \frac{2\|\mathbf{u}\|^2\|\mathbf{q}\|^2}{\lambda^2 m K_1{}^2}},$$

$$= \Theta(\frac{\lambda^2 m K_1{}^2}{2\|\mathbf{u}\|^2\|\mathbf{q}\|^2}),$$

where the first inequality is by $e^{-2\beta_{Q,k}^{(t)}{}^2/\|\boldsymbol{b}_k\|^2} \leq 1$; the second inequality is by the induction hypothesis of case (ii); the last equality is by the small $\lambda$ in Condition 1 for a sufficiently large $C$. Then we observe that

$$\lambda \geq \Theta(-\frac{4\|\boldsymbol{b}_k\|^2}{K_1} \mathbb{E}[\sum_{e \in [\pm]} \sum_{i \in [m]} \mathbf{r}_i \beta_{O_{(i,\cdot)},k}^{(t)} \mathbb{E}_{n \in \mathcal{V}_k^e}[\ell_n'{}^{(t)} \mathbb{1}_{O_{(i)}}^n{}^{(t)}(\sum_{j \in S_{n,k}^+} (\sigma_S^{(t)})_j^n)(\sum_{j \in S_{n,k}^-} (\sigma_S^{(t)})_j^n)]]),$$

where the inequality is by $-\mathbb{E}[\ell'(t)] \leq 1$,

$$\mathbb{E}[\sum_{e \in [\pm]} \sum_{i \in [m]} \mathbf{r}_i \mathbb{E}_{n \in \mathcal{V}_k^e}[\mathbb{1}_{O_{(i)}}^n{}^{(t)}]] = \sum_{e \in [\pm]} \mathbb{E}[\sum_{i \in \mathcal{W}_{k,n}^e(t)} /m] \leq 2,$$

as well as the induction hypothesis upon $e\beta_{O_{(i,\cdot)},k}^{(t)}$ in case (ii). Thus $\beta_{Q,k}^{(t+1)} \leq \Theta(\beta_{Q,k}^{(t)})$, which support our proving goal in this induction process:

$$\mathbb{E}[\beta_{Q,k}^{(t+1)}] = O(\|\mathbf{u}\|\sqrt{\log(\frac{2\|\mathbf{u}\|^2\|\mathbf{q}\|^2}{\lambda^2 m K_1{}^2})}).$$

In addition, we can see that even if we suggest the MLP's $e\beta_{O_{(i,\cdot)},k}^{(t)}$ is growing in a fastest linear-level speed, it require at least $\Theta(\frac{1+\gamma}{\lambda})$ to reach the maximum admissible value $\frac{\|\mathbf{q}\|^2}{2\lambda m K_1}$. Meanwhile, we see that even when considering the fast speed of the increasing attention, by the asymptotic perperty 1 discussed in Lemma 36, we see that we still require $\Theta(\frac{\|\mathbf{u}\|\|\mathbf{q}\|}{\lambda K_1\sqrt{m}})$ to reach the highest admissible correct attention score $\frac{1}{1 + \frac{\lambda^2 m K_1{}^2}{2\|\mathbf{u}\|^2\|\mathbf{q}\|^2}}$.

Therefore, we can have some appropriately small constants $C_5$, and when the iteration number is more than $T_2 = C_5 \min\{\frac{1+\gamma}{\lambda}, \frac{\|\mathbf{u}\|\|\mathbf{q}\|}{\lambda K_1\sqrt{m}}\}$, we need to consider the impact of regularization. $\qquad\square$

**Lemma 38.** *The scale of the coefficients will finally be stabilized at a considerable level:*

$$\alpha_{O_{(i,\cdot)},k}^{(T^*)} \leq e\beta_{O_{(i,\cdot)},k}^{(T^*)} = \Theta(\log(\frac{\|\mathbf{q}\|^2}{m\lambda K_1})),$$

$$\beta_{Q,k}^{(T^*)} = \Theta(\|\mathbf{u}\|\sqrt{\log(\frac{\|\mathbf{u}\|^2}{\lambda K_1}\log(\frac{\|\mathbf{q}\|^2}{m\lambda K_1}))}),$$

$$\mathbb{E}[(\sum_{j \in S_{n,k}^e} (\sigma_S^{(T^*)})_j^n)] = \Theta(\frac{1}{1 + \frac{\lambda K_1}{\|\mathbf{u}\|^2}\log(\frac{m\lambda K_1}{\|\mathbf{q}\|^2})}).$$

*where $e\beta_{O_{(i,\cdot)},k}^{(t)}$ represents $m\mathbf{r}_i\beta_{O_{(i,\cdot)},k}^{(t)}$. That is, we consider the positive growth of $|\beta_{O_{(i,\cdot)},k}^{(t)}|$.*

*Proof.* Recall the last discussion in Lemma 29, we see that as $\mathbb{E}[(2\sum_{l \in S_{n,k}^e} (\sigma_S^{(t)})_l^n - 1)e\beta_{O_{(i,\cdot)},k}^{(t)}]$ getting larger and larger, it will finally reach the scale of $\alpha_{O_{(i,\cdot)},k}^{(t)}$, which has updated in a feeble speed controlled by initialization when the neuron fell into the neuron set $\mathbb{E}_{n \in \mathcal{V}_k^e}[\mathcal{U}_{k,n}^e(t) \cap (\mathcal{W}_{k,n}^{-e}(t) - \mathcal{U}_{k,n}^{-e}(t))]$. After $\alpha_{O_{(i,\cdot)},k}^{(t)} \leq$

$\mathbb{E}[(2\sum_{l\in S_{n,k}^e} (\sigma_S^{(t)})_l^n - 1)e\beta_{O_{(i,\cdot)},k}^{(t)}]$, the neuron would change into the neuron set $\mathbb{E}_{n\in\mathcal{V}_k^e}[\mathcal{U}_{k,n}^e(t) - (\mathcal{W}_{k,n}^{-e}(t) - \mathcal{U}_{k,n}^{-e}(t))]$. As such, the $\alpha_{O_{(i,\cdot)},k}^{(t)}$ would again increase at a normal speed, which is even faster than $e\beta_{O_{(i,\cdot)},k}^{(t)}$ due to the update rules and the fact that $\|\boldsymbol{c}_k\| > \|\boldsymbol{d}_k\|$. As such, the neuron set $\mathbb{E}_{n\in\mathcal{V}_k^e}[\mathcal{U}_{k,n}^e(t) - (\mathcal{W}_{k,n}^{-e}(t) - \mathcal{U}_{k,n}^{-e}(t))]$ would again fell back into the neuron set $\mathbb{E}_{n\in\mathcal{V}_k^e}[\mathcal{U}_{k,n}^e(t) \cap (\mathcal{W}_{k,n}^{-e}(t) - \mathcal{U}_{k,n}^{-e}(t))]$, where the update speed is again feeble. And it will increase until $\mathbb{E}[(2\sum_{l\in S_{n,k}^e} (\sigma_S^{(t)})_l^n - 1)e\beta_{O_{(i,\cdot)},k}^{(t)}]$ catch up.

Besides, we see that the expected attention score will grow up considerably, where we can see that there exist some constant $\widetilde{c} > 1/2$, $\widetilde{c} < \mathbb{E}[(\sigma_S^{(t)})_l^n] \leq 1$. As such, ultimately we have $\mathbb{E}[(\sum_{l\in S_{n,k}^e} (\sigma_S^{(t)})_l^n - \sum_{l\in S_{n,k}^{-e}} (\sigma_S^{(t)})_l^n)] = \Theta(1)$, $\alpha_{O_{(i,\cdot)},k}^{(T^*)} \leq e\beta_{O_{(i,\cdot)},k}^{(T^*)}$ and $\mathbb{E}[\mathbf{A}_t^{k,e}] = \Theta(\mathbb{E}[e\beta_{O_{(i,\cdot)},k}^{(t)}])$. Then following the process in Lemma 37 we can obtain the results. Here we omit this part since the proving procedure is the same to Lemma 37, despite we see $\mathbb{E}[(\sum_{l\in S_{n,k}^e} (\sigma_S^{(t)})_l^n - \sum_{l\in S_{n,k}^{-e}} (\sigma_S^{(t)})_l^n)] = \Theta(1)$ and $\mathbb{E}[\mathbf{A}_t^{k,e}] = \Theta(\mathbb{E}[e\beta_{O_{(i,\cdot)},k}^{(t)}])$. $\qquad\square$

Again, similar to Lemma 36, we can have asymptotic property when considering the decaying impact of the learning rate, as well as the cross-entropy loss. We directly provide the following two lemmas. Due to the similarity of the proof procedures of Lemma 35 and Lemma 36, we omit the proofs of the following two lemmas as well as the constant details for simplicity.

**Lemma 39.** *(Asymptotic Property 2). If we consider the impact of the decaying learning rate at the second stage and do not consider the decaying of cross-entropy loss, for some constants $\overline{c}, \overline{d}, \underline{c}, \underline{d}$ regarding $K_1, \gamma, \|\mathbf{u}\|, \|\mathbf{q}\|, \kappa_{\boldsymbol{x}}, \kappa_{\boldsymbol{y}}$, we will have*

$$\underline{y}(t) \leq \beta_{Q,k}^{(t)} = \beta_{K,k}^{(t)} \leq \overline{y}(t), \quad \underline{z}(t) \leq \sum_{i\in\mathbb{E}[\mathcal{W}_{k,n}^{\pm}(t)]} \mathbf{r}_i \cdot e\beta_{O_{(i,\cdot)},k}^{(t)} \leq \overline{z}(t),$$

*for all $t \geq T_2$. Here, $\overline{y}(t), \underline{y}(t), \overline{z}(t), \underline{z}(t)$ are the unique solutions of the following ODE System respectively*

$$\frac{1}{2}(\text{Ei}(2\underline{y}(t)^2) + \text{Ei}(-2\underline{y}(t)^2) + 4\log(\underline{y}(t))) = \overline{a}\left(\text{Li}_2\left(\frac{\overline{d} + \overline{c}t}{-\gamma\overline{c} - \overline{c} + \overline{d}}\right) + \log(\overline{c}t + \overline{d})\log\left(\frac{\overline{c}(\gamma + t)}{\overline{c}\gamma - \overline{d}}\right)\right)$$
$$+ \frac{1}{2}(\text{Ei}(\log(\frac{\sigma_S^*}{1 - \sigma_S^*})) + \text{Ei}(\log(\frac{1 - \sigma_S^*}{\sigma_S^*}))) + 4\log(\beta_{QK}^-),$$
$$\underline{z}(t) = \underline{b}c'(2\sigma_S^* - 1)(t - T_1),$$
$$= \underline{a}\left(\text{Li}_2\left(\frac{\underline{d} + \underline{c}t}{-\gamma\underline{c} - \underline{c} + \underline{d}}\right) + \log(\underline{c}t + \underline{d})\log\left(\frac{\underline{c}(\gamma + t)}{\underline{c}\gamma - \underline{d}}\right)\right)$$
$$+ \frac{1}{2}(\text{Ei}(\frac{\sigma_0^2\|\mathbf{u}\|^4}{2}) + \text{Ei}(-2\frac{\sigma_0^2\|\mathbf{u}\|^4}{2})) + 4\log(\sigma_0/2\|\mathbf{u}\|^2),$$
$$\overline{z}(t) = \overline{b}t + \frac{\kappa}{8},$$

*where*

$$\text{Li}_2(x) = -\int_0^x \frac{\ln(1 - t)}{t} dt.$$

*Additionally, we would have asymptotic property that*

$$\lim_{t\to+\infty} y(t)^2 = \lim_{t\to+\infty} \Theta(\log(\log^2(t))), \quad \lim_{t\to+\infty} \frac{\mathbb{E}[\mathbb{E}_{n\in\mathcal{V}_k^{y_{Sn}}}[\sum_{j\in S_{n,k}^{y_{Sn}}} (\sigma_S^{(t)})_j^n]]}{\frac{\log^4(t)}{1 + \log^4(t)}} = \Theta(1).$$

**Lemma 40.** *(Asymptotic Property 3). If we put our sight on the long period and take the decaying property of the $-\mathbb{E}[\ell'(t)]$ into account, for some constants $\overline{a}, \overline{b}, \overline{c}, \overline{d}, \overline{j}, \underline{a}, \underline{b}, \underline{c}, \underline{d}, \underline{j}$, we will have*

$$\underline{y}(t) \leq \beta_{Q,k}^{(t)} = \beta_{K,k}^{(t)} \leq \overline{y}(t), \quad \underline{z}(t) \leq \sum_{i\in\mathbb{E}[\mathcal{W}_{k,n}^{\pm}(t)]} \mathbf{r}_i \cdot e\beta_{O_{(i,\cdot)},k}^{(t)} \leq \overline{z}(t),$$

*for all $t \geq T_2$. Here, $\overline{y}(t)$, $\underline{y}(t)$, $\overline{z}(t)$, $\underline{z}(t)$ are the unique solutions of the following ODE System respectively*

$$\frac{1}{2}(\text{Ei}(2\underline{y}(t)^2) + \text{Ei}(-2\underline{y}(t)^2) + 4\log(\underline{y}(t))) = \overline{a}(\frac{\text{Li}_2(\frac{\overline{j}(\overline{d}+\overline{c}t)}{-\overline{b}-\overline{d}+\overline{d}\overline{j}})}{\overline{c}\overline{j}} + \frac{\log(\overline{c}t+\overline{d})\log(\frac{\overline{b}+\overline{c}\overline{j}t+\overline{d}}{\overline{b}+\overline{d}-\overline{d}\overline{j}})}{\overline{c}\overline{j}})$$

$$+\frac{1}{2}(\text{Ei}(\log(\frac{\sigma_S^*}{1-\sigma_S^*})) + \text{Ei}(\log(\frac{1-\sigma_S^*}{\sigma_S^*}))) + 4\log(\underline{\beta_{QK}^-}),$$

$$\underline{z}(t) = \underline{b}\underline{c}'(2\sigma_S^* - 1)(t - T_1),$$

$$= \underline{a}(\frac{\text{Li}_2(\frac{\underline{j}(\underline{d}+\underline{c}t)}{-\underline{b}-\underline{d}+\underline{d}\underline{j}})}{\underline{c}\underline{j}} + \frac{\log(\underline{c}t+\underline{d})\log(\frac{\underline{b}+\underline{c}\underline{j}t+\underline{d}}{\underline{b}+\underline{d}-\underline{d}\underline{j}})}{\underline{c}\underline{j}})$$

$$+\frac{1}{2}(\text{Ei}(\frac{\sigma_0^2\|\mathbf{u}\|^4}{2}) + \text{Ei}(-2\frac{\sigma_0^2\|\mathbf{u}\|^4}{2})) + 4\log(\sigma_0/2\|\mathbf{u}\|^2),$$

$$\overline{z}(t) = \overline{b}t + \frac{\kappa}{8},$$

*where*

$$\text{Li}_2(x) = -\int_0^x \frac{\ln(1-t)}{t}dt.$$

*Additionally, we would have asymptotic property that*

$$\lim_{t\to+\infty} y(t)^2 = \lim_{t\to+\infty} \Theta(\log(\log^2(t))), \quad \lim_{t\to+\infty} \frac{\mathbb{E}[\mathop{\mathbb{E}}_{n\in\mathcal{V}_k^{yS_n}}[\sum_{j\in S_{n,k}^{yS_n}} (\sigma_S^{(t)})_j^n]]}{\frac{\log^4(t)}{1+\log^4(t)}} = \Theta(1).$$

It's obvious that the decaying impact of the learning rate and cross-entropy loss are at the similar order. Also, if we consider decaying learning rate, the right side of the inequality would be smaller. $z(t)$ would be in a $\Theta(\log(\log(t)))$ order when $z(t)$ get large, which will make the right side of the $y(t)$'s formula contain an intergral of $\Theta(\log\log(t))$, which is obviously slower.

## J  Exponential Convergence of 0-1 Loss

We continue our proof after Lemma 33. In this section, we assume all the events in the Section D hold, denoted as $\Upsilon_{\text{Pre}}$.

**Lemma 41.** *The Frobenius norm of $\mathbf{W}_O^y$ and its gradient can be bounded:*

$$\|\mathbf{W}_O^y\|_F^2 = O(\frac{K_1\|\mathbf{q}\|^2}{\lambda^2 m}), \quad \|\nabla_{\mathbf{W}_O^{y}{}^{(t)}}L_{\mathcal{B}_t}(\Psi^{(t)})\|_F^2 = O(\frac{K_1\|\mathbf{q}\|^2}{m}).$$

*Proof.* For $\forall i \in [m]$, by the gradient update rule in Eq.(22), as well as Lemma 4's insight we see that the lengths of the $\mathbf{W}_O^y$ on certain projection direction will continue to grow until being stuck by the regularization, which is a $\lambda$-scaled $\mathbf{W}_O^y$ itself. Due to the low-noise condition in Condition 1 with a sufficiently large $C$ as well as the isotropy of noise, the learning progress of features would be the main contributor to the F norm of NN matrices and the noise, validated in Figure 2 (iii-iv). We can consider an extreme case where all the samples in a single batch belongs to some concept $k \in [K_1]$, which we can have the upper bound of the first term of the right side of the inequality over the $k$-th concept's corresponding projection direction, and thus we can derive an upper bound

$$(\mathbf{W}_{O_{(i,\cdot)}}^y{}^{(t)}\frac{\mathbf{q}_k^\pm}{\|\mathbf{q}_k^\pm\|})^2 \leq \lambda^{-2}\frac{1}{m^2}(\|\mathbf{c}_k \pm \mathbf{d}_k\|^2 + \frac{3\sigma_\xi^2 d_\mathcal{Y}}{2}) = \Theta(\frac{\|\mathbf{q}\|^2}{\lambda^2 m^2}), \tag{62}$$

where the first inequality is by $(2\sum_{l\in S_{n,k}^e} (\sigma_S^{(t)})_l^n - 1) \leq 1$, and Lemma 6; the last equality is by the low noise condition $\sigma_\xi \leq \|\mathbf{q}\|/C\sqrt{d_\mathcal{Y}}$ in Condition 1. Then by the low noise condition as well as the data model's definition we see that all the 2-norm of the $\mathbf{W}_O^y$ is controlled by the $K_1$ concepts' corresponding lengths in projection space. Then by the definition of Frobenius norm and Eq.(22) we have

$$\|\mathbf{W}_O^y\|_F^2 \leq \Theta(\frac{K_1\|\mathbf{q}\|^2}{\lambda^2 m}), \quad \|\nabla_{\mathbf{W}_O^{y}{}^{(t)}}L_{\mathcal{B}_t}(\Psi^{(t)})\|_F^2 \leq \Theta(\frac{K_1\|\mathbf{q}\|^2}{m}).$$

$\square$

**Lemma 42.** *For* $\forall \hat{k} \in [K_1]$, $\boldsymbol{a}_{\hat{k}}^\top \mathbf{W}_Q^{x\,(t)} \boldsymbol{a}_{\hat{k}}, \boldsymbol{a}_{\hat{k}}^\top \mathbf{W}_K^{x\,(t)} \boldsymbol{a}_{\hat{k}}$ *and* $\boldsymbol{b}_{\hat{k}}^\top \mathbf{W}_Q^{x\,(t)} \boldsymbol{b}_{\hat{k}}, \boldsymbol{b}_{\hat{k}}^\top \mathbf{W}_K^{x\,(t)} \boldsymbol{b}_{\hat{k}}$ *satisfy*

$$\boldsymbol{a}_{\hat{k}}^\top \mathbf{W}_Q^{x\,(t)} \boldsymbol{a}_{\hat{k}}, \boldsymbol{a}_{\hat{k}}^\top \mathbf{W}_K^{x\,(t)} \boldsymbol{a}_{\hat{k}} = O(\sigma_0 \|\mathbf{u}\|^2),$$

$$\boldsymbol{b}_{\hat{k}}^\top \mathbf{W}_Q^{x\,(t)} \boldsymbol{b}_{\hat{k}}, \boldsymbol{b}_{\hat{k}}^\top \mathbf{W}_K^{x\,(t)} \boldsymbol{b}_{\hat{k}} = O(\|\mathbf{u}\| \sqrt{\log(\frac{(L-1)\|\mathbf{u}\|^2 \|\mathbf{q}\|^2}{\lambda^2 m})}).$$

*Moreover, the Frobenius norm of* $\mathbf{W}_Q^{x\,(t)}$ *and* $\mathbf{W}_K^{x\,(t)}$ *and its gradient can be bounded as below*

$$\|\mathbf{W}_Q^{x\,(t)}\|_F^2, \|\mathbf{W}_K^{x\,(t)}\|_F^2 = O(K_1 \log(\frac{(L-1)\|\mathbf{u}\|^2 \|\mathbf{q}\|^2}{\lambda^2 m}))$$

$$\|\nabla_{\mathbf{W}_Q^{x\,(t)}} L_{\mathcal{B}_t}(\Psi^{(t)})\|_F^2, \|\nabla_{\mathbf{W}_K^{x\,(t)}} L_{\mathcal{B}_t}(\Psi^{(t)})\|_F^2 = O(\frac{K_1(L-1)\|\mathbf{u}\|^2 \|\mathbf{q}\|^2}{m}).$$

*Proof.* By Eq.(16) and (17), we see that

$$|I_{Q,\boldsymbol{a}_{\hat{k}},\text{chaos}}^{(t)}|, |I_{Q,\boldsymbol{a}_{\hat{k}},\text{contri}}^{(t)}|, |I_{K,\boldsymbol{a}_{\hat{k}},\text{chaos}}^{(t)}|, |I_{K,\boldsymbol{a}_{\hat{k}},\text{contri}}^{(t)}| \leq \eta_t \Theta(\max\{|\boldsymbol{a}_{\hat{k}}^\top \mathbf{W}_Q^{x\,(t)} \boldsymbol{a}_{\hat{k}}|, |\boldsymbol{a}_{\hat{k}}^\top \mathbf{W}_K^{x\,(t)} \boldsymbol{a}_{\hat{k}}|\})(\|\mathbf{u}\|$$

$$\sigma_\xi \sqrt{2\log(\frac{KN}{\delta})} + \frac{1}{K}\|\mathbf{u}\|^2)^2 (\frac{\|\mathbf{q}\|/\lambda m}{m})) \leq O(\lambda \max\{|\boldsymbol{a}_{\hat{k}}^\top \mathbf{W}_Q^{x\,(t)} \boldsymbol{a}_{\hat{k}}|, |\boldsymbol{a}_{\hat{k}}^\top \mathbf{W}_K^{x\,(t)} \boldsymbol{a}_{\hat{k}}|\}).$$

Here, the first inequality is by the scaled identity initialization of $\mathbf{W}_Q^{x\,(0)}, \mathbf{W}_K^{x\,(0)}$, orthogonal relationships of vectors in Lemma 27, Lemma 6, $\sum_{l,j \in [L]} (\sigma_S^{(t)})_l^n (\sigma_S^{(t)})_j^n \leq 1/4$, Eq.(62) in Lemma 41; the second inequality is by the low noise condition $\sigma_\xi \leq \lambda m/(C\sqrt{d_X}\|\mathbf{u}\|\|\mathbf{q}\|^{1/2})$ and the large $K \geq C\|\mathbf{u}\|/(\sigma_\xi \sqrt{d_X})$ for a large $C$ in Condition 1. Thus the update of $\boldsymbol{a}_{\hat{k}}^\top \mathbf{W}_Q^{x\,(t)} \boldsymbol{a}_{\hat{k}}$ and $\boldsymbol{a}_{\hat{k}}^\top \mathbf{W}_K^{x\,(t)} \boldsymbol{a}_{\hat{k}}$ are dominated by their regularization, and thus the scale can not be better than the initialization. By Lemma 7, the conclusion holds.

On the other hand, we see that by the scaled identity initialization of $\mathbf{W}_Q^{x\,(0)}, \mathbf{W}_K^{x\,(0)}$ and orthogonal relationships of vectors in Lemma 27, the initialization of $\boldsymbol{b}_{\hat{k}}^\top \mathbf{W}_Q^{x\,(t)} \boldsymbol{b}_{\hat{k}}$ and $\boldsymbol{b}_{\hat{k}}^\top \mathbf{W}_K^{x\,(t)} \boldsymbol{b}_{\hat{k}}$ are the same, and as the gradient update is nearly symmetry, which can lead to the fact that $\Theta(\boldsymbol{b}_{\hat{k}}^\top \mathbf{W}_Q^{x\,(t)} \boldsymbol{b}_{\hat{k}}) = \Theta(\boldsymbol{b}_{\hat{k}}^\top \mathbf{W}_K^{x\,(t)} \boldsymbol{b}_{\hat{k}})$ and $\boldsymbol{b}_{\hat{k}}^\top \mathbf{W}_Q^{x\,(t)} \mathbf{W}_K^{x\,(t)} \boldsymbol{b}_{\hat{k}} = \Theta(\boldsymbol{b}_{\hat{k}}^\top \mathbf{W}_Q^{x\,(t)} \boldsymbol{b}_{\hat{k}} \boldsymbol{b}_{\hat{k}}^\top \mathbf{W}_K^{x\,(t)} \boldsymbol{b}_{\hat{k}}/\|\boldsymbol{b}_{\hat{k}}\|^2)$. By the scaled identity initialization of $\mathbf{W}_Q^{x\,(0)}, \mathbf{W}_K^{x\,(0)}$, orthogonal relationships of vectors in Lemma 27, Eq.(19) and (20) we can see that

$$|I_{Q,\boldsymbol{b}_{\hat{k}},\text{chaos}}^{(t)}|, |I_{K,\boldsymbol{b}_{\hat{k}},\text{chaos}}^{(t)}| \leq \eta_t O(\lambda \max\{|\boldsymbol{b}_{\hat{k}}^\top \mathbf{W}_Q^{x\,(t)} \boldsymbol{b}_{\hat{k}}|, |\boldsymbol{b}_{\hat{k}}^\top \mathbf{W}_K^{x\,(t)} \boldsymbol{b}_{\hat{k}}|\}).$$

$$|I_{Q,\boldsymbol{b}_{\hat{k}},\text{contri}}^{(t)}|, |I_{K,\boldsymbol{b}_{\hat{k}},\text{contri}}^{(t)}| \leq \eta_t \Theta(\max\{|\boldsymbol{b}_{\hat{k}}^\top \mathbf{W}_Q^{x\,(t)} \boldsymbol{b}_{\hat{k}}|, |\boldsymbol{b}_{\hat{k}}^\top \mathbf{W}_K^{x\,(t)} \boldsymbol{b}_{\hat{k}}|\})(\lambda + \frac{\|\mathbf{u}\|^2 \|\mathbf{q}\|^2}{\lambda m}(\sum_{j \in S_{n,k}^+} (\sigma_S^{(t)})_j^n)$$

$$(\sum_{j \in S_{n,k}^-} (\sigma_S^{(t)})_j^n))).$$

We see that $\boldsymbol{b}_{\hat{k}}^\top \mathbf{W}_Q^{x\,(t)} \boldsymbol{b}_{\hat{k}}$ and $\boldsymbol{b}_{\hat{k}}^\top \mathbf{W}_K^{x\,(t)} \boldsymbol{b}_{\hat{k}}$ will continue to grow up except always being stuck by the regularization. To see the upper bound under this situation, we consider an extreme case where all the samples in a single batch belongs to some concept $k \in [K_1]$, and there is only one demonstrations in each prompt share the semantic with the query. Then by the scaled identity initialization of $\mathbf{W}_Q^{x\,(0)}, \mathbf{W}_K^{x\,(0)}$, orthogonal relationships of vectors in Lemma 27 and Eq.(18), we can see that the growing of $\boldsymbol{b}_{\hat{k}}^\top \mathbf{W}_Q^{x\,(t)} \boldsymbol{b}_{\hat{k}}$ and $\boldsymbol{b}_{\hat{k}}^\top \mathbf{W}_K^{x\,(t)} \boldsymbol{b}_{\hat{k}}$ would satisfy the following and strive to grow up to make the equality holds, which naturally have an upper bound

$$|I_{Q,\boldsymbol{b}_{\hat{k}},\text{contri}}^{(t)}|, |I_{K,\boldsymbol{b}_{\hat{k}},\text{contri}}^{(t)}| \geq \lambda \min\{|\boldsymbol{b}_{\hat{k}}^\top \mathbf{W}_Q^{x\,(t)} \boldsymbol{b}_{\hat{k}}|, |\boldsymbol{b}_{\hat{k}}^\top \mathbf{W}_K^{x\,(t)} \boldsymbol{b}_{\hat{k}}|\} \Rightarrow$$

$$\frac{\|\mathbf{u}\|^2 \|\mathbf{q}\|^2}{\lambda m}(\sum_{j \in S_{n,k}^+} (\sigma_S^{(t)})_j^n)(\sum_{j \in S_{n,k}^-} (\sigma_S^{(t)})_j^n) \geq \Theta(\lambda) \Rightarrow$$

$$\Theta(\frac{\|\mathbf{u}\|^2 \|\mathbf{q}\|^2}{\lambda^2 m} \frac{\sum_{j \in S_{n,k}^+} e^{\boldsymbol{b}_{\hat{k}}^\top \mathbf{W}_Q^{x\,(t)} \mathbf{W}_K^{x\,(t)} \boldsymbol{b}_{\hat{k}}} \sum_{j \in S_{n,k}^-} e^{-\boldsymbol{b}_{\hat{k}}^\top \mathbf{W}_Q^{x\,(t)} \mathbf{W}_K^{x\,(t)} \boldsymbol{b}_{\hat{k}}}}{(\sum_{j \in S_{n,k}^+} e^{\boldsymbol{b}_{\hat{k}}^\top \mathbf{W}_Q^{x\,(t)} \mathbf{W}_K^{x\,(t)} \boldsymbol{b}_{\hat{k}}} + \sum_{j \in S_{n,k}^-} e^{-\boldsymbol{b}_{\hat{k}}^\top \mathbf{W}_Q^{x\,(t)} \mathbf{W}_K^{x\,(t)} \boldsymbol{b}_{\hat{k}}})^2}) \geq 1 \Rightarrow$$

$$\Theta(\frac{\|\mathbf{u}\|^2 \|\mathbf{q}\|^2}{\lambda^2 m} \frac{L-1}{(e^{\boldsymbol{b}_{\hat{k}}^\top \mathbf{W}_Q^{x\,(t)} \mathbf{W}_K^{x\,(t)} \boldsymbol{b}_{\hat{k}}} + (L-1)e^{-\boldsymbol{b}_{\hat{k}}^\top \mathbf{W}_Q^{x\,(t)} \mathbf{W}_K^{x\,(t)} \boldsymbol{b}_{\hat{k}}})^2}) \geq 1 \Rightarrow$$

$$\Theta(\frac{(L-1)\|\mathbf{u}\|^2 \|\mathbf{q}\|^2}{\lambda^2 m} e^{-2\boldsymbol{b}_{\hat{k}}^\top \mathbf{W}_Q^{x\,(t)} \mathbf{W}_K^{x\,(t)} \boldsymbol{b}_{\hat{k}}}) \geq 1$$

$$\Rightarrow \boldsymbol{b}_{\hat{k}}^\top \mathbf{W}_Q^{x\,(t)} \boldsymbol{b}_{\hat{k}}, \boldsymbol{b}_{\hat{k}}^\top \mathbf{W}_K^{x\,(t)} \boldsymbol{b}_{\hat{k}} \leq \Theta(\|\mathbf{u}\| \sqrt{\frac{1}{2}\log(\frac{(L-1)\|\mathbf{u}\|^2 \|\mathbf{q}\|^2}{\lambda^2 m})})$$

Here, the second arrow is by the definition of $\sum_{j \in S_{n,k}^+} (\sigma_S^{(t)})_j^n$ and Eq.(31); the third arrow is by our considered extreme case where there is only one demonstration in each of the prompt sample in this all-the-same-concept batch, which is considered for obtaining the upper bound; the forth arrow is by the small $\lambda$ by Condition 1, which denotes $e^{-2\boldsymbol{b}_{\hat{k}}^\top \mathbf{W}_Q^{\boldsymbol{x}(t)} \mathbf{W}_K^{\boldsymbol{x}(t)} \boldsymbol{b}_{\hat{k}}}$ should be the key contributor.

Similar to the claims in Lemma 41, here we see that as the $\lambda$ is very small, by the scaled identity initialization of $\mathbf{W}_Q^{\boldsymbol{x}(0)}, \mathbf{W}_K^{\boldsymbol{x}(0)}$, orthogonal relationships of vectors in Lemma 27, as well as the low-noise condition by Condition 1, it's safe to say that as the learning proceed, the scales of $\boldsymbol{b}_{\hat{k}}^\top \mathbf{W}_Q^{\boldsymbol{x}(t)} \boldsymbol{b}_{\hat{k}}, \boldsymbol{b}_{\hat{k}}^\top \mathbf{W}_K^{\boldsymbol{x}(t)} \boldsymbol{b}_{\hat{k}}$ would completely dominate $\boldsymbol{a}_{\hat{k}}^\top \mathbf{W}_Q^{\boldsymbol{x}(t)} \boldsymbol{a}_{\hat{k}}, \boldsymbol{a}_{\hat{k}}^\top \mathbf{W}_K^{\boldsymbol{x}(t)} \boldsymbol{a}_{\hat{k}}$, as well as $\boldsymbol{a}_k^\top \mathbf{W}_X^{\boldsymbol{x}(t)} \boldsymbol{b}_{\hat{k}}, \boldsymbol{b}_k^\top \mathbf{W}_X^{\boldsymbol{x}(t)} \boldsymbol{a}_{\hat{k}}, \boldsymbol{\nu}_r^\top \mathbf{W}_X^{\boldsymbol{x}(t)} \boldsymbol{\nu}_r, \boldsymbol{u}_w^{\perp \top} \mathbf{W}_X^{\boldsymbol{x}(t)} \boldsymbol{u}_w^\perp$, $\forall X \in \{Q, K\}, r \in [K_2], w \in [d_{\mathcal{X}} - 2K_1 - K_2]$. Collaborating with Lemma 9, we have

$$\|\mathbf{W}_Q^{\boldsymbol{x}(t)}\|_F^2, \|\mathbf{W}_K^{\boldsymbol{x}(t)}\|_F^2 \leq \frac{K_1}{\|\mathbf{u}\|^2} \max\{(\boldsymbol{b}_{\hat{k}}^\top \mathbf{W}_Q^{\boldsymbol{x}(t)} \boldsymbol{b}_{\hat{k}})^2, (\boldsymbol{b}_{\hat{k}}^\top \mathbf{W}_K^{\boldsymbol{x}(t)} \boldsymbol{b}_{\hat{k}})^2\} = O(K_1 \log(\frac{(L-1)\|\mathbf{u}\|^2 \|\mathbf{q}\|^2}{\lambda^2 m})).$$

On the other hand, we see that the maximum gradient F norm on a single batch comes from the maximum changes of the $\boldsymbol{b}_{\hat{k}}^\top \mathbf{W}_Q^{\boldsymbol{x}(t)} \boldsymbol{b}_{\hat{k}}^2$ (or $\boldsymbol{b}_{\hat{k}}^\top \mathbf{W}_K^{\boldsymbol{x}(t)} \boldsymbol{b}_{\hat{k}}^2$). As we see that the extreme case of the growing is every concept $k \in [K_1]$ has been fully learned such that even a batch full of the same concept can not let the corresponding concept's feature grow. In this case, we see that the maximum gradient F norm should be at the order of $\|\lambda \mathbf{W}_Q^{\boldsymbol{x}(t)}\|_F$ (or $\|\lambda \mathbf{W}_K^{\boldsymbol{x}(t)}\|_F$). Thus

$$\|\nabla_{\mathbf{W}_Q^{\boldsymbol{x}(t)}} L_{\mathcal{B}_t}(\Psi^{(t)})\|_F^2, \|\nabla_{\mathbf{W}_K^{\boldsymbol{x}(t)}} L_{\mathcal{B}_t}(\Psi^{(t)})\|_F^2 = O(\lambda^2 K_1 \log(\frac{(L-1)\|\mathbf{u}\|^2 \|\mathbf{q}\|^2}{\lambda^2 m}))$$
$$= O(\frac{K_1(L-1)\|\mathbf{u}\|^2 \|\mathbf{q}\|^2}{m}),$$

where the equality is by $g(x) = \log(x) \leq O(x), x > 1$. The proof is completed.

**Remark 5.** *Worth noting that this upper bound, as well as the upper bound of $\|\mathbf{W}_{O_{(i,\cdot)}}^{\boldsymbol{y}}{}^{(t)} \boldsymbol{q}_k^\pm / \|\boldsymbol{q}_k^\pm\|\|$ in Lemma 41, are looser in the order of $K_1^{-2}$ and $K_1^{-1}$ compared to those of $\beta_{Q,k}^{(t)} = \beta_{K,k}^{(t)}$ and $e\beta_{O_{(i,\cdot)},k}^{(t)}$ in Lemma 4. This suits the intuition and statistics since in the practical training setting, for $B \geq 1$ we can see that sometimes the samples of a batch all belong to one concept, or sometimes their are not any particular concept in a single batch, especially when $B$ is small. Therefore, unless we have the situation where even when every prompt sample of a batch belong to the same concept the regularization can stuck the growing, there is still chance for that concept's features to be learned. In contrast, the expectation considers every concept's sample appear in every batch scaled by a "soft weight" in the order of $\Theta(1/K)$. As the attention 's gradient contain MLP, its order would be $\Theta(1/K^2)$. Besides, we see that this lemma's result contains the scale of $L-1$, which comes from the extreme case discussion where there is only one demonstration in each prompt sample that share the semantic to those of query. In contrast, when considering the expectation, the number of two opposite semantics is the same, under which the $L/2$ would be eliminated in the numerator and denominator. Last but not least, when estimating the real cases, we have scaled the derivative of $-\ell'$ to its maximum 1, we do so because in real cases due to the imbalanced prompt samples in a single batch, it would be inconvenient to consider it is contributed by severel elements like $e\beta_{O_{(i,\cdot)},k}^{(t)}, \alpha_{O_{(i,\cdot)},k}^{(t)}$. This actually indirectly demonstrates the superiority of considering expectations.*

$\square$

**Lemma 43.** *(Restatement of Proposition 2) $\forall t \geq \hat{T}$, when $\|\Psi'^{(t)} - \mathbb{E}(\Psi'^{(t)})\|_F \leq \nu$ holds, we have $L_{\mathcal{D}^*}^{0-1}(\Psi'^{(t)}) = L_{\mathcal{D}^*}^{0-1}(\mathbb{E}(\Psi'^{(t)}))$. Here, $\|\Psi'\|_F^2 := \|\mathbf{W}_Q^{\boldsymbol{x}}\|_F^2 + \|\mathbf{W}_K^{\boldsymbol{x}}\|_F^2 + \|\mathbf{W}_O^{\boldsymbol{y}}\|_F^2$.*

*Proof.* By Lemma 33, we see that our convergence of 0-1 loss is based on the intermediate result that $\mathbb{E}[\mathbf{A}_t^{k,e}] \geq \kappa$, which will ensure that $\mathbb{E}[y_{S_n} \cdot f(\mathbb{E}(S_n), \mathbb{E}(\Psi^{\hat{T}}))] \geq \kappa/2$. Therefore, when conditioned on $\mathbb{E}[\mathbf{W}_Q^{\boldsymbol{x}(t)}], \mathbb{E}[\mathbf{W}_K^{\boldsymbol{x}(t)}]$, a minimum admissible disparity between $\mathbf{W}_O^{\boldsymbol{y}(t)}$ and $\mathbb{E}[\mathbf{W}_O^{\boldsymbol{y}(t)}]$ corresponds the the minimum admissible disparity between $\mathbf{W}_{O_{(i,\cdot)}}^{\boldsymbol{y}}{}^{(t)} \boldsymbol{c}_k, \mathbf{W}_{O_{(i,\cdot)}}^{\boldsymbol{y}}{}^{(t)} \boldsymbol{d}_k$ and $\alpha_{O_{(i,\cdot)},k}^{(t)}, \beta_{O_{(i,\cdot)},k}^{(t)}$, where would consequently cause $\mathbb{E}_{\mathcal{V}_k^e}[\mathbf{A}_t^{k,e}] \leq \kappa/2$ that could have potential to deteriorate the 0-1 loss. Given that $\kappa/2 \geq \sqrt{2}\sigma_1 \|\mathbf{q}\|$ by Lemma 23, the decomposition in Eq.(33) as well as Lemma 9, we see that for some $k \in [K_1]$, the minimum admissible disparity can be written as

$$\Theta(\|(\sqrt{2}\sigma_1\|\mathbf{q}\|)^2 (\frac{\boldsymbol{c}_k^\top}{\|\boldsymbol{c}_k\|^2})^\top \frac{\boldsymbol{c}_k^\top}{\|\boldsymbol{c}_k\|^2}\|_F) = \Theta(\sqrt{2}\sigma_1\|(\|\mathbf{q}\|)^2 (\frac{\boldsymbol{c}_k^\top}{\|\boldsymbol{c}_k\|^2})^\top \frac{\boldsymbol{c}_k^\top}{\|\boldsymbol{c}_k\|^2}\|_F) \geq \Theta(2\sqrt{2}/(1+\kappa_{\boldsymbol{y}})\sigma_1).$$

Therefore, we see that when conditioned on $\mathbb{E}[\mathbf{W}_Q^{\boldsymbol{x}(t)}], \mathbb{E}[\mathbf{W}_K^{\boldsymbol{x}(t)}]$, the minimum admissible disparity between $\mathbf{W}_O^{\boldsymbol{y}(t)}$ and $\mathbb{E}[\mathbf{W}_O^{\boldsymbol{y}(t)}]$ to not worsen the 0-1 loss is $\Theta(2\sqrt{2}/(1+\kappa_{\boldsymbol{y}})\sigma_1)$.

On the other hand, when conditioned on $\mathbb{E}[\mathbf{W}_O^{y\,(t)}], t \geq T'$, we compute the minimum admissible disparity between $\mathbf{W}_Q^{x\,(T')}$, $\mathbf{W}_K^{x\,(T')}$ and $\mathbb{E}[\mathbf{W}_Q^{x\,(T')}] = \mathbb{E}[\mathbf{W}_K^{x\,\hat{T}}]$. Considering all the activated neurons, when $\sum_{i \in \mathcal{W}_{k,n}^e(t)} \mathbb{E}[\mathbf{r}_i(2\sum_{l \in S_{n,k}^e} (\sigma_S^{(\hat{T})})_l^n - 1)\beta_{O_{(i,\cdot),k}}^{(\hat{T})}] = 0$, we should have $\sum_{i \in \mathcal{W}_{k,n}^e(t)} \mathbb{E}[\mathbf{r}_i \alpha_{O_{(i,\cdot),k}}^{(\hat{T})}] \geq 0$ otherwise some of the neurons must be deactivated, which is contradicted by the definitions of $\mathcal{W}_{k,n}^e(t)$. In this case we can magnify the impact of $\sum_{i \in \mathcal{W}_{k,n}^e(t)} \mathbb{E}[\mathbf{r}_i(2\sum_{l \in S_{n,k}^e} (\sigma_S^{(\hat{T})})_l^n - 1)\beta_{O_{(i,\cdot),k}}^{(\hat{T})}]$ by considering $\sum_{i \in \mathcal{W}_{k,n}^e(t)} \mathbb{E}[\mathbf{r}_i \alpha_{O_{(i,\cdot),k}}^{(\hat{T})}] = 0$. As such, the minimum admissible disparity would be the case where $\boldsymbol{b}_{\hat{k}}^\top \mathbf{W}_Q^{x\,(\hat{T})} \boldsymbol{b}_k$ and $\boldsymbol{b}_{\hat{k}}^\top \mathbf{W}_K^{x\,(\hat{T})} \boldsymbol{b}_k$ both differ from $\beta_{Q,k}^{(\hat{T})} = \beta_{K,k}^{(\hat{T})}$ by the amount of $\underline{\beta_{QK}^-}$. Recall the definition of $\underline{\beta_{QK}^-}$ in Lemma 34, and collaborating with Lemma 9, we have the minimum admissible disparity be

$$\sigma_0(1 - \kappa_{\boldsymbol{x}})e^{-\log(5Km/\delta)\frac{\sigma_1^2 \|\mathbf{u}\|^4 (1 + e^{-\sigma_0^2 \|\mathbf{u}\|^2})}{(1 - e^{-\sigma_0^2 \|\mathbf{u}\|^2})}} . \text{ Recall}$$

$$\nu := \min\{2\sqrt{2}\sigma_1/(1 + \kappa_{\boldsymbol{y}}), \sigma_0(1 - \kappa_{\boldsymbol{x}})e^{-\log(5Km/\delta)\frac{\sigma_1^2 \|\mathbf{u}\|^4 (1 + e^{-\sigma_0^2 \|\mathbf{u}\|^2})}{(1 - e^{-\sigma_0^2 \|\mathbf{u}\|^2})}}\},$$

the proof is completed.

$\square$

**Lemma 44.** *For $t \in \{1, \cdots, T\}$, for $\mathbf{W} \in \{\mathbf{W}_Q^x, \mathbf{W}_K^x, \mathbf{W}_O^y\}$ and $X \in \{Q, K, O\}$ it follows that*

1. $\|\mathbf{W}^{(t+1)} - \mathbf{W}_t^{(t+1)}\|_F \leq \Theta(\frac{K_1^{1/2}\|\mathbf{q}\|((L-1)^{1/2}\|\mathbf{u}\| + 1)}{m^{1/2}}\eta_t)$,

2. $\|\mathbf{W}^{(s+1)} - \mathbf{W}_t^{(s+1)}\|_F \leq (1 - \eta_s\lambda)\|\mathbf{W}^{(s)} - \mathbf{W}_t^{(s)}\|_F, \forall s \geq t + 1$,

3. $\sum_{t=0}^T \|D_X^t\|_\infty^2 \leq \Theta(\frac{K_1\|\mathbf{q}\|^2((L-1)\|\mathbf{u}\|^2 + 1)}{m\lambda^2(\gamma + T)})$.

*Proof.* We provide the proof by extending the techniques in [34, 33, 36] to Hilbert-Schmidt space, whose inner product is defined by trace. First we note that $\eta_0 = \frac{2}{\gamma + 1} \leq \min\{1/(L_{\text{Logist}} + \lambda), 1/2\lambda\}$, where $L_{\text{Logist}}$ is the $L$-smooth Lipschitz constant of cross-entropy loss $\ell(\cdot)$, which is 1. The first statement can be shown as follows. Since by definition we see that $\mathbf{W}^{(t)} = \mathbf{W}_t^{(t)}$, we only need to check the maximum disparity of the gradient in a single iteration update, then by Lemma 41 and Lemma 42 we readily obtain the results.

For the second statement, following the proof in [94, 34], we see that the Lipschitz smoothness of cross-entropy loss denotes that

$$\langle \nabla_{\mathbf{W}} L_{\mathcal{B}}(\Psi) - \nabla_{\mathbf{W}'} L_{\mathcal{B}}(\Psi), \mathbf{W} - \mathbf{W}' \rangle \geq \frac{1}{L_{\text{Logist}}}\|\nabla_{\mathbf{W}} L_{\mathcal{B}}(\Psi) - \nabla_{\mathbf{W}'} L_{\mathcal{B}}(\Psi)\|_F^2. \tag{63}$$

Then we have that for $s \geq t + 1$,

$$\begin{aligned}
\|\mathbf{W}^{(s+1)} - \mathbf{W}_t^{(s+1)}\|_F^2 &= \left\|(1 - \eta_s\lambda)\left(\mathbf{W}^{(s)} - \mathbf{W}_t^{(s)}\right) - \eta_s\left(\partial_g l\left(g_s, Z_s\right) - \partial_g l\left(g_s^t, Z_s\right)\right)\right\|_F^2 \\
&= (1 - \eta_s\lambda)^2 \left\|\mathbf{W}^{(s)} - \mathbf{W}_t^{(s)}\right\|_F^2 - 2\eta_s(1 - \eta_s\lambda) \cdot \\
&\quad \left\langle \nabla_{\mathbf{W}^{(s)}} L_{\mathcal{B}_s}(\Psi^s) - \nabla_{\mathbf{W}_t^{(s)}} L_{\mathcal{B}_s}(\Psi^s), \mathbf{W}^{(s)} - \mathbf{W}_t^{(s)} \right\rangle \\
&\quad + \eta_s^2 \|\nabla_{\mathbf{W}^{(s)}} L_{\mathcal{B}_s}(\Psi^s) - \nabla_{\mathbf{W}_t^{(s)}} L_{\mathcal{B}_s}(\Psi^s)\|_F^2 \\
&\leq (1 - \eta_s\lambda)^2 \left\|\mathbf{W}^{(s)} - \mathbf{W}_t^{(s)}\right\|_F^2 - \eta_s\left(\frac{1}{L_{\text{Logist}}} - \eta_s\right) \cdot \\
&\quad \left\|\nabla_{\mathbf{W}^{(s)}} L_{\mathcal{B}_s}(\Psi^s) - \nabla_{\mathbf{W}_t^{(s)}} L_{\mathcal{B}_s}(\Psi^s)\right\|_F^2 \\
&\leq (1 - \eta_s\lambda)^2 \left\|\mathbf{W}^{(s)} - \mathbf{W}_t^{(s)}\right\|_F^2
\end{aligned}$$

where we utilize the Eq. (63) and conditions on learning rates. Utilizing this statement, the stable property of stochastic gradient descent has been shown. Again following the techniques in [34, 33, 36], we now obtain the bound: for $t \in \{1, \ldots, T\}$,

$$\left\|\mathbf{W}^{(T+1)} - \mathbf{W}_t^{(T+1)}\right\|_F \leq \Theta(\frac{K_1^{1/2}\|\mathbf{q}\|((L-1)^{1/2}\|\mathbf{u}\| + 1)}{m^{1/2}}\eta_t) \prod_{s=t+1}^T (1 - \eta_s\lambda). \tag{64}$$

From the following inequality,

$$\prod_{s=t+1}^{T}(1-\eta_s\lambda) = \prod_{s=t+1}^{T}\frac{\gamma+s-2}{\gamma+s} < \frac{\gamma+t}{\gamma+T}$$

where the last inequality hold clearly by expanding the product, the right hand side of the Eq.(64) is upper bounded as follows

$$\Theta\big(\frac{K_1^{\frac{1}{2}}\|\mathbf{q}\|((L-1)^{\frac{1}{2}}\|\mathbf{u}\|+1)}{m^{\frac{1}{2}}}\big)\eta_t\prod_{s=t+1}^{T}(1-\eta_s\lambda) \le \Theta\big(\frac{K_1^{\frac{1}{2}}\|\mathbf{q}\|((L-1)^{\frac{1}{2}}\|\mathbf{u}\|+1)}{m^{\frac{1}{2}}}\big)\frac{\eta_t(\gamma+t)}{\gamma+T}$$

$$= \frac{\Theta\big(2\dfrac{K_1^{\frac{1}{2}}\|\mathbf{q}\|((L-1)^{\frac{1}{2}}\|\mathbf{u}\|+1)}{m^{\frac{1}{2}}}\big)}{\lambda(\gamma+T)}.$$

We finally obtain the desired bound:

$$\sum_{t=0}^{T}\|D_t\|_\infty^2 \le \sum_{t=0}^{T}\Theta\big(\frac{K_1\|\mathbf{q}\|^2((L-1)\|\mathbf{u}\|^2+1)}{m\lambda^2(\gamma+T)^2}\big) \le \Theta\big(\frac{K_1\|\mathbf{q}\|^2((L-1)\|\mathbf{u}\|^2+1)}{m\lambda^2(\gamma+T)}\big).$$

$\square$

**Remark 6.** *Utilizing this lemma, the exponential convergence over the 0-1 loss is readily obtained.*

## K   Out-of-Distribution Generalization

**Lemma 45.** *OOD 1: Master of Polysemy of Words.* *During testing, The prompt length $L^*$ can be any positive integer. The $\mathcal{D}_z^*$ can have any new probability distribution that differs from the training distribution, satisfying that each prompt has at least one co-concept $k \in [K_1]$, with equal chance to have positive or negative semantic labels. Additionally, a single $(\boldsymbol{x}, \boldsymbol{y}) \sim \mathcal{D}_x^* \times \mathcal{D}y^*$ pair can appear in at least $\|\boldsymbol{z}\|_0$ concept-specific prompts/tasks. Importantly, all of the tasks in this new distribution $\mathcal{D}^*$ enjoy Bayes-Optimal test error $L_{\mathcal{D}^*}^{0-1}(\Psi^{(T^*)}) \le \varepsilon$.*

This lemma demonstrate the strong OOD Generalization ability of transformer utilizing multi-concept semantics, suggesting the efficiency transformer to conduct unseen ICL tasks just by its learned knowledge on the two non-orthogonal dictionaries. Also, this lemma showcases an intriguing phenomenon since it allows multiple concepts with comparable chance along word-demo pairs - even with the same input-output pair and query, the model can produce diverse responses when provided varying contextual (concept / task) information. For instance, with the prompt "Japan: Sakura; China:", the LLM may output "Penoey" (national flower) or "Panda" (national symbol), reflecting different conceptual (task) interpretations. Both answers are right since they are all the co-concept tasks. Interestingly, adding another demonstration like "Japan: Sakura, France: Iris germanica, China:" stabilizes the response to "Penoey", since the only co-concept is left to be "national flower". In our theory, we make an elementary explanation to this flexible, context-sensitive in-context learning (ICL) behavior by attributing it to the transformer's ability to harness multi-concept semantics.

**Lemma 46.** *OOD 2: Innovation.* *During testing, the distribution of $\mathcal{D}_x^* \times \mathcal{D}_y^*$ can enjoy data shift. Specifically, suggest we now have a new $\mathbf{M}^*$ and $\mathbf{Q}^*$ to define new $\mathcal{D}_x^*, \mathcal{D}_y^*$. Specifically, $\forall k \ne k' \in [K_1], k_2 \in [K_2]$, we let*

$$M_{2k-1}^* = \boldsymbol{\mu}_k^{+*} = \boldsymbol{a}_k^* + \boldsymbol{b}_k^*, \quad M_{2k}^* = \boldsymbol{\mu}_k^{-*} = \boldsymbol{a}_k^* - \boldsymbol{b}_k^*,$$

$$Q_{2k-1}^* = \boldsymbol{q}_k^{+*} = \boldsymbol{c}_k^* + \boldsymbol{d}_k^*, \quad Q_{2k}^* = \boldsymbol{q}_k^{-*} = \boldsymbol{c}_k^* - \boldsymbol{d}_k^*,$$

$$M_{k_2+2K_1}^* = \boldsymbol{\nu}_{k_2}^*, \quad Q_{k_2+2K_1}^* = \mathbf{0},$$

*where*

$$\boldsymbol{a}_k^* \in conic(\{\frac{\boldsymbol{\mu}_k^+ + \boldsymbol{\mu}_k^-}{2}\}_{k=1}^{K_1}), \quad \boldsymbol{b}_k^* \in conic(\{\frac{\boldsymbol{\mu}_k^+ - \boldsymbol{\mu}_k^-}{2}\}_{k=1}^{K_1}),$$

$$\boldsymbol{c}_k^* \in conic(\{\frac{\boldsymbol{q}_k^+ + \boldsymbol{q}_k^-}{2}\}_{k=1}^{K_1}), \quad \boldsymbol{d}_k^* \in conic(\{\frac{\boldsymbol{q}_k^+ - \boldsymbol{q}_k^-}{2}\}_{k=1}^{K_1}),$$

$$\boldsymbol{\nu}_{k_2}^* \in (span(\boldsymbol{\mu}_1^+, \boldsymbol{\mu}_1^-, \boldsymbol{\mu}_2^+, \boldsymbol{\mu}_2^-, \cdots, \boldsymbol{\mu}_{K_1}^+, \boldsymbol{\mu}_{K_1}^-))^\perp,$$

*satisfying*

$$\|\boldsymbol{b}_k^*\| \ge \|\boldsymbol{a}_k^*\| = \Theta(\|\mathbf{u}\|), \quad \|\boldsymbol{d}_k^*\| \ge \|\boldsymbol{c}_k^*\| = \Theta(\|\mathbf{q}\|), \quad \boldsymbol{\nu}_{k_2}^* = \Theta(\|\mathbf{u}\|),$$

*and $\{\boldsymbol{a}_k^*, \boldsymbol{b}_k^*\}_{k=1}^{K_1}, \{\boldsymbol{c}_k^*, \boldsymbol{d}_k^*\}_{k=1}^{K_1}$ are two collections of pair wise orthogonal vectors. Then we can have a corresponding new prompt distribution $\mathcal{D}_S^* = \sum_{k=1}^{K_1}\big(\pi_k^{+*}\mathcal{P}_{k,L^*+1}^{+*} + \pi_k^{-*}\mathcal{P}_{k,L^*+1}^{-*}\big)$. Again, the model enjoys Bayes-Optimal test error $L_{\mathcal{D}^*}^{0-1}(\Psi^{(T^*)}) \le \varepsilon$.*

This lemma suggest that transformer-mlp structure empower ICL ability in solving task involving semantics ("knowledge") originally from other co-concept prompt's training distribution. This cross-concept semantic "understanding" ability ensure the transformer perform an specific OOD ability.

For example, when we show a prompt "Isaac Newton:Today I designed a machine to capture sunlight; Thomas Edison:" to GPT o1, we would obtain an answer "Today I invented a lamp that shines without fire." During training, even when the concept "Inventors and Their Inventions" may not co-appear with the concept "Fabricate a story" with high chance, the transformers empower the ICL to perform this interesting Out-of-Distribution task. We believe this can serve as an attempt to explain the innovation power of LLM [30, 95, 96] grounded in the linear geometric property of LLM representation, since most of the innovative outcomes of human being generates from cross-concept "Knowledge Intersection", and as it is not an easy task for human specialist to master cross-domain knowledge, we claim that LLM can help innovation by leveraging cross-domain knowledge when deduction over unseen structured task. Similarly, for multi-model scenarios, [86] have shown that compositing different concepts did enable OOD generalization (e.g. "blue square apples" in the Figure 1a in [86]).

This lemma seeks to elementarily explain why LLMs' ICL can excel in complex tasks when using evolutionary strategies, especially when the LLM's latent representation based on language only partially captures the relevant features. Such tasks include algorithm design [97, 4], heuristics [3], acquisition functions [98], and solutions to combinatorial optimization problems [99]. Although the resulting solutions may often seem counterintuitive to human experts, a possible explanation is that transformers can perform ICL in OOD scenarios by leveraging weighted combinations of their updated "understanding" (i.e., changing the identified underlying concepts in the evolution process) of new demo-query pairs, such as randomly sampled TSP instances. These understandings are rooted in the latent structures of the problem instances and can be effectively updated by evolutionary strategies that selectively refine and discard certain outcomes.

*Proof.* Proof of Proposition 1. By Proposition 2, we only need to check the expected 0-1 loss $L_{\mathcal{D}^*}^{0-1}(\mathbb{E}[\Psi']) = 0$. Denote $\mathbb{E}[\mathcal{M}_{y_{S_n}}] \subseteq [2K_1]$ as the expected index set denoting the expected shared concept-specific features by the query and one demonstration. By definition in the Lemma, as the semantic combination is conic combination, we see that $\mathbb{E}[\mathcal{M}_{y_{S_n}}]$ will be either a collection of odd (corresponding to positive label) or even (corresponding to negative label) numbers, and all of the combination of the features and labels in one prompt are corresponding to the same real value label without "self-conflict". By Lemma 38, we see that the coefficients are all at a substantial scale at $T^*$. Then by the condition on $z$ and Eq. (31), we can readily check that even when the probability of the fraction of demonstrations sharing the co-concept label semantic with query is feeble (but at least one), utilizing the same set of notations, we still have

$$\mathbb{E}_{n \in \mathcal{D}^*}\left[\sum_{l \in \mathbb{E}[S_{n,\hat{k}}^{y_{S_n}}]} (\sigma_S^{(T^*)})_l^n\right]$$

$$\geq \Theta\left(\frac{L^*/2 e^{\sum_{\hat{k} \in \mathbb{E}[\mathcal{M}_{y_{S_n}}]} \frac{\beta_{Q,\hat{k}}^{(T^*)} \beta_{K,\hat{k}}^{(T^*)}}{\|b_{\hat{k}}\|^2}}}{L^*/2\left(e^{\sum_{\hat{k} \in \mathbb{E}[\mathcal{M}_{y_{S_n}}]} \frac{\beta_{Q,\hat{k}}^{(T^*)} \beta_{K,\hat{k}}^{(T^*)}}{\|b_{\hat{k}}\|^2}} + e^{(K-1)\sigma_0^2 \|u\|^2 - \sum_{\hat{k} \in \mathbb{E}[\mathcal{M}_{y_{S_n}}]} \frac{\beta_{Q,\hat{k}}^{(T^*)} \beta_{K,\hat{k}}^{(T^*)}}{\|b_{\hat{k}}\|^2}}\right)}\right) \tag{65}$$

$$\geq \Theta\left(\frac{\frac{\|u\|^2}{\lambda K_1} \log(\frac{\|q\|^2}{m\lambda K_1})}{\frac{\|u\|^2}{\lambda K_1} \log(\frac{\|q\|^2}{m\lambda K_1}) + e^{(K-1)\sigma_0^2\|u\|^2}}\right)$$

$$\gg 1/2,$$

where the equality and inequality is by worse-case consideration over $\mathcal{D}_z^*$, a small $\sigma_0$ and $\lambda$ in Condition 1 with a sufficiently large $C$, as well as the requirement $\|b_k^*\| \geq \|a_k^*\| = \Theta(\|u\|)$. Besides, by $\|d_k^*\| \geq \|c_k^*\| = \Theta(\|q\|)$, Eq.(65), Lemma 4, Eq.(5) and Lemma 2, we have that

$$\mathbb{E}_{n \in \mathcal{D}^*}\left[\sum_{i \in \mathcal{W}_{n,\hat{k}}^{y_{S_n}}} r_i\left(\alpha_{O_{(i,\cdot)},\hat{k}}^{(T^*)} + y_{S_n}\left(2 \sum_{l \in \mathbb{E}[S_{n,\hat{k}}^{y_{S_n}}]} (\sigma_S^{(T^*)})_l^n - 1\right)\beta_{O_{(i,\cdot)},\hat{k}}^{(T^*)}\right)\right] \geq \Theta(\kappa),$$

Collaborating with Lemma 43, the poof is completed. □

