# OpenReview forum: "Provably Transformers Harness Multi-Concept Word Semantics for Efficient In-Context Learning"
_NeurIPS.cc/2024/Conference — NeurIPS 2024 poster_

### Official Review · Reviewer_Bq3t · 2024-07-13

**Soundness:** 3
**Presentation:** 2
**Contribution:** 3
**Rating:** 6
**Confidence:** 4

**Summary:**

This paper provides a theoretical analysis of in-context learning (ICL) in transformer models. The authors prove exponential convergence of the 0-1 loss for a three-layer transformer (attention + MLP) trained on a concept-specific prompt distribution. They also demonstrate how the model can leverage multi-concept semantics for out-of-distribution generalization. The analysis connects the geometric properties of concept-encoded representations to ICL capabilities.

**Strengths:**

- First work to show exponential convergence of 0-1 loss for a realistic transformer model: The authors prove exponential convergence for a three-layer model with **softmax** attention and **ReLU** MLP, which is more practical than previous simplified models. This is a significant theoretical advancement.
- Solid mathematical analysis: The proof techniques seem rigorous and well-developed. The authors leverage advanced techniques to handle challenges like softmax attention and logistic loss, which have been difficult to analyze theoretically.
- Experiments support theoretical findings: The simulations in Section 6 validate key aspects of the theory, like the evolution of attention weights and MLP coefficients.
- Good motivation and connections to empirical observations: The authors motivate their work well by connecting to empirical findings on LLM representations (e.g., within-concept positive inner products). This grounds the theory in practical observations.

**Weaknesses:**

- Limited training setup: The paper only trains $W_Q, W_K, W_O$, keeping other weights fixed. This is quite restrictive compared to full transformer training and may limit the applicability of the results.
- Overly specific data model: The concept-specific prompt distribution seems quite structured and simple. It's not clear how well this captures the complexity of real language data. A linear hyperplane can solve all tasks, including OOD ones, in this setup.
- Lack of non-linear task consideration: The analysis focuses on essentially linear classification tasks. It would be more compelling to see how the model handles tasks requiring non-linear decision boundaries or composition of multiple concepts.

**Questions:**

- How sensitive are the results to the specific data distribution assumptions? Would the exponential convergence hold under more general conditions?
- Have you considered extending the analysis to training all weights in the network? What challenges would this introduce?

**Limitations:**

In Appendix J, the authors acknowledge that the data model may need refinement to better align with practical scenarios. They also note that adding more attention layers could make the model more realistic. These are fair limitations to point out.

---

> ### Author Rebuttal · Authors · 2024-08-03
>
> Thank you for the insightful review. We greatly appreciate your acknowledgment of our theoretical advancements, rigorous analysis, well-grounded connections to empirical observations, and fair assessment of the limitations we address.
>
> **Q: Limited training setup & challenges of training all weights.**
>
> **A**: Thank you for raising this point. The reason we left some matrices fixed (e.g., zero entries, $W_V$ and $a$) was to simplify the coupled gradient dynamics and enable rigorous theoretical analysis, which is a common practice in deep learning theory [1-2]. We remark that this restriction does not fundamentally limit the functionality of the transformer with sufficient sample complexity in our learning problem (e.g. one $W_V$ here already can take the full job of $W_O W_V$), which is validated by experiments.
>
> We appreciate your encouragement to consider training all weights. Indeed, it’s known that adding more trainable layers could **lower the sample complexity**, especially for harder learning targets that exhibit hierarchy [3-4]. Extending the analysis to training all weights would certainly pose additional challenges since there’re many gradient updates of various matrices simultaneously. This would require simplification techniques such as layer-wise training and reparameterization [3], as well as special structural assumptions on the concept classes (e.g., information gap) and algorithm configurations (e.g., different regularizers at different phases) [4].
>
> **Q: Alignment of the data model to real-world & applicability to general conditions.**
>
> **A**: Thanks for raising the points. We would like to address your concerns as follows.
>
> - **Empirical Relevance**. Mathematical theories often simplify models to reveal intrinsic capabilities. To better align with the real world, we leverage a sparse coding approach, proposed as a suitable theoretical setup for capturing language polysemy [5], and in our case, seen as a special prompt version of the recognized LDA [6]. Our purpose is to understand how transformers leverage latent linear geometry for certain OOD tasks, modeled after empirical observations on LLM latent structure [7-9]. The practical meanings of the OOD samples are partially validated in [9-10], where different settings show that the combination of concepts forms new meaningful semantics. While there is room for improvement, such as considering hierarchy, we believe our theory stands out and provides an initial positive response to the research question in [11], which asks whether the observed LLM latent geometry can explain their OOD extrapolation abilities.
> - **Applicability**. The exponential convergence result hinges on the hard low-noise condition [12]. Intuitively it suggests that when the target pattern is sufficiently “low-noise”, the exponential convergence with a suitable learning rate can be achieved. We note that strict assumptions are common in theoretical studies to enable rigorous analysis. While our current theory holds under Condition 1, we believe there is scope to relax these assumptions and develop theories for more general data distributions. Looking ahead, a key future direction is to explore analytical techniques that can handle more general conditions.
>
> **Q: Lack of non-linear task & composition of multiple concepts**
>
> **A**: Thank you for this insightful feedback. We would like to address your concerns as follows:
>
> - **Linearity of LLM Representations**. Recent work has shown that high-level concepts in language models tend to be encoded linearly [7-8]. Our theory aims to connect this observed linear structure to the transformer's capability on certain OOD tasks.
> - **OOD Tasks with Multiple Concepts**. We note that our proposed OOD tasks allow for the composition of multiple co-concepts, as stated in the second point of Proposition 1.
> - **Handling Non-linear Tasks**. Given that our transformer model includes non-linear MLPs, which can theoretically handle non-linear tasks [13], it is feasible to consider non-linear tasks. In this context, we expect that the attention mechanisms would still assign more weight to words in demonstrations that share the most similar "non-linear patterns" with the query to complete ICL tasks.
>
> We believe our focus on empirically grounded linear modeling has allowed us to provide the first theories linking multi-concept geometry to the OOD capabilities of transformers, marking an important step forward. However, we agree that exploring whether and how transformers can excel in certain non-linear tasks is another crucial direction for our future endeavors. We appreciate you highlighting this insightful point.
>
> **Reference**
>
> [1] Jelassi et al. Vision Transformers provably learn spatial structure. NeurIPS 2022
>
> [2] Huang et al. In-Context Convergence of Transformers. ICML 2024
>
> [3] Tian et al. JoMA: Demystifying Multilayer Transformers via Joint Dynamics of MLP and Attention. ICLR 2024
>
> [4] Allen-zhu and Li. Backward feature correction: How deep learning performs deep (hierarchical) learning. COLT 2023
>
> [5] Arora et al. Linear algebraic structure of word senses, with applications to polysemy. TACL 2018
>
> [6] Blei et al. Latent Dirichlet Allocation. NIPS 2001
>
> [7] Park et al. 2023: The linear representation hypothesis and the geometry of large language models. ICML 2024
>
> [8] Jiang et al. On the origins of linear representations in large language models. ICML 2024
>
> [9] Yamagiwa et al. Discovering universal geometry in embeddings with ICA. EMNLP 2023
>
> [10] Park et al. The Geometry of Categorical and Hierarchical Concepts in Large Language Models. ICML Workshop MI 2024
>
> [11] Reizinger et al. Position: Understanding LLMs Requires More Than Statistical Generalization. ICML 2024
>
> [12] Massart and Nedelec. Risk Bounds for Statistical Learning. AISTATS 2006
>
> [13] Kou et al. Matching the Statistical Query Lower Bound for k-sparse Parity Problems with Stochastic Gradient Descent. arXiv:2404.12376

---

> > ### Comment · Reviewer_Bq3t · 2024-08-10
> > **Thank you and raise confidence score from 3 to 4**
> >
> > Thank you for your response. I have checked all rebuttals, and they have addressed my concerns. Thus, I raised my confidence score from 3 to 4.
> > The discussion about the **Linearity of LLM Representations** is interesting. The PDF Figure 1 looks nice. It would be good to include them in the main body.

---

> > > ### Author Response · Authors · 2024-08-11
> > > **Thank You for Your Feedback and Our Planned Revisions**
> > >
> > > Dear Reviewer Bq3t,
> > >
> > > Thank you for your positive feedback. We are delighted and encouraged that our rebuttals have addressed all your concerns and that you have raised your confidence score. It is an honor to receive your recognition of the value Figure 1 adds to our presentation. We will appropriately include both the discussion and Figure 1 in the main body based on your constructive suggestion.
> > >
> > > Once again, we truly appreciate your support and insightful comments.
> > >
> > > Warm regards,
> > >
> > > Authors of Submission 15661

---

### Official Review · Reviewer_L2Bx · 2024-07-13

**Soundness:** 3
**Presentation:** 2
**Contribution:** 3
**Rating:** 6
**Confidence:** 1

**Summary:**

This paper tries to understand the mechanisms that explain the emergence of in-context learning.The starting point of their approach is the observation that the embeddings have geometric properties to encode within-concept and cross-concept relationships. Their goal is to connect this geometric properties with the ability to conduct efficient in-context learning. They first describe a data distribution that is sparse coding prompt distribution and then describe their model which is a 3-layer transformer. They show that such a model trained with SGD converge in an exponential speed to the 0-1 Bayes optimal test error. Lastly, they show that their learned model can utilize the multi-concept semantics to conduct unseen ICL tasks.

**Strengths:**

I think that the results are pretty strong and I liked the proposition 1 which shows how the transformer can use its multi-concept semantics knowledge to perform in-context learning on unseen tasks.

**Weaknesses:**

I do not have any knowledge on the topic and so this is why my evaluation is going to be very shallow. Even though the theorems look sound to me, I think the authors could improve the presentation of their results by giving some proof sketch (+drawing) in the introduction.

**Questions:**

I think the authors should improve the presentation of their theory by giving more insights and intuition + simplified proof sketch.

**Limitations:**

The authors did not mention the limitations of their work.

---

> ### Author Rebuttal · Authors · 2024-08-03
>
> Thank you for your positive feedback and recognition of the strength of our theoretical results, particularly Proposition 1 which demonstrates how transformers can leverage their multi-concept semantic knowledge to perform effective unseen ICL tasks. We appreciate your insightful assessment of the significance of our work.
>
> **Q: knowledge on the topic & more insights and intuitions for presentation.**
>
> **A**: We sincerely appreciate your feedback and consider it an honor to share more background on our topic with you. We also value your advice and aim to provide additional insights and intuitions as follows:
>
> - **Understanding LLM**. In recent years, the remarkable power of LLMs has prompted both practitioners and theoreticians to explore their underlying mechanisms. Practitioners validate their claims through large-scale experiments [1-2], while theoreticians use simplified models for analysis [3-6]. A natural practice is that the theoreticians employ mathematics to explain the phenomena observed by practitioners.
> - **Research Gap and Our Contributions**. On the one hand, empirical findings suggest that the emergent power of LLMs mainly stems from their ICL ability [1], which is partially attributed to data properties and transformer structures [2]. On the other hand, existing theories are often based on unrealistic settings (e.g., noise-free orthogonal data, simplified transformer architectures like linear/QK-combined/ReLU/MLP-free/infinite-dimensional models, and square or hinge loss) [3-6]. This leaves room for providing explanations for emergent ICL capabilities in more realistic settings. To advance this understanding, we theoretically demonstrate that transformers leverage latent linear geometry for certain OOD tasks. We model the data based on empirical observations of LLM latent structures [7-9] and consider realistic non-linear transformer architectures and losses (softmax attention, ReLU MLP, and cross-entropy (logistic) loss). Using advanced analytical techniques, we showcase an **exponential** convergence rate for our non-trivial setup, going beyond the linear or sublinear rates achievable in previous simplified setups [3-5] due to their technical limitation.
>
> Interestingly, our work addresses a research question raised in Question 5.1.4 of the ICML 2024 position paper [10], which inquires whether the observed linear latent geometry of LLMs can explain their OOD extrapolation capabilities. We believe our results provide an **initial positive response** to this question.
>
> **Q: provide (simplified) proof sketch(+ drawing) in introduction.**
>
> **A**: Thank you for the valuable feedback. We appreciate your suggestion to enhance the accessibility of our proof sketch with visual aids. In response, we have prepared Figure 1 for inclusion in the introduction, which would be collaborated with a simplified proof sketch in our revised manuscript.
>
> **Simplified Poof Sketch**. Please refer to our illustration (Figure 1) in the new PDF. The Idempotent Operator Techniques depicted allow us to rigorously analyze the model's learning dynamics by examining matrix lengths across orthogonal components, whose practical meanings are partially validated in [11]. In Sections 5.2-3, we extend expectation-variance reduction techniques [12-13]. By treating conditional expectations as Doob martingales, we exploit exponential convergence properties, deriving the rate under low-noise conditions. Our theory suggests that the transformer's learned knowledge, characterized by specific lengths and cross-concept orthogonality, is essential for OOD extrapolation, particularly in prioritizing words that share components with queries.
>
> **Q: The authors did not mention the limitations of their work.**
>
> **A**: We would like to clarify that we have a Limitations Section in Appendix J. There is potential to enhance our modellings to better align with real-world scenarios, such as increasing attention layers or considering the Markovianity of the prompt distribution. We will highlight these limitations in the main body and move Appendix J to Appendix A in our revisions.
>
> **Summary**
>
> We are sincerely grateful for the reviewer’s constructive feedback. Should the reviewer have any further suggestions or wish to discuss any points in more detail, we would be delighted to continue this productive exchange. Once again, we deeply appreciate the reviewer’s time and valuable comments.
>
> **Reference**
>
> [1] Lu et al. Are emergent abilities in large language models just In-Context Learning? ACL 2024
>
> [2] Chan et al. Data Distributional Properties Drive Emergent In-Context Learning in Transformers. NeurIPS 2022
>
> [3] Zhang et. al. Trained transformers learn linear models In-Context. JMLR 2024
>
> [4] Kim and Suzuki. Transformers learn nonlinear features In-Context: nonconvex mean-field dynamics on the attention landscape. ICML 2024
>
> [5] Li et al. How do nonlinear transformers learn and generalize in In-Context Learning? ICML 2024
>
> [6] Chen et al. Training dynamics of multi-head softmax attention for In-Context Learning: emergence, convergence, and optimality. COLT 2024
>
> [7] Park et al. 2023: The linear representation hypothesis and the geometry of large language models. ICML 2024
>
> [8] Jiang et. al. On the origins of linear representations in large language models. ICML 2024
>
> [9] Li et. al. How do transformers learn topic structure: towards a mechanistic understanding. ICML 2023
>
> [10] Reizinger et al. Position: Understanding LLMs Requires More Than Statistical Generalization. ICML 2024
>
> [11] Yamagiwa et al. Discovering universal geometry in embeddings with ICA. EMNLP 2023
>
> [12] Nitanda and Suzuki. Stochastic gradient descent with exponential convergence rates of expected classification errors. AISTATS 2019
>
> [13] Yashima et. al. Exponential convergence rates of classification errors on learning with SGD and random feature. AISTATS 2021

---

> > ### Comment · Reviewer_L2Bx · 2024-08-13
> > **Post rebuttal response**
> >
> > I thank the reviewers for their rebuttal that has addressed most of my concerns. I appreciate the diagram (Figure 1) that allows to get a better sense of the proof sketch and I hope it will be included to the paper. I increase my score by one point.

---

> > > ### Author Response · Authors · 2024-08-13
> > > **Grateful Response to Reviewer's Positive Feedback**
> > >
> > > Dear Reviewer L2Bx,
> > >
> > > We're delighted that you've found our rebuttals have met your satisfaction, and we're encouraged that you've raised the score. We're happy that you found the attached Figure 1 useful, and we would be sure to include it appropriately in the main body of our manuscript.
> > >
> > > Once again, we appreciate your invaluable time and comments.
> > >
> > > Warmest regards,
> > >
> > > Author of Submission 15661

---

### Official Review · Reviewer_dQGe · 2024-07-14

**Soundness:** 3
**Presentation:** 3
**Contribution:** 4
**Rating:** 7
**Confidence:** 4

**Summary:**

This paper performs an in-depth analysis of the optimization dynamics of a simplified Transformer on a sparse coding prompt model. It manages to show the exponential 0-1 loss convergence on this non-convex loss. Experiments on synthetic data verify the convergence.

**Strengths:**

1. This paper is very original and investigates the interesting question of how a Transformer learns shared concepts within context.
2. This paper shows the clear OOD generalization of the Transformer model when trained on the sparse coding distribution.

**Weaknesses:**

1. The notation in the proof sketch is very heavy and not very easy to track. It would improve the paper if more explanation is given. Also, some of the statements seem counterintuitive. For example:

a) Definition 4 states that 'with prob 1 - $\delta$, $L_{D*}(\Phi) = 0$'. What is the source of randomness here?

b) Proposition 4 states minimization of the loss to 0, which seems highly surprising at first pass. This seems like a specialty due to 0-1 test loss considered here and should be highlighted.

2. Some of the legends in Figure 1 is left unexplained and in general the graph may need more explanations on why it matches Lemma 1.

**Questions:**

1. It seems that the proof considering mini-batch SGD is showing a concentration to the population gradient descent by Lemma 3. As mentioned in the intro, one of the key properties of the theoretical claim here is that it doesn't require a large batch size. Why is the technique here so robust to batch size?

**Limitations:**

The limitation is adequately addressed.

---

> ### Author Rebuttal · Authors · 2024-08-03
>
> We are immensely grateful for your thoughtful and insightful feedback on our work. Your recognition of the originality, soundness, and excellent contributions of our paper is deeply encouraging. We found your comments to be highly professional and constructive, and we would like to respond as follows.
>
> **Q: more explanation for the notation of the proof sketch.**
>
> **A**: Thanks for your suggestions. We would be happy to add more insights and illustrations (Figure 1 in the PDF file) to interpret the notations.
>
> The key definitions are:
>
> - $a_k$ is the mean vector of $\mu_k^+$ and $\mu_k^-$, which is denoted as the "concept vector"; $b_k$ is denoted as the "task-specific vector", where $b_k=\mu_k^+ - a_k$. $c_k$ and $d_k$ are defined similarly with regard to $q_k^+$ and $q_k^-$.
> - The coefficients $\alpha_{Q, k}$ and $\beta_{Q, k}$ are defined through the idempotent decomposition of $W_Q$. This allows us to show that ${(W_K\mu_k^{\pm e})}^{T}W_Q\mu_k^e = \alpha_{Q,k}\cdot\alpha_{K,k}\pm\beta_{Q,k}\cdot\beta_{K,k}$ by simple calculations, which pave the way for us to examine the evolution of attention weights by tracking the dynamics of the coefficients. $\alpha_O$ is defined similarly. Our theory then builds up the relationship between the test error of the expected model and the expected coefficients. This would facilitate us to establish the 0-1 loss exponential convergence by closely examining the evolution of these coefficients.
>
> **Q: source of randomness.**
>
> **A**: The primary source of randomness here stems from two factors: the uncertainty in the model's initialization, and the tail behavior of the Gaussian noise, as detailed in Appendix B.1. Our overparameterization conditions ensure that we can control, with high probability 1-δ, the volumes and directional lengths of the matrix parameters, as well as the influence of the noise. Without careful control of these preliminary properties, our theoretical results would not be able to hold rigorously. We appreciate you noting this important detail.
>
> **Q: test error considered in Proposition 4 should be highlighted.**
>
> **A**: We appreciate your intuition about our results. You are correct that, similar to [1], the consideration of the 0-1 loss and treating cross-entropy loss as a surrogate is crucial to establishing the convergence in our Proposition 4. We will certainly highlight this aspect in our revised manuscript.
>
> Another key point is that our proposition examines the expected model behavior at each time step, where the randomness here stems from the stochastic batch sampling in SGD. Given the isotropic noise and balanced positive/negative labels in our setting, the expected model can be viewed as being trained by noise-free gradient descent on a balanced dataset. From this perspective, the expected model's test error can be shown to converge to zero very rapidly, as long as the lengths of the neural network matrices grow sufficiently along the critical directions.
>
> We appreciate your reminder and would highlight the points with interpretations in our revised manuscript.
>
> **Q: more explanation for Figure 1.**
>
> **A**: Thank you for the helpful suggestion. We will replace the existing Figure 1 with a new one (Figure 2 in the supplementary pdf), and ensure the caption provides detailed descriptions.
>
> Specifically, the new Figure 2 includes:
>
> 1. A plot demonstrating the exponential convergence of the test error, validating our theory.
> 2. A plot showing that the correct attention weights for both concepts' classification tasks converge to 1, aligning with the last conclusion of Lemma 1.
> 3. A plot verifying our Lemma 1, as it shows the products $\alpha_{Q, s}^{(t)}\cdot\alpha_{K, s}^{(t)}$ are all non-increasing, while the values of $\beta_{Q, s}^{(t)}\cdot\beta_{K, s}^{(t)}$ can grow sufficiently.
> 4. A plot supporting the claims in Lemma 1 regarding the sufficient growth of the$\lvert\beta_{O_{(i, \cdot)},k}^{(t)}\rvert$ and $\alpha_{O_{(i, \cdot)}, k}^{(t)}$, as well as their relationships.
>
> As our analysis demonstrates the convergence between the expected model and the SGD-updated model, the new Figure 2 clearly illustrates how the empirical results align with and validate the theoretical claims our theories. Additionally, we include supplementary Figure 3 of OOD scenarios to further collaborate our theories. In the final manuscript, we will provide extended experiments and descriptions in the appendix.
>
> **Q: robustness to batch size.**
>
> **A**: The key reason that our technique is robust to batch size, compared to prior work [2], is that we introduce standard techniques from the theoretical literature on exponential convergence [1] to our settings. Specifically, rather than directly using the batch size (at least $\varepsilon^{-2}$, where $\varepsilon$ is the test error) to bound the gradient difference between the expected model and the SGD-updated model via Hoeffding's inequality, as done in [2], our analysis first considers the test error convergence of the expected model. We then examine the rate of convergence for the test error difference between the expected model and the SGD-updated model at each iteration, which could be exponential due to the hard low-noise condition [1]. Crucially, our analysis considers the extreme case where the batch size can be as small as 1, as in [1], to provide an upper bound on the test error. This allows our technique to be robust to the batch size, in contrast with the larger batch size requirements in prior work.
>
> **Reference**
>
> [1] Nitanda and Suzuki. Stochastic gradient descent with exponential convergence rates of expected classification errors. AISTATS 2019
>
> [2] Li et al. How do nonlinear transformers learn and generalize in In-Context Learning? ICML 2024

---

> > ### Comment · Reviewer_dQGe · 2024-08-07
> >
> > Thank the authors for the detailed response. Regarding Weakness 1.a, I still find the argument confusing, if I understand correctly $\Psi$'  refers to a specific weight of the model, and $\Phi*$ contains all the weights that can minimize the 0-1 loss. Why does this have anything to do with noise or initialization?

---

> ### Author Response · Authors · 2024-08-07
> **Clarification on Definition 4 and the Source of Randomness**
>
> Dear Reviewer dQGe,
>
> Thank you very much for your follow-up. We would like to provide an **edited** version of our response for further clarification.
>
> In the original Definition 4, $\Phi^*$ represents all the estimators that can achieve zero 0-1 loss with high probability at least $1-\delta$.
>
> For clarity, we propose removing Definition 4 and directly stating in subsequent results (Proposition 2 and Lemma 1-2) that with high probability at least $1-\delta$, certain results will hold. The reason for these high-probability statements is to avoid extreme cases of initialization and noise, under which our subsequent results would not rigorously hold. The successful evolution of the estimators (whether the expected $E[\Psi']$ or the SGD-updated $\Psi'$) relies on a non-extreme, non-zero initialization.
>
> We apologize for any confusion and appreciate your continuous patience and constructive feedback. We will include more clarifications in our revisions to make this clearer. Please let us know if this addresses your concern.
>
> Warmest regards,
>
> Authors of Submission 15661

---

### Official Review · Reviewer_qjJG · 2024-07-14

**Soundness:** 2
**Presentation:** 2
**Contribution:** 3
**Rating:** 6
**Confidence:** 2

**Summary:**

The paper investigates how transformer-based large language models (LLMs) leverage their in-context learning (ICL) capabilities through the lens of multi-concept semantics. The authors cited limitations in existing theoretical work, which uses oversimplified models and unrealistic loss functions, leading to only linear or sub-linear convergence rates. To address this, the authors present an analysis of a three-layer transformer model, consisting of one attention layer and a ReLU-activated feedforward network, trained with logistic loss. The analysis included an examination of the learning dynamics of the transformer model and proved exponential convergence of the 0-1 loss in this complex setting, demonstrating that the transformer can achieve Bayes optimal test error with a logarithmic number of iterations.
The paper also demonstrated how multi-concept encoded semantic geometry enables transformers to perform efficiently on out-of-distribution ICL tasks, explaining their success in leveraging the polysemous nature of words for diverse, unseen tasks. The paper included empirical simulations to support these theoretical findings.

**Strengths:**

The paper brings a new theoretical perspective linking multi-concept word semantics with in-context learning

**Weaknesses:**

The paper is sometimes hard to follow. More concise and clearer presentation, especially the theoretical content, would be helpful.
Lack of empirical experiments with more realistic datasets to show relevance in real-world setting.

**Questions:**

NA

**Limitations:**

Yes

---

> ### Author Rebuttal · Authors · 2024-08-03
>
> Thank you for acknowledging our work for linking the multi-concept word semantics to transformer’s ICL capability.
>
> **Q: More concise and clearer presentation, especially the theoretical content, would be helpful.**
>
> **A**:  We very much appreciate your suggestion. We will provide more explanations and figures to improve the overall readability. The following are some explanations (which will be extended in the final version).
>
> - **Definitions and Technical Challenge**. As mentioned in the paper, we leverage a sparse coding approach suitable for capturing language polysemy [1-2] and incorporate observed latent geometric properties of LLMs [3-5], distinguishing our work from previous studies that assume idealistic, noise-free orthogonal settings [6]. Our learning problem considers practical non-linear transformer architectures and losses (softmax attention, ReLU MLP, and cross-entropy loss), contrasting with the oversimplified settings (linear/QK-combined/ReLU/MLP-free/infinite-dimensional transformers with square or hinge loss) in prior work [7-8].
> - **Theoretical Achievement**. Our theorem demonstrates an **exponential** convergence rate for our non-trivial setup, going beyond the linear or sublinear rates achievable in previous simplified setups [6-7] due to their technical limitation. Intuitively, our theory suggests the transformer's learned knowledge after training, with certain lengths and cross-concept orthogonality, enables its capability for certain OOD extrapolation, such that the test prompts can enjoy different lengths, various distributions of latent concepts, and even shifts in word semantics.
> - **Proof Sketch**. We prepare Figure 1 to visualize our Idempotent Operator Techniques in the new PDF file. This technique enables us to rigorously analyze the learning dynamics by conducting scale analyses on different orthogonal components of the data, whose practical meanings are partially validated in [9]. In addition, in Sections 5.2-3, we extend standard expectation-variance reduction techniques [10] to our setting. We treat the conditional expectations of the NN matrices as Doob martingales, and by constructing martingale difference sequences and exploiting the exponential convergence property of the tails, we derive the exponential convergence rate under low-noise conditions.
>
> **Q: Alignment to Real-world.**
>
> **A**:  Thank you for your feedback. We would like to address your concerns as follow.
>
> 1. **Empirical Relevance**. Our setting of non-orthogonal dictionaries is inspired by practical observations [4-5, 9], where the LLM's latent geometry exhibits within-concept positive inner products and cross-concept orthogonality, which is also theoretically validated by [3]. Furthermore, according to [1-2], the sparse coding approach is a suitable setup for modelling the polysemy of language.
> 2. **Data Modelling in Feature Learning Theory**. Mathematical theories often deal with simplified modelling, aiming to reveal certain **intrinsic capabilities** of models. There is a rich literature that adopts modelling **similar to ours**, such as studying self-supervised contrastive learning [2], or classification upon orthogonal dictionaries [6].
> 3. **Alignment to the Real-World Setting**. In contrast to prior theories that considered oversimplified settings like noise-free orthogonal feature dictionaries [6] or linear/QK-combined/ReLU/MLP-free/infinite-dimensional transformers [7-8], our setup with non-orthogonal dictionaries, non-linear transformers, and cross-entropy loss is more aligned with practical scenarios. Our goal is to demonstrate the capability of transformers to utilize latent linear geometry for certain OOD tasks. The practical meanings of the OOD samples are partially validated in [9] and Theorem 8 in [11], where different settings show that the combination of concepts forms new meaningful semantics. We believe this provides an **initial positive response** to Question 5.1.4 in the ICML 2024 position paper [12], which asks whether the observed latent geometry of LLMs can explain their OOD extrapolation abilities.
>
> In summary, grounded in practical observations, we study this non-trivial learning problem with **comparatively realistic settings**, and is an important step forward to reveal the ICL emergent capability of transformer [13].  Looking ahead, one of our important future directions include exploring more realistic model settings, such as incorporating additional attention layers, as well as considering the hierarchy of language in our theoretical analysis.
>
> **Reference**
>
> [1] Arora et. al. Linear algebraic structure of word senses, with applications to polysemy. TACL 2018
>
> [2] Wen and Li. Toward understanding the feature learning process of self-supervised contrastive learning. ICML 2021
>
> [3] Li et al. How do transformers learn topic structure: towards a mechanistic understanding. ICML 2023
>
> [4] Park et al. 2023: The linear representation hypothesis and the geometry of large language models. ICML 2024
>
> [5] Jiang et al. On the origins of linear representations in large language models. ICML 2024
>
> [6] Li et al. How do nonlinear transformers learn and generalize in In-Context Learning? ICML 2024
>
> [7] Chen et al. Training dynamics of multi-head softmax attention for In-Context Learning: emergence, convergence, and optimality. COLT 2024
>
> [8] Zhang et al. Trained transformers learn linear models In-Context. JMLR 2024
>
> [9] Yamagiwa et al. Discovering universal geometry in embeddings with ICA. EMNLP 2023
>
> [10] Nitanda and Suzuki. Stochastic gradient descent with exponential convergence rates of expected classification errors. AISTATS 2019
>
> [11] Park et al. The Geometry of Categorical and Hierarchical Concepts in Large Language Models. ICML Workshop MI 2024
>
> [12] Reizinger et al. Position: Understanding LLMs Requires More Than Statistical Generalization. ICML 2024
>
> [13] Lu et al. Are emergent abilities in large language models just In-Context Learning? ACL 2024

---

> > ### Comment · Reviewer_qjJG · 2024-08-13
> > **Thank you**
> >
> > I thank the authors for their response. The authors have edit the manuscript to address my core concerns, I have increased the score accordingly.

---

> > > ### Author Response · Authors · 2024-08-13
> > > **Thank you for your recognition**
> > >
> > > Dear Reviewer qjJG,
> > >
> > > We're greatly encouraged that our rebuttals have been effective in addressing all your core concerns and that you have raised the scores accordingly. We are honored by your recognition and look forward to incorporating the necessary changes.
> > >
> > > Thank you for your invaluable time and comments throughout this process.
> > >
> > > Warmest regards,
> > >
> > > Authors of Submission 15661

---

### Author Rebuttal · Authors · 2024-08-03

Dear ACs and Reviewers,

Thank you again for all of your positive and constructive feedback! We are truly encouraged to see so many well-recognized comments on our work, such as the **innovative angle** (Reviewer qjJG, Reviewer dQGe), **advanced techniques** (Reviewer dQGe, Reviewer Bq3t), **significant theoretical achievement** (Reviewer L2Bx, Reviewer Bq3t), **practical modelings** (Reviewer Bq3t), **sound analysis** (Reviewer L2Bx, Reviewer Bq3t), and **empirically-grounded motivation** (Reviewer Bq3t).

Your insights have greatly helped us strengthen our manuscript. Alongside addressing your comments point-by-point, we have attached a new PDF featuring illustrations of our proving technique (Figure 1) and additional experiments with detailed descriptions (Figures 2-3). We particularly thank Reviewer dQGe for the reminder to provide more interpretation for the notations and Reviewer L2Bx for suggesting visual aids.

Your feedback assures us that we have successfully achieved our main goal: understanding how transformers leverage latent linear geometry for certain out-of-distribution tasks, modeled after the empirical observations on LLM latent structure [1-3]. Interestingly, the research gap we’re addressing aligns with a research question raised in the ICML 2024 position paper [4], regarding whether the observed linear latent geometry of LLMs can be leveraged to explain their OOD extrapolation abilities. By emphasizing your points in our revisions, we believe we can excel as an important step forward and continue contributing to the community in future directions inspired by your valuable comments.

Should the reviewer have any further suggestions or wish to discuss any points in more detail, we would be more than delighted to continue our productive exchange. Once again, we deeply appreciate the reviewer’s time and valuable comments.

Warmest regards,

Authors of Submission 15661

**Reference**

[1] Yamagiwa et al. Discovering universal geometry in embeddings with ICA. EMNLP 2023

[2] Park et al. 2023: The linear representation hypothesis and the geometry of large language models. ICML 2024

[3] Jiang et. al. On the origins of linear representations in large language models. ICML 2024

[4] Reizinger et al. Position: Understanding LLMs Requires More Than Statistical Generalization. ICML 2024

---

### Decision · Program_Chairs · 2024-09-25

**Decision:**

Accept (poster)

**Comment:**

This paper adds to the growing body of working trying to formalize in-context learning of large language models. Despite lower-than-average confidence scores, the reviewers unanimously praised the paper for its originality, theoretical treatment of ICL, and its empirical results. The main concerns pertain to the clarity and presentation of the paper and to some of the assumptions required by its theoretical formulation.

Overall, I agree with the reviewers that this is paper provides an interesting and compelling formalism of LLMs through the lens of mutli-concept word semantics, that leads to a novel and sound interpretation of ICL. Despite its limited evaluation and simplified setting, I believe its strengths outweigh its weaknesses. However, the authors should definitely take into account the reviewer's comments regarding clarity and presentation, since the paper in its current form is unnecessarily opaque.